# Moculus: an immersive virtual reality system for mice incorporating stereo vision

Linda Judák[1,2,10], Gergely Dobos[3,10], Katalin Ócsai[2,4,10], Eszter Báthory[2], Huba Szebik [2], Balázs Tarján [2,5], Pál Maák[6], Zoltán Szadai[1,2], István Takács [1], Balázs Chiovini[1,7], Tibor Lőrincz[1,2], Áron Szepesi[1,2], Botond Roska[2,8,9], Gergely Szalay[1,2,10] & Balázs Rózsa [1,2,7,10] ✉

Due to technical roadblocks, it is unclear how visual circuits represent multiple features or how behaviorally relevant representations are selected for long-term memory. Here we developed Moculus, a head-mounted virtual reality platform for mice that covers the entire visual field, and allows binocular depth perception and full visual immersion. This controllable environment, with three-dimensional (3D) corridors and 3D objects, in combination with 3D acousto-optical imaging, affords rapid visual learning and the uncovering of circuit substrates in one measurement session. Both the control and reinforcement-associated visual cue coding neuronal assemblies are transiently expanded by reinforcement feedback to near-saturation levels. This increases computational capability and allows competition among assemblies that encode behaviorally relevant information. The coding assemblies form partially orthogonal and overlapping clusters centered around hub cells with higher and earlier ramp-like responses, as well as locally increased functional connectivity.

Historically, single-neuron recordings in V1 revealed that cells operate as simple feature detectors[1]; however, brain state-dependent changes such as arousal, attention, locomotion[2–5] and visual learning[6–9] can also modify V1 activity. Several-day-long visual discrimination tasks increased the number of responsive neurons to rewarded gratings (~8–20%) and modestly to nonrewarded gratings (<10%)[6,7,9]. Without discrimination, using a single cue, neuronal activity was upregulated by ~50%[5]. Some studies, however, found no change in orientation preference or neuronal response enhancement[8,10], and even suppression of neighboring orientations[10] or unrewarded stimuli[10]. Such discrepancies in visual-learning studies are common and may stem from factors like neuronal connectivity, representation sparsity, tuning sharpness, learning phases[11], representation stability[12], response variability or changes in spatial distribution of networks after learning[10]. The need for prolonged training protocols may also contribute; although neural circuits can support rapid learning[13], several days are often needed for reliable visual learning, during which factors like sensor expression levels may change, affecting recordings. Additionally, while V1 representations are relatively stable over short timescales, they drift and fluctuate over days[14–18], interfering with the measurement of the direct effects of learning.

To investigate cortical plasticity, virtual reality (VR) combined with head-fixed recordings offers a promising alternative to measurements in freely moving animals, allowing the use of complex recording systems with high spatial and temporal resolution, which are typically too heavy[19] for recordings in freely moving animals. Advances

¹Laboratory of 3D Functional Network and Dendritic Imaging, Institute of Experimental Medicine, Budapest, Hungary. ²BrainVisionCenter Research Institute and Competence Center, Budapest, Hungary. ³Bay Zoltán Nonprofit for Applied Research, Budapest, Hungary. ⁴Department of Algebra and Geometry, Institute of Mathematics, Budapest University of Technology and Economics, Budapest, Hungary. ⁵Doctoral School, Semmelweis University, Budapest, Hungary. ⁶Department of Atomic Physics, Budapest University of Technology and Economics, Budapest, Hungary. ⁷Faculty of Information Technology and Bionics, Pázmány Péter Catholic University, Budapest, Hungary. ⁸Department of Ophthalmology, University of Basel, Basel, Switzerland. ⁹Institute of Molecular and Clinical Ophthalmology Basel, Basel, Switzerland. ¹⁰These authors contributed equally: Linda Judák, Gergely Dobos, Katalin Ócsai, Gergely Szalay, Balázs Rózsa. ✉e-mail: rozsabal@koki.hu

in computational power have enabled real-time creation of complex virtual environments[7,20]. Early rodent systems used flat monitors with predefined optic flow[21], whereas later systems with panoramic projectors incorporated treadmill-based closed-loop feedback. These panoramic projectors are routinely used at present but, regardless of the surface geometry of the visualization device, the 3D space of the VR is first projected into two dimensions and then presented on the planar or curved surface of two-dimensional (2D) monitor or dome screens. This lack of stereoscopic vision limits the immersive nature of the experience. While humans with their high-abstract computing capacity can switch between 2D and 3D representations, this can be a challenge for rodents.

Here we present a compact, head-mounted VR system, Moculus, with behavioral feedback covering the extended visual field of view of mice without interfering physically with the continuous visual flow and immersion during experiments. Moculus offers stereoscopic vision with distortion correction and separate rendering for each eye. This provides depth perception and creates a 3D illusion of the visual world in a compact design compatible with many in vivo recording systems. We have validated stereoscopic vision and fast visual immersion using abyss tests. Combined with 3D acousto-optical (AO) measurements[19,22,23], Moculus enables visual learning protocols based on 3D virtual corridors with 2D grating patterns and 3D objects with grating and nongrating patterns that are >200-fold faster than classical learning protocols. Our validation focused on rapid and parallel formation of reinforcement- and control-zone-coding assemblies during visual discrimination and the stochastic nature of vision.

## Results

### Optomechanical design of Moculus

To cover the full field of view of mice (horizontal, 184.9–284.2°; vertical, 91.2°)[24] with optimal quality and negligible spherical and chromatic aberration, we used an optical model of an adult Bl6 mouse eye[25], combined it with multiple projection optics, and performed detailed optical modeling with parametric optimization in the ZEMAX model (Figs. 1a–d and 2, Extended Data Fig. 1g and Supplementary Note 5). The maximal field of view of the retina was measured using an enucleated albino mouse eye on a 3D-printed resin eye holder (Fig. 2a–e). Aberrations were introduced using irregular surfaces[25] and iteratively optimized with ray tracing to produce sharp projection images with minimal aberrations along the curved retina in the model eye (Fig. 2g). We then computed the chromatic and spherical aberration, tangential field curvature, and the point-spread function as a function of distance from the center along the surface of the retina of a model mouse eye to compare different projection optics (Fig. 2h,i).

Our optical modeling and measurements identified an optical assembly that fulfilled the criteria listed above, as a combination of a custom lens and a diffractive phase shifter (Moculus-S; Fig. 1a–d and Extended Data Fig. 1). The optimized biconvex lens, made of N-BK7 glass, was 1.5-mm thick with radii of curvatures of 1.9 mm and 4.26 mm. The phase plate's radial shifts were optimized up to the fifth coefficient,

resulting in complementary phase coefficients of −283, 404, −259, 163 and −49 radians for the second, fourth, sixth, eighth and tenth power of the radius measured from the center. The optimal distances between the phase plate and the cornea, and between the lens and plate, are 0.5 mm and 0.3 mm, respectively. Moculus-S is based on a 0.6-inch microdisplay (Extended Data Fig. 2a). Alternatively, a bidirectional Fraunhofer display[26] was used for simultaneous image projection and eye movement recording (Extended Data Fig. 2b,h,i). We also developed a simplified version without the phase plate (Moculus-XL; Fig. 2g–i and Extended Data Fig. 1).

An ideal VR headset should tolerate variable eye distances, angles and sizes of animals without interfering with behavior, such as whisking, which could disrupt VR immersion. It must also be compatible with different recording methods. To address these challenges, we employed generative design. First, we created a generalized anatomical model of the mouse body using high-resolution 3D scans taken under anesthesia (Fig. 1a, Extended Data Fig. 1c and Supplementary Note 6). The mouse's real-scale point cloud was captured with a 3D scanner and a structured-light depth sensor, and post-processed with Poisson reconstruction and remeshing[27]. Next, we added constraints based on the recording devices' geometry and the mouse's anatomy, iteratively reducing collisions.

Our approach resulted in a symmetric mounting system with mechanical arms on each side, allowing vertical and horizontal movement, along with a rotating joint for holding the case with the projection optics and display (Moculus-S (Fig. 1a,b) and Moculus-XL (Extended Data Fig. 1a–f)), providing five degrees of mechanical freedom. To measure and correct the angle, position and rotation errors relative to the eyes, we used a calibration device in place of the removable display modules. Z-focusing was validated through retinal imaging with a large-sensor photo camera.

### Distortion correction and optical validation

Due to the large field of view, curved retina and small size, the optical system exhibits substantial pincushion distortion that cannot be corrected by optical design alone. There are three main categories of distortion which are amplified at the edges (Fig. 2d,f and Extended Data Fig. 1i)[28]. To correct this, we used the Brown–Conrady model used in computer vision[29], enabling precise pixel-level correction in real-time. A shader adjusted the rendered images before display, effectively canceling out the lens distortions (Fig. 2f)[28]. The corrected image was calculated using ray tracing, shown on the Moculus's display, and projected onto the retina (Fig. 2e). Naturalistic images demonstrated high imaging quality at the retina (Fig. 2g). Validation was conducted by capturing the corrected retinal image from an enucleated mouse eye with a digital camera and comparing it to the simulation (Fig. 2a–e).

The optical model was validated by examining the image projection of the optical system in both directions. In the forward direction, drifting bars and grids were rendered, presented on the display of Moculus, and projected onto the retina of an enucleated mouse eye through the lens assembly, showing consistent field of view, focus

**Fig. 1 | Moculus, a head-mounted VR display with binocular vision provides full visual immersion. a**, Schematic of Moculus with a head-fixed mouse, a craniotomy and a headplate. **b**, Exploded view of the left side of Moculus: a mount with five degrees of mechanical freedom, a display holder case, a microdisplay, a projection lens and a phase plate. **c**, Cross-section of the Moculus optical path in the ZEMAX model with the projection lens, phase plate and the mouse eye model. **d**, Cross-section view of Moculus with an enucleated albino mouse eye used to measure the monocular field of view, focal plane, image quality and distortion properties. The dashed red line marks the border of the retinal image projection. **e**, Two different projections are calculated and rendered for the left and right eyes. **f**, Left and right display images were rendered at different angles to create stereo vision during the experiments. **g–k**, Abyss experiment to investigate immersion. An untrained mouse immediately stopped

(zero velocity) or recoiled (negative velocity) in Moculus at the edge of the abyss. Mice running on a linear treadmill stopped at the edge of an abyss displayed at the end of a 3D virtual corridor (**g**). 3D overviews of the stereo vision test with Moculus (**h**). The same abyss test but with a classical monitor (**i**). Average velocity for Moculus (red, $n = 7$ trials) and dual monitor (blue, $n = 10$ trials) systems for an example mouse around the edge (at 0 s) of the abyss (**j**). Velocity ratio at the abyss edge showing significant differences between Moculus and dual monitor experiments (bars indicate mean ± s.e.m., $n = 6$ mice, $P = 0.026$, two-tailed Mann–Whitney $U$-test) (**k**). The velocity ratio for the single-monitor system is also plotted ($n = 4$ mice), showing a significant difference compared to the Moculus ($P = 0.0095$, $U$-test), but not significant compared to the dual monitor experiments ($P = 0.9143$, $U$-test). Detailed statistics are in Source Data.

plane, image quality and distortion between the optical simulation and the real projection (Fig. 2a–e). In the backward direction, a sharp retinal image was captured in vivo by back illuminating with a green LED through the cranial window and detecting the projection at the display plane using a 1-inch camera sensor. For Moculus-S, which has a bidirectional display, image formation was also tested directly using the display's original driver (Extended Data Fig. 2b). The optical system produced a sharp retinal image, ensuring perfect focus on the retina

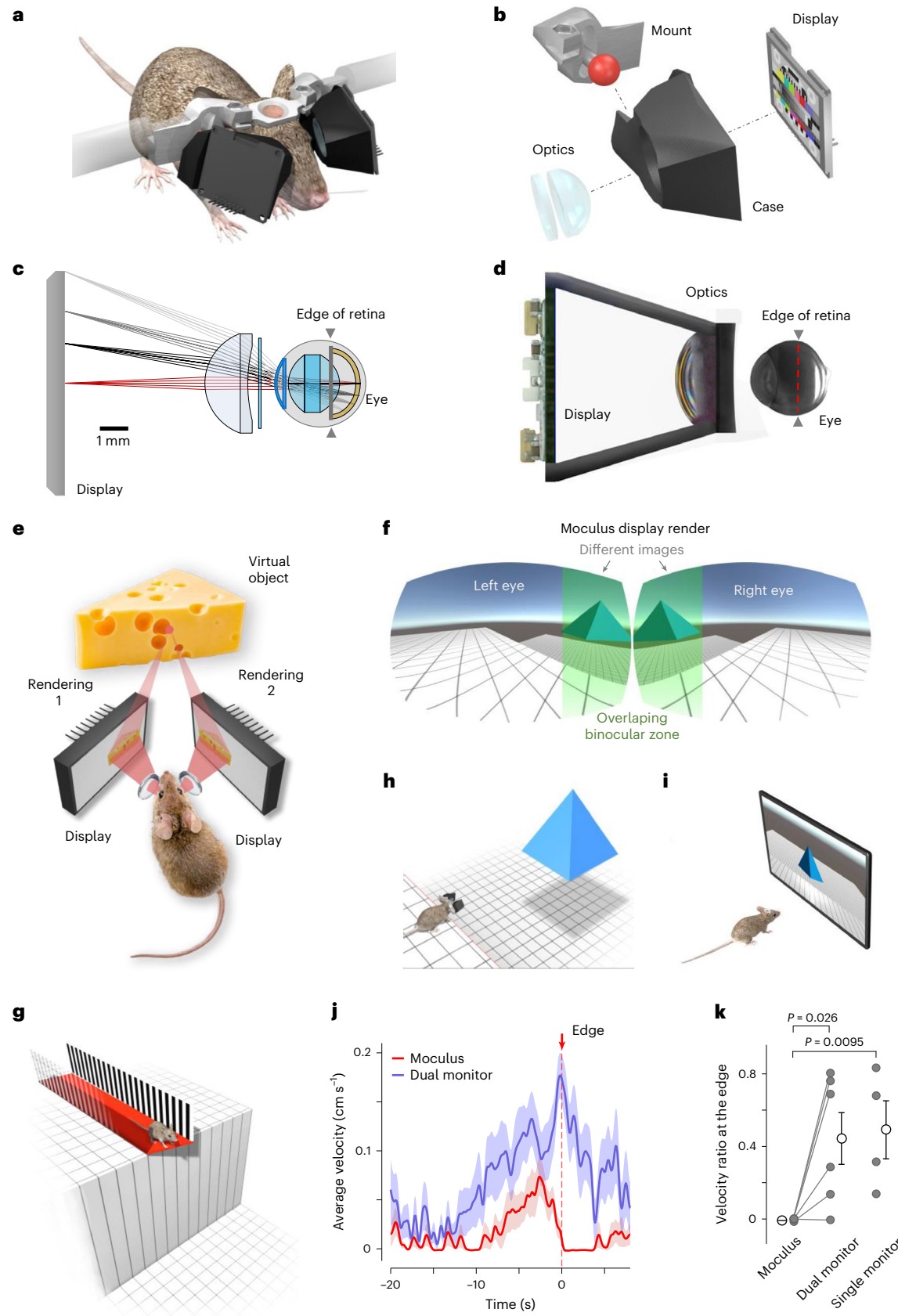

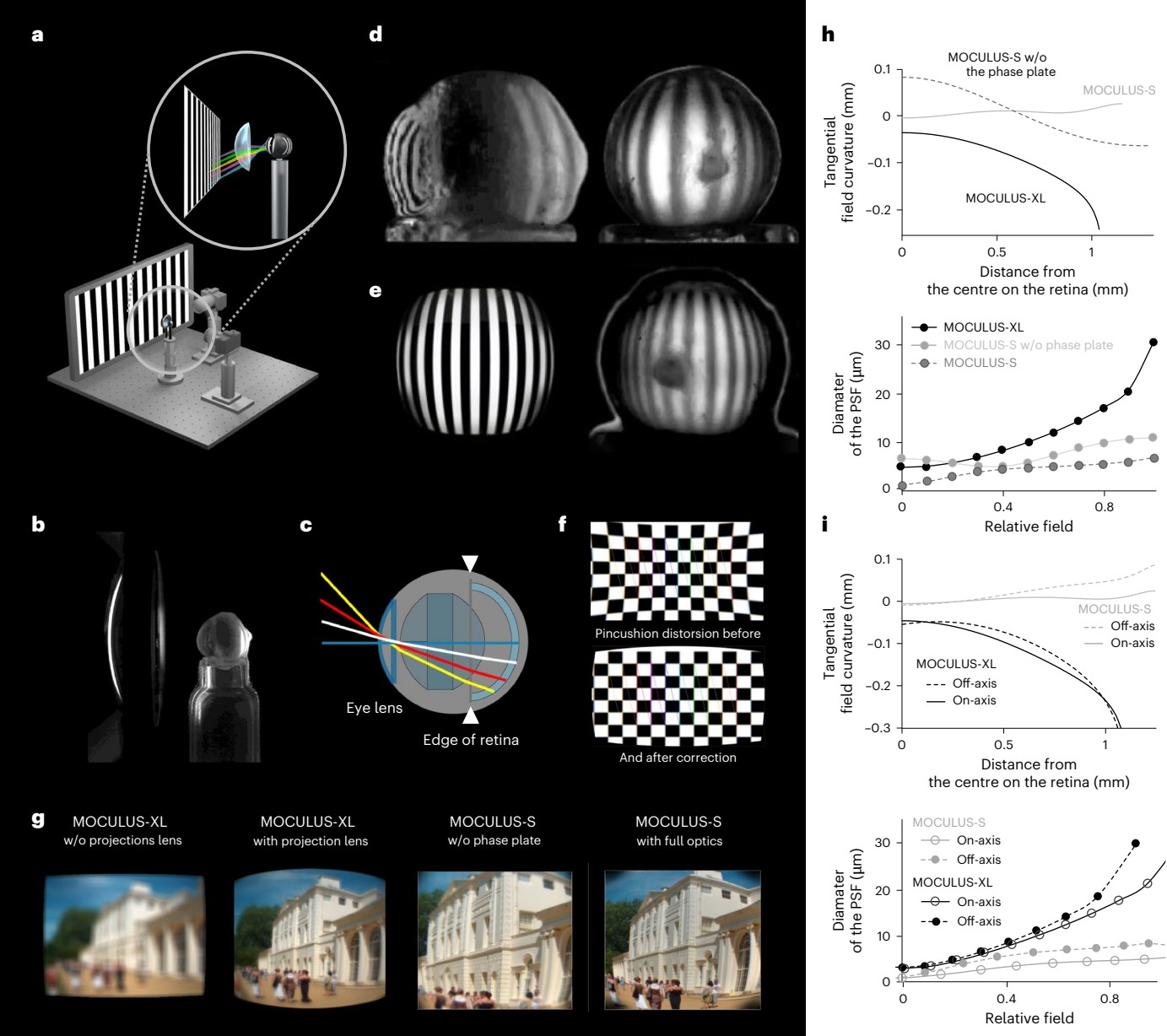

**Fig. 2 | Validation of the optical design of Moculus using an enucleated albino mouse eye. a**, The experimental setup with the enucleated eye, a projection lens, a monitor screen and two cameras. The enucleated eye was imaged through the two cameras (lateral and back view). The projection lens focused the screen on the retina. **b**, The enucleated albino eye with a test lens (Supplementary Video 2). **c**, ZEMAX model of the mouse eye. **d**, Lateral and back view of the enucleated albino mouse eye during projection. The gray patch indicates optical nerve stump. **e**, Ray-traced simulation of retinal projection with the distortion-corrected optical system of Moculus with a denser grid size (left). Corresponding image of the enucleated albino mouse eye with the distortion-corrected figure on the retina (right). **f**, Pincushion distortion of the Moculus-XL optical system (top). Corrected pincushion distortion of the optical system (bottom). **g**, Ray-traced simulation of the distortion-corrected optical system in ZEMAX during projection of an image onto the retina of the model mouse eye with Moculus-XL

and Moculus-S. Moculus-XL without (w/o) the projection lens and Moculus-S w/o the phase plate are also shown. **h**, Tangential field curvature from the surface of the retina as a function of distance from the center of the retina for Moculus-S and Moculus-XL (top). Moculus-S is also shown w/o the phase plate (dashed). Diameter of the point-spread function (PSF) as a function of relative field of view for Moculus-S and-XL (bottom). Moculus-S is also shown w/o the phase plate. **i**, Tangential field curvature from the surface of the retina as a function of distance from the center of the retina for Moculus-S and-XL in cases of off-axis and on-axis beams (top). Diameter of the PSF as a function of relative field of view for Moculus-S and-XL in cases of off-axis and on-axis beams (bottom). On-axis retinal position is when the optical axes of the eyes and the optical systems coincide and off-axis position represents 7.5° rotation of the eye, which is the maximum naturally occurring deflection relative to the central position.

(Extended Data Fig. 2h). The optomechanical design was further validated through retinotopic mapping (Extended Data Fig. 1l–r).

**Virtual environment, rendering and validation of immersion**

We used the Unity3D game engine to create VR environments (Fig. 1e,f and Extended Data Fig. 3), with 3D mazes, including corridors, abysses

and walls with various 2D patterns oriented for perspective correction (Fig. 3a,c). 3D objects with (Fig. 4d,e) or without (Fig. 1f,h) grating and nongrating patterns (Extended Data Fig. 4), were added to support stereo vision, depth perception and full immersion. We also used a flying bird of prey to trigger freezing and escaping behavior as a test of mouse vision (Extended Data Fig. 3a,g and Supplementary Video 1).

During the stereographic rendering of the virtual environment, a virtual mouse (the Moculus character) navigated in the space, with images for the left and right displays rendered from its viewpoint (Fig. 1e–g, Extended Data Fig. 3a,f,g and Supplementary Video 1). The VR platform communicated with the imaging system, lick port, linear or circular treadmills and other input/output peripherals. Motion was recorded using a linear or circular treadmill with an optical encoder.

The key requirement of a VR system is to create a sense of presence, where the subject perceives the virtual environment as real[30,31]. To test this, we implemented an elevated maze test in the VR, where mice had to stop at the edge of a cliff to avoid 'falling' (abyss test; Fig. 1f,h). This experiment was repeated with single and dual monitor arrangements (Fig. 1i), where mice were more likely to run across the gap (Fig. 1j,k and Extended Data Fig. 5e–g). We added 3D objects to the VR, to enhance 3D vision and depth perception (Fig. 1f,h and Extended Data Fig. 3). Mice that ran slowly and did not reach the edge of the virtual cliff were excluded from analysis ($n = 2$ of 17 mice). Of the remaining mice, $n = 6$ were tested with both Moculus and dual monitors, $n = 4$ with a single monitor only and $n = 5$ with Moculus only. The average velocities decreased near the edge of the abyss (Fig. 1j and Extended Data Fig. 5e–g). Speed reduction at the edge was quantified as the ratio of the average velocity before and after the edge in 2-s intervals and was significant with Moculus ($P = 0.0095$, Moculus versus dual monitor and $P = 0.026$ versus single monitor; two-tailed Mann–Whitney $U$-test), indicating better depth perception compared to monitors (Fig. 1k and Extended Data Fig. 5e).

Some mice even moved backward to avoid 'failing' (negative speed at the edge; Extended Data Fig. 5b,c), confirming immediate immersion with Moculus. In contrast, running speeds of mice did not decrease to zero, or even showed transient increases with the single or dual monitor systems (Fig. 1j and Extended Data Fig. 5e–g), indicating lack of fear of depth. A control experiment, where the abyss location was randomized while keeping the length of the maze wall constant, confirmed that mice stopped only at the cliff edge (Extended Data Fig. 5a–g). In summary, our data indicate that the mice perceived their virtual environment as real, with effective depth perception, when using Moculus, unlike standard VRs with large monitors[31].

## Moculus is combined with 3D microscopy

Next, we investigated how Moculus facilitates learning by combining it with a treadmill and 3D AO two-photon microscopy (Fig. 3a,b), enabling neuronal network imaging with orders-of-magnitude higher temporal resolution and signal-to-noise ratio than standard 2D laser microscopy (Supplementary Table 1). To allow neuropil decontamination and maximize the signal-to-noise ratio of the individual trials during in vivo measurement, we used a high-NA objective at the cost of a limited maximal field of view (~650 μm)[22], and combined 3D chessboard scanning (Fig. 3b) with motion correction.

We designed a pattern-discrimination task in Moculus, where mice navigated in a virtual corridor with a natural grayscale pattern interrupted with by 2D frames with grating patterns at randomized positions, oriented in 3D with perspective correction according to the geometry of the 3D corridor (0°, 45° and 135° zones; Fig. 3c). One zone (0°) was associated with an aversive stimulus (airpuff), if the mouse did not escape within 3.5 s. Randomized teleportation into the aversive zone ensured a definite time interval for decision-making before the aversive stimulus. Bootstrapping analysis confirmed that visual learning was faster with Moculus than with standard monitor systems (Fig. 3g and Extended Data Fig. 5k–n). Mice trained with Moculus performed better after 30–40 min, with the effect sustained over subsequent training days.

## Visual learning within 20–40 min using 3D objects

In the first phase of the task (0–10 min training, 'early learning'; Fig. 3e), mice showed a significant increase in velocity at the end of both aversive and control zones ($P = 0.0074$ and 0.01, respectively, one-tailed paired $t$-test, detailed statistics are in Source Data), but with no significant difference between the two ($P = 0.14$), indicating they detected task-relevant visual information but could not discriminate between the two orientations. Learning was quantified before the onset time of the airpuff to avoid indirect effects. In the later phase (30–40 min, 'after learning'; Fig. 3e), velocity increased further in both zones, with a significant difference at the end of the aversive zone compared to control zones ($P = 0.023$) and the difference persisted on subsequent experimental days (Extended Data Fig. 5h–n). These data show that mice trained using Moculus could partially, but significantly, discriminate between orientations after only 40 min of training.

To validate rapid visual learning with more natural components of the rodents' natural environment, we introduced 3D objects in the VR and used positive reinforcement (Fig. 4d,e). Two reward locations were introduced in the reward zone: the second reward was provided continuously at the end of the zone. In contrast, the first reward was only given if the mouse started licking before the second reward location. The licking rate increased rapidly before the reward zone ('anticipatory licking') during the first experimental day and remained stable during the following experimental days, as shown in 10-min intervals (Extended Data Fig. 6). Performance was more stable than with the previous protocol using a negative reinforcement signal. Mice learned to locate a reward zone (defined as task engagement) in 20 min, which is about 288–432-fold faster learning speed than with classical monitor systems (Moculus, 20 min; 2D monitors, '4–6 days')[5].

Next, reward and control zones both with 3D objects with grating patterns were added in the real 3D virtual corridor (Fig. 4d,e). Anticipatory licking rate increased significantly at the reward zone after just

**Fig. 3 | Fast visual learning with Moculus related to population level neuronal response. a**, Schematic of the measurements combining Moculus with 3D AO imaging and a treadmill. **b**, Maximal intensity projections of a z-stack with somata selected for 3D chessboard scanning ($n = 85$, GCaMP6f-labeled neurons). **c**, Top, walls of the VR in Moculus had a neutral pattern with zones of gratings (0°, 45° and 135°) in randomized positions. The 0° zone was associated with an aversive stimulus (airpuff) if mice did not escape within 3.5 s (bottom). **d**, Average 3D Ca²⁺ responses in the aversive (0°) and control (45°) zones during pre-learning, early-learning (0–10 min) and late-learning (30–40 min) phases in the same mouse. Neurons were sorted by response amplitudes at the zone ends. Dashed red lines indicate the zone ends. Gratings indicate visual stimulation zones. **e**, Average (mean ± s.e.m.) running speed during pre-learning ($n = 10$ trials), early- (0–10 min, $n = 9$ trials) and late- (30–40 min, $n = 7$ trials) learning phases in the aversive zone (0°, red) for the same mouse with speed also shown for the control zones (45°; gray; before learning: $n = 16$ trials, 0–10 min: $n = 12$ trials, 30–40 min: $n = 7$ trials). Blue asterisk and bar indicate significant speed difference between the control and aversive zones (Student's $t$-test, $P$ threshold for significance was

0.05). Red bars indicate airpuffs. **f**, Average (mean ± s.e.m.) amplitude of the ramp-like component during pre-learning ($n = 850$ transients and 10 trials), early (0–10 min, $n = 765$ transients and 9 trials) and late (30–40 min, $n = 595$ transients and 7 trials) learning phases in the aversive (red) and control (gray) zones in the same mouse (pre-learning, $n = 1,360$ transients and 16 trials; 0–10 min, $n = 935$ transients and 11 trials; 30–40 min, $n = 595$ transients and 7 trials; two-way analysis of variance (ANOVA) for 0–40 min: $P_{learning} = 6.9 \times 10^{-12}$, $P_{AVERvsCTRL} = 0.001$, $P_{interaction} = 7.9 \times 10^{-4}$, $P_{model} = 3.2 \times 10^{-12}$; for day 5 all $P < 10^{-10}$). **g**, Training scores for visual learning with Moculus (blue, $n = 6$ mice) versus a monitor system (red, $n = 3$ mice) (left). Average running speeds ($n = 5$ mice, mean ± s.d.) as a function of time of learning (two-way ANOVA for 0–40 min: $P_{learning} = 1.80 \times 10^{-5}$, $P_{model} = 1.97 \times 10^{-5}$, $P_{AVERvsCTRL} = 0.00756$; for day 5 all $P < 10^{-6}$) (right). **h**, Corresponding population averages ($n = 5$ mice, mean + s.d.). Ramp amplitudes integrated from −1 s to −0.4 s in the aversive (red) and control (gray) zones as a function of time of learning. Exact $P$ values can be found in Source Data. A two-tailed $t$-test was used as the statistical test. Box-and-whisker plots show the median, 25th and 75th percentiles, range of nonoutliers and outliers (**g,h**).

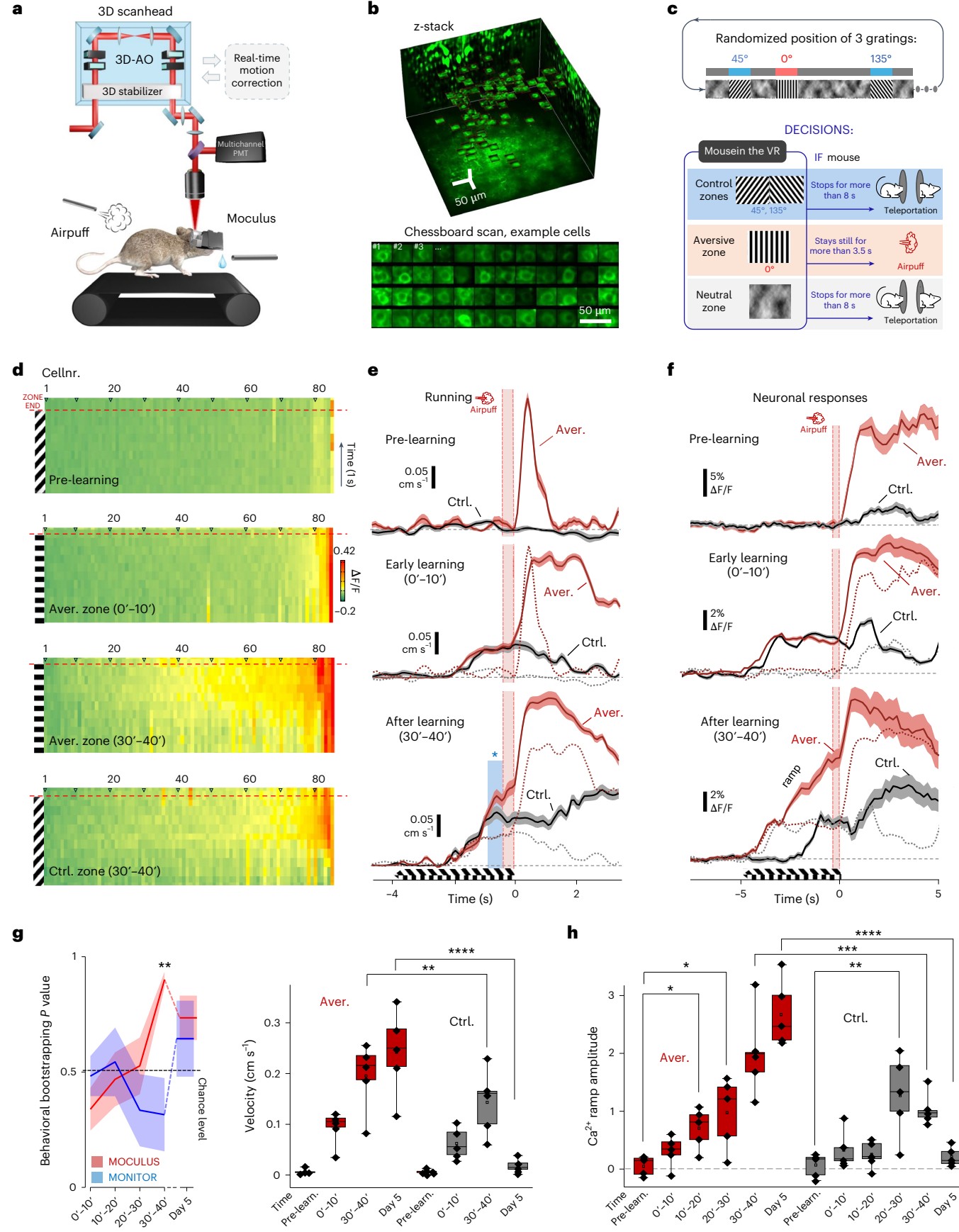

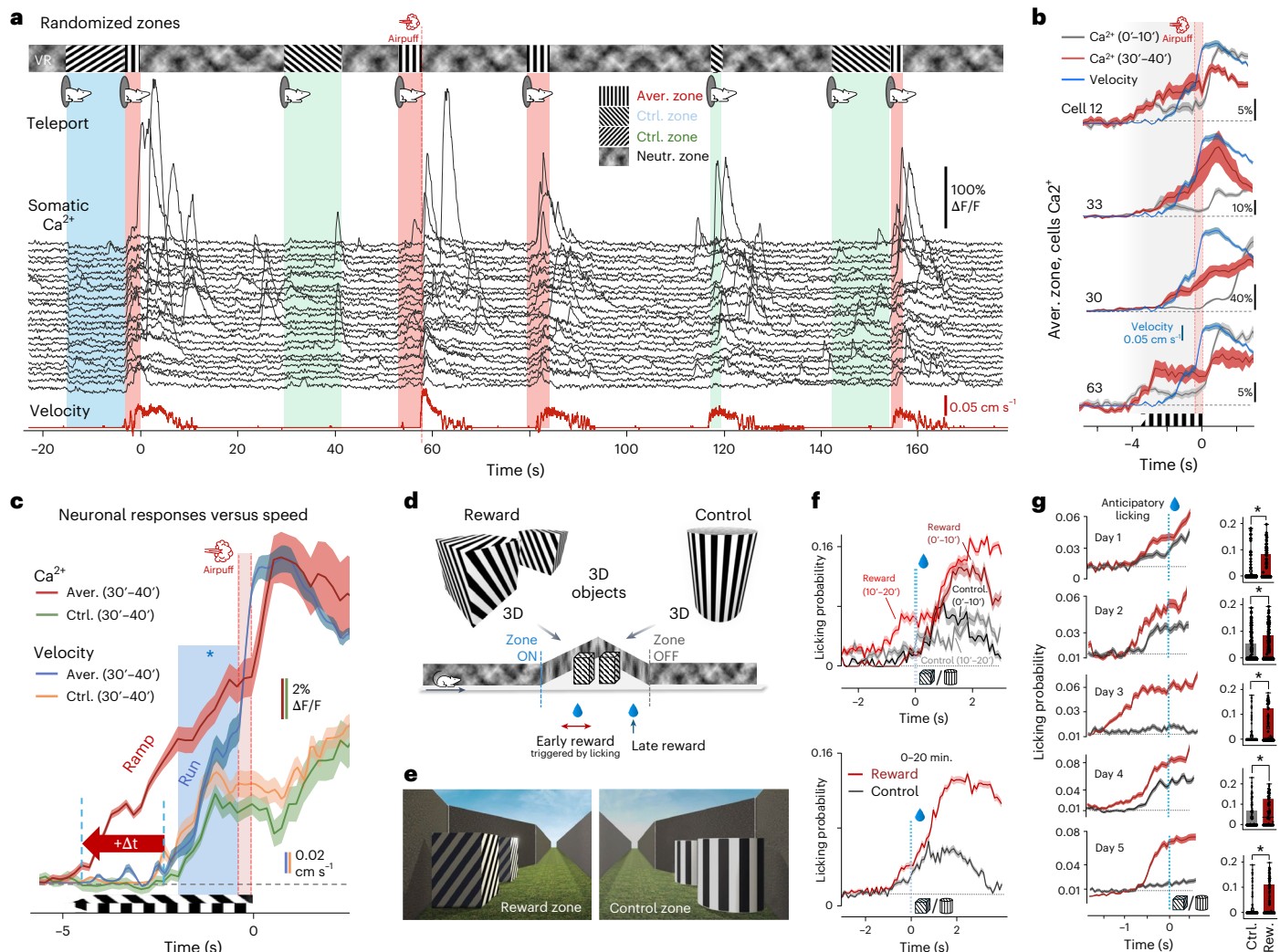

**Fig. 4 | Fast visual learning with Moculus in 3D virtual corridors. a**, Example transients (*n* = 24 from 85 cells) with the combined 3D AO measurement of the mouse shown in Fig. 3b–f in the 3D virtual corridor decorated with 2D patterns oriented and rendered according to the perspective of the true 3D corridor. Colored bars indicate the control and aversive zones. Teleportation and airpuff are indicated with the same icons as in Fig. 3c. **b**, Average (mean ± s.e.m.) Ca²⁺ responses of four example neurons from Fig. 3d during the early (0–10 min) and late (30–40 min) learning phases in the aversive zone. Transients were aligned to the end of the zones (0 s) before averaging. The corresponding average running speed is indicated in blue. **c**, Average velocity and population responses during the late (30–40 min) learning phase overlaid from Fig. 3e,g indicate time delay. **d**, Schematic of visual discriminative learning with 3D objects in virtual corridors. V-shaped notches and grating patterns marked reward (3D cuboids with oblique grating) and control zones (3D cylinders with vertical grating).

The first reward was given after initial licking, while the second reward was provided if the first reward was missed. **e**, Examples of reward and control zones from the mice's viewpoint. **f**, Licking rate at the beginning of the reward and control zones (0 s) after 0–20 min learning from an exemplified mouse (bottom). The same as the bottom panel but separated for the first and second 10-min intervals (top). **g**, Licking rates at the beginning of the reward (red) and control (gray) zones (0 s) after 0–20 min (day 1) and 0–30 min training (day 2–5). Inset, bar graphs showing average anticipatory licking rates (mean ± s.e.m.) before entering control and reward zones. Data from *n* = 5 mice, minimum 146 and 150 trials per day in control and reward zones, (mean = 161.8 and 168.8, s.d. = 19.79 and 15.69). Asterisk indicates a significant difference between the two zones (one-tailed *t*-test, $P_{day1}$ = 0.032, $P_{day2}$ = 0.0067, $P_{day3}$ = 1.41 × 10⁻⁸, $P_{day4}$ = 0.00038 and $P_{day5}$ = 1.80 × 10⁻⁸).

20–30 min training compared to the control zone (*P* = 0.01, Wilcoxon signed-rank test; Fig. 4f). Visual learning was stable and maintained during the next experimental days (Extended Data Fig. 4g). With classical monitor systems, mice learned the visual discrimination task in 3–6 days[6,9,32] and anticipatory signals appeared after 5–6 days training[9], indicating that visual discriminative learning with Moculus using 3D corridors with 3D objects is more than 144–288-fold faster. Finally, we found that 3D objects with nongrating patterns, floating in the upper middle part in the binocular zone in the control and reward zones, were able to induce discriminative learning in ~20 min, with learning remaining stable over the following days (Extended Data Fig. 4a,b).

## Competition between aversive-zone and control-zone coding during fast learning

We analyzed how rapid learning with Moculus is represented at the level of V1 neuronal assemblies. We used the first training protocol with grating patterns for comparability with existing literature[6,7,9,10]. The fast time course of Moculus-based learning allowed to examine before and after periods within a single session. At the beginning of learning (0–10 min, 'early learning'), only a small fraction of neurons was active relative to the data recorded before learning ('before learning'; Fig. 3d,f,h), reflecting sparse coding theory[33], with no significant difference between the two zones (*P* = 0.867, two-tailed, paired *t*-test). In the second phase (30–40 min, 'after learning'; Fig. 3d,f,h), population

coding was enhanced in parallel with the significant increase in velocity ($P = 0.045$; Fig. 3e,g and Extended Data Fig. 5h–n): both the number of responsive cells and the amplitude of neuronal responses increased similarly in both zones to a near-saturating activation ratio (Figs. 3d and 5a–e); however, responses became significantly larger at the end of the aversive zone compared to the control zone ($P = 0.015$; Fig. 3d,f,h and 5a–e). Most active cells showed a ramp-like activity increase after learning, starting a few seconds before the end of the aversive zone (Figs. 3d and 5a,b,e). The ramp-like activity increase was also evident at the population level (Fig. 3d,f,h) and maintained during subsequent experimental days (Fig. 3h and Extended Data Fig. 7). Similar to anticipatory signals, ramp-like neuronal activity could be generated 'de novo' (Fig. 3f,h), but ramp signals emerged also in the control zone, involved neurons in a near-saturating ratio, and were not associated with specific visual patterns of the neutral zones, as shown for anticipatory signals[9].

During the ramp period the activation ratio quickly increased from a baseline of 4.7% to near saturation (97.6–100%) before returning to baseline (Extended Data Fig. 8a–c). The periodic motion in the VR resulted in average and baseline activation ratios of 38.3 ± 4.0% and 2.67 ± 0.76%, respectively (mean ± s.e.m.; Extended Data Fig. 8c). Unlike previous studies with visual discrimination where only ~8–20%[6,7,9,34] of neurons responded to reinforcement-associated gratings or showed no significant change[8,10], we found that 80.1 ± 5.9 % of the cells were activated at the end of the aversive zone (Fig. 5e and Extended Data Fig. 8d–i). Similarly, while previous studies reported modest (<10%) or no activation for the control cues[10], we observed 70.0 ± 7% of the cells activated in the control zone (Fig. 5e). At the population level, 44% of cells coded more for the aversive zone and 14% for the control zone, indicating a dominance over control zone coding (Fig. 5e and consistent with Figs. 3–5). These findings show that fast learning can generate anticipatory signal-like ramp components within ~30 min by recruiting neurons that encode both the aversive and control visual cues at near-saturating levels with a dominance of aversive encoding.

Our data indicated that the ramp-like increases is an efficient metric of fast visual learning. Therefore, we quantified the temporal development of this component. The average of the ramp components across the population increased significantly within the first 30 min of training (Fig. 3h), in both the aversive and the control zones ($P = 0.046$ and $P = 0.037$, respectively, two-tailed, paired $t$-test), but were not significantly different between the two zones ($P = 0.592$); however, at the late phase of learning (30–40 min), the ramp-like component increased significantly in the aversive zone compared to the control zone ($P = 0.015$; Fig. 3h). This trend in pattern separation continued during subsequent experimental days, until the ramp signals disappeared in the control zone but were maintained in the aversive zone (Fig. 3h and Extended Data Fig. 7). This continuous transition, which resulted in dominance of aversive-zone coding over control-zone coding, was accompanied by a similar transition in behavior (Fig. 3g and Extended

Data Fig. 5h–n). In summary, these data indicated that aversive-zone coding acquired the encoding from control-zone coding during learning and became the dominant coding in the V1 region, in parallel with improvements in behavior.

## Coding assemblies validated by decoder form partially overlapping clusters in 3D

Both aversive- and control-zone coding recruited almost all neurons (Fig. 3d), which became evident when activity after learning was subtracted from the data recorded before learning and significance was calculated at the level of individual neurons (two-tailed, paired $t$-test; Fig. 5a,b). Thus, the question arises whether population coding is sufficiently orthogonal. To address this, we sorted neurons as a function of their ramp amplitudes at the end of the aversive zone (Fig. 5c). Although neuronal ramp-like components were correlated between the aversive and control zones, responses of the same cells in the control zone fluctuated (Fig. 5c), indicating that coding itself has also an orthogonal component. Consistent with this, average activity of only three cells (defined as a decoder function) from the top end of the distribution (Fig. 5c) was able to separate aversive- and control-zone-coding assemblies at the level of individual trials (Extended Data Fig. 9a–e). Specifically, if the integral of the decoder function (from −4 to −2.5 s) was above a given threshold, mice were in the aversive zone. The orthogonality was also evident at the population level: decoder functions generated from the average activity from three to seven cells were able to separate aversive and control-zone-coding assemblies at the level of individual trials with high reliability (Fig. 5d; 97.5 ± 2.5%, $n = 5$).

The distribution of the learning-associated ramp-like responses of individual neurons showed a jump at an amplitude threshold at the top of the amplitude range as indicated by the first derivative (Fig. 5c,d), suggesting robust learning in a subpopulation (Fig. 5c,d). Sensory modalities like orientation or direction tuning are often coded in well-orchestrated spatial clusters[35]. Supporting, newly formed spatial receptive fields of hippocampal neurons have been shown to recruit co-active presynaptic neurons, which jointly encode the physical location of the animal[13]. We found that both aversive and control orientation coding neurons above the amplitude threshold also form spatial clusters in V1 (Fig. 5f,g), indicating overlapping and orthogonal coding in space (Fig. 5f) and time (Fig. 5b–e). These neurons can be defined as hub cells, initiated responses over 1 s earlier and with higher amplitudes than those below the threshold (Fig. 5h), and their average response (Extended Data Fig. 9f, g) mirrored population's ramping activity (Fig. 3f), likely due to the locally increased connectivity of hub cells, enabling them to collect, integrate and, thus, mimic local network activity[36]. Accordingly, hub cells with higher ramp-like components showed increased cross-correlation in spontaneous activity outside the aversive zone (Fig. 5i), indicating stronger interconnections, because higher cross-correlation in spontaneous activity was shown to be associated with higher functional connectivity[37]. The ten

**Fig. 5 | Competing neuronal networks form clusters centered around hub cells with higher and earlier ramp-like responses and locally increased connectivity. a**, Neurons from Fig. 3d. were analyzed by subtracting early (0–10 min) from late (30–40 min) 3D Ca²⁺ responses, aligned to the zone ends (dashed), sorted by amplitude differences. **b**, Responses below the 2 × s.e.m. threshold are indicated with gray. **c**, Integrals (from −1 s to −0.3 s) of the 3D Ca²⁺ responses at the aversive and control zone ends, sorted by aversive zone amplitude. A triangle indicates a jump in the amplitude distribution. **d**, Similar to **c** but for $n = 5$ mice (mean ± s.e.m.), with the amplitude threshold (black triangle). The gray trace shows the first derivative. **e**, Responses aligned to zone ends (dashed red), averaged, control zone responses subtracted and sorted by difference (top). Representative cells with aversive and control zone preference (AVER and CTRL coding) from the two ends of the distribution (middle). Blue curve indicates the average running speed. Average activation ratio (left, $n = 5$ mice, light colors show ±s.e.m.) and dominant coding cells (right) (bottom).

**f**, Aversive (red) and control (green) zone-coding neurons above the amplitude threshold formed spatial clusters (hub cells). Enlarged spheres indicate neurons in both groups. **g**, Average Euclidian distance among hub cells (red) versus coding random neurons (blue) in $n = 5$ mice (mean ± s.e.m.) (left). Normalized distances for hub cells and random pairs (right). **h**, Average activity (mean ± s.e.m.) of the ten hub cells (red) that coded the aversive zone, preceded population activity (blue) by >1 s. Transients were aligned to the zone end before averaging. **i**, Cross-correlation matrix of neuronal activity at the end of fast learning (30–40 min) for Fig. 3d cells (left). Cross-correlation coefficient (mean ± s.e.m.) of spontaneous activity for hub cells (red), for the ten cells with the lowest ramp-like components (gray) and for the rest (blue); $P$ values indicate two-tailed $t$-tests (right). **j**, Cross-correlation for the ten best and worst learner cells from an example mouse, with significance by two-tailed $t$-test (left). Average cross-correlation matrix ($n = 5$ mice), showing differences in cross-correlation and connectivity between best (red) and worst (gray) learners and the rest (middle–right).

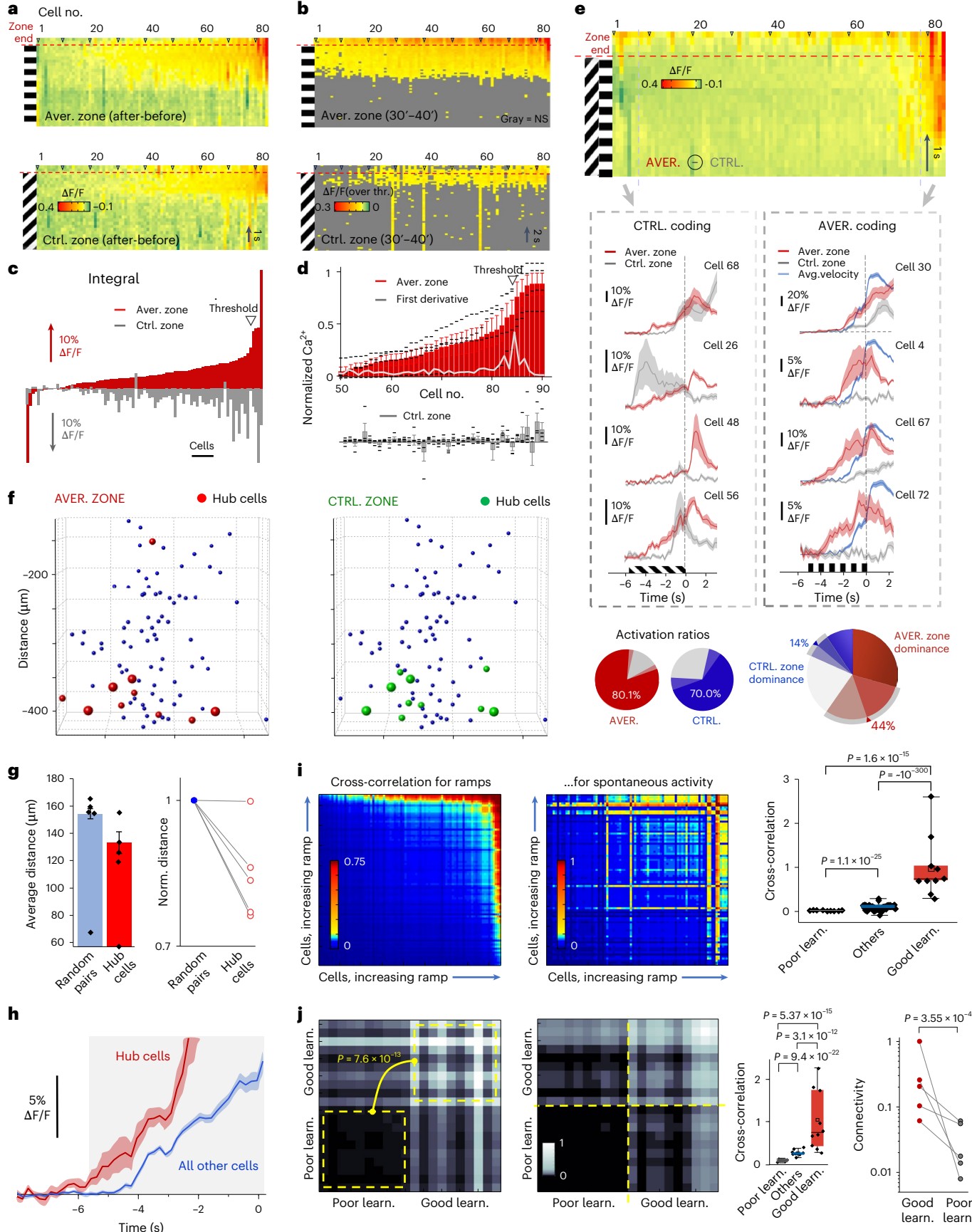

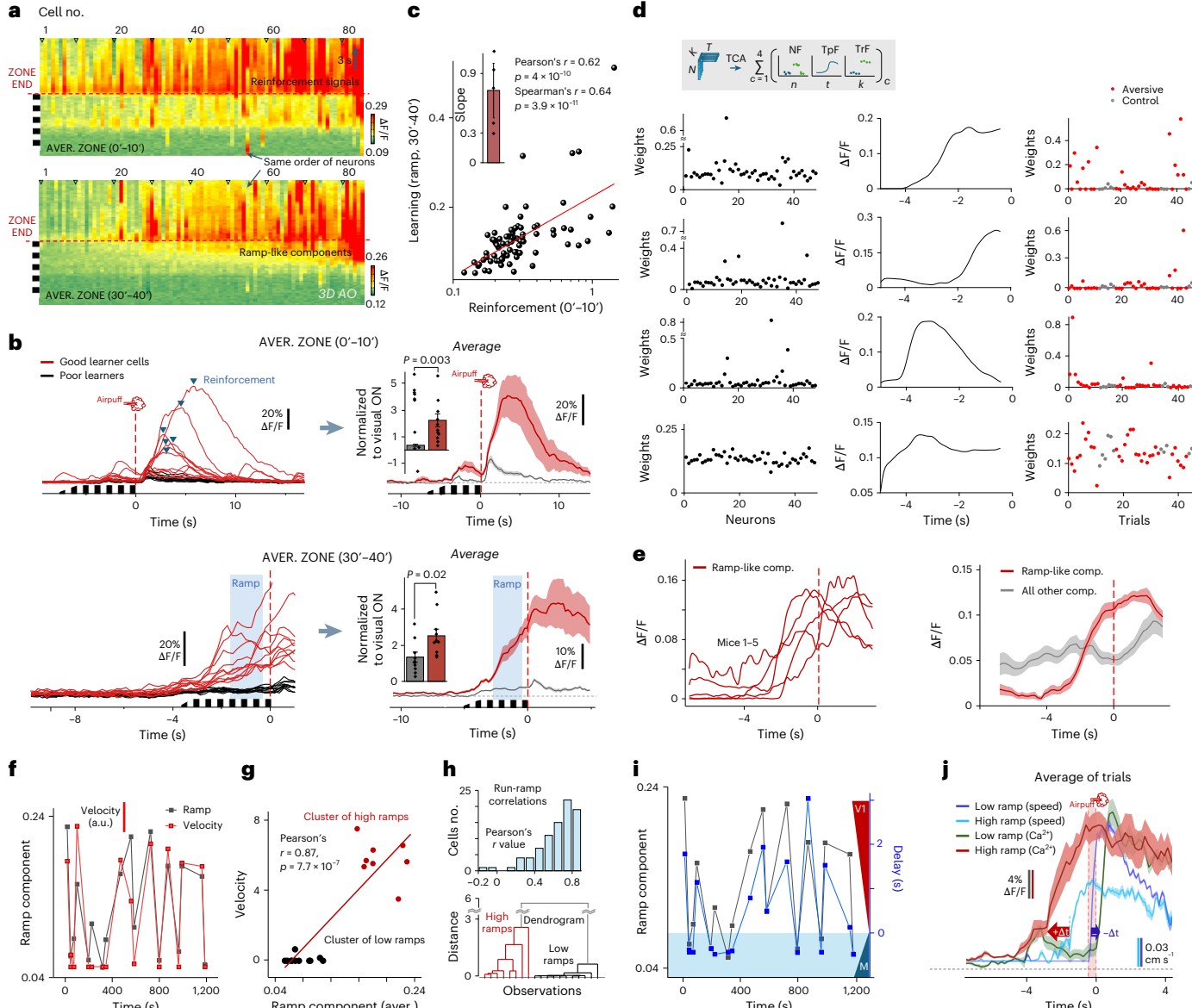

**Fig. 6 | Learning is driven by reinforcement. a**, Responses of individual neurons sorted by reinforcement-associated amplitudes at early (0–10 min) and late (30–40 min) learning phases in an example mouse. **b**, Correlation of the learning-associated ramp-like components with reinforcement signals recorded 30 min earlier. Average responses of the ten best and ten worst learner cells at the early phase of learning (top left). Average traces (mean ± s.e.m., $n$ = 10 good and 10 poor learner cells) (top right). Inset, average peak amplitudes normalized to the visual ON responses. The same but for the late phase (30–40 min) (bottom). Neurons with higher early reinforcement signals (red) had stronger ramp-like components. $P$ values indicate two-sample, two-tailed $t$-tests. **c**, Linear correlation between the late ramp-like signals (30–40 min) and earlier reinforcement responses (0–10 min, second component in Extended Data Fig. 9o) at the level of individual neurons (Person's, two-tailed, $r$ = 0.62, $P$ = 4 × 10⁻¹⁰). Inset, slope values (mean ± s.e.m., $n$ = 5 mice). **d**, Rank of four TCA neural, temporal and trial factors. First latent prefers the aversive zone and resembles ramp-like neuronal responses (two-sample, two-tailed $t$-test, $P$ = 0.033,

Mann–Whitney, $P$ = 0.015). Second latent correlates with motion and resembles the running speed in Fig. 4c. Third and fourth latents relate to visual ON responses and average visual stimuli, respectively with low motion correlation (corr.coef$_{3rd\&4th}$ = −0.08 and 0.15) and no zone preference ($P_{3rd}$ = 0.204 and $P_{4th}$ = 0.703). **e**, Left, the first latent with ramp-like kinetics was found in all mice ($n$ = 5). Right, corresponding average (mean ± s.e.m.), with other latents in gray. **f–j**, Causal link between V1 computation and motor activity fluctuates trial-to-trial. **f**, Average velocity and ramp-like component at aversive zone ends. **g**, Velocity versus average ramp components. Pearson's (two-tailed) $r$ = 0.87 and 0.98 for 20–40 and 30–40 min, respectively. $K$-means clustering identified low- and high-ramp groups. **h**, Top, distribution of Pearson's $r$ in individual cells. Bottom, dendrogram from cluster analysis in **g**. **i**, The ramp components from **f** with delays between the ramp-like component and running. **j**, Average population activity (mean ± s.e.m.) in the aversive zone (red) with running triggered by the V1 activity (light blue) during high-ramp trials. In contrast, motor activity (dark blue) triggers V1 (green) during low-ramp trials.

neurons with the highest and lowest ramp-like learning-signal amplitude at the late phase of learning (30–40 min) can also be defined as 'good' and 'poor' learners, respectively. We found that poor learners on average had lower (corr$_{avg}$ = 0.049) and good learners had higher (corr$_{avg}$ = 1.00) normalized cross-correlation coefficient than the rest

of the population (corr$_{avg}$ = 0.221) (Fig. 5j). In summary, neurons above the amplitude threshold share hub cell characteristics; they are (1) spatially and (2) temporally clustered; (3) have earlier and (4) larger response amplitudes; (5) more functional connections than the rest of the population; and (6) mimic the population's activity.

**TCA confirmed the learning-associated ramp-like component**

To dissect further the underlying dynamics of the ramp-like learning-associated component (Figs. 3f,h, 4c and 5h) at the level of individual neurons and trials (Figs. 4b and 5e), we used non-negative tensor component analysis (TCA)[38] with a framework of alternating rank-k non-negativity constrained least squares based on the block principal pivoting method. We identified a ramp-like latent temporal component that was represented approximately five times more strongly by the assembly coding the aversive zone compared to the control zone (first component; Fig. 6d,e). Note that the first component closely resembles the average ramp-like neuronal responses in the aversive zone (Figs. 3f and 4c). The nonzero trial components of second latent had a higher correlation with the integral of motion per trial (compare to Figs. 4c and 6j and Extended Data Fig. 10a) and third component resembled the average visual ON responses in kinetics (compare to Fig. 6b and Extended Data Fig. 9l,o). Outliers were not the consequence of overfitting (Extended Data Fig. 9h–n). These data also confirmed that the learning-related ramp-like component is not purely a consequence of the increased motor activity as the two effects can be separated even with a linear trial-by-trial model (Supplementary Note 4).

**Reinforcement and learning correlates at individual neurons**

The ability of activity-dependent plasticity to imprint new memory traces into the V1 region[39] or the hippocampus[13] through optogenetic activation has already been validated. As long-term potentiation and long-term depression as the main neural substrates of learning have been shown to be based on activity-dependent synaptic plasticity, which is formed after over ~30 min[40], we correlated the amplitude of learning (the ramp-like component) to the reinforcement signals recorded 30 min earlier at the beginning of learning (Fig. 6a–c). Neurons with higher reinforcement signals (good learners) expressed significantly higher learning-associated responses ~30 min later than poor learners ($P = 0.02$, two-tailed, two-sample $t$-test; Fig. 6a,b). In addition, while poor learners had a monocomponent reinforcement response, which overlapped well with the normalized running speed and could be fitted with a single exponential, the good learner cells had an additional second component after reinforcement (Extended Data Fig. 9o). This second, but not the first component correlated with the velocity triggered by reinforcement (Extended Data Fig. 9o), indicating that the second component is proportional to reinforcement[41]. To better understand the correlation between reinforcement and learning at the level of individual cells, we compared the second component measured at the beginning of learning with the ramp-like component recorded at the end of learning and found a strong correlation at the level of individual neurons (Fig. 6c). Our data indicated that reinforcement signals can contribute to the formation of long-term changes in activity at the level of individual neurons in the V1 region. Long-term neuronal activity increase appeared within ~30 min, which is on the timescale of the subcellular mechanisms of long-term plasticity[40,42]. The activity increase and the corresponding behavioral response after learning was maintained during the subsequent experimental days (Fig. 3g,h and Extended Data Figs. 5j,n and 7), supporting the emergence of the long-lasting form of plasticity (Supplementary Note 3).

**Variability in the causality between V1 and motor activity**

Our data suggested that the activity increases in the V1 region are not simply a reflection of running (Supplementary Note 1). Accordingly, neuronal activity began to rise 2.45 s before running speed increased (Fig. 4c). To explore the temporal separation between V1 computation and running, we analyzed trial-to-trial fluctuation (Fig. 6f), identifying two trial groups (high- and low-ramp trials; Fig. 6g,h) with a reversed causal link between V1 computation and motor activity. In high-ramp trials, ramp-like activity in the V1 was high and preceded running at the level of average responses, individual trials and individual neuronal

responses (Fig. 6j, Extended Data Fig. 10 and Supplementary Fig. 1). In contrast, during low-ramp trials, running preceded V1 activity on average, and at the level of individual trials and single cells. This suggest that the causality between V1 and motor activity varies; during high ramps, V1 activity in cooperation with other brain regions may generate ramp-like activity increases and drive motor cortex and running, while during low ramps, running, induced by somatosensory inputs (airpuff), may trigger V1 activity indirectly (Extended Data Fig. 10). Thus, vision during the intermediate phase of learning can be stochastic, with cue recognition, ramp-like activity and behavioral responses fluctuating between 'failures' and 'successes'; however, further investigation is required to understand the casual link between visual and motor cortex and the contribution of other cortical areas.

## Discussion

Moculus provides an immersive experience of the visual environment for rodent experiments by offering a perspective-corrected 3D virtual world with 3D objects, stereoscopic binocular vision and a large field of view to avoid visual interference by static laboratory objects. Depth avoidance appeared readily in Moculus without preconditioning or haptic feedback. Because of its compact modular design, Moculus can be adapted to any craniotomy position and recording device (for example, multiphoton microscopes) and can open a new horizon in our understanding of behaviorally relevant neuronal computation. The current design has certain inherent limitations (such as potential contact with some whiskers, field of view restrictions and limited eye tracking), which have not affected the presented results. To demonstrate the benefits of Moculus, we focused on visual learning and have revealed unexpected network mechanisms.

The main issue with current VR systems in neuroscience is their reliance on 2D projections of a virtual corridor, assuming that mice, like humans, possess advanced visual computation to reconstruct the full 3D environment from 2D projected images. Contrary to this hypothesis, we demonstrated that mice have only a low level of understanding of the different components of 2D projected images. Moreover, adding more natural 3D components generated more than 200-fold faster task engagement and discriminative learning than classical monitor systems[5,6,43]. The best performance was achieved when multiple 3D features were simultaneously combined in a natural way. Moculus demonstrated that visual learning in rodents can occur within tens of minutes, aligning with real-life scenarios—contrary to current dogma. Combining Moculus with 3D multiphoton measurements[19,22,23] has enabled identification of the underlying mechanisms of visual learning in a single session, avoiding artifacts like sensor concentration changes or day-to-day rearrangement in network activity[14–18].

While several factors influence V1, including arousal[2,3,5], locomotion[4], visual flow and self-motion[44], local mismatch, prediction[45], global reinforcement through disinhibition[9,41,46], learning[5–10,17] and categorization of visual inputs[3,43,47], we focused solely on visual learning to highlight Moculus' contribution to understanding visual computation. Key findings include (1) high neuronal activation during visual discrimination; (2) formation of overlapping cue-associated neuronal assemblies; (3) competition between these clusters; (4) correlation between reinforcement signals and learning at the level of single cells; and (5) stochastic trial-to-trial fluctuation in V1 computation and motor activity. In the first phase of learning, both running speed and neuronal responses increased for both control- and reinforcement-associated cues, indicating that mice began processing the visual information but could not yet discriminate between different gratings. Contrary to the sparse coding model[33], nearly all neurons are transiently activated to near saturation during visual discrimination, maximizing cortical representation, computational capacity and energy consumption[34] transiently. While representation of multiple cues correlated and expanded to near saturation, with neurons participating in multiple representations simultaneously, orthogonality

also emerged. Unlike previous studies, the activity of just three to seven neurons could reliably predict behavior. After 30 min, aversive cue representation dominated, with more neurons showing stronger, earlier responses in the aversive zone, mirrored by a behavioral speed increase. In the final phase, coding assemblies became more distinct, with reinforcement-associated cues dominating, while control cue representation became sparse, reflecting a 'winner-takes-all' dynamic.

In the hippocampus[13], the local circuit architecture of pyramidal cells and interneurons began to facilitate the emergence of the new information, more specifically a new location, which was coded at the beginning by a single newly formed place cell. Similarly, we show rapid formation of local circuits in V1 during learning of multiple visual cues. Because visual learning and computation were based on the emergence of spatiotemporally organized clusters with higher functional connectivity, presynaptic partners of a given V1 neuron should share similar visual tuning as a sign of previous visual experience. Indeed, the existence of locally tuned presynaptic networks has already been demonstrated[19,48].

Moculus' improved learning speed and immersive nature could enable more complex behavioral protocols that currently take over 60 days[49]. This may open new horizons, for example, in our understanding of the network mechanisms of low-level and high-level visual computation and learning, more-difficult cognitive functions and social interactions.

## Online content

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

# Methods

## Animals

All experiments were approved by the Animal Care and Experimentation Committee of the Institute of Experimental Medicine (IEM) and the Hungarian Scientific Ethics Council for Animal Experiments of the State Secretariat for Food Chain Supervision (approval no. PE/EA/54-02/2019). Adult Thy1-Cre–FVB/AntFx mice (8–18 weeks old, both sexes, $n = 47$, RRID:IMSR JAX:006143) were used, re-derivate and bred in the Medical Gene Technology Unit of IEM. Mice were housed in small groups (2–4 mice per home cage) in a temperature-controlled environment (at 23 C°) with a 12-h reverse light cycle, 45–65% humidity and enriched with rotary disks, cardboard rolls and extra nesting materials. Mice received food and water ad libitum. For experiments with water rewards, mice were water-restricted, maintaining body weight above 85% of their initial weight.

## AAV labeling and surgical procedure

An approximately $1 \times 1$-mm in the primary visual cortex was labeled with a Cre-dependent, neuron-specific GCaMP6f calcium sensor[22]. Mice were anesthetized with a mixture of fentanyl, midazolam and medetomidine (0.05 mg, 5 mg and 0.5 mg kg$^{-1}$ body weight, respectively) and a 0.5-mm hole was drilled over the V1 region (2.0 mm lateral and 0.5 mm anterior to the lambda). A glass micro-pipette (~10-μm tip diameter) back-filled with 500 nl vector solution was used for slow injection (10 nl s$^{-1}$ for 30 nl, then 1 nl s$^{-1}$) into the cortex at 450 μm depth using a microsyringe pump (Micro4 Microsyringe Pump Controller and Nanoliter 2010 Injector, World Precision Instruments). The AAV9.Syn.Flex.GcaMP6f.WPRE.SV40 vector (RRID: Addgene_100833), produced in the Laboratory of 3D Functional Network and Dendritic Imaging of the IEM using the GCaMP6f plasmid (plasmid #100833) acquired from Addgene. The stock solution had a $5.2 \times 10^{14}$ vector genomes per ml titer (measured using qPCR with a primer set designed against the ITR (inverted terminal repeat) region of the rAAV genome). It was diluted 8–10× in Ringer's lactate to ~$6 \times 10^{13}$ vector genomes per ml for stable labeling. Two to three weeks after injection, a round craniotomy centered on the injection site was performed under similar anesthesia and covered with a double coverslip[22]. After injection and surgery, animals received a wake-up cocktail (Revertor: CP-Pharma, 2.5 mg 10 g$^{-1}$ body weight; flumazenil: Fresenius Kabi, 2.5 mg 10 g$^{-1}$ body weight; and Nexodal: G.L. Pharma, 1.2 mg 10 g$^{-1}$ body weight). Baytril (Bayer, 2.5%, 200 μl 10 g$^{-1}$ body weight) was administered post-intervention to reduce inflammation. For the next 3 days, animals were treated with Melosolute (CP-Pharma, 1 mg 10 g$^{-1}$ body weight) diluted in Ringer's lactate to reduce postoperative discomfort.

## Mouse eye optical test

Enucleated eyeballs from CD-1 (RRID: IMSR_CRL:022) albino mice were used to examine the optical properties of the eye lens by projecting visual patterns through them (Fig. 2). These mice were chosen for their transparent retinal surface due to unpigmented pigment epithelium. Images were captured from the posterior and lateral sides using a Nikon D7200 DSLR camera with a 105-mm f1.8 macro lens (AF-S VR Micro-Nikkor 105 mm f/2.8 G IF-ED, Nikon) and a 35-mm CMOS sensor, at a resolution of $1,720 \times 860$ pixels (Supplementary Video 2). Main eye aberrations were calculated by analyzing the distortion of parallel grating patterns projected through the lenses using Moculus-XL or Moculus-S screens, at the same lens-eye distance as in behavioral tests.

## Immersion task (abyss test)

No previous habituation was used for the mice engaged in the immersion protocol. Mice ran in a virtual linear maze that ended in an abyss, where they had to stop at the edge of a cliff to avoid virtual falling (as represented in Fig. 1e). A 3D blue pyramid (or other objects) was added to the VR, placed beyond the edge of the cliff to enhance 3D vision and real depth perception. Slowing down at the edge of the abyss was quantified by determining the average velocity 2 s before and 2 s after mice reached the virtual edge of the abyss. The moment when mice reached the abyss edge was defined as the moment when mice passed the spatial coordinate corresponding to the edge. Alternatively, if this point was not reached, then it was defined as the beginning of the time period when mice stopped for at least 2 s at a point from where the edge of the abyss was already visible. If mice did not reach this point within 20 s, they were teleported back to the beginning of the corridor and the trial was marked as a failed trial. The velocity ratio at the edge was calculated by the ratio of the average speeds from all trials before and after the abyss. The significance of the difference both for the average curves and for the individual mice was determined as the $P$ value of the unpaired $t$-test. The numbers of failed trails were determined for all measurements, but failures were not included in the $P$ value calculation of the $t$-test.

## Driver electronics of the head-mounted display

We used three different displays in Moculus: (1) an organic light-emitting diode (OLED) microdisplay for Moculus-S (SXGA120, eMagin; Extended Data Fig. 2a); (2) a bidirectional OLED SVGA display (EBCW1020A BiMi EvalKit, Fraunhofer; Extended Data Fig. 2b) also for Moculus-S, also used for tracking eye trajectories; and (3) a 2K full HD display (LS029B3SX02, Sharp; Extended Data Fig. 2c) with a MIPI driver (TC358870XBG, Toshiba) for Moculus-XL. The SVGA OLED display of Moculus-S (0.6-inch diagonal, EBCW1020A BiMi EvalKit, Fraunhofer) features high refresh rates (60 or 120 Hz for bidirectional and conventional display modes, respectively) and dynamic range. The backlit LCD display was replaced with OLED, in which the black pixels emit no luminance (unlike LCDs), providing crucial contrast for prey animals, as black pixels emit no luminance unlike LCDs. Two displays were selected for Moculus-S: the SXGA120 ($1,292 \times 1,036$ pixels, 12-μm pitch, 69% fill factor; Fig. 1a and Extended Data Fig. 2a) and the bidirectional Fraunhofer display (Extended Data Fig. 2b), both with integrated photodiodes to track eye movements during experiments (Extended Data Fig. 2g–i). This is crucial due to the Moculus design, where display holders cover the optical field of mice, preventing the addition of external eye-tracking cameras. A dedicated display controller and power supply board were developed for the eMagin display (Extended Data Fig. 2d–f). Primary design specifications of the controller module were compactness and flexible wires. The extension board has four wires for the power supply (Extended Data Fig. 2d–f). Two dedicated printed circuit boards (PCBs) were developed: PCB panel A with the power supply and the VGA connector, and PCB panel B with a simplified original schematic except for the VGA connector. We simplified the original PCB board from eMagin (because it had 0 Ω resistors and some components were not used and soldered) and designed a much smaller device. The power supply connectors and the VGA signal connectors (J3 and J4 on PCB panel A, Extended Data Fig. 2d; and J2 and J3 on PCB panel B, Extended Data Fig. 2e) are simple pin connectors with gold-plated surfaces for better connection. PCB panel A (Extended Data Fig. 2d) converts the VGA signal from a standard VGA connector to the simple pin connector, generates the appropriate voltages from the power source (4 V, 3.3 V or 3 V) and connects them to the pin connector. Adjustable low-drop positive voltage regulators (LD1117 series, TME) were used to generate the positive 3 V and 4 V. A potentiometer was added to both circuits for fine tuning of the voltage levels. A fixed positive voltage regulator (AMS1117-3.3, TME) was used to generate 3.3 V. These voltage regulators can provide up to 800 mA of output current, which is adequate for the display according to the datasheet. To generate the negative 3 V, the previously generated positive 3 V was used with a charge pump voltage converter (TC7660, TME) to invert the voltage level. The recommended input voltage range of the power supply was 7–12 V.

Moculus-XL is based on a driver (TC358870XBG HDMI 1.4 (3D stereo) MIPI CSI-2 TX4 Data Lanes × 1ch driver, Toshiba) combined

with a 2K full HD display (1,920 × 1,080 pixels, 60 Hz, LS029B3SX02, Sharp; Extended Data Fig. 2c), which has been used in head-mounted human stereo displays.

## Latency of the VR display

The delay of the VR pipeline or photon-to-photon latency, was measured based on the method of Matthew Warburton[52] using a high frame-rate camera (BFS-U3-17S7M-C, Teledyne FLIR, Integrated Imaging Solutions) capturing both the initial treadmill movement signal and the subsequent virtual environment response. The time difference between movement and the virtual image update was measured in frames and converted to time using the refresh rate. Moculus's photon-to-photon latency was 22 ms, comparable to existing systems: FreemoVR (60–75 ms)[53] and domes for rodents (97 ms)[54]. This latency aligns with industry standards for state-of-the-art human VR systems like Oculus Rift (20.8 ms), Vive (30.8 ms) and Valve Index (38.5 ms)[52]. Rendering for latency measurement used binocular 90 Hz OLED displays (2,160 × 1,200) powered by an NVIDIA GeForce GTX1660 GPU, Intel Core i7 13,700K and 16 GB RAM. Future optimizations will focus on reducing response times via graphical acceleration, parallel processing and improved data sampling to enhance the immersive experience.

## Firmware

The firmware on the Arduino MKR Zero controls TTL (transistor–transistor logic) outputs, monitors trigger signals and reads optical encoder data. Written in Arduino IDE, the source code is available in Supplementary Data File 3, Arduino Firmware. The Arduino reads disc displacement every 3 ms via the ADNS-3090 chip using its built-in Serial Peripheral Interface library. It communicates with the PC running the VR environment, sending data through a USB every 3 ms in a format compatible with Unity. The Arduino sends three values: a counter showing milliseconds since the last trigger, velocity from the ADNS-3090 chip (Extended Data Fig. 2j) and the trigger state (1/0). Displacement is measured in counts, with the encoder set to 3,500 counts per inch.

## Unity environment

Using game engines like Unity3D allowed for the creation of controllable, photorealistic mazes with diverse patterns, corridors, abysses, walls with different patterns (for example, grids and bars) and 3D objects, enhancing stereo vision, depth perception and immersion (Fig. 1e,f and Extended Data Fig. 3). Unity3D's ray-tracing technology provides real-time reflections, shadows, perspective and custom shaders for distortion correction. Integrating the Moculus 'avatar' enabled stereographic rendering and communication with various peripherals, including the imaging system, lick port, treadmill and more, exporting data to the experiment's database. Our software is open source (https://github.com/axolotlWorks/Moculus) for reproducibility and further development by other researchers. We selected Unity as the development environment due to its hierarchical framework, managing VR objects such as terrain features, light sources, event scripts and virtual avatars of the mouse (see 'Nomenclature of the Unity structure'). These objects form scenes within the interactive virtual environments. The Moculus virtual avatar can be integrated into arbitrary virtual environments by adding it to the scene. The avatar acts as a digital twin of the mouse, reflecting the virtual world based on its position relative to the linear or circular treadmill (Gramophone). Using stereographic rendering, Moculus's left and right displays are projected from the virtual mouse's viewpoint, aligning with the physical setup (Fig. 1e–g, Extended Data Fig. 3 and Supplementary Video 1). Each mouse eye has a virtual camera (left_eye and right_eye) that renders the environment with the same field of view and optical axis as the real mouse vision (Fig. 1f and Extended Data Fig. 3). Parameters of the rendering can be configured on the Inspector panel in Unity Editor with a dedicated input field for 'field of view', the 'baseline optical axis' of the mouse that defines the physical orientation of the screens and the $k_1$, $k_2$ and

$k_3$ distortion correction values to compensate for the barrel distortion of the optical system.

## Nomenclature of the Unity structure

**Transform position.** The Moculus avatar has a coordinate 'transform' field on the Inspector panel; this matrix describes the absolute position and the rotation of the digital twin in the scene. The user should set the initial (start) position of the character, then the velocity inputs from the treadmill update the position of the character in the virtual maze.

**State zones.** State zones are boxes (trigger colliders) that function as the trigger of an event. Users can initialize the position and the boundaries of the trigger zone and assign several functions, like start position and end position, which teleports the character back to the start position, reward zone (activate water reward), aversive zone (activate airpuff), delay, etc.

**Rigidbody.** The 'Rigidbody' component extends the Moculus avatar (gameObject) with physical properties like gravity, friction and collision with the virtual environment to achieve realistic behavior of the character.

**Ortographic spectator camera.** An external orthographic spectator camera is attached and follows the character from a bird's eye view to give visual insight to the user on a control monitor into the actions during the behavior experiment.

**Mesh.** Visual representation of the experimental mouse. The mesh renderer component displays the appearance of the virtual character, which helps the researcher to get visual feedback about the position and actual behavior of the virtual mouse character.

**Environmental design.** The scenes are composed of optimized low-polygon 3D mesh models with a material that defines the appearance of the objects and a collider that describes the physical properties and interactions.

**Illumination.** Illumination of the virtual environment is important for a realistic, lifelike surrounding, but the amount of light has special importance for rodents as well. The contrast and direction in luminance are crucial aspects for the behavior of prey animals, whether they flee or feel safe in dark places[55], and are thus decisive in most behavioral experiments. Researchers can customize the position and parameters of any number of light sources in the Inspector panel to achieve the required effect according to the experiment design. The ray-tracing rendering allows the use of a variety of light sources such as sunlight (directional light), spotlight or area lights with different strengths and directions of illumination. The light sources are under the 'environment' in the hierarchy, with configurable parameters of 'range', 'spot angle', 'color' and 'intensity'.

**Shadows.** OLED display technology can create a real dark environment where, if a scene is shady, like a tunnel or shelter, the corresponding pixels of the display do not emit light. With LCD technology, there is always a backlight and, thus, a true dark environment is impossible to produce.

## Display settings and calibration

In head-restrained experiments, headplate position varies with brain region, compensated by Moculus's six degrees of mechanical freedom, allowing lens alignment with eye axes at removed displays but only at a certain level of accuracy.

Eye position is adjusted visually or with a CMOS sensor. To correct remaining errors, the rendering method tracks display position errors to adjust software and ensure accurate virtual environment

representation. To measure the angle, position and rotation errors relative to the eyes, a calibration device is used that replaces the removable display modules and has a rod in the center as an indicator and two scale bars. In the first step, the rod is set to point toward the center of the eye and then the two angles between the rod and the case are read using the scale bars. The rotation of the display is measured relative to the suspension arm. The calculated actual rotation and orientation values of the physical displays are transferred to the software to calculate and set the exact field of view of the virtual cameras, that is, eyes, accordingly.

## Testing stereo vision

To evaluate stereo vision, we conducted three different tests with an abyss, one with a flying bird, one with 3D oriented 2D patches and three with 3D objects, where stereo vision was tested on the following components of the 3D virtual corridor:

- ceiling in the flying bird test (Supplementary Video 1 and Extended Data Fig. 3g),
- ceiling in the discrimination test with floating 3D objects and nongrating patterns (Extended Data Fig. 4a,b),
- wall in the task engagement test with 3D objects (Extended Data Fig. 6),
- wall in the discrimination test with 3D objects (Fig. 4d–g and Extended Data Fig. 4c,d),
- wall in the classical discrimination test with 3D oriented 2D frames with grating patterns (Fig. 3),
- walls versus floor in the elevated maze test (Extended Data Fig. 5a–d),
- floor and walls in the abyss test (Fig. 1).

## Distortion correction in details

To maximize the field of view in Moculus, the lens needs to warp the image onto the curved retinal surface. Due to the exceptionally large field of view, the curved surface of the retina and the relatively small size of the device, the optical system has substantial pincushion distortion that cannot be corrected with optical design alone. This projection error is amplified at the edge of the lens (Fig. 2). There are three main categories of distortion: barrel, pincushion, and mustache (Extended Data Fig. 1i)[28]. Most head-mounted displays for humans work around the issue by applying a distortion correction effect on the rendered image before displaying it. This distortion correction is made with the same method that is used in computer vision, known as the Brown–Conrady model[29,56]. According to the model, the designed optical system of the Moculus has pincushion distortion (Fig. 2f and Extended Data Fig. 1i):

$$x_d \approx x_u(1 + (K_1 \times r^2) + (K_2 \times r^4) + (K_3 \times r^6)) \qquad (1)$$

$$y_d \approx y_u(1 + (K_1 \times r^2) + (K_2 \times r^4) + (K_3 \times r^6)) \qquad (2)$$

where $x_d$, $y_d$ are the distorted point coordinates; $x_u$, $y_u$ are the undistorted point coordinates; $K_1$, $K_2$, and $K_3$ are the radial distortion coefficients; and $r$ indicates the normalized distance from the center. In the Brown–Conrady model, we calculated the radial distortion coefficients ($K_i$), which was used to reverse the distortion. We acquired the values of $K_i$ using ray tracing of a chessboard pattern in our ZEMAX optical model. Then, using the rendered and distorted calibration grid (chessboard) sample, we ran the calibration algorithm to calculate the coefficients by implementing the Levenberg–Marquardt method, which is a nonlinear least-squares fitting algorithm[57]. A negative $K_i$ value typically indicates a barrel distortion, whereas a positive value indicates a pincushion distortion (Extended Data Fig. 1i). A mix of positive and negative $K$ values indicates a mustache distortion. The Brown–Conrady model was implemented in Unity (3D game engine) using a fragment shader

(a program that runs on the graphics processor) that allows precise correction at a pixel level. In this way, we were able to quickly modify the rendered images from every frame just before displaying them on the screen in stereo mode using the shader in the high-definition rendering pipeline. In this way, the distortion became the inverse distortion function of the lens and they canceled each other out; thus the image that the animal perceives was undistorted (Fig. 2f)[28]. The distortion-corrected image was calculated in the optical model using ray tracing and was shown on the Moculus display and projected onto the retina (Fig. 2e). We also used naturalistic images to demonstrate the high imaging quality at the retina (Fig. 2g). For in vitro validation, the distortion-corrected retinal image was captured at the back of the enucleated mouse eye by a digital camera as before and was compared to the simulation (Fig. 2a–e).

## Mouse velocity recordings

The spatial movement of the mice is recorded with a single-dimension locomotion tracking device (Gramophone, Femtonics). Gramophone consists of a head-fixation system (Extended Data Fig. 1), a freely rotating disk on which the restrained animal can stand or run. A rotary encoder from a modified optical mouse (uRage reaper 3090, Hama) was used to detect the rotation of the disk. The circuit board of the optical mouse was removed so that it did not interfere with the optical encoder (Extended Data Fig. 2j). The displacement signal from the rotary encoder is integrated with an ADNS-3090 chip (Avago Technologies) that communicates with the micro-controller via a Serial Peripheral Interface. To extract the data, wires from a ribbon cable were soldered to the pins required for communication and activating the chip. The other end of the cable was connected to a 10-pole male ribbon cable connector. In this way, the electronics of the communication are separated on another board to keep the system modular and of low height. Position data were sent to the computer through USB serial communication (see 'Firmware' section). The device reports the rotation speed as a change in position every 3 ms.

## Imaging of whiskers

Whisker imaging[50] was conducted using a high frame-rate global shutter grayscale sensor (BFS-U3-17S7M-C, 200 Hz, Teledyne FLIR, Integrated Imaging Solutions), with a 16 mm f1.8 lens (LM16HC, 1′′, 5MP C-Mount Lens, Kowa) under $2 \times 220$ lumen illumination from two angles (front and below, 480 LUX at mouse level). A post-processing computer vision pipeline written in GLSL (OpenGL Shading Language) was used. Shaders are executed on a GPU to process the rendering pipeline, specifically how vertices and pixels (fragments) are processed and displayed on the screen. Our method processes pixel data to determine the final color of each pixel in the rendered image. For Moculus-S and -XL, 33% and 53% of whiskers reached the device, respectively, of these, some made continuous contact (Moculus-S, 33% and Moculus-XL, 53%), while the rest touched the device at a certain frequency (Moculus-S, $27.84 \pm 4.74$ Hz and Moculus-XL, $30.8 \pm 11.97$ Hz, mean ± s.d.; Supplementary Videos 3 and 4).

## Visual discrimination task/visual stimulation

Before the head-restrained visual discrimination task, mice were habituated to the environment for three consecutive days, starting with head-fixation for 5, 15 and 25 min on days 1, 2 and 3. Some mice had previous training in other visual discrimination tasks. During the task, mice ran in an infinite virtual corridor, where wall movement speed was controlled by their velocity on the treadmill (Figs. 3a,c and 4a). The initial corridor had a neutral background without defined edges to avoid interference with the task but still allowed motion perception. After a random period, mice were teleported to zones with neutral backgrounds interrupted by gratings of different orientations. In aversive zones, mice had 3.5 s to exit before receiving mild airpuffs (0.4 s long pulse every 0.8 s) until they left. Airpuffs stopped immediately if the

mouse exited the zone. Control zones had no aversive stimuli. There was one aversive and two control zones; the second control zone was usually introduced after a few days of training. Teleportation chances were equal (50% for two zones and 33% for three). Grating zones were followed by the neutral zone, so the mice remained in the neutral zone until the next teleportation. This happened on average every 20 s. To prevent passive behavior, mice were also teleported to the center of random zones with equal probability if they did not move for 8 s.

Additionally, we used visual conditioning with positive reinforcement. We trained mice to associate a visual feature with a water reward by dividing the virtual space into a neutral, rewarded and control zone within an infinite corridor. In the rewarded zone, two reward locations were set (based on Pakan et al. 2018)[5]: one fixed at the end and another conditional at the beginning, triggered only if mice licked (Fig. 4d–g). The second reward was only provided if the animal failed to trigger the first reward or showed no licking after reward delivery. Position, velocity and licking rates were recorded. The control zone was structured like the reward zone but without water rewards. Zones were marked with distinct 3D objects: two diagonally striped cubes for the reward zone and a vertically striped cylinder for the control (Extended Data Fig. 4c,d). Alternatively, 3D objects were placed along the medial line of the visual field appearing slightly above the animal and consisted of two cubes with a dotted pattern and a single pyramid with a stellated pattern (Extended Data Fig. 4a,b).

### Behavioral analysis
Running epochs during the visual discrimination task were analyzed using a bootstrapping method, comparing speed differences between control and aversive zones to random fluctuations (Extended Data Fig. 5k–m). This analysis was carried out separately for each mouse and day, accounting for variability. Velocity curves around teleportations were collected, and a label vector was generated for control (1) and aversive (2) trials, then shuffled 1,000 times to determine speed differences for shuffled cases. This distribution was compared to the original speed difference, using dynamic time warping in a 1.5 s interval before the first airpuff to exclude its direct effect. The bootstrapping *P* value was the proportion of shuffled differences lower than the original. The chance level was set at 0.5. For inter-day training analysis (Fig. 3g and Extended Data Fig. 5k–n), the training day was divided into four equal parts and analyzed separately.

A decoder providing orthogonal binary output for neuronal assemblies was defined using a subpopulation of neurons sorted by response differences between aversive and control zones (Extended Data Fig. 9a–e). The responses of selected neurons were integrated and averaged before airpuffs at zone ends. A threshold value of this decision function separated aversive and control zone assemblies in individual trials.

### Three-dimensional two-photon imaging
Two-photon imaging was conducted using a 3D AO microscope (Femto3D Atlas, Femtonics) with high-NA objectives (×16 CFI LWD Plan Fluorite Objective 0.80 NA, Nikon or ×20 XLUMPLFLN Objective 1.00 NA, Olympus). The high NA limited the field of view but maximized spatial resolution, ensuring the most effective removal of neuropil contamination. Neurons were labeled with the GCaMP6f $Ca^{2+}$ sensor via AAV vectors. A z-stack was recorded initially, layer 2/3 neurons were detected using MATLAB-based code (MES, Femtonics) with manual adjustment[22]. We used the 3D chessboard scanning mode of the 3D AO microscope (Fig. 3b) combined with motion correction to allow neuropil decontamination and to maximize the signal-to-noise ratio of the individual trials and cells during in vivo measurements. 3D AO microscopy (Fig. 3a,b) enabled neuronal network imaging with orders-of-magnitude higher temporal resolution and higher signal-to-noise ratio than raster scanning of the same volume (see equation S84 in Szalay et al.)[22]. On average, 97.2 ± 17.4 regions of interest (ROIs) were recorded per animal

(mean ± s.d., range 65–119) in 3D with the chessboard scanning mode of the AO microscope. In brief, each ROI's center was recorded simultaneously at a scanning speed of 14.31 ± 2.96 Hz (mean ± s.d., range 11.2–20.5 Hz) with frames generated by using 20 lines with a 36-μs AO dwell time, achieving a 40,468-fold faster scanning speed (17.14 Hz) compared to point-by-point raster scanning of the same volume (2,359 s for 512 × 512 × 300 pixels). The 3D chessboard scanning was 164.66-times faster and had 12.5-times higher signal collection efficiency than combined resonant and piezo 3D scanning with the same parameters (Supplementary Table 1). During this comparison, the piezo-actuator available was used (P-725, Physik Instrumente), which provides about 30 ms settling time and 300 Hz resonant frequency at the required 600 μm z-scanning range. While multiple-layer scanning was speed-limited by the piezo settling time, continuous sine wave driving allowed for efficient volume scanning below the resonant frequency (0.8 Hz < 300 Hz; Supplementary Table 1). In summary, 3D chessboard scanning offers orders-of-magnitude higher measurement speed and signal collection efficiency than the resonant scanning methods, which in combination with motion artifact elimination achieved the high signal-to-noise ratio of the $Ca^{2+}$ transients in our measurements.

Time synchrony between the Moculus and the 3D AO microscope was achieved using a dual-trigger transistor–transistor logic signal. The microscope sent a 50-ms trigger 1 s after measurement start and 1 s before the end. These triggers were saved with behavioral data for synchronization, allowing behavioral data to be automatically cropped and aligned with imaging data during analysis.

### Motion correction in 3D
Motion correction of 3D chessboard scanning data used local spatial information around cells within square ROIs, as previously described[22]. In brief, the original motion correction code, available at MathWorks' file exchange (https://www.mathworks.com/matlabcentral/fileexchange/18401-efficient-subpixel-image-registration-by-cross-correlation), was modified to handle the file format of our acquisition software MES (see above) and the special features of our imaging strategy (measuring the local surrounding of the cells in 3D, instead of a full 3D frame). During motion correction, every subfield (chessboard piece) was rigidly motion corrected separately but the algorithm used the global displacement information as well as the original location of cells within the scanning volume. During motion correction, consecutive measurement units were aligned to the maximum projection of the first measurement unit, thus the same ROI mask could be used on all measurement units.

### Imaging data analysis
Motion correction, ROI selection, data export, background and $\Delta F/F_0$ calculation and data filtering, as well as real-time and offline visualization were performed using MES software (Femtonics) written in $C^{++}$ and MATLAB. Motion correction used an offline algorithm (see previous section), with remaining artifacts smoothed with partial Gauss filtering or interpolation. The $\Delta F/F$ calculation[51] was recreated in MATLAB (Supplementary Data File 4). In brief, normalized $\Delta F/F$ transients are calculated as $(F(t)-F_0)/F_0$, where $F_0$ is determined by the 8% percentile of all the data points for each cell and day.

### Elimination of neuropil contamination
In the first approach, cells were manually outlined with polygons on motion-corrected background images following maximum intensity projection in time, allowing precise identification of active cells regardless of their activation timing (Fig. 3b). The same cell masks were used across all measurement units on the given measurement day. Transients were automatically exported using a MATLAB script (Supplementary Data File 5). The advantage of 3D chessboard scanning is that x, y and z coordinates can be precisely aligned with the center of somata. Therefore, unlike classical 2D laser-scanning methods, z axis

offsets can be optimized to minimize the contribution of the neuropil. Neuropil contamination was estimated from surrounding areas, with 60% of the neuropil trace subtracted from raw Ca²⁺ traces before $\Delta F/F$ calculation[58]. In the second approach, with similar results, Suite2p (https://www.suite2p.org/) was used for automatic cell detection and neuropil subtraction. Suite2p successfully identified putative cells on most chessboard pieces, occasionally detecting multiple cells per field. To adapt these results, originally designed for full-field scanning, for our data, neuropil masks were analyzed pixel-by-pixel for all detected ROIs, pixels outside the chessboard or overlapping with other ROIs were omitted, then neuropil was recalculated for a proper neuropil correction. We observed a similar high activation ratio with Suite2p (94.11%) and our MES analysis program ($t$-test: 97.6%, Wilcoxon signed-rank test: 97.1%) at the end of the aversive zone with a small contribution of neuropil signaling (Suite2p: 3%; MES with $t$-test: 3%, MES with Wilcoxon signed-rank test: 5.9%).

## Automatic ROI and neuropil detection using Suite2p

Suite2p is a robust open-source tool for automated cell detection and removal of neuropil contamination, making it valuable for functional fluorescent imaging, including during behavioral experiments. As our raw data are stored in the commonly used HDF5 format, Suite2p operates seamlessly on data aligned to a chessboard matrix format without special preprocessing; however, we encountered notable artifacts (for example, neuropil boundaries encroaching into adjacent frames), which required post hoc corrections. These issues arose because Suite2p does not natively account for the chessboard borders, leading to contaminated pixels (from motion or from neighboring chessboard fields, which may be distant in the scanning volume), appearing as noise in the neuropil or even in the ROI data. To address this, we only used the automatically detected ROI and neuropil pixel coordinates and weights from Suite2p. Each pixel was examined individually, and we recalculated the ROIs and neuropil transients using only valid pixels (Supplementary Fig. 2). This calculation involved the following two steps:

(1) During offline motion correction, when ROIs move out of the imaging frame during scanning, the 'missing' pixels (those that fall outside the imaging field due to displacement) are replaced with either zero (0) or NaN (not a number) values. In full-field imaging, these zero-value pixels appear at the image edges, representing a minor portion of the total image, and can be effectively managed by tools such as Suite2p; however, with 3D chessboard scanning, zero-value pixels also emerge in more central regions of the image along the individual chessboard field edges within the m-by-n chessboard scan matrix. To address this issue, we separately collected the pixel coordinates identified as part of the ROI by Suite2p and extracted the corresponding fluorescent transients for each pixel. As expected, some of these calcium transients reached zero at certain points. To mitigate this artifact, we excluded transients containing zero values from further analysis. Subsequently, the mean raw Ca²⁺ response for each ROI was calculated as a weighted average of the remaining pixels, using the weight factors determined by the Suite2p algorithm.

(2) The second issue arises from the chessboard boundaries, which Suite2p is unable to detect or account for. Specifically, the neuropil of the detected ROIs, as identified by Suite2p, frequently extends beyond the edges of individual frames within the 3D chessboard scan matrix, spilling into adjacent frames positioned in different locations within 3D space. To resolve this, we re-examined the single-pixel data from the raw measurements and excluded pixels that were outside the chessboard frame of the neuron in question from further analysis. The neuropil transient was then calculated as the average of the remaining pixels.

With the ROI and neuropil transients determined as described above, the same method could be used for neuropil correction and $\Delta F/F$ calculation as for the manually selected ROIs. Specifically, 60% of the neuropil signal was subtracted from the raw fluorescent signal, and a silent region of the transient was used as the baseline for $\Delta F/F$ computation.

The pixel-level post-processing algorithm is included in the Supplementary Data (Supplementary Data File 7).

Finally, we recalculated activation ratios using the new algorithm. After learning, the neuronal activation ratio was 94.11% (one-tailed $t$-tests, after 20–40 min of training). Without subtracting neuropil contamination, the activation ratio increased by 3% reaching 97.14% (one-tailed $t$-tests). Additionally, recalculating activation ratios with the Wilcoxon signed-rank test yielded 97.1% without neuropil subtraction and 91.2% with neuropil subtraction. These values closely align with our original MES software-based pipeline results (97.6%).

The small relative differences in activation ratios calculated with and without subtracting the neuropil contamination ($t$-test: 97.14%/94.11% = 1.034 → 3.4%, for a Wilcoxon signed-rank test: 97.1%/91.2% = 1.064 → 6.4%) can be attributed to the precise $z$ axis alignment enabled by AO microscopy, which allows for accurate focusing on the center of somata, minimizing neuropil contamination during somatic measurements. In contrast, most neuroscience laboratories use imaging across one or multiple focal planes, accepting neurons with a wider range of $z$ axis offsets (for example, the $z$ axis offset threshold was 18.5 µm in Oldenburg et al.)[59]; however, the PSF of two-photon excitation extends about 4–5 times more along the $z$ axis, leading to greater neuropil contamination in these classical measurements with broader $z$ axis offsets.

## Tensor component analysis

Normalized Ca²⁺ transients were extracted for each neuron and cropped to 5 s before the zone end. After Gaussian filtering, the data were organized into a 3D matrix (tensor) with neurons in the first dimension, temporal transient in the second and trials in the third. TCA was performed separately for each mouse using alternating rank-$k$ non-negativity-constrained least squares based on a block principal pivoting method. We determined ranks in a way that components do not overfit. Note that the parameters of the TCA were optimized to find the most dominant components, rather than generating a full reconstruction of the tensor with very low error. Trial components were correlated with mouse velocity using Pearson correlation.

## Statistics and reproducibility

All values were reported as mean ± s.e.m., unless otherwise stated. Box-and-whisker plots show the median, 25th and 75th percentiles, range and outliers. Statistical significance was tested using one-way or two-way Student's paired or unpaired $t$-test and Pearson's correlation for normally distributed data and Mann–Whitney, Wilcoxon signed-ranks, two-way ANOVA with interaction analysis, Kolmogorov–Smirnov tests and Spearman's rank correlation for other distributions. Significance was set at $P < 0.05$. Significance is indicated by *$P < 0.05$, **$P < 0.01$ and ***$P < 0.001$. To confirm normal distribution of velocity data, Gaussian fitting was implemented following a frequency count of the velocity data (failures were excluded):

$$y = y_0 + \frac{A}{w \times \left(\frac{\pi}{4 \times \ln(2)}\right)^{-0.5}} \cdot e^{-4 \times \ln(2) \times (x - x_c)^2 / w^2}$$

where $x_c$, $A$ and $w$ parameters were fitted with a probability of $P = 4.77 \times 10^{-11}$, $P = 3.65 \times 10^{-5}$, $P = 4.77 \times 10^{-11}$, respectively. In addition, the Gaussian fit resulted in an ANOVA regression $P$ value of $P = 2.36 \times 10^{-6}$ and $R^2$ value was 0.92887. These data confirmed that the boundary

conditions for the Student's *t*-test were fulfilled. Two-photon images used as representation were replicated successfully for different animals, with identical results. Statistical calculations were performed in Excel (Microsoft), Origin Pro (OriginLab) and MES (Femtonics). Detailed statistics are in Source Data.

**3D models used on the Figures as representation**
The following 3D models are open source or license fees are cleared by the co-authors:

- 3D model of laboratory mouse: https://andrewnoske.com/wiki/3D_model:_laboratory_mouse (Fig. 3a and Extended Data Figs. 1 and 3a).
- 3D model of an eagle: https://free3d.com/3d-model/falcon-3342.html (Extended Data Fig. 3a).
- Mouse Unity model: https://assetstore.unity.com/packages/3d/characters/animals/mammals/mouse-animated-190614 (Fig. 1a,e,h,I, Extended Data Fig. 3f,g and Supplementary Video 1).
- Eagle Unity model: https://assetstore.unity.com/packages/3d/characters/animals/birds/bald-eagle-lite-56908 (Extended Data Fig. 3g and Supplementary Video 1).
- All other representative models used on the main or supplementary figures were generated by the co-authors using Blender or Unity, alternatively rendered from the production documentation in SolidWorks.

**Reporting summary**
Further information on research design is available in the Nature Portfolio Reporting Summary linked to this article.

## Data availability
All the data presented in the figures can be found in individual Excel files attached to the article, including Extended Data. Any additional information requested related to the presented findings is available from the corresponding author (rozsabal@koki.hu) upon request. Source data are provided with this paper.

## Code availability
Our Moculus software is open source (https://github.com/axolotl-Works/Moculus) for reproducibility and further development by other researchers. MATLAB scripts used during the analysis are attached among the Supplementary Data Files. These scripts were only tested with MATLAB 2021b. Unity 5 was used for generating and running the 3D virtual environments.

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

## Acknowledgements
R.B. is supported by ERC-682426 (VISONby3DSTIM), GINOP-2016, NKP-2017, VKE-2018, GINOP-2021-00061, PM/20453-15/2020, KFI-2018-097, AMPLITUDE, GYORSÍTÓSÁV-2021-04, GINOP_PLUSZ-00143, GYORSÍTÓSÁV-2022-064, NL-2022-012, KK-2022-05, ED-2021-00190 and ED-2022-00208. S.G. is grateful for the PD scholarship (NKFIH/143650). We thank D. Hillier, G. Katona and T. Tompa for helpful comments on our manuscript. We thank P. Nagyidai, B. Szőke and S. Kovács for their dedicated work on the distortion correction shader implementation, G. Horváth for the treadmill and inputs integration and B. Lontai for the virtual environments.

## Author contributions
The project was initiated by B. Rózsa and G.D. Experiments were designed by B. Rózsa, G.D. and G.S. and carried out by E.B., G.S., L.J., I.T., H.S., B.T., Z.S. and B.C. Moculus electronics and software was developed by E.B., G.D. and I.T. Analysis was carried out by G.S., L.J., H.S., T.L., K.Ó. and B. Rózsa. MES software was written by K.Ó. AAV vector technology was provided by B. Roska and Á.S. The optical models and measurements were made by P.M. The project was supervised by B. Rózsa. The manuscript was written by B. Rózsa with comments from all authors.

## Competing interests
G.D., B.R. and G.S. are inventors on patent PCT/HU2020/050029. B.R. is a founder of Femtonics and a member of its scientific advisory board. The remaining authors declare no competing interests.

## Additional information
**Extended data** is available for this paper at https://doi.org/10.1038/s41592-024-02554-6.

**Correspondence and requests for materials** should be addressed to Balázs Rózsa.

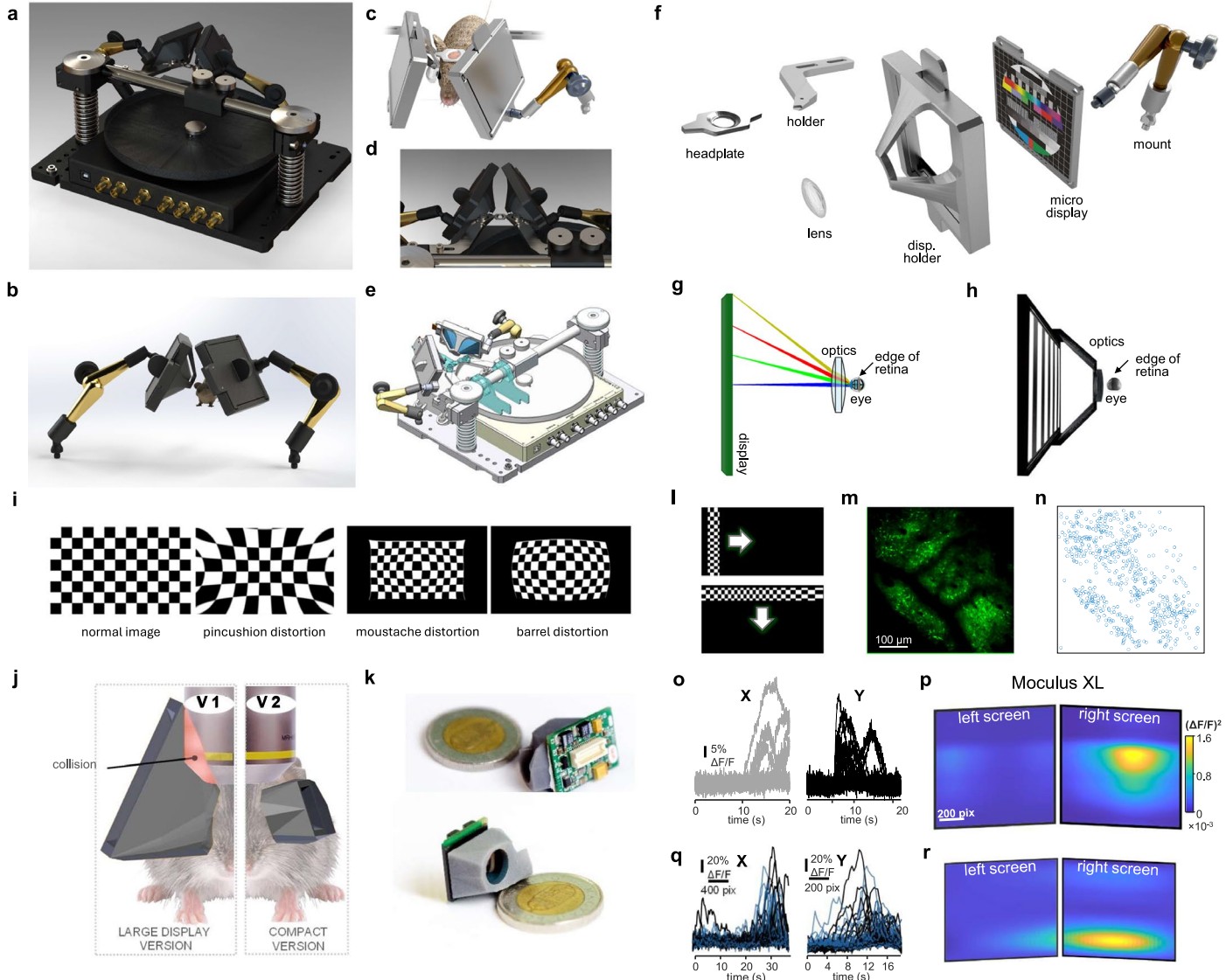

**Extended Data Fig. 1 | Design of the large-display version of Moculus (Moculus-XL) with a rotating treadmill (Gramophone) and validation with retinotopic mapping. a**, Photo of Moculus-XL with the Gramophone. **b**, 3D rendering of Moculus-XL with a model mouse. **c**, 3D rendering of Moculus-XL with a head-fixed mouse model, showing a craniotomy window and headplate. **d**, 3D rendering highlighting the headplate and headplate-holder components with the Gramophone and Moculus-XL. **e**, SolidWorks model of Moculus-XL from a different angle, with two 3D-printed alignment plates (turquoise). **f**, Exploded view of Moculus-XL showing components like the mount, microdisplay, display holder, ½' lens, and headplate. The molded silicone shading material is omitted for clarity. **g**, Optical path of Moculus-XL, including a plano-convex lens (NBK-7, diameter of 12.7 mm and focal length of 25 mm) and a mouse eye model. **h**, Lateral view of Moculus-XL with an enucleated albino mouse eye. **i**, Different types of distortion corrections implemented into the Moculus software. **j**, Rendered images to compare the large (V1, Moculus-XL) and compact (V2, Moculus-S)

versions. **k**, Moculus-S compared in size to a coin (100 Hungarian Forint) that is similar in diameter (23.8 mm) to a quarter dollar (24.3 mm). **l-r**, Retinotopic mapping. **l**, A bar with a chessboard pattern moved vertically or horizontally at 0.5 Hz. **m**, Neuronal responses were recorded using multi-layer imaging (3–5 layers) in V1, with a field-of-view of 460–580 µm. The image shows an example layer. **n**, Example of automatically detected neurons from the single layer shown in panel **m**. **o**, Neuronal responses to bar movement along the x or y axis. Each curve shows the average response of a cell over 45 repetitions. Only cells with a signal-to-noise ratio over four are included. **p**, Average x and y response traces from panel **o** are projected onto displays, showing the response amplitudes where the bars passed. **q**, Measurements were repeated with bars moving in both directions to avoid response effects. Blue curves are for left-to-right and top-to-bottom movements; black curves are for right-to-left and bottom-to-top movements. **r**, Same projection as panel **p** but for measurements shown in panel **q**.

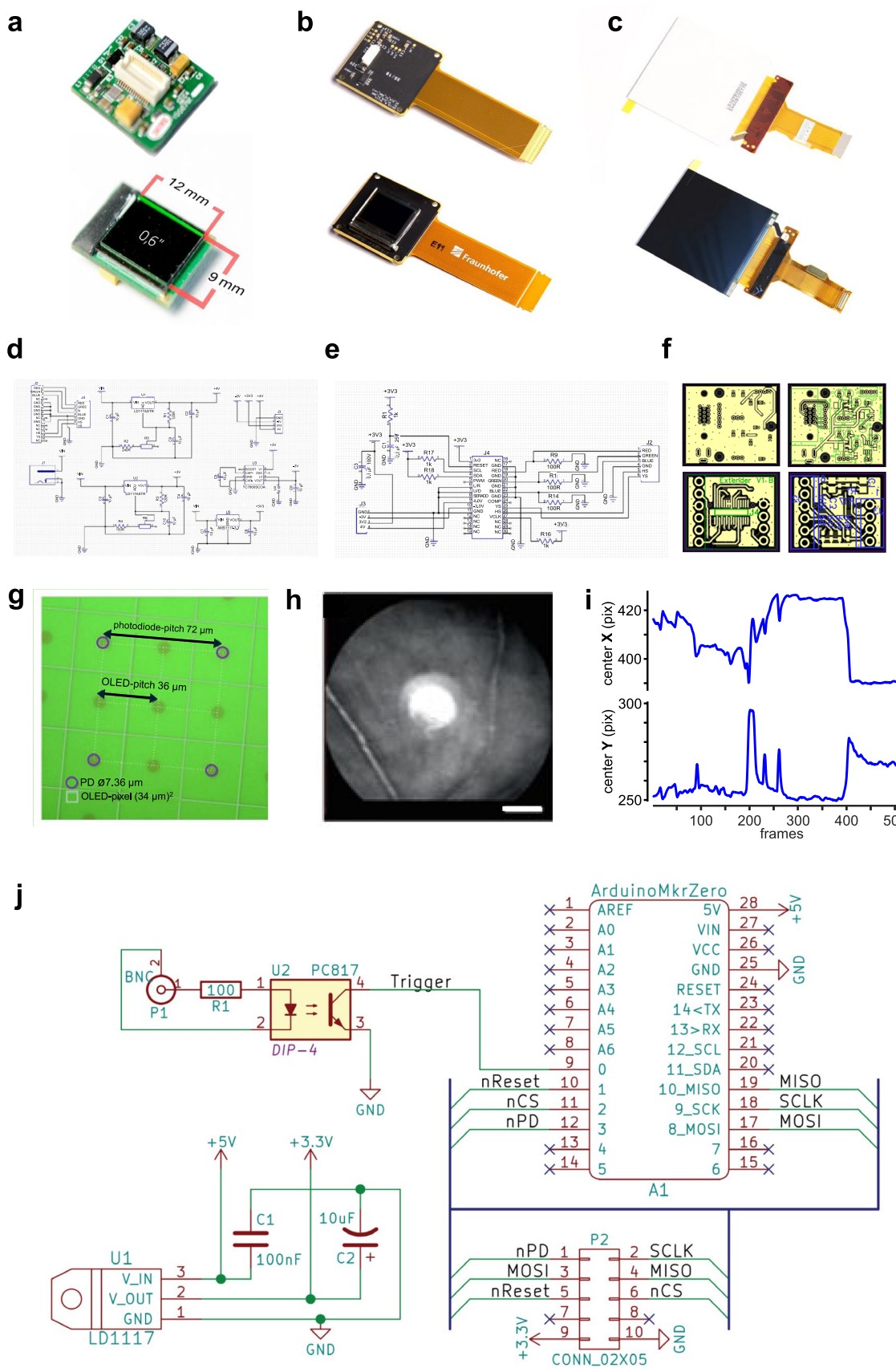

**Extended Data Fig. 2 | See next page for caption.**

**Extended Data Fig. 2 | Electronic design of Moculus allows eye tracking and communication with Gramophone. a-c**, The three versions of the displays used in Moculus. **a**, The OLED microdisplay used for Moculus-S **b**, The bidirectional OLED SVGA display (EBCW1020A BiMi EvalKit, Fraunhofer, from Vogel et al.)[26] that can also be used to track the trajectory of the mouse's eye. **c**, The Toshiba MIPI driver with 2 K full HD display used for Moculus-XL. **d**, Left, wiring diagram of the custom-developed printed circuit board (PCB panel A) with a power supply unit and the VGA driver components of Moculus-S. **e**, Simplified version of the wiring diagram of the display driver board (PCB panel B) integrated into the eMagin microdisplay used for Moculus-S, for which the custom driver panel (shown in panel **d**) was developed. **f**, Both sides of the board layouts of the custom-developed board (top: PCB panel A) and the driver unit (bottom: PCB panel B). The green frame at the left shows the orientation of the display connector. **g**, Arrangement of the OLEDs and photodiodes on the bidirectional microdisplay of Moculus-S. **h**, Retinal and optic nerve image of a mouse captured with the bidirectional display and optical system of Moculus-S. **i**, The bidirectional display enables simultaneous image projection and eye movement recording in the mouse. The relative position of the eyes in the x and y directions over time is shown. Displacement is calculated from the center of the optic nerve in retinal images (as in panel **h**). **j**, A small controller circuit with a ribbon cable connector was created to communicate with the Gramophone's (Extended Data Fig. 1) rotary optical encoder. The circuit uses an Arduino MKR Zero to read the sensor data and manage power. Connections are made via a printed circuit board, with a BNC connector for trigger input.

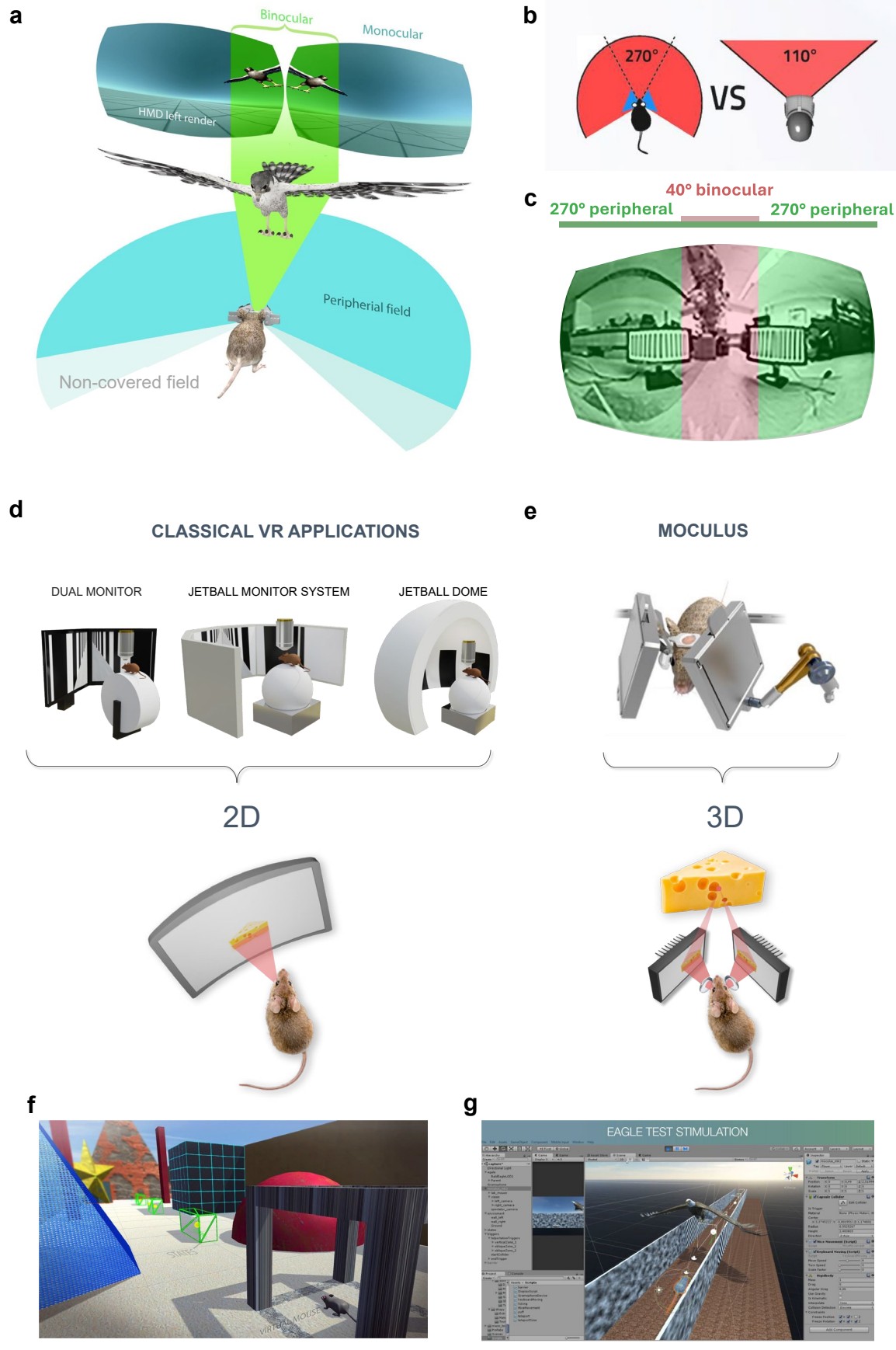

**Extended Data Fig. 3 | See next page for caption.**

**Extended Data Fig. 3 | Stereo vision in Moculus with a large field of-view.**
**a**, Schematic of stereo vision in Moculus with an approaching bird used to test
the vision of the mice before experiments. Note that in the overlapping binocular
zone (green), which is part of the VR observed by both eyes simultaneously,
two different images were rendered and projected into the left and right eyes
according to the two different view angles of the two eyes. The monocular zone
and the non-covered field are shown in blue and gray, respectively (see also Fig.
1e, f). **b**, Mice have larger fields of view than humans. **c**, Image rendered from
the viewpoint of a mouse and then projected into 2D. **d-e**, Difference between
classical VR applications and Moculus. **d**, All classical VR applications with two or
more monitors or a dome generate a 2D projected image from the 3D world of the
VR independently of the geometry of the surface: plane or curved. **e**, In contrast,
using Moculus, the VR is projected into two view angles according to the two eye
positions and orientations and then the two projection images are shown on the
two displays to generate real stereo vision. Therefore, mice are presented
with visual information that closely resembles their natural vision.
**f-g**, Virtual environment rendered in the Unity game engine. **f**, The Unity game
engine enabled us to implement complex, colored 3D objects, corridors, gaps,
and even moving objects in the Moculus VR. **g**, Graphical user interface (GUI) of
the Unity game engine with a moving eagle used as a vision test for mice at the
beginning of the experiments (Supplementary Movie 1).

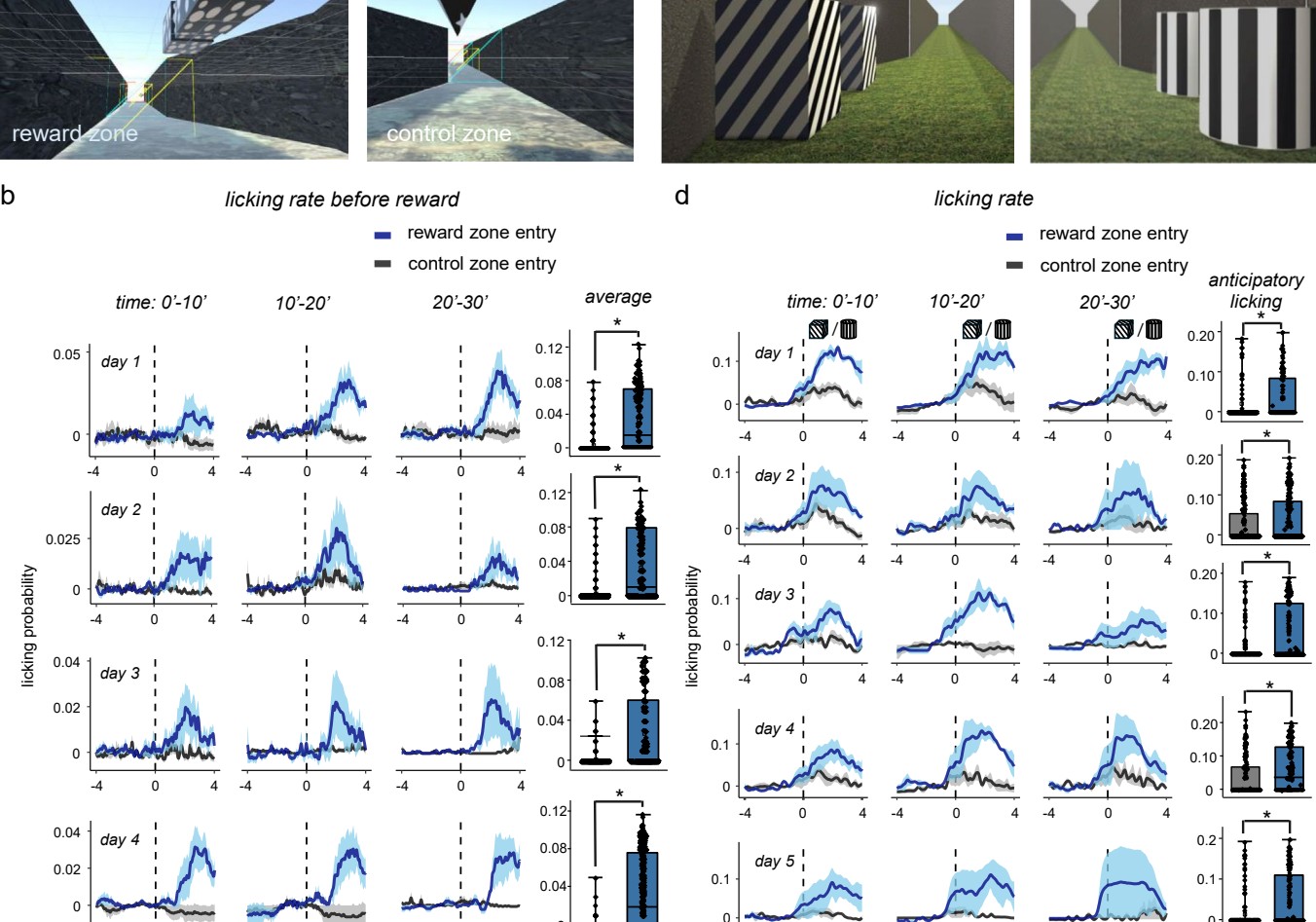

**Extended Data Fig. 4 | Rapid discriminative learning with true 3D objects in the binocular zone with non-grating and grating patterns. a**, Examples of a reward and a control zone from the viewpoint of mice. The reward zone was cued with a cube with a dotted pattern, while the control zone contained a cone decorated with stars. Both objects floated close to the middle bisector plane in the upper part of the binocular zone. Regardless of whether the 3D objects differed from those in Fig. 4d, the same reward protocol with two reward locations was applied: the second reward was provided at the end of the reward zone. The first reward was only given if mice started licking between the beginning of the zone and the location of the second reward. The second reward was only provided if the animal failed to trigger the first reward or showed no licking after reward delivery. **b**, Average licking rate before reward (mean ± SEM) timed to the entry into the control and rewarded zones (0 s) across multiple training days, broken into 10-minute intervals, comparing control (unrewarded, black) and reward (blue) zones. Only licking data before reward are included. Bar graphs showing

average licking rates (mean ± SEM) in the control zone and in the reward zone before reward. Data from $n$ = 6 mice, minimum 281 and 338 trials/day in reward and control zones, respectively (mean ± SD = 348.75 ± 49.02 and 395.75 ± 61.9, respectively). *: significant difference between the two zones, two-tailed t-test ($p_{day1}$ = 7.39×10$^{-37}$, $p_{day2}$ = 8.22×10$^{-30}$, $p_{day3}$ = 1.30×10$^{-33}$, $p_{day4}$ = 7.94×10$^{-54}$). **c-d**, The same protocol as in Fig. 4d, e. **d**, Left, licking rate (mean ± SEM) timed to the entry into the control and rewarded zones across multiple training days, broken into 10-minute intervals, comparing control (unrewarded, black) and reward (blue) zones. Data are from the measurements shown in Fig. 4d–g. Right, bar graphs showing average anticipatory licking rates (mean ± SEM) before entry into the control (black) and reward (blue) zones. Data from $n$ = 5 mice, minimum 146 and 150 trials/day in control and reward zones, respectively (mean ± SD = 161.8 ± 19.79 and 168.8 ± 15.69, respectively). *: significant difference between the two zones (one-tailed t-test, $p_{day1}$ = 0.032, $p_{day2}$ = 0.0067, $p_{day3}$ = 1.41×10$^{-8}$, $p_{day4}$ = 0.00038, $p_{day5}$ = 1.80×10$^{-8}$).

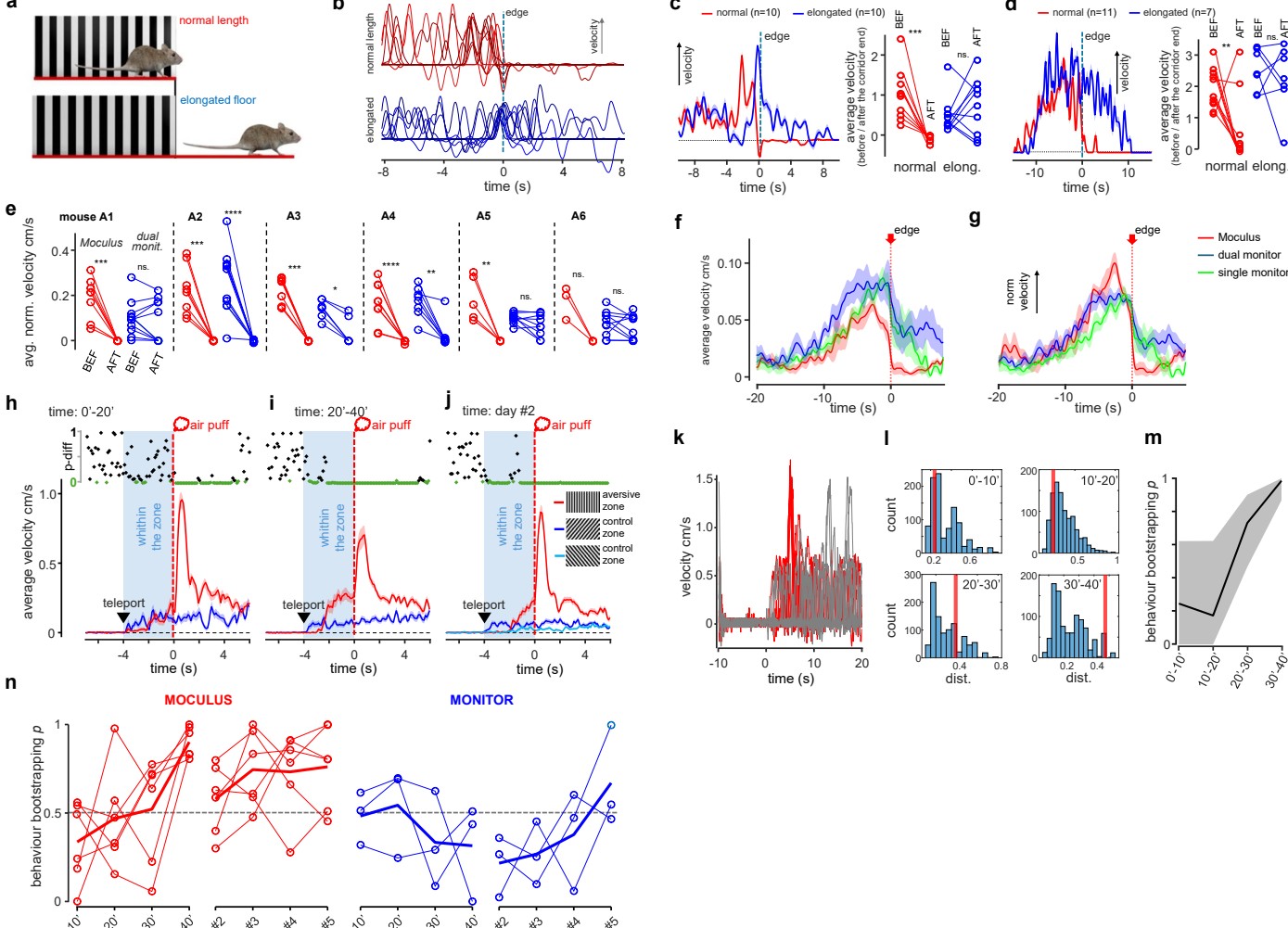

**Extended Data Fig. 5 | Abyss experiment with variable corridor length.**
**a**, Schematic of the two methods: i) floor and wall length equal, as in Fig. 1; ii) elongated floor. **b**, Example mouse. Running curves with normal floor (red) and elongated floor (blue). **c**, Left: average running speed (mean ± SEM) with normal (red) and elongated (blue, n = 10-10 trials) floor for the mouse in **b**. Right: Average speeds before (BEF) and after (AFT) the abyss for normal and elongated floors. Speed difference was significant with normal but not with elongated floor (Wilcoxon signed-rank: w = 0.0186 for normal n = 11 mice; w = 0.813 for elongated, n = 7 mice). **d**, the same as **c**, but for multiple mice (w = 0.002 vs. w = 0.232, Wilcoxon). **e-g**, Moculus (red) vs. monitor-based VR (blue, green) systems. **e**, Average running speeds before (BEF) and after (AFT) the abyss (n = 6 mice, A1-A6, two-tailed Wilcoxon: *p < 0.05, **p < 0.01, ***p < 0.001, ****p < 0.0001; exact p-values for Moculus-XL: $p_{A1}$ = 1.55×10$^{-4}$, $p_{A2}$ = 1.55×10$^{-4}$; $p_{A3}$ = 5.83×10$^{-4}$; $p_{A4}$ = 4.11×10$^{-5}$; $p_{A5}$ = 0.0079; $p_{A6}$ = 0.1; for dual monitor: $p_{A1}$ = 0.469, $p_{A2}$ = 8.11×10$^{-5}$; $p_{A3}$ = 0.0152; $p_{A4}$ = 0.0011; $p_{A5}$ = 0.079; $p_{A6}$ = 0.7). **f**, Average running transients (mean ± SEM) for Moculus-XL (red), dual monitor (blue), and single monitor

(green, n = 9/6/4 mice). **g**, Same as **f**, but normalized to velocity 2 s before abyss edge. **h-n**, Visual learning with stable memory over time in corridor with grating patterns. **h**, Average running speed (mean ± SEM) after randomized teleportation into control (blue) and aversive (red) zones during early learning (0–20 minutes). Transients aligned to zone ends. Inset: significant velocity differences (two-tailed t-test; green: p < 0.05, black: p > 0.05). **i**, Same as panel **h** for late learning (20–40 minutes). **j**, Same as **h** and **i**, for the second training day. **k**, Example running curves (raw data) aligned to teleport time (0 s) in aversive (red) and control (gray) zones. **l**, Distance between running curves measured by Dynamic Time Warping. Red: distances between average aversive and control curves. Blue: distances between shuffled pairs, with 10,000 shuffles per 10-minute sections. **m**, Example bootstrapping–based analysis showing fast visual learning within 40 minutes with Moculus. **n**, Summary of learning with Moculus vs. traditional monitor on day one and consecutive days. Mice performed better with Moculus (Moculus/monitor: n = 6/3 mice, Mann–Whitney, $p_{30'-40'}$ = 0.0238; $p_{0'-10'}$ = 0.381, $p_{10'-20'}$ = 0.714, $p_{20'-30'}$ = 0.381, $p_{day2}$ = 0.0476, $p_{day3}$ = 0.0238, $p_{day4}$ = 0.0952, $p_{day5}$ = 0.714).

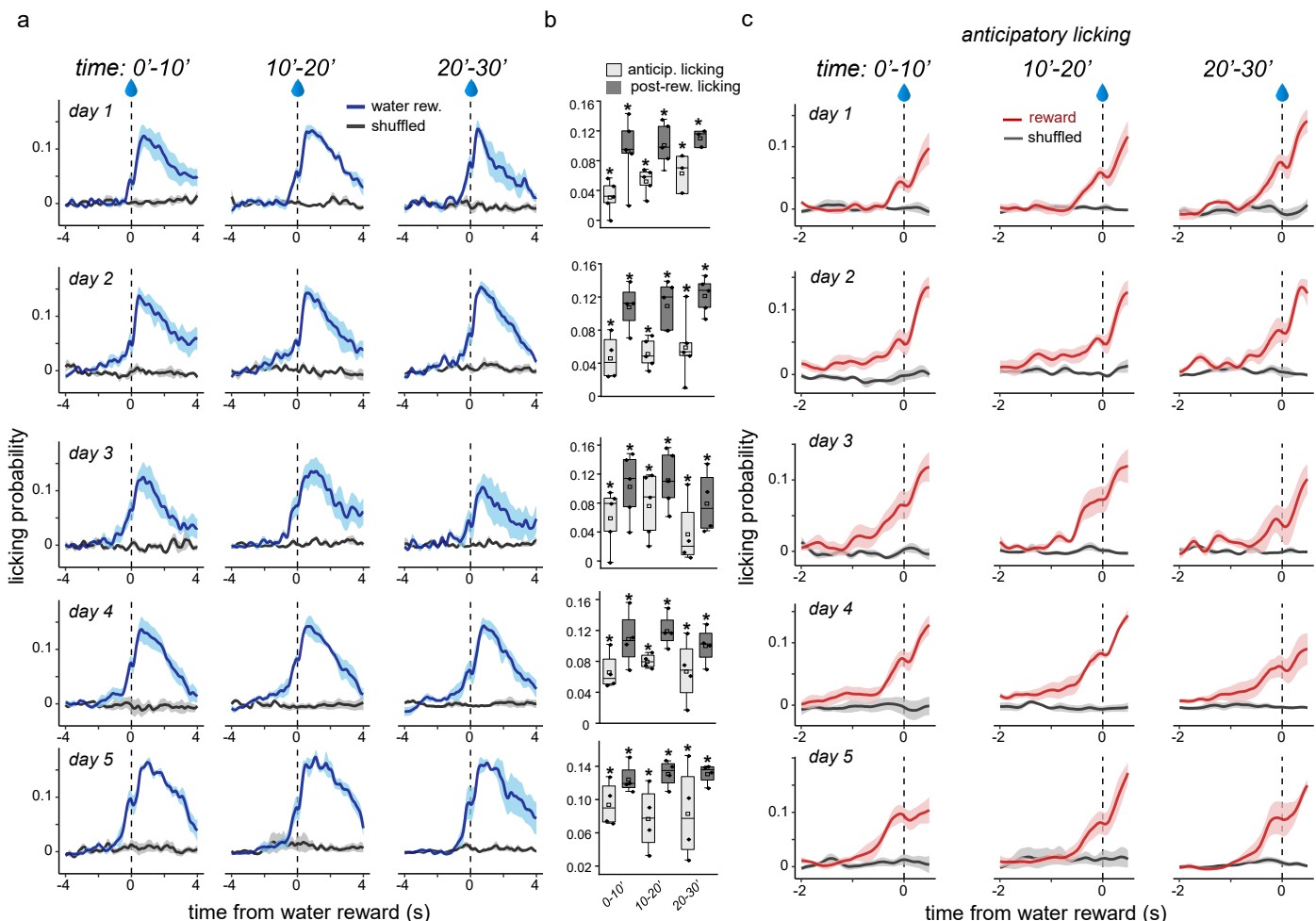

**Extended Data Fig. 6 | Fast task engagement with Moculus using 3D object as a cue for the reward zone. a**, Average licking rate (mean ± SEM) timed to a visually cued water reward (Fig. 4d,e) across multiple training days, broken to 10-minutes intervals, compared to shuffled data. **b**, Bar graphs showing averaged licking rates before the reward zone (anticipatory licking, light gray) and in the reward zone (gray). Data from *n* = 5 mice, minimum 150 trials/day (trials/day: mean=205.6, SD = 43.41). *: indicates significant difference compared to

the baseline period (two-sided paired t-test, *p* < 0.05, for exact p-values, see **Source Data Files**). **c**, The same data as in a but with a shorter timeframe. Average licking rate (mean ± SEM) timed to a visually cued water reward across multiple training days, broken into 10-minute intervals (red), compared to shuffled data (black). Data from *n* = 5 mice, minimum 150 trials/day (trials/day: mean= 205.6, SD = 43.41).

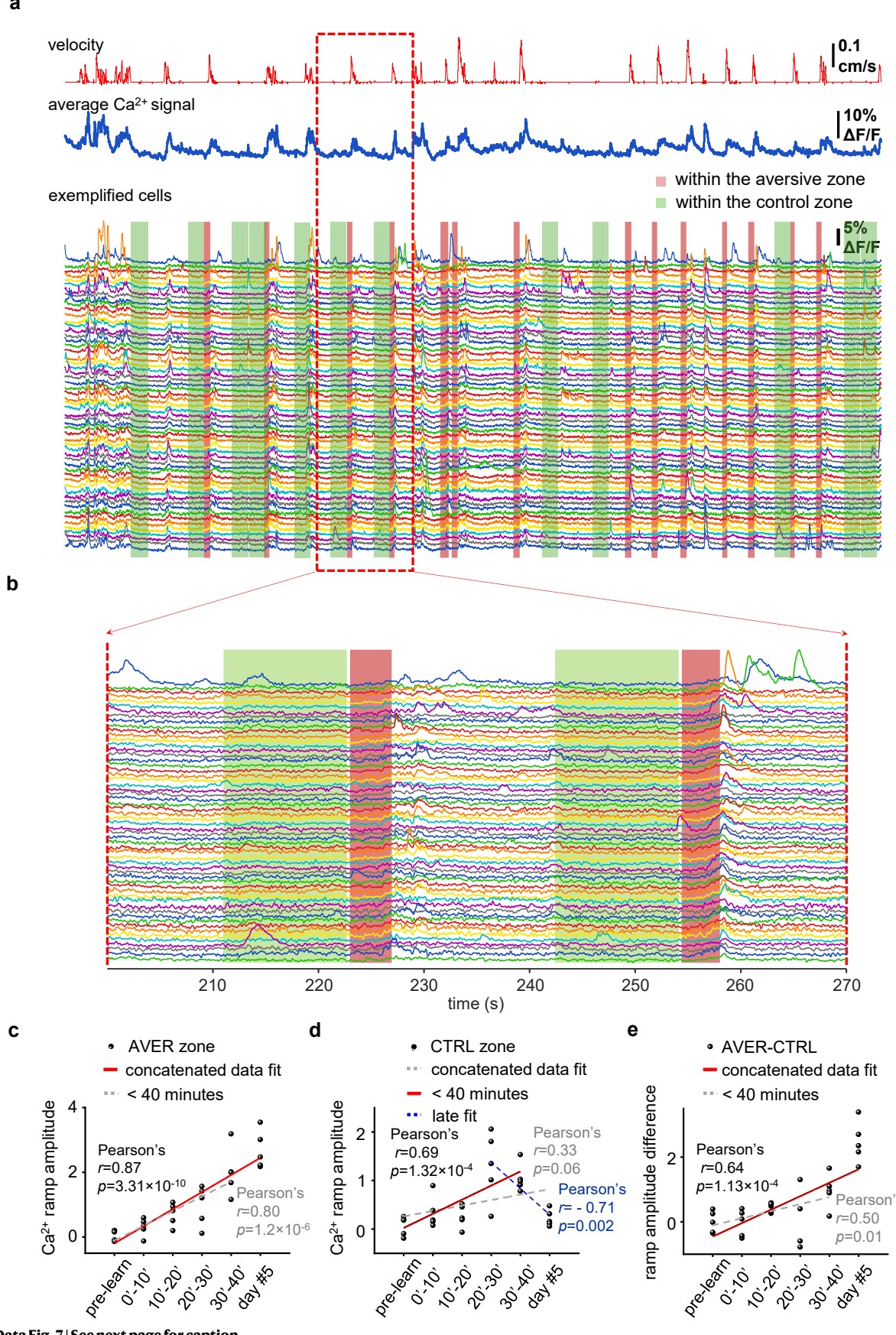

**Extended Data Fig. 7 | See next page for caption.**

**Extended Data Fig. 7 | Visual learning as a function of time. a-b**, Neuronal activity after 5 days training. **a**, Individual color-coded somatic Ca$^{2+}$ responses from an example mouse after 5 days learning using Moculus-XL with negative reinforcement (airpuff). The blue curve shows the average population response. Red and green bars indicate aversive and control zones, respectively. Corresponding running speed is shown in red. **b**, Similar to panel **a** but with a shorter timeframe. Neuronal responses are elevated at the end of the aversive zones but not in the control zones. **c-e**, Learning in the aversive and control zones as a function of time. **c**, Average (mean ± SEM) amplitude of the ramp-like components as a function of time in the aversive zone from Fig. 3h ($n = 5$ mice). The red line is Multi-Data Linear Fit to Concatenated Data (Origin Pro, OriginLab) showing high correlation between ramp-like components and time during learning (ANOVA $p = 3.31×10^{-10}$, Pearson's $r = 0.87$). The gray dashed line is the same but without day #5 (ANOVA $p = 1.2×10^{-6}$, Pearson's $r = 0.80$).

**d**, Same as panel **c**, but in the control zone. The correlation was lower in the control zone than in the aversive zone (0–40 minutes: Pearson's $r_{CTRL}=0.69$ v.s. Pearson's $r_{AVER} = 0.80$) and ANOVA test of the Multi-Data Linear Fit was significant only during the first 40 minutes (ANOVA $p = 1.32×10^{-4}$) but not when day 5 was added (ANOVA $p = 0.06$). Multi-Data Linear Fit to Concatenated Data of the last 3 timepoints (blue dashed line) showed a significant decline in the learning-associated ramp-like component in the control zone (Pearson's $r = -0.71$, ANOVA $p = 0.002$). **e**, Same as panel **c**, but with the differences in the ramp components between the aversive and control zones. The differences in the ramp components were significantly correlated with time (Pearson's $r = 0.64$, ANOVA $p = 1.13×10^{-4}$, within 40 minutes: Pearson's $r = 0.50$, ANOVA $p = 0.01$). These data indicate that mice learned the difference between the aversive and control visual cues within 40 minutes. (For **c-e**, all Pearson's tests are two-tailed).

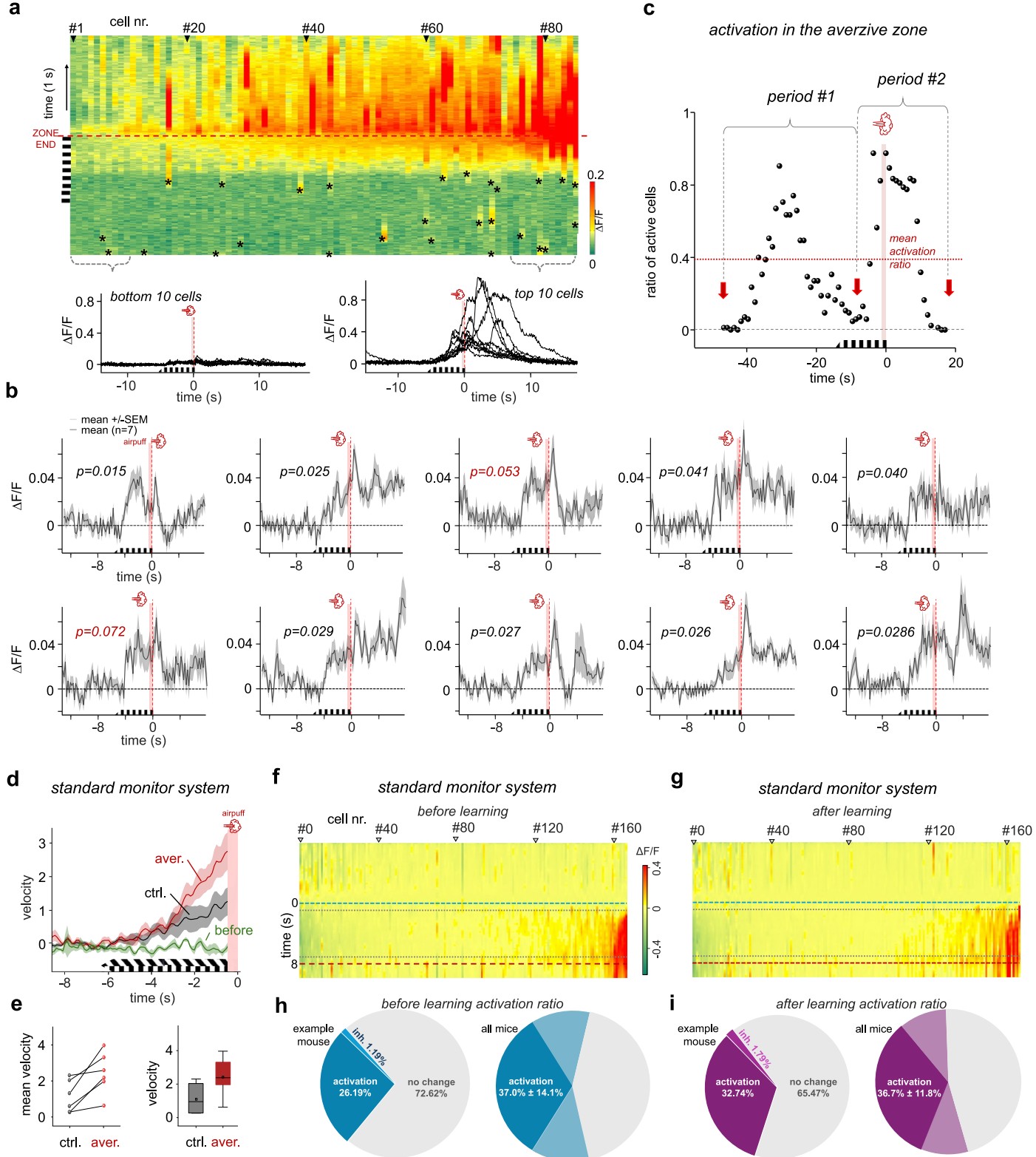

**Extended Data Fig. 8 | See next page for caption.**

**Extended Data Fig. 8 | Average neuronal responses and activation ratios at the end of the aversive zone after learning (30–40 minutes) with Moculus (a-c) and conventional monitor systems (d-i). a**, Neuronal responses at the aversive zone end from Fig. 3d, sorted by ramp components with a LUT set for small responses. Top, Low neuronal activation ratio in the baseline period allowed clear identification of spontaneous activities (*) which were excluded before baseline calculation. Bottom, responses from the top and bottom 10 cells. **b**, Responses (mean ± SEM) of the 10 cells (from **a**) with the smallest amplitudes. Significant (black) and non-significant (red) $t$-test $p$ values indicate 97.6% activation ratio (83/85 active cell). The exact p values are shown in the figure. Recalculation for the two nonsignificant cells from −3,300 ms to −400 ms yielded significant $P$ values: $P = 0.046$, $P = 0.02$, indicating 100% activation. **c**, Active cell ratio after 30–40 minutes training at 1-second intervals. Middle red arrow indicates the baseline period; other two arrows indicate returns to baseline. Activation ratio increased from 4.7% baseline to 100%, peaking at the airpuffs. Mean activation in periods #1-2 was 38.3 ± 4.0%, and baseline was 2.67 ± 0.76%. Therefore,

baseline-corrected activation ratios were 94.93% (= 97.6 − 2.67%) and 97.33% (=100 − 2.67%). The periodic motion in the VR explains the activation ratio increase at about minus 30 s. **d-i**, Similar training protocol as in Fig. 3 but with standard monitors. **d**, Running speed (mean ± SEM, $n = 6$ mice) before (green) and after 3–5 days training (red), also in the control zones (black). **e**, Mean running speeds from −1,000 ms to 0 ms before the airpuff were significantly increased in aversive vs. control zones (Wilcoxon, $w = 0.018$; Mann–Whitney, $U = 0.03$, paired $t$-test, $p = 0.008$). **f**, Neuronal responses from an example mouse during grating stimulations (0°), averaged over three days, ordered by peak amplitude. Dashed lines indicate stimulation phases (moving gratings: 1–7 s, static: 0–1 s & 7–8 s). **h**, Mean activation ratio during moving gratings for an example mouse (left) and all mice (right, $n = 6$, light colors: ±SEM). **g**, The same as **f**, but after 3–5 days training. **i**, Same as **h** but after learning. Activation ratio after learning showed a small increase in the example mouse, but was not significant overall ($n = 6$ mice, before: 37.0 ± 14.1%, after: 36.7 ± 11.8%, two-tailed Wilcoxon, $w = 0.6875$).

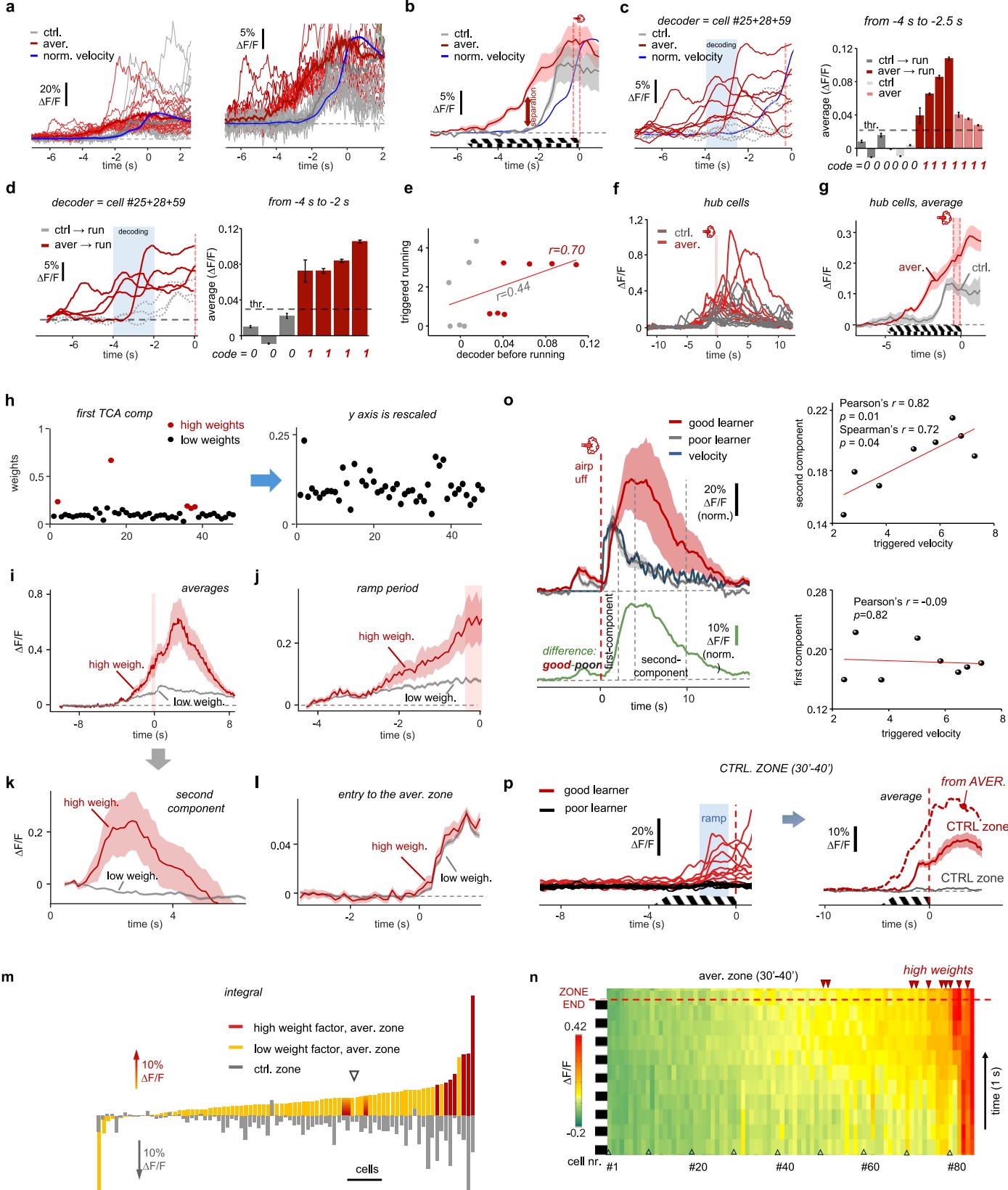

**Extended Data Fig. 9 | See next page for caption.**

**Extended Data Fig. 9 | Coding. a-g**, a three-neuron decoder provides orthogonal binary output for the assemblies coding the aversive and control zones. **a**, Average activity of the top 18 neurons from histogram in Fig. 5c in aversive (red) and control zones (gray). Note the gap in responses at about −2.5 s, preceding velocity (blue) increase. **b**, Corresponding average responses. **c**, Left, summed activity of three neurons (=#25 + #28 + #59) from the 18 cells in aversive (red) and control (gray dotted) zones. Right, activity of the three cells integrated from -4 to -2.5 s defined a decoder with orthogonal outputs for the control and aversive zones, before running speed increase (mean ± SEM). Dark colors: high-ramp trials. Light colors: low-ramp trials. **d**, Similar to **c** but during high-ramp trials only. **e**, Correlation between running and decoder performance for all trials (Pearson's $r = 0.44$) and for high-ramp trials (red dots, $r = 0.70$). **f-g**, Average response of hub cells mimics average ramping activity. **f**, Mean responses of the 10 aversive and 10 control hub cells (Fig. 5f) at zone ends. **g**, Average responses (mean ± SEM) of hub cells showing similarity to Fig. 3f. **h-n**, Weight factors

analysis. **h**, Left, weight factors of the first TCA component (Fig. 6d). Right, after rescaling. **i**, Average responses (mean ± SEM) of neurons with high and low weight factors from **h**. **j**, Same as **i** but rescaled. Curves are presented as mean values ± SEM, **k**, Responses from **i**, with a new baseline showing a second peak component linked to reinforcement. **l**, Same neuronal responses as in **i** show overlapping ON response. **m**, The same as Fig. 5c, but neurons with high (red) and low (orange) weight factors indicated. Most high-weight cells align with hub cells. **n**, Panel from Fig. 3d showing neurons with high weight factors (triangle). **o-p**, Relationship between reinforcement and learning. **o**, Left, Responses of the 10 best (red) and 10 worst (black) learners (from Fig. 6b) at early learning normalized to the visual ON response, averaged (mean ± SEM) and subtracted (green); running speed is overlaid; definition of the first and second components is indicated. Right, correlation between velocity and response components. **p**, The same as the bottom part of Fig. 6b but for the control zone. The exact Pearson's r values are shown in the figure and in the Source Data File.

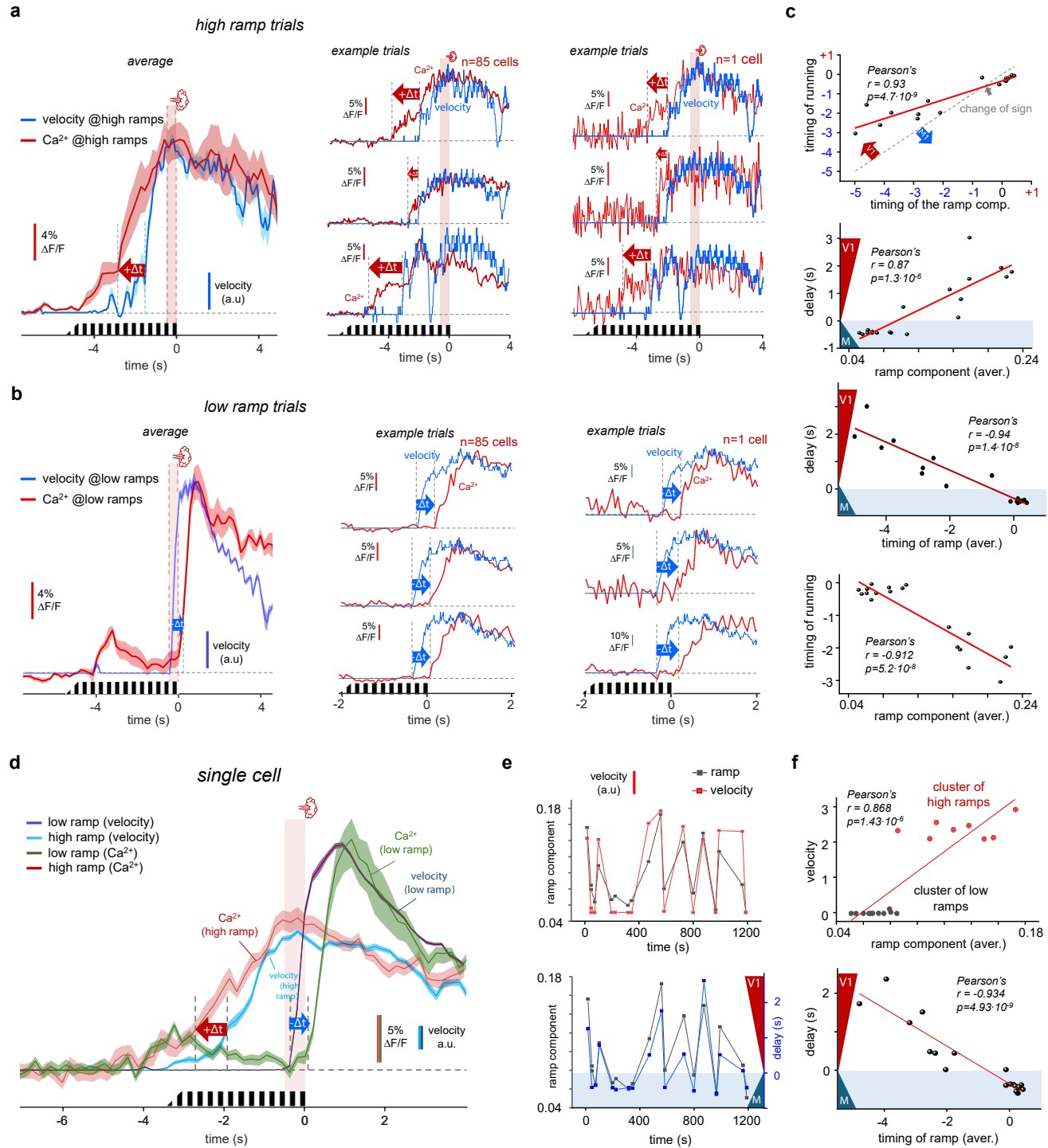

**Extended Data Fig. 10 | Stochastic fluctuation of vision: V1 activity and running dynamics during high-ramp and low-ramp trials.** (see Supplementary Note 1 for details). **a**, Left, average neuronal activity and running speed during high ramps in the aversive zone (mean ± SEM), similar to Fig. 6j but normalized to show relative time delay. Middle, average activity (*n* = 85 cells: red) and running (blue) during three representative high-ramp trials. Right, activity and running speed for one example cell. Red arrows indicate that neuronal activity precedes running during high ramps. **b**, Same as panel **a** but for low-ramp trials (mean ± SEM). Blue arrows indicate that running precedes neuronal activity. **c**, Top: Correlation between running start time and the average ramp onset times for individual trials, dashed line (x = y) and red arrow indicate trials where

V1 activity preceded running (Pearson's *r* = 0.93). Middle top: Delay between neuronal activity and running vs. ramp amplitude (Pearson's *r* = 0.87). Red and blue triangles indicate trials where V1 activity preceded or followed motor activity, respectively. Middle bottom: Delay between neuronal activity and running vs. ramp timing (Pearson's *r* = -0.94). Bottom: running onset vs. ramp amplitude. **d**, Similar to Fig. 6j, but for a single neuron. Calcium activity and velocity curves were presented as mean values ± SEM. **e**, Similar to Fig. 6f and 6i, but for neuron in panel **d**. **f**, Same as Fig. 6g and the third row of panel **c** but for neuron in panel **d**. Due to character limitations, the exact Pearson's r values are provided in the figure.

# Reporting Summary

## Statistics

For all statistical analyses, confirm that the following items are present in the figure legend, table legend, main text, or Methods section.

| n/a | Confirmed | |
|---|---|---|
| ☐ | ☒ | The exact sample size (*n*) for each experimental group/condition, given as a discrete number and unit of measurement |
| ☐ | ☒ | A statement on whether measurements were taken from distinct samples or whether the same sample was measured repeatedly |
| ☐ | ☒ | The statistical test(s) used AND whether they are one- or two-sided<br>*Only common tests should be described solely by name; describe more complex techniques in the Methods section.* |
| ☒ | ☐ | A description of all covariates tested |
| ☒ | ☐ | A description of any assumptions or corrections, such as tests of normality and adjustment for multiple comparisons |
| ☐ | ☒ | A full description of the statistical parameters including central tendency (e.g. means) or other basic estimates (e.g. regression coefficient) AND variation (e.g. standard deviation) or associated estimates of uncertainty (e.g. confidence intervals) |
| ☐ | ☒ | For null hypothesis testing, the test statistic (e.g. *F*, *t*, *r*) with confidence intervals, effect sizes, degrees of freedom and *P* value noted<br>*Give P values as exact values whenever suitable.* |
| ☒ | ☐ | For Bayesian analysis, information on the choice of priors and Markov chain Monte Carlo settings |
| ☒ | ☐ | For hierarchical and complex designs, identification of the appropriate level for tests and full reporting of outcomes |
| ☐ | ☒ | Estimates of effect sizes (e.g. Cohen's *d*, Pearson's *r*), indicating how they were calculated |

*Our web collection on statistics for biologists contains articles on many of the points above.*

## Software and code

Policy information about availability of computer code

| | |
|---|---|
| Data collection | All two-photon experiments were performed with a 3D-AO twophoton microscope (ATLAS, Femtonics Ltd.) ang Matlab-based software Mes 5 version. The spatial movement of the mice is recorded with a single-dimension locomotion tracking device (Gramophone, Femtonics Ltd). |
| Data analysis | Calcium imaging data were analysed using Matlab and Matlab-based software Mes 5 version (Femtonics), OriginPro (Originlab) and Excel (Microsoft). Zemax 13 (release 2.) software was used for the optical models in the manuscript. The source Data File was entered in an excel spreadsheet Microsoft Excel (Microsoft 365 MSO, 2407 buildversion16.0.17830.20056, 64 bit). |

For manuscripts utilizing custom algorithms or software that are central to the research but not yet described in published literature, software must be made available to editors and reviewers. We strongly encourage code deposition in a community repository (e.g. GitHub). See the Nature Portfolio guidelines for submitting code & software for further information.

# Data

Policy information about availability of data

All manuscripts must include a data availability statement. This statement should provide the following information, where applicable:
- Accession codes, unique identifiers, or web links for publicly available datasets
- A description of any restrictions on data availability
- For clinical datasets or third party data, please ensure that the statement adheres to our policy

All data provided in the figures are now included in the source data file that is uploaded along with the Extended Data files. Any additional information requested related to the presented findings (including raw data or methods) is available from the corresponding author upon request.

# Research involving human participants, their data, or biological material

Policy information about studies with human participants or human data. See also policy information about sex, gender (identity/presentation), and sexual orientation and race, ethnicity and racism.

| Reporting on sex and gender | N/A |
|---|---|
| Reporting on race, ethnicity, or other socially relevant groupings | N/A |
| Population characteristics | N/A |
| Recruitment | N/A |
| Ethics oversight | N/A |

Note that full information on the approval of the study protocol must also be provided in the manuscript.

# Field-specific reporting

Please select the one below that is the best fit for your research. If you are not sure, read the appropriate sections before making your selection.

☒ Life sciences  ☐ Behavioural & social sciences  ☐ Ecological, evolutionary & environmental sciences

For a reference copy of the document with all sections, see nature.com/documents/nr-reporting-summary-flat.pdf

# Life sciences study design

All studies must disclose on these points even when the disclosure is negative.

| Sample size | No sample size calculation was performed as there were no similar data available to apriory determine the standard error of the experimental parameters. Sample size were chosen based on similar studies using comparable approaches (J.Cichon et al., 2015; Geiller et al.,2020; A.Klioutchnikov,2022). |
|---|---|
| Data exclusions | Data were not excluded from the analysis of the imaging data. 2 from 17 animals were excluded in the abbys test as they moved to slow to reach the edge of the cliff. As it detailed in the manuscript. |
| Replication | All details of the technical implementation is explained in details in the manuscript for reproductibility. |
| Randomization | Randomization was not performed because all mice were assigned to a single group per experiment type. |
| Blinding | Blinding was not possible as experimental conditions were evident from the image data. |

# Reporting for specific materials, systems and methods

We require information from authors about some types of materials, experimental systems and methods used in many studies. Here, indicate whether each material, system or method listed is relevant to your study. If you are not sure if a list item applies to your research, read the appropriate section before selecting a response.

## Materials & experimental systems

| n/a | Involved in the study |
|-----|----------------------|
| ☒ | ☐ Antibodies |
| ☒ | ☐ Eukaryotic cell lines |
| ☒ | ☐ Palaeontology and archaeology |
| ☐ | ☒ Animals and other organisms |
| ☒ | ☐ Clinical data |
| ☒ | ☐ Dual use research of concern |
| ☒ | ☐ Plants |

## Methods

| n/a | Involved in the study |
|-----|----------------------|
| ☒ | ☐ ChIP-seq |
| ☒ | ☐ Flow cytometry |
| ☒ | ☐ MRI-based neuroimaging |

# Animals and other research organisms

Policy information about studies involving animals; ARRIVE guidelines recommended for reporting animal research, and Sex and Gender in Research

| Laboratory animals | All experiments were conducted in accordance with the Animal Care and Experimentation Committee of the Institute of Experimental Medicine (IEM) and the Hungarian Scientific Ethics Council for Animal Experiments of the State Secretariat for Food Chain Supervision (approval reference numbers PE/EA/54-02/2019). Thy1-Cre FVB/AntFx adult mice (RRID:IMSR_JAX:006143, re-derivate and bred in the Medical Gene Technology Unit of IEM, 8-18 weeks old) of both sexes were used (n=47) and were housed in a temperature-controlled environment on a 12 h reverse light cycle in small groups (2-4 mice/home cage) in an enriched environment with rotary discs, cardboard rolls, and extra nesting material, at 23C° with an humidity of 45–65% and were provided food and water ad libitum. Mouse eye tissue of CD1 albino mice (RRID: IMSR_CRL:022) were used to the optical tests. |
|---|---|
| Wild animals | No wild animals were used in this study. |
| Reporting on sex | Thy 1-Cre FVB/AntFx adult mice (8-18 weeks old) of both sexes were used (n=47) to the in vivo experiments. To the mouse eye optical test, to directly validate the optical properties of the lens of the mouse eyeball we used CD1 albino mouse. |
| Field-collected samples | No field-collected samples were used in this study. |
| Ethics oversight | All experiments were conducted in accordance with the Animal Care and Experimentation Committee of the Institute of Experimental Medicine (IEM) and the Hungarian Scientific Ethics Council for Animal Experiments of the State Secretariat for Food Chain Supervision (approval reference numbers PE/EA/54-02/2019). |

Note that full information on the approval of the study protocol must also be provided in the manuscript.

# Plants

| Seed stocks | N/A |
|---|---|
| Novel plant genotypes | N/A |
| Authentication | N/A |

