## [Peer Review File · Nature Methods]

Moculus: an immersive virtual reality that revealed competition of cortical clusters during on-demand visual learning

Corresponding Author: Dr Balazs Rozsa

Version 0:

Decision Letter:

10th Jan 2024

Dear Dr Rozsa,

Thank you for your patience during the review process and discussion of the situation with my colleagues. Your Article, "Competition of cortical clusters during on-demand visual learning in immersive virtual reality", has been seen by three reviewers. As you will see from their comments below, although the reviewers find your work of considerable potential interest, they have raised a number of concerns. We are interested in the possibility of publishing your paper in Nature Methods, but would like to consider your response to these concerns before we reach a final decision on publication.

We therefore invite you to revise your manuscript to address these concerns. Specifically, there are concerns about the analysis of the imaging data and concerns about the conclusions. Furthermore, a stronger case for why your approach is enabling over other display systems should be made.

I would also like to mention a few issues that came up in our discussions within the editorial team. Currently, the manuscript is very focused on the neuroscientific findings. For Nature Methods, we do want the focus of the manuscript to be on the developed technology. Please do make sure to shift the balance of the manuscript accordingly.

Let me point out that our policies regarding related manuscripts under consideration elsewhere have been violated. Our policy states: "If you submit a related manuscript to any other journal **while the submission to Nature Methods is under consideration**, you must send us a copy of the related manuscript and details of its progress towards publication." While we have decided to be lenient here, I do want to make sure that you are aware that this is a special situation and that we will not tolerate this again.

In addition, I'd like to mention the issue of novelty. The iMRSIV system obviously compromises the novelty of your technology. However, since your manuscript has been under review when the iMRSIV system was published, we have decided to be lenient about concerns regarding novelty.

* include a point-by-point response to the reviewers and to any editorial suggestions

* please underline/highlight any additions to the text or areas with other significant changes to facilitate review of the revised manuscript

- * address the points listed described below to conform to our open science requirements
- * ensure it complies with our general format requirements as set out in our guide to authors at www.nature.com/naturemethods
- * resubmit all the necessary files electronically by using the link below to access your home page

Link Redacted

We hope to receive your revised paper within two months. If you cannot send it within this time, please let us know. In this event, we will still be happy to reconsider your paper at a later date so long as nothing similar has been accepted for publication at Nature Methods or published elsewhere.

OPEN SCIENCE REQUIREMENTS

REPORTING SUMMARY AND EDITORIAL POLICY CHECKLISTS

DATA AVAILABILITY

All novel DNA and RNA sequencing data, protein sequences, genetic polymorphisms, linked genotype and phenotype data, gene expression data, macromolecular structures, and proteomics data must be deposited in a publicly accessible database, and accession codes and associated hyperlinks must be provided in the "Data Availability" section.

Please include a "Data availability" subsection in the Online Methods. This section should inform readers about the availability of the data used to support the conclusions of your study, including accession codes to public repositories, references to source data that may be published alongside the paper, unique identifiers such as URLs to data repository entries, or data set DOIs, and any other statement about data availability. At a minimum, you should include the following statement: "The data that support the findings of this study are available from the corresponding author upon request", describing which data is available

upon request and mentioning any restrictions on availability. If DOIs are provided, please include these in the Reference list (authors, title, publisher (repository name), identifier, year). For more guidance on how to write this section please see: <http://www.nature.com/authors/policies/data/data-availability-statements-data-citations.pdf>

CODE AVAILABILITY

Please include a “Code Availability” subsection in the Online Methods which details how your custom code is made available. Only in rare cases (where code is not central to the main conclusions of the paper) is the statement “available upon request” allowed (and reasons should be specified).

MATERIALS AVAILABILITY

ORCID

Nature Methods is committed to improving transparency in authorship. As part of our efforts in this direction, we are now requesting that all authors identified as ‘corresponding author’ on published papers create and link their Open Researcher and Contributor Identifier (ORCID) with their account on the Manuscript Tracking System (MTS), prior to acceptance. This applies to primary research papers only. ORCID helps the scientific community achieve unambiguous attribution of all scholarly contributions. You can create and link your ORCID from the home page of the MTS by clicking on ‘Modify my Springer Nature account’. For more information please visit <http://www.springernature.com/orcid>

Best regards,
Nina

Nina Vogt, PhD
Senior Editor
Nature Methods

Reviewers' Comments:

Reviewer #1:

Remarks to the Author:

Learning leads to changes in neural coding in primary visual cortex, but previous studies have reported distinct or conflicting results. Judák, Dobos, Ócsai, et al. propose that one possibility for variability in the results of these studies is the relatively long time it takes to train animals on head-fixed visual tasks. They propose that an immersive virtual reality system would reduce training time and provide the necessary control over the visual environment to investigate learning in visual circuits. To this end, they design Moccus, a compact head-mounted virtual reality system that is compatible with existing systems to measure neural activity, such as two-photon imaging. Through the use of an aversive learning paradigm, they report that neurons in V1 rapidly develop anticipatory responses that precede the onset of locomotion to avoid an air puff. The authors find partially overlapping populations of neurons in V1 that code for the aversive and control conditions, and report a set of “hub” neurons that potentially drive responses in their respective clusters.

This work provides an important step forward in the development of better techniques for head-fixed virtual reality in rodents. The design, characterization, and calibration of the VR system is very impressive. The behavioral and imaging data appear to be of high quality, and the use of “the abyss” to test for immersiveness of the VR system is particularly compelling. There are significant issues regarding data analysis; most critically, as the authors point out, the apparent result that 100% of recorded neurons display coding for the aversive cue is surprising. Additional steps should be taken to ensure that appropriate neuropil

subtraction and calculations of responsive cells are being employed, especially given that much of the subsequent analysis and results hinge on this particular analysis. The manuscript could be substantially improved by addressing the following points:

-It is not entirely clear what is meant by “near saturating activation ratio” in the sentence “...both the number of responsive cells and the amplitude of the individual neuronal responses increased in both zones...to a near-saturating activation ratio.” Specifically, the term activation ratio (which I am not familiar with as a standard in physiology literature) does not appear to be explicitly defined. Based on other parts of the manuscript, activation ratio seems to refer to the fraction of neurons responsive to a particular cue (e.g. aversive zone). Assuming saturation means that 100% of neurons are responding to one of these cues (“we demonstrated that activation ratios during visual discrimination can reach saturating levels (100%) and were found over ca. 80% and 70% on average for the reinforcement-associated and control cues, respectively”), that is particularly surprising as only a fraction of V1 neurons are typically responsive to visual stimuli, and not all are modulated by locomotion/arousal. This raises some concern here about the analytical threshold for activation (2 SD above mean, which may not be sufficiently strict). A recent study (King, Ledochowitsch, Buice, & de Vries, *eNeuro* 2023) showed that this type of analysis overestimates the number of responsive neurons (in their case to saccades), and after implementing bootstrapping methods their estimated number of responsive neurons drops from about 50% down to 9%. Based on the raw data from Fig2d, there appear to be a significant number of neurons with little to no response, which would suggest implementing alternative analyses of responsiveness would provide a more accurate measure. The authors should utilize a more rigorous analysis, ideally akin to the King paper above, to determine the responsiveness of neurons.

-The high fraction of responsive neurons and apparent high activity correlation between cells also brings concern that the neuropil signal has not been successfully subtracted. Based on the methods, it appears that the authors used a custom set of analyses to perform motion correction and neuropil subtraction. Ideally they would utilize existing software, such as CalmAn (Giovannucci et al., *eLife* 2019) or Suite2p (Pachitariu et al., *bioRxiv* 2016), which have been used to analyze many published datasets. If the format of the data (chessboard scans vs. typical 2D image scans) preclude them from using the graphical user interface of such packages, they could incorporate the algorithms utilized by these software packages into their code (see Giovannucci methods, e.g., for modeling neuropil in different types of datasets). Ultimately, a quality control measure such as isolation distance should be implemented to determine the success of cell extraction (see Stringer & Pachitariu, *Current Opinion in Neurobiology* 2019).

-It is mentioned that the S system interfered less with the whiskers than the XL system, however there does not appear to be any quantification of the interaction between the whiskers and the system. If possible, the authors should quantify which whiskers contact the system and with what frequency.

-“At the beginning of learning (0-10 minutes, ‘early learning’), only a small fraction of neurons were active in both the control and aversive zones relative to the data recorded before learning...” is this saying fewer neurons were active in early learning versus before learning? The data are not shown for the “before learning” period in Fig2d. One would expect the responses to be similar before learning and during early learning. Continuing this sentence: “...and responses were not significantly different between the two zones” does this specifically refer to the average response? One would expect that different neurons would respond to the different zones given the distinct orientations present in the visual stimuli.

-Fig4f seems to suggest there is significant overlap in the hub cells between control and aversive conditions. Given the differences in averaging ramping activity between the two conditions, how do the authors explain this strong overlap in hub cells?

-Regarding the sentence “Therefore, the direction of causality between V1 and motor activity is also changing: V1 activity triggers running during high ramps and, in contrast, running triggers V1 activity during low ramps,” given the lack of causal perturbations to neural activity here, the authors might consider rewording this sentence as it is not known whether V1 activity causes running. The same is true for other sentences, such as “In summary, our data indicate that network activity in the V1 during high ramps is not a simple reflection of motor activity but rather represents local computations within V1 that triggers running with a significant delay.”

-For the sentence “In summary, our data indicated that reinforcement signals can contribute to the formation of long-term potentiation of neuronal activity at the level of individual neurons in the V1 region.,” the LTP terminology is typically reserved for synaptic measurements, thus the authors might choose different wording such as “long term changes in activity.”

-For the Discussion section titled “Transiently saturating activation ratios maximize computational power,” is there a reference for the math used to calculate computational power? This seems like an oversimplified measure given other implementations of estimating computational power (e.g., Crumiller et al., *Frontiers in Neuroscience* 2011). It also seems to be based on some assumptions about the maximal firing rates of neurons in frontal cortex in acute brain slices, which are likely different from typical V1 firing rates in vivo. Ultimately, this section does not seem necessary, and given the issues mentioned above with overestimating the number of active neurons, is not providing useful information.

-What is the lag of the VR system, i.e., the time from data input of mouse running speed vs. stimulus presentation change on the screen? How does the measured lag compare to other existing VR systems? The authors should discuss this.

-Fig1j: units (cm/s?) on y-axis

-Fig2d: the x-axis should be more clearly labeled - the “scale bar” with “cells” is not quantitatively informative. Numbers on the x-axis would be ideal, and number of cells/animals in the figure legend.

-Fig 2g: red dotted line (before learning aversive response) is missing from the middle panel

-The manuscript should be thoroughly edited for clarity. For instance, some sentences could use word choice or grammatical improvements, such as the subheading "Aversive-zone coding takes the encoding from control-zone coding during learning."

Reviewer #2:

Remarks to the Author:

One of the reasons for conflicting results from neurophysiological studies of visual learning with animals may come from the fact that training visual tasks can take rather long so that network mechanisms may already be modified before core experiments are conducted. There is a need for a behavioural protocol for visual tasks that is fast. For research questions involving binocular vision this furthermore requires real stereo images.

To address these shortcomings the authors developed Moccus, a head-mounted VR system for mice. It has an extended visual field of view, and provides stereoscopic vision with depth perception.

Experimental tests using this system provided convincing evidence, using the abyss test, that animals experience stereoscopic vision. Furthermore, training was achieved about 100-times faster than using traditional protocols.

One result of this in-depth engineered visual stimulation device was the uncovering of a circuit mechanisms of visual learning not observed before in mice.

The success of this new approach has its foundation in the meticulous engineering of this new instrument based on state-of-the-art optical modelling including measurements of a mouse eye's optical properties. This led to the design of an optimized optical assembly using a combination of a biconvex lens and a diffractive phase shifter. Beyond optics, the authors also made special efforts to optimize the geometry of the device to fit to the head of a mouse, and to correct by software optical distortions of the VR projection over the curved retina of a mouse.

The experimental tests with Moccus in comparison to single or dual monitor VR scenarios convincingly demonstrate that this device allows for a new level of realistic visual VR stimulation. Thus, it opens new areas of research hitherto thought impossible with head-fixed animals.

Furthermore, the fast training with this device makes it possible that within the less than one hour time span that is normally available as a session duration with a water deprived mouse an entire learning event can take place and be experimentally recorded. This additional significant achievement will likely trigger a whole new generation of experiments on learning and neuronal plasticity. Not surprisingly, the validation experiments for Moccus already uncovered network mechanisms not observed before.

The highly significant paper is very well written. Changes are not necessary.

Reviewer #3:

Remarks to the Author:

In this manuscript, Judak and colleagues present a novel head-fixed VR system for mice that enables full visual view stereoscopic stimuli presentation and performs behavioral experiments and details investigation of the neural correlates of visual context avoidance learning in the visual cortex.

The group developed state-of-the-art custom projection optics into the mouse retina to achieve impressive quality visual projection of the microdisplay onto the full retina with minimal distortions. Due to the small size of the projection optics they could provide independent stimulation of each eye in the compact form factor that allows for large imaging objectives over the neocortex. The advantage of the new design of the VR is illustrated with a cliff detection task (in comparison to a dual monitor setup). The rest of the paper is dedicated to the demonstration of the new visual contextual avoidance task, rapid within single session learning in Moccus, and a very detailed analysis of the neural activity in the visual cortex (recorded with 2 photon microscopy) associated with this learning. The authors discover several interesting features of population activity that are correlated with learning and argue that this was only possible due to enhanced properties of Moccus compared to conventional VR. However, the heavy focus of the paper on the neural correlates of the single-task learning in the Moccus setup goes into so many details that it fails to provide a comparison to neural activity during learning in the conventional setup, making it less of a demonstration of the new methodology and more of an investigation of the visual contextual avoidance learning in the visual cortex. While a comparison of the behavioral performance and learning dynamics between Moccus and a two-monitor conventional VR is shown, it is not at all clear what in the task used for visual learning is critical for fast learning, as the task relies on orientation grating textures and no specific 3D virtual environment features, or stereo vision, or interaction with the environment or any other aspects that Moccus could be particularly beneficial for. At the same time, in my view, the richness and complexity of the uncovered neural correlates of the task are so high that even four main figures and a dozen supplementary figures do not provide a complete and coherent story (see comments below).

Overall, as the comments below show, I am not seeing the fast learning of the new task as the best use-case or validator of the advantages of Moccus. Having only visual cliff paradigm left as a VR-feature benefiting behavior, I am not seeing a very strong case for the method and its advantages. Clearly, stereo-VR could be very beneficial, small form-factor as well, but they are not demonstrated beyond the cliff experiment.

That being said, I am highly impressed by the quality of the methodological side of the work and care for details in optics design, mechanics design, distortions corrections etc. If not for the specific method at hand, but for the rigorous approach to the method development this paper is a prime example for the field. I can't judge whether opto-acoustic 2 photon scanning is completely novel and can be featured as part of the methodological toolbox, but it is certainly providing high-quality population

data. A much more focused on the demonstration of the beneficial features of the Mculus would make a very strong methods paper. However, the major portion of the paper goes in a very different direction and is much more suitable for the research article.

The edge of abyss experiment demonstrating the immersive properties of the Mculus for the naive mice is working impressively well. It would be useful to know whether further training and habituation of the mice in the single and double monitor setups lead to improved performance. To be fair, it is not clear if these setups represent the state-of-the-art head-fixed VR, as dome-based systems present less inconsistent and distracting visual features such as edges of the screens etc, and have larger visual fields of view. It is worth noting that depth perception is also natively emerging in the freely-moving VR systems (Stowers et al 2017, Del Grosso et al 2017), thus Mculus appears to be comparable in immersive power with those even in the absence of head movements. Thus while authors do show good performance in their setup, comparison to a very particular and likely suboptimal conventional setup doesn't make a strong case for Mculus as a game-changer. However, it is certainly a very good and economical alternative to the existing dome-like systems.

In contrast to freely-moving VR the improved visual stimulation of the Mculus under head-fixation does not provide a sense of full spatial understanding of the virtual space. A greater level of immersion into the VR should be coupled with a better spatial understanding of the environment, which is not demonstrated here. It is well-accepted in the field of spatial cognition that it is the congruence between idiothetic and allothetic cues that enables coherent spatial representation and, ultimately, understanding of space. Rats (Aronov et al Neuron 2014) and mice (Chen et al eLife 2018) in the body/head-fixed VR setup only obtain 2D spatial representations (place, grid and head-direction tuning) when their head is allowed to rotate. In fact, this points out that idiothetic vestibular cues are critical for immersiveness. Linear translational idiothetic cues are lacking under head fixation and, hence, full immersion is not possible. Strong response to an abyss, while impressive, doesn't recapitulate full immersion in the VR space in general.

Faster learning of the contextual avoidance in the Mculus system compared to the conventional Monitor-based setup is a nice added benefit of the system. It would be nice to show how the behavioral performance grows for both mouse groups within a day across 40 min of training as well as across all days, not just day 1 and day 5, as current figure 2f shows and more than Ext Fig9g shows. It would be important to demonstrate that learning performance is staying high for the Mculus group and when it catches up in the Monitor group. Showing in which 10-minute training block each mouse is reaching the threshold of good performance would lead to more nuanced statistics than average over-time performance across all mice in a group. Ext Fig9g seems to show that the performance of the Mculus group mice is not very stable and they do not stay at a good performance on day 2 or even later. In fact, only 4 out of 6 mice reached high performance on day 5, and only 2 on day 2. This is in contrast to the shaded confidence plot that Fig2f is showing. At the same time, the Monitor group seems to be showing non-random below-chance performance. Both this unexplained "significant" bias in behavior (freezing in the aversive zone?) and very small group size (3 mice) prevent, in my view, any possibility for statistical comparison between the groups. Showing behavioral and neural responses for the Monitor mice (as Fig2e,g) would allow the reader to assess the type of responses that this group has and group statistics (as f,h) would be useful as well. It is not clear from the current exposition how and to what extent the neural engagement of V1 in the Monitor group is different even if they do not learn as fast. Finally, as Fig3 exposes, the behavior is highly variable across trials even during periods of apartment high performance on average (authors refer to stochasticity of visual perception, which I don't find justified until other factors are ruled out). It would thus be necessary to expose if and why sensory experience in Mculus is different from conventional VR, what this variability is related to and if it contributes to the trial-average performance assessment.

On a more conceptual level, while this task is demonstrating rapid visio-spatial learning, the main claim is that without the advanced features of Mculus it would not be possible. However, for this particular task, a head-fixed setup is not required, as contextual avoidance paradigms are well established in freely-moving rodents and large scale electrophysiological (especially with the Neuropixel probes), 1PM (Inscopix, Miniscope) and even 2PM imaging are all possible. I could not discern from the paper why does one need Mculus for studying this specific learning paradigm. Moreover, there is no comparison of the discovered aspects of neural correlates of learning with the ground truth for the head-fixed VR - freely moving equivalent of the task. Is learning of the task faster than in the freely-moving analog? Are neural correlates matching those in the freely-moving analog?

Is ramp-like firing rate increase associated with running velocity increase in the aversive zone compared to control? If well-established effect of velocity cannot be regressed out, I don't see how the term "encoding" for the aversive/control zones can be applied. Similarly, quantification of the time-dependent effect given the increase in velocity is not justified. All arguments for differential coding in Fig4 are also confounded majorly by the velocity modulation, which is not identical for different neurons, hence there is a slight difference btw aversive and control zones in activity rate (e.g. Fig4c). According to Fig2e till about 1 sec before the end zone the velocity is not different between aversive and control zones on average, but the gain in firing rate is maximal in exactly after this point (Fig4a,b), when velocity in the aversive zone starts increasing compared to control.

The logic of the exposition of the results is not very easy to follow. The authors start by presenting the average behavior (velocity) and neural activity (population Ca imaging) comparing aversive and control zones (Fig2); then they show that both velocity and neural activity fluctuate from trial to trial even at the end of 1st day learning and are associated with neural activity being pre- and postdictive to motor activity depending on the trial (Fig3); then they turn to the analysis of the neural variability based on the trial-average responses in aversive and control zones splitting cells into hub, good and poor learners (Fig4); then they link early trials neural responses to emergence of the ramp and, finally, attempt to capture neuron/trial/time variability using TCA. I did not get a sense of coherent and focused approach to dissect the neural phenomenon, but rather combination of many interesting and diverse results which do not compile into a coherent picture. While authors use many terms "saturation", "sparse", "orthogonal" etc, they often do not clearly link to the proper statistics or theory and rather distract from the

understanding of results. Control and proper regression of velocity are coming much after the terms “coding” get applied to changes of the firing rate that well parallel changes in velocity. Another example of rushed conclusions is that TCA analysis, that shows very few cells having non-zero loading on components with diverse temporal and equally sparse trial loadings. The meaning of these factors/components is very different from the interpretation authors make. It is not appearing as a robust low-dimensional and comprehensive summary of the large variability in the neural data/behavior, but an overfitting result. In general, the notion of coding and other theoretical concepts appears to be too strongly pushed throughout the paper with no validation or reference to theoretical work. It is only at Fig4 that contribution from locomotion becomes clear and is acknowledged and anticipation of reward or motor confounds can become separated.

Authors observe in aversive and control stimulus areas significant trial-to-trial variability of the activation of V1 neurons, normally having orientation tuning in the retinal space. Could this be related to the trial-to-trial variability of the pupil orientation of the left and right eyes while mouse is running through the grating pattern which results in the differential orientation-tuning-based drive by distinct grating patterns? Some authors (e.g. A. Meyer, C. Neil) have demonstrated independent movement of the eyes by mice, which in the current paradigm would likely be comparable. It would be important to control whether and to what extent the degree and consistency of the eye movement by mice in the 2 monitor (or conventional dome VR setup) differs from that in the Mculus setup.

Another aspect of the specific implementation of the task that is not discussed, but could play a role, is that control zones are composed of +/- 45 degree stripes orientation, while the aversive zone is composed of 0 degree orientation. This would result in a differential optic flow effect and, potentially, differential eye movement induced. Did the authors test whether a different, counterbalanced, design of the control/aversive orientations of the grating give rise to a comparable behavioral and neural effects? Does a non-grating pattern work equally well? It would be important to test the generality of the visual context when arguing for the principles of the visual learning neural correlates.

Frame and refresh rate, and cohesion with other senses are important factors that contribute to immersion into the VR. It would be worth discussing how miniaturization of the VR setup led to compromised (frame rate or contrast of the micro screen) performance of the VR in other dimensions compared to state-of-the-art possible with larger performant projectors and screens.

Minor comments:

Authors emphasize that whiskers are not affected by their optic contraption. But this is also the case for the standard head-fixed VR. Moreover, false tactile feedback from parts of the setup is no less false than a lack of feedback when the mouse approaches virtual boundaries. Thus optimized mechanical design is a nice feature, but its advantage needs to be demonstrated. If authors want to emphasize this feature there needs to be a demonstration of the behavioral paradigm that makes use of the whiskers in combination with visual VR.

Ref 62 is incomplete.

While the improved 2PM imaging technique is impressive, it doesn't appear to be a critical and integral component of the focus of the paper - Mculus. I would therefore move some of the details of the methods in this section to the results, unless they are important to understand the validation of the Mculus setup.

Fig2d - please label x-axis. It appears to be a number of neurons, based on legends, but more explicit labeling would help.

Results exposition jumps from Fig2 to Fig4 and then back to Fig3, consider swapping figures.

Version 1:

Decision Letter:

Our ref: NMETH-A53725A

6th Aug 2024

Dear Dr. Rozsa,

Thank you for your patience and for submitting your revised manuscript "Competition of cortical clusters during on-demand visual learning in immersive virtual reality" (NMETH-A53725A). As I mentioned before, the manuscript has now been seen by the original referees and their comments are below. We have also discussed your responses to remaining concerns of reviewer #1 with this reviewer. They remain concerned about the data analysis. As there are no remaining concerns about the technology, we'll be happy in principle to publish your manuscript in Nature Methods, pending minor revisions to satisfy the referees' final requests and to comply with our editorial and formatting guidelines.

Specifically, I would like to ask you to either remove some of the offending data analysis or to add a strong disclaimer that the analysis is non-standard.

TRANSPARENT PEER REVIEW

Nature Methods offers a transparent peer review option for new original research manuscripts submitted from 17th February 2021. We encourage increased transparency in peer review by publishing the reviewer comments, author rebuttal letters and editorial decision letters if the authors agree. Such peer review material is made available as a supplementary peer review file. **Please state in the cover letter ‘I wish to participate in transparent peer review’ if you want to opt in, or ‘I do not wish to participate in transparent peer review’ if you don’t.** Failure to state your preference will result in delays in accepting your manuscript for publication.

Please note: we allow redactions to authors’ rebuttal and reviewer comments in the interest of confidentiality. If you are concerned about the release of confidential data, please let us know specifically what information you would like to have removed. Please note that we cannot incorporate redactions for any other reasons. Reviewer names will be published in the peer review files if the reviewer signed the comments to authors, or if reviewers explicitly agree to release their name. For more information, please refer to our [FAQ page](https://www.nature.com/documents/nr-transparent-peer-review.pdf).

ORCID

Best regards,
Nina

Nina Vogt, PhD
Senior Editor
Nature Methods

Reviewer #1 (Remarks to the Author):

The revised manuscript “Competition of cortical clusters during on-demand visual learning in immersive virtual reality” by Judák, Dobos, Ócsai et al. addresses a significant amount of the comments from reviewers. The custom implementation of an alternative cell extraction method and detailed characterization of whisker contacts with the VR system are impressive. Two main issues remain, detailed here:

1) The custom implementation of Suite2p on a “non-traditional” dataset was clearly not a trivial process, and the authors are applauded for their efforts. They report the result of using this cell extraction method: “In summary, our original method, the Suite2p algorithm without (87%) and with subtraction of neuropil contamination (84%), gave a neuronal activation ratio very similar to our original pipeline (97.6%, Ref. Fig. 5f).” However, I would argue that 84% and 97.6% are not very similar numbers, and that the results with Suite2p analysis appear closer to the trends apparent in the raw data from Figure 3D (the ramping effect is not visible in 100% of neurons). While the individual effect of neuropil subtraction was relatively small (~3%) on their measurements of activation ratio, a perceived lack of change in results by the authors may not be the best argument for use of custom code over a tool accepted by the field that promotes consistency across studies. Given the ability to apply this tool to their custom dataset, the use of this standard technique seems more appropriate than custom analyses.

2) The in-depth discussion of the King et al. paper is genuinely appreciated. In the course of this discussion, however, two issues arise. First, data are apparently being removed from the baseline period in the calculation of activation ratio (“intervals with spontaneous neuronal events were neglected” and “periods with spontaneous events (black stars in Ref. Fig. 2a) were removed before the baseline was calculated.”). It is not clear that this is a scientifically justifiable practice, and will certainly bias results toward “activation” of neurons since the recalculated baselines will inherently be lower. Second, one-sample one-tailed t-tests are being used to calculate whether neurons show significant activation. The use of a one-tailed t-test inherently assumes that neural activity will go in one direction (in this case, up). Neural activity could go up, down, or show no change, and it is the authors’ assumption that the response should go up, so the choice in statistical test could bias the result, and a two-tailed test would be more appropriate. Furthermore, if one-sample t-tests will be used on every neuron, multiple comparisons will need to be accounted for to prevent false positives. In this case, with approximately 85 neurons, the significance level should be on the order of $p < 0.0006$ using Bonferroni correction.

In sum, there appears to be a very interesting and real result about changes in neural activity with very rapid learning here, however it is suggested that the authors use caution not to overinflate the incidence of this effect across the neuronal population and instead rely on standard statistical techniques to determine the prevalence of this signal in the population.

Reviewer #2 (Remarks to the Author):

The authors have addressed the reviewers concerns comprehensively. I have no further suggestions for improvement.

Reviewer #3 (Remarks to the Author):

The authors made a significant improvement to the manuscript and addressed most of my concerns.

Version 2:

Decision Letter:

29th Oct 2024

Dear Balázs,

I am pleased to inform you that your Article, "Moculus: an immersive virtual reality that revealed competition of cortical clusters during on-demand visual learning", has now been accepted for publication in Nature Methods. The received and accepted dates will be September 13th, 2023 and October 29th, 2024. This note is intended to let you know what to expect from us over the next month or so, and to let you know where to address any further questions.

Over the next few weeks, your paper will be copyedited to ensure that it conforms to Nature Methods style. Once your paper is typeset, you will receive an email with a link to choose the appropriate publishing options for your paper and our Author Services team will be in touch regarding any additional information that may be required. It is extremely important that you let us know now whether you will be difficult to contact over the next month. If this is the case, we ask that you send us the contact information (email, phone and fax) of someone who will be able to check the proofs and deal with any last-minute problems.

Please note that *Nature Methods* is a Transformative Journal (TJ). Authors may publish their research with us through the traditional subscription access route or make their paper immediately open access through payment of an article-processing charge (APC). Authors will not be required to make a final decision about access to their article until it has been accepted. [Find out more about Transformative Journals](https://www.springernature.com/gp/open-research/transformative-journals)

If you are active on Twitter/X and haven't already done so, please e-mail me your and your coauthors' handles so that we may tag you when the paper is published.

You can now use a single sign-on for all your accounts, view the status of all your manuscript submissions and reviews,

access usage statistics for your published articles and download a record of your refereeing activity for the Nature journals.

Best regards,
Nina

Nina Vogt, PhD
Senior Editor
Nature Methods

** Visit the Springer Nature Editorial and Publishing website at http://editorial-jobs.springernature.com?utm_source=ejP_NMeth_email&utm_medium=ejP_NMeth_email&utm_campaign=ejp_Nmeth for more information about our career opportunities. If you have any questions please click [here](mailto:editorial.publishing.jobs@springernature.com).

Open Access This Peer Review File is licensed under a Creative Commons Attribution 4.0 International License, which permits use, sharing, adaptation, distribution and reproduction in any medium or format, as long as you give appropriate credit to the original author(s) and the source, provide a link to the Creative Commons license, and indicate if changes were made. In cases where reviewers are anonymous, credit should be given to 'Anonymous Referee' and the source.

Reviewers' Comments:

Reviewer #1:

Remarks to the Author:

Learning leads to changes in neural coding in primary visual cortex, but previous studies have reported distinct or conflicting results. Judák, Dobos, Ócsai, et al. propose that one possibility for variability in the results of these studies is the relatively long time it takes to train animals on head-fixed visual tasks. They propose that an immersive virtual reality system would reduce training time and provide the necessary control over the visual environment to investigate learning in visual circuits. To this end, they design Moccus, a compact head-mounted virtual reality system that is compatible with existing systems to measure neural activity, such as two-photon imaging. Through the use of an aversive learning paradigm, they report that neurons in V1 rapidly develop anticipatory responses that precede the onset of locomotion to avoid an air puff. The authors find partially overlapping populations of neurons in V1 that code for the aversive and control conditions, and report a set of “hub” neurons that potentially drive responses in their respective clusters.

This work provides an important step forward in the development of better techniques for head-fixed virtual reality in rodents. The design, characterization, and calibration of the VR system is very impressive. The behavioral and imaging data appear to be of high quality, and the use of “the abyss” to test for immersiveness of the VR system is particularly compelling.

1.1. We thank the referee for her/his positive comment on our manuscript.

There are significant issues regarding data analysis; most critically, as the authors point out, the apparent result that 100% of recorded neurons display coding for the aversive cue is surprising.

1.2. In the previous version of the manuscript, we demonstrated that $80.1 \pm 5.9\%$ of the recorded neurons exhibited coding for the aversive zone. A similar high activation ratio (almost 90%) in the V1 region of the cortex has already been demonstrated in previous work⁶ for a similar reinforcement learning protocol (see the ratios of task-related cells in **Ref. Fig. 1E.**, The authors found $88 \pm 3\%$ of cells to be task related by the end of the training, defined as a significant change in the response after the reward zone onset (see ‘*Most V1 Layer 2/3 Neurons Display Task-Related Responses after Learning*’ chapter and Figure S1 in Paken et al.)⁶.

Redacted

The main difference between our study and that of Pagan et al. is that they used a reward instead of an aversive stimulus. Nevertheless, previous research has shown that both reward and aversive stimuli elicit similar cortex-wide activation during learning: about 83% and 85% of the VIP cells responded to reward and punishment, respectively⁷. We can therefore expect the activation ratio for aversive stimuli to be just as high (~80%) as for rewarding stimuli.

Pagan et al. used only a single grating pattern in combination with a background, thus their protocol can be defined as a simplified visual discrimination protocol or task engagement (**Ref. Fig. 1A**). In

contrast, with multiple grating patterns, we performed a conventional visual discrimination task in 3D and demonstrated that similar high activation ratios (70%-80%) for all grating patterns that were to be discriminated.

See also our response in section 1.3.

Additional steps should be taken to ensure that appropriate neuropil subtraction and calculations of responsive cells are being employed, especially given that much of the subsequent analysis and results hinge on this particular analysis. The manuscript could be substantially improved by addressing the following points:

-It is not entirely clear what is meant by “near saturating activation ratio” in the sentence “...both the number of responsive cells and the amplitude of the individual neuronal responses increased in both zones...to a near-saturating activation ratio.” Specifically, the term activation ratio (which I am not familiar with as a standard in physiology literature) does not appear to be explicitly defined. Based on other parts of the manuscript, activation ratio seems to refer to the fraction of neurons responsive to a particular cue (e.g. aversive zone). Assuming saturation means that 100% of neurons are responding to one of these cues (“we demonstrated that activation ratios during visual discrimination can reach saturating levels (100%) and were found over ca. 80% and 70% on average for the reinforcement-associated and control cues, respectively”), that is particularly surprising as only a fraction of V1 neurons are typically responsive to visual stimuli, and not all are modulated by locomotion/arousal. This raises some concern here about the analytical threshold for activation (2 SD above mean, which may not be sufficiently strict). A recent study (King, Ledochowitsch, Buice, & de Vries, eNeuro 2023) showed that this type of analysis overestimates the number of responsive neurons (in their case to saccades), and after implementing bootstrapping methods their estimated number of responsive neurons drops from about 50% down to 9%. Based on the raw data from Fig2d, there appear to be a significant number of neurons with little to no response, which would suggest implementing alternative analyses of responsiveness would provide a more accurate measure. The authors should utilize a more rigorous analysis, ideally akin to the King paper above, to determine the responsiveness of neurons.

1.3. We thank the referee for drawing our attention to this important and complex issue. We have changed the lookup table of the aversive responses in **Fig. 2d** of the previous version of the manuscript to more clearly show small responses (**Ref. Fig. 2a**). To improve the visibility of small responses, the baseline was reset at the level of individual neurons. In the majority of the studies, the baseline is calculated within a predetermined interval of fixed length, which is relatively long in order to minimize the contribution of spontaneous activity. We used the same approach but intervals with spontaneous neuronal events were neglected. Next, neurons were sorted according to their ramp-like response amplitudes before the time of the airpuff (**Ref. Fig. 2a**). Unlike the upper end of the distribution, the mean responses of the lowest 10 cells of the distribution were not visible (**Ref. Fig. 2a**), so we have plotted the mean Ca^{2+} transients (mean \pm SEM) of these 10 neurons for clarity (**Ref. Fig. 2b**).

Activation ratio at the end of the zones was defined as the ratio of the neurons with significant activity increase before the airpuff onset relative to the spontaneous event-free baseline period. If, in the first step, the mean responses of each trial were calculated from $-1,600$ ms to 600 ms before the airpuff, and in the second step, one sample one-tail t-tests were performed⁷⁻¹⁰, 8 of the 10 cells showed significantly increased responses (**Ref. Fig. 2b**), which means a 97.6% activation ratio (83 / 85 cells). However, the t-test recalculated for the non-significant two cells in a wider time interval (from $-3,300$ ms to -400 ms) results in significant P values: $P = 0.046$, $P = 0.02$, indicating a 100% activation ratio. During these calculations, periods with spontaneous events (black stars in **Ref. Fig. 2a**) were removed before the baseline was calculated.

Ref. Fig. 2. Average neuronal responses and activation ratios at the end of the aversive zone at the end of learning (30-40 minutes). **a**, Average neuronal responses at the end of the aversive zone from Fig. 3d, 5a,b. Responses were sorted according to their ramp-like component and are shown with a LUT set for small responses. Top, intervals with spontaneous activities (*) were excluded during baseline calculation. Bottom, average neuronal responses from the top (10 cells) and bottom (10 cells) ends of the distribution are shown. **b**, Average responses (mean±SEM) of the 10 cells with the smallest response amplitude from amplitude distribution shown in panel a. Significant and non-significant t-test P values are shown in black and red, respectively. If, in the first step, the mean responses of each trial were calculated from -1600 ms to 600 ms before the airpuff, and in the second step, with one sample, one-tail t-tests were performed⁷⁻¹⁰, 8 of the 10 cells showed significantly increased responses, which means a 97.6% total activation ratio (83 significantly active / 85 all cells). However, the t-test recalculated for the non-significant two cells in a wider time interval (from -3300 ms to -400 ms) results in significant P values: $P=0.046$, $P=0.02$ indicating a 100% activation ratio. **c**, The ratio of significantly active cells after 30-40 minutes training calculated at 1-second intervals. The average activation ratio calculated for the whole interval is indicated by the red dotted line. The red arrow in the middle indicates the interval defined as the baseline. The two other red arrows indicate the intervals where activity had fallen to the baseline level. Zero time indicates the time when mice left the aversive zone. The activation ratio increased rapidly during the ramp period from a baseline activation of 4.7% to a saturating activation ratio of 100%, peaked at airpuff, and then returned to the baseline value. The periodic nature of the motion with the presence in the previous zones explains the increase in activation ratio before the baseline period at about minus 30 s. The mean activation ratio in periods #1 and #2 was $38.3 \pm 4.0\%$ (red dotted line). The average activation ratio calculated for the three baseline periods (red arrows) was $2.67 \pm 0.76\%$. Therefore, to calculate a baseline activity-corrected activation ratio, we can

subtract 2.67% from the 97.6% and 100% activation ratios calculated above, giving baseline-corrected ratios of 94.93% and 97.33%, respectively.

We then calculated the activation ratios as a function of time. The average neuronal activity in the interval from -1.6 s to -0.6 s before the end of the aversive zone (0 s) was calculated and compared with the average response in a baseline period of a 3 s duration with a paired one-tail t-test. We then shifted the interval [-1.6 s to -0.6 s] in steps of one second along the x-axis in both directions and repeated the same paired t-test to calculate the activation ratio (**Ref. Fig. 2c**). After a gradual increase in the ramping period, which corresponds to the average increase in neuronal activity shown in **Fig. 2g** of the previous version of the manuscript, activation ratio peaked at the time of the airpuff and then decreased to the baseline. However, the activation ratio also increased before the baseline interval and peaked at around 80-90% at around -30 s, before returning to the baseline value at around -45 s. **Fig. 3a** in the previous version of the manuscript explains this phenomenon: motion in VR systems generally show a periodic pattern, which means that mice repetitively return to the same zones where neuronal activation ratio is increasing, so that the peak at -30 s can be explained by increase in activity when mice went to the previous zones. Consistent with this, the average spatial distribution of mice in the previous control and aversive zones peaked from -50 s to -28 s and the end of this interval coincided with the peak in activation ratio at around -30 s. In summary, the presence in the previous zones explains the increase in activation ratio before the baseline period at about -30 s.

Behavior can be considered as a periodic pattern in VR reality systems as described above. In **Ref. Fig. 2c** the activation ratio reached three times the activation ratio of the baseline interval (indicated with red arrows in **Ref. Fig. 2c**) separating two intervals of the motion (defined as period #1 and #2 in **Ref. Fig. 2c**). As inter zone time varies between two successive zones, period #1 is longer than period #2. The average activation ratio in period #1 was $28.4 \pm 3.4\%$. This means that use of different randomization methods (e.g., shuffling or bootstrapping) to determine the baseline activity in a wider temporal window than our short baseline interval (indicated with the middle red arrow in **Ref. Fig. 2c**) would result in an activation ratio of 28.4%.

The average activation ratio in periods #1 and #2 was $38.3 \pm 4.0\%$ (red dotted line in **Ref. Fig. 2c**). This means that if, ignoring the periodic nature of movement in the behavioral protocols, we calculate a baseline activation ratio by using different shuffling or bootstrapping methods outside the short baseline interval we have defined, we can expect a relatively high activation ratio of about $38.3 \pm 4.0\%$.

Chase W. King et al.¹¹ used a similar approach to define activation ratio: they selected a short interval of 333 ms before and after the saccades, averaged the data and compared the means using a pairwise statistical test (Wilcoxon signed-rank test). This approach resulted in a high activation rate of 52.7%, similar to that in our study. However, in the next step, to identify neurons with robust saccade responses, they used a stricter bootstrapping procedure that reduced activation ratio to 9.4%. The difference between the two numbers, which was taken as a baseline activity in their study, is 43.3% (= 52.7% - 9.4%), which is very close to the average activation ratio we can calculate from our data by shuffling ($38.3 \pm 4.0\%$ for period #1+#2, and $28.4 \pm 3.4\%$ for period #1). This high baseline activity value (43.3%), which has been subtracted by the bootstrapping method, is well reflected in individual neuronal responses before saccades: see for example Fig. 2B¹¹ (**Ref. Fig. 3**). However, subtracting this relatively high baseline neuronal activity from saccade-associated responses using different bootstrapping methods would increase the ratio of non-significant saccade responses (failures). Similarly to our study, there were conditions at the level of individual saccades when neuronal activity was higher (indicated by red asterisks in **Ref. Fig. 4**). In these cases, the preceding baseline activity was lower. This inverse relationship between the neuronal responses associated with saccades and the

preceding baseline activity can also be observed at the level of average response groups: if, before time zero, the mean response of a group of saccades was below the baseline calculated by a bootstrapping method, the saccade-associated responses were higher (see the red arrows in **Ref. Fig. 4**). These data suggest that the relatively high average brain activity, which can also be determined by different shuffling methods, is transformed into a condition where two network states, a high-amplitude highly synchronic state and low-amplitude low synchronic state, alternate in different behavioral protocols when motion is periodic. Consistent with this, Pagan et al. 2018 show a similar effect (**Ref. Fig. 1**): when a subset of neurons became responsive in the V1 region during visual learning, neuronal activity not only increased in the rewarded location but also decreased in the preceding baseline period (**Ref. Fig. 1A**). This rearrangement of network activity into high and low states is also visible at the level of individual cells (note the decreased spontaneous activity in the baseline period after learning **Ref. Fig. 1A**).

Redacted

Similar to Pagan et al.⁶, in our study, periods with a low activation ratio appeared repeatedly between zones (e.g., indicated by red arrows in **Ref. Fig. 2c**). This allowed the baseline to be defined for these periods. We prefer this definition because, without taking into account previous network states, increases in activity associated with previously visited zones would mix high-activity intervals in the calculation of the baseline activation ratio during bootstrapping. The average activation ratio calculated for the three baseline periods (red arrows in **Ref. Fig. 2c**) was $2.67 \pm 0.76\%$. Therefore, to calculate a baseline activity corrected activation ratio, we can subtract 2.67% from the 97.6% and 100% activation ratios calculated above, giving a baseline-corrected activation ratio of 94.93% and 97.33%, respectively. Bootstrapping does not change the average baseline activation ratio if it is limited to the three baseline periods.

Redacted

We added **Ref. Fig. 2** as a new figure (**Extended Data Fig. 18**).

In the Supplementary Information we say:

“Extended Data Fig. 18. Average neuronal responses and activation ratios at the end of the aversive zone at the end of learning (30-40 minutes). a, Average neuronal responses at the end of the aversive zone from **Fig. 3d**. Responses were sorted according to their ramp-like component and are shown with a LUT set for low responses. Top, Intervals with spontaneous activities (*) were excluded during baseline calculation. Bottom, average neuronal responses from the top (10 cells) and bottom (10 cells) end of the distribution are shown. **b,** Average responses (mean±SEM) of the 10 cells with the lowest response amplitudes from the amplitude distribution shown in panel **a**. Significant and non-significant t-test P values are shown in black and red, respectively. If in the first step the mean responses of each trial were calculated from – 1,600 ms to 600 ms before the airpuff, and in the second step, one sample, one-tail t-tests were performed⁷⁻¹⁰, 8 of the 10 cells showed significantly increased responses, which means a 97.6% activation ratio (83 / 85 cells). However, a t-test recalculated for the non-significant two cells in a wider time interval (from – 3,300 ms to – 400 ms) results in significant P values: P =0.046, P =0.02

indicating 100% activation ratio **c**, The ratio of significantly active cells after 30-40 minutes training calculated at 1-second intervals. The average activation ratio calculated for the whole interval is indicated by the red dotted line. The red arrow in the middle indicates the interval defined as the baseline. The two other red arrows indicate the intervals where activity has fallen to the baseline level. Zero time indicates the time when mice left the aversive zone. Activation ratios increased rapidly during the ramp period from a baseline activation ratio of 4.7% to a saturating activation ratio 100%, peaked at airpuff, and then returned to the baseline value. The periodic nature of the motion present in the previous zones explains the increase in activation ratio before the baseline period at about minus 30 s. The mean activation ratio in periods #1 and #2 was $38.3 \pm 4.0\%$ (red dotted line). The average activation ratio calculated for the three baseline periods (red arrows) was $2.67 \pm 0.76\%$. Therefore, to calculate a baseline activity-corrected activation ratio, we can subtract 2.67% from the 97.6% and 100% activation ratios calculated above, giving baseline-corrected activation ratios of 94.93% and 97.33%, respectively."

In the manuscript we now say:

"To further quantify coding, we determined the activation ratio^{7,12} in a shifting 1,000 ms -long interval with baseline corrected transients (baselines were calculated after neglecting the interval of spontaneous activity). The activation ratio increased rapidly during the ramp period as a function of time from a baseline value of 4.7% to a near saturating activation ratio (97.6%-100%) and returned to the baseline (**Extended Data Fig. 18**). The periodic nature of motion in the VR resulted in average and baseline activation ratios of $38.3 \pm 4.0\%$ and $2.67 \pm 0.76\%$, respectively (**Extended Data Fig. 18c**). In contrast to all previous studies with visual discrimination, where the number of neurons responsive to the reinforcement-associated grating was only ca. 8-20%¹⁻³ or did not change significantly^{3,13,14}, we found that most ($80.1 \pm 5.9\%$) of the cells were activated on average ($n=5$ mice) at the end of the aversive zone during the fast-learning protocol with Mocolus (**Fig. 5e**, bottom, $n=5$ mice). Similarly, in contrast to the modest ($<10\%$)^{1-3,12} or missing activation ratio for the control cue^{3,13,14} reported previously, we found that $70.0 \pm 7\%$ of the cells were activated on average in the control zone (**Fig. 5e**, bottom, $n=5$ mice). Supporting these previous studies with the low or missing activation ratio, we obtained missing activation ratios when our training protocol was repeated with a classical monitor system (**Extended Data Fig. 19**)."

-The high fraction of responsive neurons and apparent high activity correlation between cells also brings concern that the neuropil signal has not been successfully subtracted. Based on the methods, it appears that the authors used a custom set of analyses to perform motion correction and neuropil subtraction. Ideally they would utilize existing software, such as CalmAn (Giovannucci et al., eLife 2019) or Suite2p (Pachitariu et al., bioRxiv 2016), which have been used to analyze many published datasets. If the format of the data (chessboard scans vs. typical 2D image scans) preclude them from using the graphical user interface of such packages, they could incorporate the algorithms utilized by these software packages into their code (see Giovannucci methods, e.g., for modeling neuropil in different types of datasets). Ultimately, a quality control measure such as isolation distance should be implemented to determine the success of cell extraction (see Stringer & Pachitariu, Current Opinion in Neurobiology 2019).

1.4.1 As requested by the reviewer, we integrated the Suite2p software package into our analysis pipeline and re-analyzed our data because its format did not allow use of the graphical user interface.

To avoid neuropile contamination, we use the Suite2p algorithm on the image by exporting it to TIFF or by directly reading the HDF5 file (our.mesc file format is a valid HDF5 file). For this analysis, after motion correction was performed in our imaging control software (MES, Femtonics Ltd.), the imaging data from chessboard fields were converted to a 2D mosaic for each timeframe and then exported into

a multilayer, scaled TIFF file. Suite2p successfully identified a putative cell on the vast majority of chessboard pieces (occasionally detecting multiple cells per chessboard field). If less than 50% of a cell was visible at the edge of a chessboard field, the respective cell was manually omitted from further analysis. From the output file (F.mat) automatically generated by the suite2p GUI, we extracted both the raw fluorescent curves and the neuropil curve for each ROI separately. The neuropil information was incorporated by subtracting 50% of the neuropil curve, which had been previously filtered by a Gauss filter with a temporal bandwidth of 2 frames.

Suite2p was able to detect not only the central neurons of individual frames of the chessboard scanning but also other neurons typically close to the edges of the frames of chessboard scanning, where brain movement caused by heartbeat, breathing or physical motion can generate a larger artefact (**Ref. Fig. 5**). As a result, the signal-to-noise ratio of neuronal responses decreased; whereas 3D chessboard scanning produced a standard deviation of 0.023 ± 0.011 (mean \pm SD, $\Delta F/F$) with our analysis pipeline in the baseline period, Suite2p produced a larger mean standard deviation: 0.21 ± 0.13 (mean \pm SD, $\Delta F/F$) in the same baseline period. However, the higher noise amplitude was in the amplitude range of ramp-like responses. Therefore, to reduce the effect of increased noise, neurons with a standard deviation greater than $0.3 \Delta F/F$ were removed before calculating the effect of neuropil contamination. This approach preserved only 80.4% of the active neurons detected with the Suite2p algorithm.

Note that this was a primary analysis using the automatic function of Suite2p. However, we see several solutions to reduce the noise level of suite2p data, which will include new 3D scanning methods and real-time motion corrections. This information lies beyond the scope of the present report and will be discussed in more detail in a separate manuscript that is currently being composed. Briefly, the Suite2p algorithm can be used on the images by exporting to TIFF or directly reading the HDF5 files (our .mesc file format is a valid HDF5 file). During this process, we encountered two additional problems: 1) Cells were lost at the limit of the scanning volume of the 3D AO microscopy. 2) The Suite2p algorithm generated large amplitude drops in responses in about 10% of the neurons when used in combination with 3D chessboard scanning. The first problem was caused by a decrease in radial laser intensity as a function of distance from the main optical axis and was successfully addressed by rewriting the electronics firmware. The laser intensity can now be precisely adjusted with the 3D AO scanner in wide field-of-view. This method is already compatible with the background correction algorithm of the Suite2p. Our analysis pipeline is not sensitive to the field curvature of the baseline fluorescence. As Suite2p was developed for images with field-of-view, the second problem may be due to the small size of the frames of the chessboard scans, which are more sensitive to brain motion, but we could not find the error in the algorithm. However, we were able to rule out the possibility that the amplitude drop was the result of neuronal inhibition. Consequently, all cells with these negative artifacts (about 10% of the cells) were excluded from the analysis above. In summary, although compatibility with chessboard scanning requires further development, we have added Suite2p to the Methods section as a technique that is compatible with wide field-of-view 3D AO imaging using Multi-Layer Imaging, Tilted Frame Scanning, High Speed Arbitrary Frame Scanning, and other methods.

Ref. Fig. 5. Comparison of the original analysis pipeline with Suite2p. a, Numbered regions detected by the integrated Suite2p method. **b**, Average responses (mean \pm SEM) of cell #27 at the end of the aversive zone after learning (30-40 minutes) with and without subtraction of neuropil contamination

were not significantly different. **c**, The same as Panel **b** but for a different cell. Responses with and without neuropil contamination were different. **d**, Population averages with and without subtraction of neuropil contamination show a non-significant trend. **e**, *t*-test *P*-values for individual neurons detected with the Suite2p method (*n*=87 cells) showed decreased significance when neuropil contamination was subtracted. **f**, Activation ratios calculated with our integrated analysis program (3D-AO) and with the Suite2p method, with and without subtraction of neuropil contamination.

We next investigated the contribution of neuropil activity to the responses of the neurons below the noise threshold. Although the activity of a few cells showed a remarkable neuropil contribution during the ramp-like activity and after airpuff (**Ref. Fig. 5c**), neuropil activation in the majority of neurons generated no significant differences in amplitude (**Ref. Fig. 5b**). Although neuropil contamination increased population responses after airpuff and during the ramp-like period, the effect was not significant (**Ref. Fig. 5d**). We investigated whether this non-significant tendency in the ramp-like period might affect the activation ratios by calculating the activation ratios during the ramp-like period (from -1.4 to -0.4 s) using a two-tailed one sample *t*-test (see also our previous paper⁷). This produced an activation ratio of 87% when neuropil activation was not subtracted in the Suite2p program, a value close to the activation ratio of 97.6% generated by our original analysis pipeline in the same ramp-like period (**Ref. Fig. 5f**). When the neuropil was subtracted with the Suite2p algorithm, the significance level decreased, meaning that the *P*-values calculated for the ramp-like period increased at the level of individual neurons (**Ref. Fig. 5e**); 3% of the cells lost significance, resulting in an activation ratio of 84%. These data indicate that, similar to the weak effect on mean calcium transients, neuropil activity has only a modest effect on the neuronal activation ratio in the ramp-like period. In summary, our original method, the Suite2p algorithm without (87%) and with subtraction of neuropil contamination (84%), gave a neuronal activation ratio very similar to our original pipeline (97.6%, **Ref. Fig. 5f**).

1.4.2. AO microscopy allows precise adjustment of the z coordinates, which can result in lower neuropil contamination

The relatively low contribution of neuropil signals to the data recorded with 3D chessboard scanning can be explained by the improved precision of the measurements with 3D AO microscopy. To illustrate this problem, we can take a model cell with a diameter of 10 μm (**Ref. Fig. 6a**). The boundary between the model cell and the neuropil is illustrated with a dashed red line. The point spread function (PSF) of the 3D AO microscope with a high NA (NA=1.0) objective is approximately $0.45 \mu\text{m} \times 2.5 \mu\text{m}^{15}$. During 3D laser scanning, the PSF of the microscope is convolved with the cell. The convolution of the PSF and the border layer of the neuropil on the surface of the model cell is illustrated in the x-z plane in **Ref. Fig. 6a**. The pink color shows the region where the product of the PSF and the neuropil boundary layer is not zero. Consequently, the neuropil signal will contribute to neuronal responses in the pink regions but not in the central green regions indicated as 'uncontaminated' (**Ref. Fig. 6b**). The expansion of the uncontaminated region along the z axis is only 5 μm in our model cell, indicating the need for high precision in z coordinate adjustments. As the PSF is elongated along the z axis, the contribution of the neuropil is asymmetric: it is higher for imaging planes that are further from the central plane of the neurons (plane A in **Ref. Fig. 6b**) and lower when the imaging plane is aligned with the center plane of the cell (plane B in **Ref. Fig. 6b**). The wide bandwidth of acousto-optical deflectors allows a spatial accuracy of around 50-100nm when setting the x, y, and z coordinates¹⁶, allowing us to minimize neuropil contribution. In addition, minimizing the contribution of the neuropil was the main reason for choosing a high NA lens rather than a wide field of view lens at the cost of reducing NA.

The majority of neuroscience laboratories perform imaging in one or more focal planes simultaneously and accept neurons in a relatively wide range of z-axis offsets (for example, the z-axis

offset threshold was 18.5 μm in Ref. Fig. 6c), making neuropil subtraction essential in these measurements. In summary, one of the main advantages of 3D chessboard scanning is that x, y, and z coordinates can be precisely aligned with the center of somata and, as a result, not only can the range of z-axis offsets be minimized to reduce the contribution of the neuropil, but the contribution of neighboring neuronal activity can also be decreased.

Ref. Fig. 6. Illustration of the contribution of neuropil signals during laser scanning. a, Schematic of the recording of a model cell with a diameter of 10 μm using the PSF of an 3D AO microscope^{15,16}. The boundary between the model cell and neuropil is indicated with a dashed red line. During laser scanning, the PSF is convolved with the model cell, which is illustrated in the x-z plane. As the PSF is asymmetric ($x=0.45 \mu\text{m} \times z=2.6 \mu\text{m}$), which is a characteristic of two-photon microscopy, this will result in an asymmetric distribution for regions with and without neuropil contamination. The pink color indicates the spatial distribution of the center of PSFs that partially excite neuropil, while the spatial location of the centers of PSFs that do not reach the neuropil region is shown in green. **b**, The same spatial distribution of PSF centers with and without neuropil contamination as in panel a, but two imaging planes are also indicated. If the model cell is imaged in plane B, which is the center plane of the model cell, the neuropil contamination, which is the ratio of pink and green areas, is low. However, using a more distant plane (plane A) results in a high contamination ratio. **c**, Redacted

In the manuscript we say:

“The advantage of 3D chessboard scanning is that x,y, and z coordinates can be precisely aligned with the center of somata. Therefore, unlike classical 2D laser scanning methods, z-axis offsets can be optimized to minimize the contribution of the neuropil.”

“We also employed Suite2p (an open-source library, available at <https://www.suite2p.org/>) for automatic cell detection and neuropil subtraction. Suite2p successfully identified putative cells, occasionally detecting multiple cells, on the vast majority of chessboard pieces. We observed a similar high activation ratio at the end of the aversive zone (87%) and a small contribution of neuropil signaling (3%).”

-It is mentioned that the S system interfered less with the whiskers than the XL system, however there does not appear to be any quantification of the interaction between the whiskers and the system. If possible, the authors should quantify which whiskers contact the system and with what frequency.

1.5. Thank you for your valuable observation. We have quantified the interaction between the S and XL version of Mocolus and the whiskers.

Various recordings of the mice were made from different angles while using different versions of our virtual reality systems (Mocolus S and XL versions). The small size of the S system makes it perfectly visible with a camera placed in front of a mouse whose whiskers were in contact with the screen while running or resting. However, in the case of the larger XL version, the interaction of the mouse whiskers with the system was more visible with a different and more specialized camera position, i.e., mounted at the bottom and used with a transparent Gramophone disc for better visibility.

The summary of these recordings can be found as **Supplementary Movie 3** (recording with the Mocolus XL using two camera positions) and **Supplementary Movie 4** (recording with Mocolus S using two camera positions). To identify the whiskers from the video, we used a whisker tracking software and identified the whisker based on the nomenclature used in the literature, see **Ref. Fig 6-4**¹⁸⁻²⁰. In our case, widely used software could not track whisker movement or determine frequency. Therefore, we developed our own detection method. Imaging of the whiskers was performed using a global shutter grayscale sensor with high framerate (200 Hz, Teledyne FLIR Integrated Imaging Solutions, BFS-U3-17S7M-C), and a lens with a focal length of 16 mm and an aperture of f1.8 (1" diameter, 5MP C-Mount Lens, Kowa lenses, LM16HC) under constant 2×220 lumen illumination from two different light sources at different angles (front and below, providing 480 LUX illumination measured at the position of the mouse). We implemented a postprocessing computer vision pipeline written in GLSL (OpenGL Shading Language). Shaders are executed on a Graphics Processing Unit to process the rendering pipeline, specifically how vertices and pixels (fragments) are processed and displayed on the screen. Our method consists of processing pixel data to determine the final color of each pixel of the rendered image.

Redacted

We used the following functions and processes for clear and comprehensible viewing from the two viewpoints and for the analysis of collisions of the whiskers and the devices:

- **Variables:** `uniform float u_scaleFactor;`, `uniform float u_contrast;`, `uniform float u_brightness;`, `uniform float u_width;`, `uniform float u_height;` are uniform variables used to control various aspects of the shader's operation, such as scaling, contrast, brightness, and the dimensions of the texture. The parameters enhance the images resulting in a sharp, contrasted, and well-visible image, ideal for examination purposes and as an input for edge-detection.
- `AddBrightness`, `AddContrast`, `owndFdx`, `owndFdy`, and `ownFwidth` are custom functions used to modify the color to detect the edges.
- `AddBrightness` and `AddContrast` adjusted the brightness and contrast of a colors.
- The `owndFdx` and `owndFdy` function computed the difference in color along the x and y directions and were used for gradient-based effects.
- `ownFwidth` calculated the total color change to estimate the magnitude of the gradient at a point.
- `EdgeDetection` based on the aforementioned gradient magnitude to determine edges on the input frames and scales this value by `u_scaleFactor` and adjusts the contrast according to `u_contrast`.
- The `main` function is the entry point of the image processing algorithm. It first retrieves the color of the current pixel from the first texture and applies a heatmap based on the intensity of the pixel's red channel (`color.x`).
- The `fragColor`, the output color of the fragment, is set by combining the heatmap color with the edge detection to visualize the highlighted edges of the whiskers segmented from the background.

According to the referee's request, *"the authors should quantify which whiskers contact the system and with what frequency"*, we have quantified, based on the videos, the whiskers that made contact with the S and XL versions of Mocusus and with what frequency (**Table S2**, see below). Based on the videos, we could divide mouse whiskers into three groups: i) those that always touch Mocusus ('full'), ii) those that never touch Mocusus ('no touch'), iii) those that touch Mocusus with a certain frequency. In the case of Mocusus S, 33% of all whiskers reached the device, while in the case of the Mocusus XL

this ratio was higher (53% of all whiskers). This is why we considered it important to miniaturize the Moccus device during development.

whisker label	Moccus S				Moccus XL			
	front view		bottom view		front view		bottom view	
	touch/freq. (Hz)	video label	touch/freq. (Hz)	video label	touch/freq. (Hz)	video label	touch/freq. (Hz)	video label
α	full	2	no touch	-	not visible	-	not visible	-
A1	no touch	-	no touch	-	not visible	-	60.00	3
A2	no touch	-	no touch	-	full	3	not visible	-
A3	32.60	1	full	1	full	2	12.38	1
A4	no touch	-	no touch	-	full	1	once	0
β	full	3	no touch	-	not visible	-	not visible	-
B1	32.60	5	25.86	4	full	8	not visible	-
B2	30.61	4	18.98	3	full	9	27.27	5
B3	27.27	6	37.50	2	full	4	23.47	4
B4	no touch	-	no touch	-	full	5	25.47	2
γ	full	7	26.78	8	not visible	-	full	10
C1	no touch	-	26.78	5	not visible	-	26.86	6
C2	no touch	-	26.78	6	not visible	-	27.27	7
C3	no touch	-	28.84	7	not visible	-	33.33	8
C4	no touch	-	no touch	-	36.84	6	27.27	9
C5	no touch	-	no touch	-	38.70	7	no touch	-
δ	no touch	-	no touch	-	not visible	-	full	12
D1	no touch	-	24.19	10	no touch	-	no touch	-
D2	no touch	-	23.07	9	no touch	-	no touch	-

Table S2. Interaction between the whiskers and the Moccus S and XL systems. (Related to **Supplementary Movie 3 and 4**). Using the standardized whisker nomenclature in **Ref. Fig 6-4**, we summarize which whiskers touched the Moccus S and XL devices with what frequency. ‘No touch’ means that a particular whisker did not touch the devices. The term ‘full’ refers to whiskers constantly touching the device.

We have added **Supplementary Movie 3 and 4**.

In the manuscript we now say:

“Imaging of the whiskers was performed using a global shutter grayscale sensor with high framerate (200 Hz, Teledyne FLIR Integrated Imaging Solutions, BFS-U3-17S7M-C), and a lens with a focal length of 16 mm and an aperture of f1.8 (1” diameter, 5MP C-Mount Lens, Kowa lenses, LM16HC) under constant 2×220 lumen illumination from two different light sources at different angles (front and below, providing 480 LUX illumination measured at the position of the mouse). We implemented a postprocessing computer vision pipeline written in GLSL (OpenGL Shading Language). Shaders are executed on a Graphics Processing Unit to process the rendering pipeline, specifically how vertices and pixels (fragments) are processed and displayed on the screen. Our method consists of processing pixel data to determine the final color of each pixel of the rendered image. In the case of Moccus S and XL, 33% and 53% of all whiskers reached the device, respectively, of which a part of the population made continuous contact with the device (Moccus S: 33 %, Moccus XL: 53 % of all whiskers), while the rest

touched the device at a certain frequency (Moculus S: 27.84 ± 4.74 Hz, Moculus XL: 30.8 ± 11.97 Hz, mean \pm SD, **Supplementary Movie 3 and 4**).”

-“At the beginning of learning (0-10 minutes, ‘early learning’), only a small fraction of neurons were active in both the control and aversive zones relative to the data recorded before learning...” is this saying fewer neurons were active in early learning versus before learning? The data are not shown for the “before learning” period in Fig2d. One would expect the responses to be similar before learning and during early learning. Continuing this sentence: “...and responses were not significantly different between the two zones” does this specifically refer to the average response? One would expect that different neurons would respond to the different zones given the distinct orientations present in the visual stimuli.

1.6.1. We thank the referee for the helpful comment. In **Fig. 2h** of the previous version of the manuscript, we demonstrated that the amplitude of ramp-like activity increased continuously as a function of learning time in the first 30 minutes, in both the aversive and control zones (**Ref. Fig. 7**). This increase was a non-significant tendency during 0-10 minutes of training (aversive zone: paired t-test, $P=0.177$, Mann-Whitney test $U=0.095$; control zone: paired t-test, $P=0.18$, Mann-Whitney test $U=0.40$) but became significantly higher than the pre-learning data after 10-20 minutes training in the aversive zone (paired t-test, $P=0.026$, Mann-Whitney test $U=0.021$). It nevertheless remained non-significant in the control zone (paired t-test, $P=0.10$, Mann-Whitney test $U=0.21$, one tailed paired t-test, $P=0.051$, one tailed Mann-Whitney test $U=0.105$). After 20-30 minutes of training, ramp-like activity showed a significant increase over pre-learning data in both zones (aversive zone: one-tailed paired t-test, $P=0.02693$, one-tailed Mann-Whitney test $U=0.03$; control zone: one-tailed paired t-test, $P=0.0184$, Mann-Whitney test $U=0.00609$). We have added the missing information to **Fig. 2h** (see **Ref. Fig. 7**).

Ref. Fig. 7. This is Fig 2h of the previous version of the manuscript.

1.6.2. “The data are not shown for the “before learning” period in **Fig. 2d**.”

Following the referee’s request, we have added before-learning data to **Fig. 2d** (**Ref. Fig. 8**):

Ref. Fig. 8. The average 3D Ca^{2+} responses before learning from the mouse shown in Fig. 2d.

1.6.3. “One would expect that different neurons would respond to the different zones given the distinct orientations present in the visual stimuli.” This was also our initial expectation. However, we found, similarly to Pakan et al, 2018 (Ref. Fig. 1)⁶, that the ratio of task-related cells, which are the aversive and control zone coding cells in our experiments, increased to a very high level close to saturation: the activation ratios in the aversive and control zones were 80.1% and 70.0%, respectively (Ref. Fig. 9a), suggesting that the proportion of neurons responding in both zones is high. When maximum orthogonality is assumed, the calculated overlap between the aversive and control zone coding populations is 50.1% [50.1%=70.0%-(100%-80.1%)]. In contrast, assuming a maximum overlap in coding, the overlap is 70.0% [70.0%=minimum (80.1%;70.0%)]. Indeed, the coding of the aversive zone and control zone overlapped (Ref. Fig. 9b). This overlap in coding is also reflected in the parallel increase in the mean ramp-like component in the two zones as a function of learning time (Ref. Fig. 7), and also in the spatial distribution of the clusters of the hub cells coding the aversive and control zones (Fig. 5f). Importantly, coding of the two zones also had an orthogonal component that was presented in a separate figure (Extended Data Fig. 12 of the previous version of the manuscript) and in the associated text.

Ref. Fig. 8-2. Overlap in neuronal coding. These panels are from Fig. 4e and c of the previous version of the manuscript.

We have added: Ref. Fig. 7 as Fig. 3h, and Ref. Fig. 8 as Fig. 3d top panel.

In the manuscript we now say at Fig. 3:

“h, Corresponding population averages ($n=5$ mice). Ramp amplitudes integrated from -1 s to -0.4 s in the aversive (red) and control (gray) zones as a function of time of learning. The amplitude of ramp-like activity increased continuously as a function of learning time in the first 30 minutes, in both the aversive

and control zones. This increase was a non-significant tendency during 0-10 minutes of training (aversive zone: paired t-test, $P=0.177$, Mann-Whitney test $U=0.095$; control zone: paired t-test, $P=0.18$, Mann-Whitney test $U=0.40$) but became significantly higher than pre-learning data after 10-20 minutes training in the aversive zone (paired t-test, $P=0.026$, Mann-Whitney test $U=0.021$), nevertheless, it remained non-significant in the control zone (paired t-test, $P=0.10$, Mann-Whitney test $U=0.21$, one tailed paired t-test, $P=0.051$, one tailed Mann-Whitney test $U=0.105$). After 20-30 minutes of training, ramp-like activity showed a significant increase over pre-learning data in both zones (aversive zone: one-tailed paired t-test, $P=0.02693$, one-tailed Mann-Whitney test $U=0.03$; control zone: one-tailed paired t-test, $P=0.0184$, Mann-Whitney test $U=0.00609$)."

-Fig4f seems to suggest there is significant overlap in the hub cells between control and aversive conditions. Given the differences in averaging ramping activity between the two conditions, how do the authors explain this strong overlap in hub cells?

1.7. We thank the referee for the helpful comment. We show below that although there is a considerable overlap in the spatial clustering of hub cells, the responses of hub cells coding the aversive and control zones reflect the differences in averaging ramping activity between the two conditions with high accuracy.

In **Ref. Fig. 10a**, we present the mean responses of the 10 aversive hub cells and 10 control hub cells described in **Fig. 4f**. When these responses were averaged (**Ref. Fig. 10b**), we obtained ramp-like responses similar to the average ramping activity (compare **Ref. Fig. 10b** to **Ref. Fig. 10d**). Accordingly, the ratio of the amplitude of the hub cells in the aversive zone compared to the control zone was 1.96 ± 0.41 (**Ref. Fig. 10b**), a similar value to the ratio of the amplitude of all cells: 1.77 ± 0.51 , which is also shown in **Fig. 2g** of the previous version of the manuscript (**Ref. Fig. 10d**). The two amplitude ratios were not significantly different (Mann-Whitney test $U=0.0661$).

The kinetics (ramp slope, timing, delay between aver. and ctr.) were also very similar for the average ramping activities and for the hub cells (compare **Ref. Fig. 10b** to **Ref. Fig. 10d**). The similarity between hub cell and whole population responses may be explained by the locally increased connectivity of hub cells as shown in **Fig. 5i,j**. Supporting this, it has been previously shown that hub cells strongly influence the synchronization of spiking activity in a large group of neurons, as their increased connectivity allows them to collect, integrate and thus mimic local network activity²¹. However, decoders also revealed an orthogonal component of the clusters coding the aversive and control zones (see the "**Decoder revealed orthogonality of the overlapping coding assemblies**" section in the manuscript, **Extended Data Fig. 12** of the previous version of the manuscript).

In summary, these data indicate that although there was a considerable overlap in the spatial clustering of hub cells (**Fig. 5f**), the difference in mean responses of hub cells coding the aversive and control zones is similar to differences in averaging ramping activity of the whole population (**Fig. 3f**).

Ref. Fig. 10. Average response of hub cells mimics average ramping activity. **a**, Mean neuronal responses of the 10 aversive and 10 control hub cells shown in Fig. 5f in the aversive and control zones, respectively. **b**, Average responses (mean \pm SEM) of the mean neuronal responses of the 10-10 hub cells coding the aversive and control zone and shown in panel a. **c**, This is Fig. 2h of the previous version of the manuscript with the new statistics. **d**, This panel is taken from Fig. 2g of the earlier version of the manuscript.

We have added Ref. Fig. 10a,b as Extended Data Fig. 17.

In the figure legend we say:

"Extended Data Fig. 17. Average response of hub cells mimics average ramping activity. **a**, Mean neuronal responses of the 10 aversive and 10 control hub cells shown in Fig. 5f in the aversive and control zones, respectively. **b**, Average responses (mean \pm SEM) of the mean neuronal responses of the 10-10 hub cells coding the aversive and control zones and shown in panel a. These data indicate that although there was a considerable overlap in the spatial clustering of hub cells (Fig. 5f), the difference in mean responses of hub cells coding the aversive and control zones is similar to differences in averaging ramping activity of the whole population (Fig. 3f)."

-Regarding the sentence "Therefore, the direction of causality between V1 and motor activity is also changing: V1 activity triggers running during high ramps and, in contrast, running triggers V1 activity during low ramps," given the lack of causal perturbations to neural activity here, the authors might consider rewording this sentence as it is not known whether V1 activity causes running. The same is true for other sentences, such as "In summary, our data indicate that network activity in the V1 during high ramps is not a simple reflection of motor activity but rather represents local computations within V1 that triggers running with a significant delay."

1.8.1 We agree with the referee that the direction of causality has not been confirmed by causal perturbation of the neuronal network. Consequently, we have reworded our statements accordingly.

In the manuscript, we now say:

“These data indicate that the direction of causality between V1 and motor activity may also vary: V1 activity in cooperation with other brain regions may generate ramp-like activity increases and trigger motor cortex and running during high ramps, while running may trigger indirect V1 activity during low ramps.”

“In summary, these data indicate that network activity in the V1 during high ramps is not a simple reflection of motion, as it starts more than 2 s earlier than motion, suggesting that computation that prepares visually-triggered motor activity, may also involve the V1 region during the visual discrimination task. However, further investigation is required to understand the contribution of other cortical areas to local V1 computation and to understand the casual link between visual and motor cortex.”

1.8.2. See also our detailed response in section **3.17**, where we clarify in detail the relationship between running and neuronal responses in eight figures (**Ref. Fig. 35-42**).

-For the sentence “In summary, our data indicated that reinforcement signals can contribute to the formation of long-term potentiation of neuronal activity at the level of individual neurons in the V1 region,” the LTP terminology is typically reserved for synaptic measurements, thus the authors might choose different wording such as “long term changes in activity.”

1.9. Thank you for the helpful comment.

In the manuscript we now say:

“In summary, our data indicate that reinforcement signals can contribute to the formation of long-term activity changes at the level of individual neurons in the V1 region.”

-For the Discussion section titled “Transiently saturating activation ratios maximize computational power,” is there a reference for the math used to calculate computational power? This seems like an oversimplified measure given other implementations of estimating computational power (e.g., Crumiller et al., *Frontiers in Neuroscience* 2011). It also seems to be based on some assumptions about the maximal firing rates of neurons in frontal cortex in acute brain slices, which are likely different from typical V1 firing rates in vivo. Ultimately, this section does not seem necessary, and given the issues mentioned above with overestimating the number of active neurons, is not providing useful information.

1.10. At the request of the referee, we have removed this section from the manuscript.

-What is the lag of the VR system, i.e., the time from data input of mouse running speed vs. stimulus presentation change on the screen? How does the measured lag compare to other existing VR systems? The authors should discuss this.

1.11. The delay of the virtual reality pipeline is measured as photon-to-photon latency based on the method of Matthew Warburton²². A high frame-rate camera (Teladyne FLIR Integrated Imaging Solutions, BFS-U3-17S7M-C) captured the initial signal of the entire pipeline that occurs when the

mouse moves on the treadmill and recorded the subsequent change in the virtual environment displayed on the screens. The differences in elapsed time between animal movement on the treadmill and the corresponding virtual image response was measured in frames and the measurement time of one frame was then calculated by considering the refresh rate (1 s/framerate). The photon-to-photon latency of the entire system was 22 ms. In the field of neuroscience, most published work does not address the latency of VR systems. The few exceptions we found are:

1. FreemoVR, a widespread standard VR solution for freely moving animals developed by John R Stowers et. al.²³ with 60-75 ms of latency,
2. 'The Dome' for freely moving locomotion for rodents produced by Manu S. Madhav et. al.²⁴, where the latency was calculated to be 97 ms.

In addition, as stated by Thurley et al.²⁵, the biological effect of latency in rodent effects is not clear: *"Also the time lag between the actions of the animal and the system's response is crucial to realistic virtual stimulation. This issue has been investigated in humans (see, e.g., Friston and Steed 2014 for an overview²⁶) but to our best knowledge there is no such study examining effects of VR lag on rodent perception and performance. This may be due to the difficulty of fully determining what appears realistic to animals, specifically to rodents²⁵."*

The current value (22 ms) is compliant with the industry standards of state-of-the-art human head-mounted-display virtual reality solutions and the competing solutions in behavior neuroscience. For instance, the Oculus Rift is a reference in human virtual reality research with 20.8 ms photon-to-photon latency, as is Vive with 30.8 ms or Valve Index with 38.5 ms²².

The rendering of the Mocus virtual environment for the latency measurement was performed on binocular 90 Hz OLED displays (2160x1200) by an NVIDIA GeForce GTX1660 GPU with 13th Gen Intel(R) Core(TM) i7 13700K and 16GB DDR4 RAM. Reducing the response time of the visualization will be a part of future work by optimizing rendering with graphical acceleration and parallelization, as well as terminating the bottleneck of temporal data sampling of movement input to improve the immersive experience.

In the manuscript we say:

"The delay of the virtual reality pipeline is measured as photon-to-photon latency based on the method of Matthew Warburton²². A high frame-rate camera (BFS-U3-17S7M-C, Teladyne FLIR Integrated Imaging Solutions) captured the initial signal of the entire pipeline that occurs when the mouse moves on the treadmill and recorded the subsequent change in the virtual environment displayed on the screens. The differences in elapsed time between animal movement on the treadmill and the corresponding virtual image response was measured in frames and then the measurement time of one frame was calculated with refresh rate as 1/framerate. The photon-to-photon latency of the entire Mocus system was 22 ms, which is similar to previously published works: FreemoVR with 60-75 ms latency²³ and dome for freely moving rodents with 97 ms latency²⁴. The current value (22 ms) is also compliant with the industry standards of state-of-the-art human head-mounted-display virtual reality solutions and the competing solutions in behavior neuroscience (for example, the Oculus Rift is a reference in human virtual reality research with 20.8 ms photon-to-photon latency, as is Vive with 30.8 ms or Valve Index with 38.5 ms)²².

The rendering of the Mocus virtual environment for the latency measurement was performed on binocular 90 Hz OLED displays (2160x1200) by an NVIDIA GeForce GTX1660 GPU with 13th Gen Intel(R) Core(TM) i7 13700K and 16GB DDR4 RAM. Reducing the response time of the visualization will be a part of future work by optimizing rendering with graphical acceleration and parallelization, as well as terminating the bottleneck of temporal data sampling of movement input to improve the immersive experience."

-Fig1j: units (cm/s?) on y-axis

1.12. We have calibrated the movement on the treadmill to express the previously used arbitrary units in SI measurement units. Displacement was recorded in Unity environment during 10 full rotations observed with markers on the treadmill. According to the measurement, 1 arbitrary unit corresponds to a physical displacement of 0.0471 cm. The y-axis has been corrected accordingly in **Fig. 1j**, **Fig. 3e, g** and **f**, **Fig. 4a** and **b**, **Fig. 5h**, **Fig. 6l**, **Extended Data Fig. 9**, **Extended Data Fig. 10**, and **Extended Data Fig. 14**.

-Fig2d: the x-axis should be more clearly labeled - the “scale bar” with “cells” is not quantitatively informative. Numbers on the x-axis would be ideal, and number of cells/animals in the figure legend.

1.13. We have modified the figure for clarity (see **Fig. 3d**).

-Fig 2g: red dotted line (before learning aversive response) is missing from the middle panel

1.14. We have corrected **Fig. 2g**.

-The manuscript should be thoroughly edited for clarity. For instance, some sentences could use word choice or grammatical improvements, such as the subheading “Aversive-zone coding takes the encoding from control-zone coding during learning.”

1.15. The manuscript has been proofread by a native English speaker.

Reviewer #2:

Remarks to the Author:

One of the reasons for conflicting results from neurophysiological studies of visual learning with animals may come from the fact that training visual tasks can take rather long so that network mechanisms may already be modified before core experiments are conducted. There is a need for a behavioural protocol for visual tasks that is fast. For research questions involving binocular vision this furthermore requires real stereo images.

To address these shortcoming the authors developed Moccus, a head-mounted VR system for mice. It has an extended visual field of view, and provides stereoscopic vision with depth perception. Experimental tests using this system provided convincing evidence, using the abyss test, that animals experience stereoscopic vision. Furthermore, training was achieved about 100-times faster than using traditional protocols.

One result of this in-depth engineered visual stimulation device was the uncovering of a circuit mechanisms of visual learning not observed before in mice.

The success of this new approach has its foundation in the meticulous engineering of this new instrument based on state-of-the-art optical modelling including measurements of a mouse eye's optical properties. This led to the design of an optimized optical assembly using a combination of a biconvex lens and a diffractive phase shifter. Beyond optics, the authors also made special efforts to optimize the geometry of the device to fit to the head of a mouse, and to correct by software optical distortions of the VR projection over the curved retina of a mouse.

The experimental tests with Moccus in comparison to single or dual monitor VR scenarios convincingly demonstrate that this device allows for a new level of realistic visual VR stimulation.

Thus, it opens new areas of research hitherto thought impossible with head-fixed animals.

Furthermore, the fast training with this device makes it possible that within the less than one hour time span that is normally available as a session duration with a water deprived mouse an entire

learning event can take place and be experimentally recorded. This additional significant achievement will likely trigger a whole new generation of experiments on learning and neuronal plasticity. Not surprisingly, the validation experiments for Moccus already uncovered network mechanisms not observed before.

The highly significant paper is very well written. Changes are not necessary.

2. We would like to thank the referee for her/his very positive comments on our work. We agree with the referee that Moccus “opens new areas of research hitherto thought impossible with head-fixed animals.”

Reviewer #3:

Remarks to the Author:

In this manuscript, Judak and colleagues present a novel head-fixed VR system for mice that enables full visual view stereoscopic stimuli presentation and performs behavioral experiments and details investigation of the neural correlates of visual context avoidance learning in the visual cortex.

The group developed state-of-the-art custom projection optics into the mouse retina to achieve impressive quality visual projection of the microdisplay onto the full retina with minimal distortions. Due to the small size of the projection optics they could provide independent stimulation of each eye in the compact form factor that allows for large imaging objectives over the neocortex. The advantage of the new design of the VR is illustrated with a cliff detection task (in comparison to a dual monitor setup). The rest of the paper is dedicated to the demonstration of the new visual contextual avoidance task, rapid within single session learning in Moccus, and a very detailed analysis of the neural activity in the visual cortex (recorded with 2 photon microscopy) associated with this learning. The authors discover several interesting features of population activity that are correlated with learning and argue that this was only possible due to enhanced properties of Moccus compared to conventional VR. However, the heavy focus of the paper on the neural correlates of the single-task learning in the Moccus setup goes into so many details that it fails to provide a comparison to neural activity during learning in the conventional setup, making it less of a demonstration of the new methodology and more of an investigation of the visual contextual avoidance learning in the visual cortex.

3.1. We thank the referee for the positive comments. Addressing this part of the comment: “that it fails to provide a comparison to neural activity during learning in the conventional setup”, In an earlier set of experiments, we investigated visual discrimination learning using the same training protocol described in the current manuscript but we used a classical monitor system instead of Moccus. Training with the aversive stimulation (mild airpuff) lasted from 3 to 5 days and the velocity increased significantly after learning in the aversive zone before airpuffs compared to the pre-learning data and the control zone (**Ref. Fig. 11a**). The neuronal activation ratio was calculated as the percentage of cells for which the mean calcium signal for all trials during the first three seconds of the visual stimulation was significantly higher than during the three second baseline period prior to visual stimulation. To increase the sensitivity of our measurements, we calculated average baseline neuronal activity over a 3-day period before training. Similarly, after 3-5 days training, we averaged 3 training days to calculate the average effect of learning at the level of individual neurons. Although the mean activation ratio increased after learning in some mice compared to pre-learning data (for example, from 26.19% to 32.74% in the exemplified mouse: **Ref. Fig. 11b-e**), the mean activation ratio was not significantly different at the level of the entire population ($n=6$ mice, before learning: $37.0\pm 14.1\%$, after learning: $36.7\pm 11.8\%$, $\text{mean}\pm\text{SEM}$, $w = 0.6875$, Wilcoxon signed rank test). The low activation ratio found in this earlier set of measurements is consistent with previous studies showing that conditioning of orientation discrimination does not alter the average distribution of orientation preference nor

enhance neuronal responses^{3,13,14}. Furthermore, some studies found a suppression in the activity of neurons tuned for neighboring orientations¹⁴ or for unrewarded stimuli¹⁴. Such discrepancies between studies of visual learning are common and related, for example, to how connectivity between neurons^{27,28}, sparsity of representations, sharpness of the orienting tuning, phases of learning²⁹, stability of cortical representations³⁰, response variability³¹, or spatial distribution of the enhanced network change following learning¹⁴. Part of the inconsistencies in visual learning studies may stem from the long duration of current training protocols. Although neural circuits can support rapid learning³²⁻³⁴, training for several days or even weeks are currently needed to develop reliable visual learning^{3,4,13,35}. During these prolonged learning protocols, several factors may hide and modify the network mechanisms underlying visual learning. First, conditions can change and affect neuronal recordings, including the expression levels of genetically encoded sensors and tissue stability. Second, neuronal networks have prominent internal dynamics and, although cortical representations are relatively stable on shorter time-scales in the V1 region³⁶, cortical representations of sensory information drift and fluctuate over a timescale of days³⁷⁻⁴⁴, which interferes with measurement of the direct effect of learning. Overcoming these challenges was our main motivation to develop Moccus and the related fast learning protocols: to retain learning and associated network effects in one session.

To ensure that the low activation ratio in the earlier set of measurements (**Ref. Fig. 11b-e**) was not due to the low spatial precision of the recording during chronic measurements, we used multiple-cube scanning (**Ref. Fig 12**), performing the scanning at the beginning of each experimental day. During multiple-cube scanning, a $50 \times 50 \times 50 \mu\text{m}$ volume was captured around each cell of interest, coordinates of recording were then realigned according to the three projection images (xy, xz, zy) of the cells using custom-written GUI (**Ref. Fig 12**). This rigorous procedure ensured that the low activation ratio observed (**Ref. Fig. 12**) was not due to difficulties in accurately locating the cells during consecutive experimental days or due to their movement out of the plane of measurement.

Ref. Fig. 11. Neuronal responses before and after learning using a conventional monitor system. The training protocol was the same as that used in Fig. 3. Briefly, the wall of the VR presented on monitor screens was decorated with a neutral gray pattern interrupted with zones of gratings (0° , 45° and 135° orientations) in randomized positions. One of the zones (0° oriented) was associated with an aversive stimulus (airpuff). To exclude passive behavior, teleportation was generated into randomized locations if mice were still for more than 8 s. **a**, Left, average running speed (mean \pm SEM, $n=6$ mice) before (green) and after 3-5 days training (red). Average running speed is also shown in the control zones. Transients were aligned to the end of the zones (0 s) before averaging. Middle and right, mean running speed from $-1,000$ ms to 0 ms before airpuff in the aversive zone was significantly different compared to control zones ($w=0.018$, Wilcoxon signed ranks test, $u=0.03$, Mann Whitney test, $P=0.008$, paired t -test). **b**, Mean neuronal responses from an exemplified mouse during visual stimulation with grating and moving grating stimulation (0° oriented) before learning. Responses for all trials on three consecutive experimental days were averaged and cells were ordered according to their maximum amplitude calculated in the first 3 seconds of the stimulation. The dashed blue line indicates the start of the stimulation with static grating stimulation (from 0 to 1 s) that continued with moving grating

stimulation (from 1 to 7 s, between the two gray lines) and finished with a static grating stimulation (from 7 s to 8 s, between the second gray and red lines). **c**, The top chart shows the mean neuronal activation ratio (the percentage of the significantly activated cells in the first 3-s period with moving gratings) from the exemplified mouse shown in panel **a**, while the lower chart is the average activation ratio for all mice ($n=6$). Light color indicates \pm SEM. **d**, The same as panel **a** but after visual learning. **e**, The same as **b** but after learning.

Ref. Fig. 12. Aligning coordinates of recorded neurons with submicrometer precision during in vivo measurements using multi-cube scanning. **a**, z-stack (500 × 500 × 500 μm) from the V1 region of an exemplified mouse labelled with GCaMP6 sensor. **b**, Each soma was scanned with a 50 × 50 × 50 μm cube at multiple times during consecutive experimental days to realign recording coordinates with submicrometer precision. **c**, Exemplified cube with the three projection images. Maximum intensity projection was calculated for the xy, yz, and xz planes. The red crosshairs represent the 3D location of the chessboard center selected automatically or manually on the first measurement days and realigned using the GUI during the realignment process on the consecutive experimental days. **d**, Multi-cube alignment protocol during an imaging session. The multi-cubes obtained on the first day were used as a reference to align cells over several days (middle panel: second day of experiments, right panel: first day of the effect measurement, after six days of training; before and after the alignment).

We have added **Ref. Fig. 11** as **Extended Data Fig. 19** to the manuscript.

“In contrast to all previous studies with visual discrimination, where the number of neurons responsive to the reinforcement-associated grating was only ca. 8-20%¹⁻³ or did not change significantly^{3,13,14}, we found that most ($80.1 \pm 5.9\%$) of the cells were activated on average ($n=5$ mice) at the end of the aversive zone during the fast-learning protocol with Mocolus (**Fig. 5e**, bottom, $n=5$ mice). Similarly, in contrast to the modest ($<10\%$)^{1-3,12} or missing activation ratio for the control *cue*^{3,13,14}

reported previously, we found that $70.0 \pm 7\%$ of the cells were activated on average in the control zone (Fig. 5e, bottom, $n=5$ mice). Supporting these previous studies with the low or missing activation ratio, we obtained missing activation ratios when our training protocol was repeated with a classical monitor system (Extended Data Fig. 19)."

While a comparison of the behavioral performance and learning dynamics between Mculus and a two-monitor conventional VR is shown, it is not at all clear what in the task used for visual learning is critical for fast learning, as the task relies on orientation grating textures and no specific 3D virtual environment features, or stereo vision, or interaction with the environment or any other aspects that Mculus could be particularly beneficial for.

We thank the referee for the helpful comment. We address this very important question in the following sections:

- 3.2.1.** A detailed explanation of the previous tests and summary of three new tests.
- 3.2.2.** Fast task engagement with true 3D objects (a new set of measurements).
- 3.2.3.** Fast visual discrimination task with true 3D objects (a new set of measurements)
- 3.2.4.** Visual discrimination task with non-grating patterns and floating true 3D objects (a new set of measurements).

3.2.1. A detailed explanation of the previous tests and summary of three new tests.

We agree with the referee that one of the main advantages of the Mculus is the ability to implement 3D objects in virtual reality, creating a more natural environment for rodents. However, as the majority of the in vivo neuroscience laboratories use virtual mazes made up of corridors and walls decorated with different grating patterns, we initially opted to utilize this "gold standard" to validate the advantages of the Mculus. Neuroscientists generally prefer corridors because mice live in tunnels in their natural habitat. In their natural behavior, mice are constantly exploring their tunnel system in search of food, such as various insects that can cross the walls of the corridor. Therefore, it is essential for the navigation and survival of mice to have a true 3D stereoscopic view of the corridor wall with different patterns, even if light levels may limit visibility at certain points in the mouse tunnel system. Consequently, we carried out several tests in our study to confirm that mice actively use 3D stereo vision in the 3D virtual corridor generated by Mculus:

- 1) We introduced a predator bird test (**Suppl. Movie 1** and **Extended Data Fig 5b** of the previous version of the manuscript, **Ref. Fig. 13a,b**). The bird flew in the binocular zone above the corridor to test the vision of mice selected for the experiments.
- 2) An abyss test was used, where the abyss intersected the entire 3D virtual corridor (**Fig 1.g**). Mice stopped at the end of the true stereo 3D virtual corridor, but not at the end of the 2D projected version of the 3D virtual corridor (**Fig. 1j,k**).
- 3) We repeated the abyss experiment with variable corridor lengths, where the length of the floor varied relative to the wall of the corridor, providing an open arm of variable length (**Ref. Fig. 13d**).
- 4) To further test the role of 3D patterns in the corridor, we added a 3D gap at the middle of the closed part of the 3D corridor (**Ref. Fig. 13c**). Mice not only stopped at the gap but moved backwards, similar to **Fig. 1g-k** of the previous version of the manuscript.
- 5) Most importantly, we implemented a visual discrimination task into the 3D corridor (**Fig. 2** of the previous version of the manuscript) using 2D frames with grating patterns oriented in 3D with perspective correction according to the geometry of the 3D virtual corridor. Mice learned the task in

40 minutes in the 3D version of the VR corridor but not in the projected 2D version of the same 3D corridor, although all parameters, for example the size of the zones with the grating pattern, were the same in the 2D and 3D versions from the point of view of the mice. This means that the only difference between the 2D corridor and the 3D corridor was the three-dimensionality of the vision provided by Mculus.

6) According to the referee's request, we performed new measurements with 3D objects to test task engagement (section 3.2.2). We found that task engagement with the new protocol was not only much faster compared to classical monitor systems, but also compared to the previous Mculus measurements with grating patterns on the wall of the 3D corridor, i.e., supporting the opinion of referee #3.

7) In a new series of measurements with 3D objects in the 3D virtual corridor (see the next point: section 3.2.3, Ref. Fig. 14), we achieved fast learning during visual discrimination (within 40 minutes), which is a result similar to that with the previous protocol with 2D grating patterns oriented in the 3D space, but the stability of the behavior was higher.

8) Finally, a new visual discrimination task with non-grating patterns and floating 3D objects resulted in fast learning in about 20 minutes (section 3.2.4).

Ref. Fig. 13. Tests used in our study with specific 3D virtual environment features. a,b The vision of mice was tested with a flying bird at the beginning of the experiments. **c**, Similar to the abyss test (Fig. 1h-k) of the earlier version of the manuscript, but with a 3D gap in the middle of the closed part of the 3D corridor. **d**, The abyss experiment with variable corridor lengths where the length of the floor varied in relation to the corridor wall (this is Extended Data Fig. 6a of the previous version of the manuscript).

3.2.2. Fast task engagement with true 3D objects. A new set of measurements.

“No specific 3D virtual environment features, or stereo vision”. We agree with the referee that other features of the 3D virtual environment, such as 3D objects, can be implemented in VR as more

appropriate components of the rodents' natural environment. Therefore, we repeated our previous visual discrimination protocol, but instead of using 2D frames with grating patterns oriented in 3D with perspective correction according to the geometry of the 3D virtual corridor, we introduced real 3D objects (**Ref. Fig. 14a,b**). V-shaped notches were generated on the wall at both sides of the 3D virtual corridor and 3D cuboids and 3D cylinders with grating patterns were added to mark the reinforcement-associated and control zones (**Ref. Fig. 14a,b**). According to the referee's question **3.14** (see later), we used water reward instead of airpuff as a reinforcement signal to decrease the variance of our measurements. Two reward locations were introduced in the reward zone. The second reward was provided at a fix spatial location: at the end of the reward zone. In contrast, the first reward was only given if mice started licking between the beginning and the center of the zone (**Ref. Fig. 14a**). The second reward was only provided if the animal failed to trigger the first reward, or showed no licking after reward delivery. We recorded position, velocity and licking rate.

In the first set of experiments, similarly to previous works from Pakan et al. 2018⁶, we introduced only the reward and neutral zones to investigate task engagement: visual learning of the reward zone location. In the second separate set of experiments, we also added the control zones to investigate visual discriminative learning (see section **3.2.3**). In the first set of experiments, there was a rapid and stable increase in the licking rate in the reward zone with a small amplitude variance within one day (measured in 10-minute intervals) and across multiple experimental days (**Ref. Fig. 15**). Task engagement formed rapidly and remained stable during consecutive experimental days, measured in 10-minute intervals. Performance was more stable than with the previous protocol using negative reinforcement (see the lower variance compared, for example, to **Extended Data Fig. 9** of the previous version of the manuscript). Moreover, licking rate before the reward zone (anticipatory licking) increased rapidly during the first experimental day (and remained stable the following experimental days **Ref. Fig. 16**). Since anticipatory signals that have previously been demonstrated as indicators of visual learning appeared only after 5-6 days training¹, our data indicate rapid task engagement and fast visual learning of the reward zone location. However, Poort et al. 2015 used both control and reward zones in a visual discriminative learning protocol¹. Therefore, for a more accurate comparison, we need to compare the speed of task engagement with Mocolus with previous studies with a single reward zone, for example to Pakan et al. 2018. In this study, mice learned to locate a reward zone along a linear corridor in 4-6 days, indicating that task engagement with Mocolus using 3D objects in the 3D virtual corridor is about 288 times faster than task engagement with classical monitor systems (task engagement time with 3D objects with Mocolus: 20 minutes, with classical 2D monitors: 4 days (5,760 minutes), ratio: $288 = 5,760 \text{ minutes} / 20 \text{ minutes}$)⁶.

Ref. Fig. 15. Fast task engagement with Moccus using true 3D objects. a, Average licking rate (mean \pm SEM) timed to a visually cued water reward across multiple training days, divided in 10-minute intervals, compared to shuffled data. **b**, Bar graphs showing averaged licking rates before the reward zone (anticipatory licking, gray) and in the reward zone (black). Data from $n=5$ mice, minimum 150 trials/day (mean=205.6, SD=43.41). * indicates significant difference compared to the baseline period (paired t-test, $P<0.05$).

Ref. Fig. 16. Fast task engagement with Moccus using 3D objects is indicated by increased anticipatory licking rate. Related to Ref. Fig. 15. Average licking rate (mean±SEM) timed to a visually cued water reward across multiple training days, divided into 10-minute intervals (red), compared to shuffled data (black). Data from n=5 mice, minimum 150 trials/day (mean= 205.6, SD=43.42).

3.2.3. Visual discrimination task with true 3D objects. New set of measurements.

In this set of experiments, we used 3D objects in the 3D virtual corridor as during the task engagement experiments (section 3.2.2), but control zones were also added to investigate visual discriminative learning (Ref. Fig. 14). Similar to the previous task engagement protocol, licking rate increased rapidly during the first experimental day and showed high stability within one experimental day (measured in 10-minute intervals) and across multiple experimental days (Ref. Fig. 14). The mean licking rate increased even before the beginning of the reward zone after 0-20 minutes training (Ref. Fig. 14c). The anticipatory licking signal was significantly higher before the reward zone than before the control zone calculated between minus 500 ms to 0 ms relative to the water reward (Wilcoxon signed rank test: $w=0.0139$; Mann-Whitney test: $u=0.0237$, Ref. Fig. 14c). At the beginning of learning (0-10 minutes) licking rates were not significantly elevated and licking rates before the reward and control zones were not significantly different. However, in the later phase of learning (10-20 minutes), there was a significant elevation before the reward zone relative to the control zone (Ref. Fig. 14c). Visual learning of 3D objects formed after 20 minutes of training was stable during the following experimental days: the anticipatory signal remained significantly higher before the reward zone than before the control zone (Ref. Fig. 14d; Ref. Fig. 17, $n=5$ mice). In previous studies, mice learnt the visual discrimination task in 3-6 days and anticipatory signals appeared after 5-6 days training¹, showing that visual discriminative learning with Moccus with 3D objects is at least 216-fold faster than with classical monitor systems (Moccus: 20 minutes; classical monitor systems at least 3 experimental days; ratio: $216=4320 \text{ minutes} / 20 \text{ minutes}$).

Ref. Fig. 14. Fast visual discrimination with 3D objects in Moccus. *a*, Schematic of the measurements. Visual discriminative learning with 3D objects. V-shaped notches on the wall on both sides of the 3D virtual corridor were generated for 3D objects with grating patterns that marked the reward and control zones (3D cuboids with oblique grating: reward zone, 3D cylinders with vertical grating: control zone). The second reward location at the end of the reward zone was fixed (late reward). The first reward, located between the center and the beginning of the reward zone, was triggered by licking. *b*, Examples of a reward and a control zone from the viewpoint of the mice. *c*, Bottom, licking rate aligned to the beginning (0 s) of the reward and control (ctrl.) zones after 20 minutes learning from a representative mouse. Top, the same as the bottom panel but confined to the first and second 10-

minute intervals. **d**, Licking rates aligned to the beginning (0 s) of the reward and control (ctrl.) zones after 20 minutes (day 1) and 30 minutes training (day 2-5). Note the stability of the anticipatory licking signal before the beginning of the zones (0 s).

Ref. Fig. 17. Fast discriminative learning with true 3D objects in the 3D virtual corridor using Mocus. **a**, Average licking rate (mean±SEM) timed to the entry into the control and rewarded zones across multiple training days, divided into 10-minute intervals, comparing control (unrewarded, black)

and reward (blue) zones. Data are from the measurements shown in **Ref. Fig. 14. b**, Bar graphs showing averaged anticipatory licking rates before entry into the control (black) and reward (blue) zones. Data from $n=5$ mice, minimum 146 and 150 trials/day in control and reward zones, respectively (mean=161.8 and 168.8, $SD=19.79$ and 15.69 , respectively). * indicate significant difference between the two zones (one tailed t-test, $P_{day1} = 0.032$, $P_{day2} = 0.0067$, $P_{day3} = 1.4 \times 10^{-8}$, $P_{day4} = 0.00038$, $P_{day5} = 1.8 \times 10^{-8}$).

3.2.4. Visual discrimination task with non-grating patterns and floating true 3D objects. New set of measurements.

We investigated whether fast discriminative learning can also be induced using 3D objects with non-grating patterns. We found that patterned objects floating close to the middle bisector plane of the binocular zone in the control and reward zones were able to induce discriminative learning in a short time (in about 20 minutes) and that learning formed this way was stable over the following experimental days (**Ref. Fig. 18**). In addition, induction of learning was faster and learning was more robust than with previous protocols (note the higher difference between licking rates in the control and reward zones in **Ref. Fig. 18** versus **Ref. Fig. 17**).

Ref. Fig. 18. Rapid discriminative learning with floating true 3D objects bearing non-grating patterns.
a, Examples of a reward and a control zone from the viewpoint of mice. The reward zone was cued with

a cube with a dotted pattern, while the control zone contained a cone decorated with stars. Both objects floated close to the middle bisector plane of the binocular zone. Regardless of whether the 3D objects were different, the same reward protocol with two reward locations was used as in **Ref. Fig. 14**: the second reward was provided at a fixed spatial location at the end of the reward zone. In contrast, the first reward was only given if mice started licking between the beginning of the zone and the second reward. The second reward was only provided if the animal failed to trigger the first reward or showed no licking after reward delivery. **b**, Average licking rate (mean±SEM) timed to the entry into the control and rewarded zones (0 s) across multiple training days, divided into 10-minute intervals, comparing control (unrewarded, black) and reward (blue) zones. Only licking data before reward are included. Bar graphs showing average licking rates (mean±SEM) in the reward and control zones before reward. Data from n=6 mice, minimum 281 and 338 trials/day in reward and control zones, respectively (mean=348.75 and 395.75, SD= 49.02 and 61.9, respectively). *: significant difference between the two zones, two-tailed t-test, P<0.05.

In the introduction we now say:

“This provides true depth perception and creates a true 3D illusion of the visual world in a compact design compatible with many in vivo recording systems. We have validated real stereoscopic vision and fast visual immersion using abyss tests and, in combination with 3D acousto-optical measurements^{15,16,45,46}, we have developed visual learning protocols based on 3D virtual corridors with 2D grating patterns and 3D objects with grating and non-grating patterns that are more than 250-fold faster than classical learning protocols.”

We have added:

Ref. Fig. 14 as new figure (**Fig. 4d-g**),

Ref. Fig. 15 as new figure (**Extended Data Fig. 11**),

Ref. Fig. 16 as new figure (**Extended Data Fig. 12**),

Ref. Fig. 17 as new figure (**Extended Data Fig. 13, Fig. 4g**),

Ref. Fig. 18 as new figure (**Extended Data Fig. 8**),

In the manuscript, we now say:

“Use of true 3D objects enhance visual learning

*To validate rapid visual learning with more natural components of the rodents’ natural environment, we introduced true 3D objects in the VR (**Fig. 4d,e**) instead of using the 2D frames with grating patterns oriented in 3D with perspective correction according to the geometry of the 3D virtual corridor (**Fig. 3c**). Two reward locations were introduced in the reward zone: the second reward was provided at a fix spatial location at the end of the reward zone. In contrast, the first reward was only given if mice started licking between the beginning and the center of the reward zone. The second reward was only provided if the animal failed to trigger the first reward or showed no licking after reward delivery. In the first set of experiments, we introduced only the reward and neutral zones to investigate task engagement (as in **Fig. 4d** but without the control zone). Licking rate increased rapidly before the reward zone (defined as anticipatory licking) during the first experimental day and remained stable during the following experimental days, as shown in 10-minute intervals (**Extended Data Fig. 11**,*

for anticipatory licking see: **Extended Data Fig. 12**). Performance was more stable than with the previous protocol using airpuff as a reinforcement signal. Mice learned to locate a reward zone (defined as task engagement) in 20 minutes, which is an about 288-432-fold faster learning speed than with classical monitor systems (Moculus: 20 minutes, 2D monitors: “4-6 days” in Pakan et al, 2018⁶).

In the second set of experiments, reward and control zones both with true 3D objects with grating patterns were added in the 3D virtual corridor to investigate discriminative learning (**Fig. 4d**). The mean licking rate increased before the beginning of the reward zone after just 0-20 minutes training: the anticipatory licking signal was significantly higher before the reward zone than before the control zone, between -500 ms to 0 ms (Wilcoxon signed rank test: $w = 0.0139$; Mann-Whitney test: $u = 0.0237$, $n = 5$ mice, **Fig. 4f**). At the beginning of learning (0-10 minutes) licking rates were not significantly elevated and licking rate before the reward and control zones were not significantly different (**Fig. 4f**). Visual learning formed in 30 minutes training and was maintained (**Fig. 4g**, **Extended Data Fig. 13b**) and stable during the next experimental days, as shown in 10-minute intervals (**Extended Data Fig. 13a**). With classical monitor systems, mice learnt the visual discrimination task in 3-6 days and anticipatory signals appeared after 5-6 days training¹, indicating that visual discriminative learning with Moculus using 3D corridors with true 3D objects is about 144-288-fold faster than with classical monitor systems (Moculus: 30 minutes; classical monitor systems at least 3-6 experimental days; ratio: $216 = 4,320 \text{ minutes} / 30 \text{ minutes}$, $288 = 8,640 \text{ minutes} / 30 \text{ minutes}$).

Finally, we investigated whether fast discriminative learning can also be induced using 3D objects with non-grating patterns. We found that patterned objects floating in the upper middle part in the binocular zone in the control and reward zones were able to induce discriminative learning in about 20 minutes and that the learning formed was stable over the following experimental days, as shown in 10-minute intervals (**Extended Data Fig. 8**). Induction of learning was faster and learning was more robust than with the previous protocols (note the greater difference between the licking rates in the control and reward zones versus the standard error of the means in **Extended Data Fig. 8** compared to **Extended Data Fig. 13**).“

In the discussion, we now say:

“Moculus: immersive VR with 3D vision that mice can understand

The main problem with the currently available and broadly used VR systems in neuroscience is that they use 2D projected view of a virtual corridor and hypothesize that mice, similar to humans, have a high level of abstract visual computation capacity that can reconstruct the entire 3D environment from flat 2D projected images. Contrary to this hypothesis, we demonstrated in this study using different tests: 1) an abyss, 2) an elevated maze, 3) a flying birds, and by experiments 4) with task engagement using 3D objects, 5) with discrimination learning using 3D oriented 2D frames with grating patterns, 6) using 3D objects with grating patterns, and 7) using 3D objects with non-grating patterns, that mice have only a low level of understanding of the different components of the 2D projected images. Moreover, these tests and experiments also validated that adding these more natural components of vision to the VR with Moculus (true 3D corridors, abysses, 3D objects in the middle part of the binocular zone), generated better visual immersion with a more than 200-fold faster task engagement than with classical monitor systems⁶. In addition, the use of true 3D corridors with 2D patterns oriented in 3D and perspective correction to match the geometry of the 3D virtual corridor has improved discriminative learning by a factor of about 144-432 compared to previous studies with 2D screens (Moculus: 30-40 minutes, 2D screens: 3-9 days¹⁻⁵). However, the best performance was achieved when multiple 3D features were simultaneously combined in a natural way: we used a true 3D virtual corridor as above, because mice in their natural habit live in tunnels, but also added 3D objects with more natural patterns than gratings placed to the upper middle part of the binocular zone, because

mice in their natural habitat approach objects from the front. The improved learning speed opens up the possibility of applying complex protocols that currently take more than 60 days⁴⁷."

At the same time, in my view, the richness and complexity of the uncovered neural correlates of the task are so high that even four main figures and a dozen supplementary figures do not provide a complete and coherent story (see comments below).

3.3. Referee #2 comment: *„Furthermore, the fast training with this device makes it possible that within the less than one hour time span that is normally available as a session duration with a water deprived mouse an entire learning event can take place and be experimentally recorded. This additional significant achievement will likely trigger a whole new generation of experiments focusing on learning and neuronal plasticity.”* Indeed, our original goal was to demonstrate that Moccus could catalyze *"a whole new generation of experiments"*. Rather than measuring, analyzing, and discussing only one neurobiological topic in detail, we showed multiple new phenomena using Moccus that raise more questions than answers and make Moccus more attractive for a larger community of researchers. Nevertheless, in response to the referee's comment, we have increased the coherence:

The previous version of the manuscript contained two, weakly connected main neuroscience topics: i) rapid formation of orthogonal neuronal assemblies during discriminative learning, and ii) stochastic vision. We began with the first topic in order to align our study with previously published works, then moved to the second story to explain the stochastic nature of the first topic and then switched back to the first topic. Following the referee's need for clarity, we now keep the first topic in one block and have shifted the second topic with the stochastic nature of vision to the end of the manuscript. Alternatively, the second topic with the stochastic nature of vision could be our starting point and we have tried this order, but without explaining the basic terms on average trials, such as the definition and the ramp-like component associated with learning, fluctuation could not be clearly defined.

Following the referee's advice, we rewrote the last paragraph of the introduction and this now serves as a "table of contents" as it lists and explains the logical order of the topics in the manuscript.

At the end of the introduction, we now say:

"To validate that Moccus and the new behavioral protocols can reveal novel circuit mechanisms, we focused on two main neurobiological topics: i) rapid and parallel formation of reinforcement-associated and control-zone coding neuronal assemblies during visual discrimination, and ii) the stochastic nature of vision. In the first topic, we have demonstrated, during the first phase of learning, that reinforcement and control cues that are simultaneously present in the VR and need to be separated by the V1 region of the cortex during a discrimination task, generate anticipatory signals with ramp-like increase in activity. These anticipatory signals are generated "de novo" independently of orientation tuning responses and, in contrast to previous studies, orchestrate the majority of V1 neurons (70-80%) into growing and partially overlapping spatiotemporal clusters centered around hub cells. These have earlier response onset times, higher amplitudes, and increased functional connectivity, and compete in representing behavior relevant information. Over time, the reinforcement cue-associated cluster dominates neuronal activity exactly when behavior also becomes significant and, in parallel, representations of control cues shrink to the level of sparse activity (low activation ratio). The rapid formation of the clusters is mediated by reinforcement: neurons that have elongated signals at the beginning of learning with a large amplitude reinforcement-associated second component will develop increased ramp-like anticipatory signals at the end of learning. This supports the idea that reinforcement in the early phase of learning correlates with the amplitude of learning at the level of individual neurons in the late phase of learning.

In the second topic, we have demonstrated that vision is a stochastic process: successes and failures alternate from trial-to-trial. This means that the ramp-like anticipatory signals in V1 are generated stochastically from trial-to-trial with a decreasing failure rate. In the case of failures, when anticipatory responses are not generated in the V1 region of the cortex, the direction of communication is reversed, and motion will trigger an indirect signal in the V1 region with short delay and low jitter. Vice versa, during successes, ramp-like neuronal activity increases with high amplitude are followed by running with a significant delay, thus indicating that anticipatory ramp-like signals represent local computation in the V1 that contributes to the generation of motion.”

Overall, as the comments below show, I am not seeing the fast learning of the new task as the best use-case or validator of the advantages of Mculus. Having only visual cliff paradigm left as a VR-features benefiting behavior, I am not seeing a very strong case for the method and its advantages. Clearly, stereo-VR could be very beneficial, small form-factor as well, but they are not demonstrated beyond the cliff experiment.

3.4. Most in vivo rodent research laboratories use some version of virtual reality systems with corridors. Neuroscientists may prefer corridors because mice live in tunnels in their natural habitat (see previous point). Until now, however, VR corridors have been realized as 2D projections on screens. The biggest advantage of Mculus is that, for the first time, it offers true 3D stereo vision for these 3D VR corridors.

Many laboratories use a 3D virtual maze consisting of corridors and walls decorated with different grating patterns, but these VR mazes are typically represented as 2D projections. This means that regardless of the geometry of the projection surface, whether it is a flat screen or a curved dome system, the three-dimensional space of the virtual reality is compressed into two dimensions and shown on a flat or curved 2D surface (**Ref. Fig. 19**).

Ref. Fig. 19. Difference between classical VR applications and Mocus. This is **Extended Data Fig. 8.** from the previous version of the manuscript. **a**, All classical VR applications with two or more displays or with a dome generate only a 2D projected image, regardless of the geometry of the surface of the device: flat or curved. **b**, In contrast, with Mocus the VR is projected at two view angles and the two projection images are then shown on the two displays to generate true stereo vision.

According to our hypothesis, humans can easily understand these 2D images unlike mice (**Ref. Fig. 20**). Briefly, when looking at a 2D scene in virtual reality (top image in **Ref. Fig. 20**), the human brain can quickly detect lines, calculate angles, fit surfaces, etc. In other words, the human brain performs complex visual calculations that generate a depth-encoded 3D model of the surrounding environment in which the coordinates of various objects (**Ref. Fig. 20**) are also estimated⁴⁸. During this visual computation, the brain has to use advanced mathematics⁴⁹, even in simplified form, to calculate various features and parameters of the 3D space. According to our hypothesis, the brains of mice do not have this advanced computational capacity and cannot compute 3D space from 2D scenes. Although mice certainly see lines, patterns and contrast, as has been validated in multiple behavioral experiments, they transform 2D scenes into a flat, two-dimensional mosaic of patterns, as illustrated in **Ref. Fig. 20**, but 3D abstract computation is missing. This means that immersion, the understanding of the 3D space as a 3D corridor decorated with different patterns that mice are indeed moving along, is also missing. To test this hypothesis, we performed eight different experiments (see section **3.2**). From the eight tests, the advantage of the abyss tests are that they are based on the natural behavior of animals: the readout of the test is immediate and they work on naïve, untrained animals. We found that unlike monitor systems, mice stopped immediately at the edge of the abyss. Moreover, mice started to move backwards, indicating that the fear of depth emerged immediately: note the negative speed at the edge of the abyss (**Ref. Fig. 21b**). In contrast, in the case of standard monitor systems,

there was a significant increase in speed at the edge of the abyss in most cases (see the peak in average speed at **Ref. Fig. 21a** and **b**), which may reflect the effect of the changing pattern/contrast on the 2D screen at the edge of the abyss. These data indicate that perception of 3D space, and distance calculation at a certain level, was formed immediately in Mculus with stereo display. The abyss test in **Fig. 1** of the previous version of the manuscript tested the 3D vision of both the floor and walls. The version of the abyss test in **Ref. Fig. 13c** examined the 3D vision of the floor and, finally, the bird image tested the “ceiling” of the 3D corridor. In summary, we carefully determined that the stereovision of Mculus was required for the 3D vision of the corridor.

We do not know what mice understand from the projected 2D scenes (**Ref. Fig. 20**), but according to our measurements understanding the 3D nature of the environment is not among the features that are understood. Therefore, immersion in the 3D virtual world does not occur with 2D screens. This may explain why mice learn to discriminate between two grating patterns rapidly under freely moving conditions [**Fig. 1b** in reference #4, **Ref. Fig. 26**], but it takes on average several days or even weeks [see e.g. **Fig. 3a** in reference #4] for the same level of visual learning with standard 2D VR systems^{3,4,13,50}. We have demonstrated that visual learning can occur within 40 minutes when using Mculus (**Fig. 2** of the previous version of the manuscript), reaching and even exceeding the learning speed of freely moving mice. We have developed three further behavior protocols and conducted experiments that further validate the need for 3D vision in fast visual learning:

- fast task engagement with true 3D objects (section **3.2.2**),
- fast visual discrimination task with true 3D objects (section **3.2.3**)
- visual discrimination task with non-grating patterns and floating true 3D objects (section **3.2.4**).

In summary, the main advantage of the Mculus is that it provides a true 3D representation of 3D virtual corridors and objects that can actually be seen by mice and can therefore serve as a more natural environment for neurobiological experiments.

**DECODING THE SCENES FROM FLAT OR CURVED
2D SCREENS OF VIRTUAL CORRIDORS**

detect lines and angles
calculate linear perspective (parallel projection)
generate a 3D space (3D coordinate system)
addig estimated positions of objectes

generate a z-coordinate encoded abstract 3D space

adding estimated positions of objects to the space

detect features
and
patches in 2D

pattern #1 pattern #2 pattern #3

pattern #4

no 3D calculation is
performed
3D abstract representation
of the space is not
generated

Ref. Fig. 20. Hypothetical differences between the visual computation of humans and mice while observing 2D scenes of a 3D virtual world. According to our hypothesis, the human brain detects different features (e.g., lines, surfaces, angles) of 2D images and calculates a perspective-corrected 3D representation of the environment in which the position of different objects is estimated. However, mice do not have such effective visual computational ability and the original 2D images are transformed

into a mosaic of 2D patterns (pattern #1, #2...) that fails to reveal the true nature of the surrounding 3D space and locate the objects in that space.

Ref. Fig. 21. Average velocity in abyss tests. *a*, Average velocity for Mocus (red, $n=7$ trials) and dual monitor (blue, $n=10$ trials) systems for an example mouse around the edge (at 0 s) of the abyss. In the absence of immersion and depth perception (i.e., when using the classical monitor system), mice ran across the gap of the virtual abyss. *b*, Abyss experiment with variable corridor length. (This is Fig. 1j and Extended Data Fig. 6d of the previous version of the manuscript.)

That being said, I am highly impressed by the quality of the methodological side of the work and care for details in optics design, mechanics design, distortions corrections etc. If not for the specific method at hand, but for the rigorous approach to the method development this paper is a prime example for the field.

3.5. We sincerely thank the referee for the positive comment on our work. Following the referees' and editor's advice, we have adjusted the balance of the manuscript by transferring the valued technical details from Supplementary Information to the main figures and texts and created 1.5 main figures from the methodological side. To compensate, neurobiological results have been moved to the Supplementary Information.

We have added Fig. 2. and Fig. 4d-g.

I can't judge whether opto-acoustic 2 photon scanning is completely novel and can be featured as part of the methodological toolbox, but it is certainly providing high-quality population data. A much more focused on the demonstration of the beneficial features of the Mocus would make a very strong methods paper. However, the major portion of the paper goes in a very different direction and is much more suitable for the research article.

3.6. We thank the referee for her/his positive comment on the high-quality population data. The advantage of the improved 3D AO scanning is shown in the **Supplementary Table 1**. The description of the 3D AO technology was already limited in the original main text. In response to the reviewer's suggestion, we further reduced the 3D AO section by moving the corresponding text blocks to the supplementary material and added an additional figure on Mocus validation and calibration to the main figures to make the manuscript more Mocus-centric.

The edge of abyss experiment demonstrating the immersive properties of the Mculus for the naive mice is working impressively well.

3.7. We thank for the referee for appreciating our experiment.

It would be useful to know whether further training and habituation of the mice in the single and double monitor setups lead to improved performance.

3.8. As the main objective of the abyss measurements was to demonstrate the intrinsic behavior of naïve, untrained animals, we did not carry out multiday measurement with this behavior protocol. We show in **Fig. 2f** and **Extended Data Figure 9** of the previous version of the manuscript that further training with a classical monitor system also led to improved performance. To better quantify the improvement, we added **Extended Data Fig. 19a** showing good behavioral performance with classical monitor systems when 3 consecutive training days were averaged after 3-5 training days. This means that after several days training with classical monitor systems, the ramp-like running speed increase at the end of the aversive zone relative to the control zone and to the before learning data was similar to the ramp-like increases obtained with Mculus after only 40 minutes training (compare **Extended Data Fig. 19a** to **Fig. 3e**). We also enlarged Fig 2f, which is now shown as **Fig. 3g**.

See also point **3.14.1**.

To be fair, it is not clear if these setups represent the state-of-the-art head-fixed VR, as dome-based systems present less inconsistent and distracting visual features such as edges of the screens etc, and have larger visual fields of view.

3.9. We agree with the referee's that the field-of-view of the dome systems is very large and reaches the total field-of-view of mice. However, Mculus has some advantages over dome systems:

- 1) Dome systems are usually combined with reward systems (**Ref. Fig. 22**) that either have distracting static objects, such as a water tube, or use a motorized arm for the water tube that is not perfectly synchronized with motion and therefore generates visual interference. Because the components of the reward system must be placed in the binocular zone, directly in front of the mouse, they cause visual interference with the projected images. Many other devices, such as patch clamp systems, can also partially block vision. Mculus, on the other hand, completely hides the reward systems and other laboratory instruments. Moreover, Mculus opens an easily accessible free space in front of the mice for other devices.

a

b

Redacted

- 2) Most laboratory-made imaging and recording systems have complex dimensions that are difficult to fit to the large projector screens of the dome systems. In contrast, Moccus, can be easily fitted to many existing in vivo setups.
- 3) Finally, true 3D stereo vision, the main advantage of Moccus, is discussed in detail in section 3.4. Briefly, although domes can cover the entire field-of-view of mice, the surface of the dome's screen is just 2D. This means that the 3D virtual world is projected in two dimensions and shown on the curved 2D surface of the dome (Ref. Fig. 19, curved surfaces can be parameterized with only two parameters and are therefore defined two-dimensionally in mathematics). According to our hypotheses and measurements (see section 3.2.1), mice cannot reconstruct the 3D space from 2D information (Ref. Fig. 20) and may simply understand the projected 3D movie in the dome as a mosaic of 2D patterns with different features and distribution; mice are not able to see the 3D virtual corridor. This may explain why visual learning in the 3D corridor is much faster with Moccus (~40 minutes) than with standard VR systems (typically 5-9 days)^{3,4,13,35}. To support the advantages of stereovision, we conducted 3 different tests with an abyss, one with a flying bird, one with 3D-oriented 2D patches, and three with 3D objects, where stereovision was tested on the following components of the 3D virtual corridor:

- *ceiling* in the flying bird test (Ref. Fig. 13a,b)
- *ceiling* in the discrimination test with floating 3D objects and non-grating patterns (Ref. Fig. 18)
- *wall* in the task-engagement test with 3D objects (Ref. Fig. 15, 16, section 3.2.1)
- *wall* in the discrimination test with 3D objects (Ref. Fig. 14, 17, section 3.2.3)
- *wall* in the classical discrimination test with 3D oriented 2D frames with grating patterns (Fig. 3)
- *walls* versus *floor* in the elevated maze test (Ref. Fig. 13d)
- *floor* in the pit test (Ref. Fig. 13c)
- *floor and walls* in the abyss test (Fig. 1).

Our data demonstrate that the 3D space was understood by mice from 2D images only when perspective-corrected images were displayed in a stereoscopic manner into the two eyes.

It is worth noting that depth perception is also natively emerging in the freely-moving VR systems (Stowers et al 2017, Del Grosso et al 2017), thus Moccus appears to be comparable in immersive power with those even in the absence of head movements.

3.10. We agree with the referee that Moccus appears to be comparable in immersive power with freely moving VR systems, even in the absence of head movements. However, the benefits of head-restraining have also been demonstrated: it has been shown in monkeys that voluntary head fixation promotes more efficient vision-learning experiments than freely moving conditions⁵¹.

In addition, we have demonstrated in two sets of experiments that Moccus can enable faster visual learning than freely moving conditions. It was shown previously that at least 4 training sessions (4 days) are required for freely moving mice to learn to discriminate two different orientations⁴, see Ref. Fig. 26. However, when a similar visual discrimination experiment was performed with 3D objects and similar grating patterns, discrimination required only 40 minutes, i.e., a single session (section 3.2.3, Ref. Fig. 14, 17), indicating that learning with Moccus was 144-fold faster than freely moving conditions (freely moving: 5760 minutes, Moccus: 40 minutes). In a second set of experiments, we

also demonstrated that visual discrimination with floating 3D objects bearing non-grating patterns required only 20 minutes (section 3.2.4, **Ref. Fig. 17**).

These data indicate that head restraining may accelerate some learning protocols, allowing learning and exploration of the underlying network mechanisms in a single session and eliminating multiple, time-dependent artifacts.

Thus while authors do show good performance in their setup, comparison to a very particular and likely suboptimal conventional setup doesn't make a strong case for Mocus as a game-changer.

3.11.1. Fast learning. One of the main advantages of Mocus is that it results in an about 144-432-fold faster learning speed than 2D screens. Although our conventional monitor system used in the comparison may be suboptimal, the learning rate achieved is in line with the literature: several previous studies similar to our work have shown that reliable visual learning with monitor systems usually requires training periods of several days or even weeks^{3,4,13,35}. More specifically, a systematic literature review of 75 publications on visual learning was conducted to determine the number of trials and/or training sessions necessary to reach criterion performance during vision-based tasks [see Supplemental Systematic Literature Review in reference⁵]. Only 18 of the 75 publications correctly reported the training speed and the number of training sessions quantitatively (**Ref. Fig. 23**). The number of training sessions (days) required to reach consistent performance ranged from 4 to 50 days (**Ref. Fig. 23**). Therefore, we agree with the referee that although our control protocol with monitor system was suboptimal, it was still close to the previously published protocols with the best performance as it resulted in significant learning in 5 days (**Fig. 2f** of the previous version of the manuscript, **Extended Data Fig. 19a**). In contrast, Mocus led to significant learning within 40 minutes with the same mouse line and the same recording conditions as the classical monitor systems. Thus, instead of the 7,200 minutes (5 days), only 40 minutes with Mocus achieved discriminative learning. However, following the referee's comment, we now compare the 40 minutes learning time with Mocus not to our measurement but to the lowest session number shown in **Ref. Fig. 23**, which was on average 4 days⁵. In the manuscript we now write that Mocus provides an about 144-fold faster learning speed. We have also added other studies (see below) with 7-9 training days and our fastest protocols with 30 minutes learning, resulting in an increase in learning speed of 144-432 compared to classical 2D screens.

Redacted

In the manuscript we now say:

“Contrary to this hypothesis, we demonstrated in this study using different tests: 1) an abyss, 2) an elevated maze, 3) a flying birds, and by experiments 4) with task engagement using 3D objects, 5) with discrimination learning using 3D oriented 2D frames with grating patterns, 6) using 3D objects with grating patterns, and 7) using 3D objects with non-grating patterns, that mice have only a low level of understanding of the different components of the 2D projected images. Moreover, these tests and experiments also validated that adding these more natural components of vision to the VR with Mculus (true 3D corridors, abysses, 3D objects in the middle part of the binocular zone), generated better visual immersion with a more than 200-fold faster task engagement than with classical monitor systems⁶. In addition, the use of true 3D corridors with 2D patterns oriented in 3D and perspective correction to match the geometry of the 3D virtual corridor has improved discriminative learning by a factor of about 144-432 compared to previous studies with 2D screens (Mculus: 30-40 minutes, 2D screens: 3-9 days¹⁻⁵). However, the best performance was achieved when multiple 3D features were simultaneously combined in a natural way: we used a true 3D virtual corridor as above, because mice in their natural habitat live in tunnels, but also added 3D objects with more natural patterns than gratings placed to the upper middle part of the binocular zone, because mice in their natural habitat approach objects from the front. The improved learning speed opens up the possibility of applying complex protocols that currently take more than 60 days⁴⁷”

3.11.2. Stereo vision and attention. The second main advantage of Mculus is that it offers natural vision by providing natural 3D visual stimulation through showing two different, perspective-corrected images that are rendered separately for the left and right displays, rather than showing the same 2D projected image for the left and right eye in the binocular zone, as in dome and classical monitor systems. While humans have a high visual abstract computational capacity and can easily switch between 2D screens and 3D movie presentations, this is more challenging for other animals (**Ref. Fig. 20**). For example, dogs have a relatively high level of attention in the real 3D world but it is difficult to get their attention on two-dimensional TV screens [*“A dog centred approach to the analysis of dogs’ interactions with media on TV screens”*⁵²]. This study revealed a considerable fraction of time when the dogs were not looking at the screens. Moreover, attention periods were very short (**Ref. Fig. 24**), suggesting that full immersion in the virtual world presented on the two-dimensional TV screens was not established. Since the visual computational capacity of mice is even less developed than that of dogs, we can expect an even lower attention level for two-dimensional TV screens.

The higher attention level may explain the faster learning speed with Mculus compared to freely moving conditions. It was shown previously that at least 4 training sessions (4 days) are required for freely moving mice to learn to discriminate two different orientations⁴. However, when a similar visual discrimination experiment was performed with 3D objects and similar grating patterns, discrimination required only 40 minutes, i.e., a single session (section **3.2.3**, **Ref. Fig. 14, 17**), indicating that learning with Mculus is 144-fold faster than in freely moving conditions (freely moving: 5,760 minutes, Mculus: 40 minutes, see section **3.10**).

Redacted

However, it is certainly a very good and economical alternative to the existing dome-like systems.

3.12. We thank the referee for the positive comment on our work. In the manuscript, we now highlight the economic benefit of Moccus. Indeed, in discussions with several PIs about the integration times and development times of customized and laboratory-made dome VR systems, some reported that it took several months or more than a year to adopt custom-developed dome-shaped VR systems to complex, large, lab-made recording systems. The compact size of Moccus allows faster integration.

In contrast to freely-moving VR the improved visual stimulation of the Moccus under head-fixation does not provide a sense of full spatial understanding of the virtual space. A greater level of immersion into the VR should be coupled with a better spatial understanding of the environment, which is not demonstrated here. It is well-accepted in the field of spatial cognition that it is the congruence between idiothetic and allothetic cues that enables coherent spatial representation and, ultimately, understanding of space. Rats (Aronov et al Neuron 2014) and mice (Chen et al eLife 2018) in the body/head-fixed VR setup only obtain 2D spatial representations (place, grid and head-direction tuning) when their head is allowed to rotate. In fact, this points out that idiothetic vestibular cues are critical for immersiveness. Linear translational idiothetic cues are lacking under head fixation and, hence, full immersion is not possible. Strong response to an abyss, while impressive, doesn't recapitulate full immersion in the VR space in general.

3.13.1 We agree with the referee that although Moccus improves visual immersion, much sensory information, such as idiothetic vestibular cues or translational and rotational idiothetic cues, is still missing. Therefore, we have changed the phrase "full immersion" to "full visual immersion" throughout the manuscript to emphasize the difference, although the phrase "full immersion" is widely accepted in the field of virtual reality research and industry without considering idiothetic cues.

In addition, in the discussion section we now say:

"Although Moccus provides full visual immersion, idiothetic cues and congruence between idiothetic and allothetic cues that enables coherent spatial representation are still missing in the case of head-restrained mice. Nevertheless, Moccus enables an intricate systematic manipulation of the 3D visual environment and, in combination with fast 3D AO imaging, showed that visual learning is possible in rodents on the timescale of tens of minutes in agreement with real-life situations – contrary to current dogma. Therefore, learning and the associated neuronal mechanisms can be investigated

within one measurement session, excluding artefacts such as day-to-day rearrangement in network activity, or changes in sensor concentration and neuronal viability.”

It is also worth noting that fast visual learning is also possible without idiothetic vestibular cues. In addition, head fixation could be an advantage in visual learning: it has been shown in monkeys that voluntary head fixation enables more efficient vision learning of complex tasks than freely moving conditions⁵¹. Moreover, Mculus provided faster reinforcement learning than freely moving conditions (see sections **3.10** and **3.13.3**).

3.13.2 Referee’s comment: *“...A greater level of immersion into the VR should be coupled with a better spatial understanding of the environment, which is not demonstrated here...”* We carried out several experiments and tests to confirm that Mculus provides a higher level of immersion associated with better spatial understanding of environmental components: we conducted 3 different tests with an abyss, one with a flying bird, and three with 3D objects in 3D virtual corridors. The following series of experiments were conducted:

- 1) In an abyss test, shown in **Fig. 1** of the previous version of the manuscript, we simultaneously tested the improved spatial understanding of two environmental components: the floor and wall, by intersecting the entire 3D virtual corridor.
- 2) We repeated the abyss experiments with variable corridor lengths. In this way, we performed an elevated maze test in which the 3D stereo visibilities of the walls and floor were tested separately. Mice stopped at the edge of the floor as in the original abyss test (**Fig. 1** of the previous version of the manuscript) but did not reduce their speed at the end of the wall, in close agreement with previous elevated maze tests (**Ref. Fig. 25**). Moreover, speed increased at the end of the wall (**Ref. Fig. 25d**).
- 3) To test for a better spatial understanding of the environment in the upper part of the 3D virtual corridor, we used a flying prayer bird (**Suppl. Movie 1** and **Extended Data Fig 5b** of the previous version of the manuscript, **Ref. Fig. 13a,b**). The bird flying in the binocular zone above the 3D corridor generated an immediate escape behavior.
- 4) In a set of experiments, floor stereovision alone was tested by introducing a 3D pit in the center of the 3D corridor (**Ref. Fig. 13c**).
- 5) In the original visual discrimination task described in the previous version of the manuscript (**Fig. 3**), we tested again the walls as an environment component: whether mice can see and understand different 2D frames with grating patterns oriented in 3D on the wall according to the perspective corrected view of the 3D corridor.
- 6) In a new set of measurements, we tested stereo vision of the wall with 3D objects (section **3.2.2**, **Ref. Fig. 15, 16**). We found that task engagement with the new protocol was not only much faster and more stable than classical monitor systems, but also compared to the previous Mculus measurements with grating patterns on the walls of the 3D corridor. These data indicate that more naturalistic objects are indeed more effective for visual learning experiments than simple 2D frames placed oriented in 3D.
- 7) In a new set of measurements using 3D objects in the visual discrimination protocol (see section **3.2.3**, **Ref. Fig. 14, 17**), we tested the stereovision of the wall. Similar to the previous protocol with 2D grating patterns oriented in the 3D space, fast learning was achieved in the 3D virtual corridor within 40 minutes, but with a higher stability.
- 8) Finally, in a new set of measurements, we investigated the upper part of the binocular zone for stereo vision with a discriminative learning test using 3D objects with non-grating patterns (section **3.2.4**). We found that patterned objects floating close to the middle bisector plane of the binocular zone in the control and reward zones were able to induce discriminative learning in a short time (in about 20 minutes) and that the learning was stable over the following experimental days (**Ref. Fig. 18**).

See also section 3.9.

Ref. Fig. 25. Abyss experiment with variable corridor length. *a-d*, Same as the experiment in Fig. 1 but the length of the floor varied relative to the wall of the corridor, providing an open arm of variable length. *a*, Schematic of the measurement with two examples: i) the floor was the same length as the walls (top), like the experiments shown in Fig. 1; ii) the floor was elongated (bottom). *b*, Corresponding running curves from an example mouse plotted as a function of time. Running speed typically dropped to zero with a negative peak and remained zero or negative after mice had reached the edge of the abyss with normal floor length (red). (Different shades of red are used because of figure visibility issues.) In contrast, there was typically no sudden speed drop when the floors were elongated (blue). *c*, Left, average running speed (mean±SEM) dropped to zero when the mice reached the edge with the wall and corridor of the same length (red, $n=11$ mice), but not when the corridor floor was extended (blue, $n=7$ mice), indicating that mice stopped because of the fear of depth and not because of the change in the contrast or pattern. Right, corresponding average running speeds before (BEF) and after (AFT) the abyss calculated in 0.5-0.5 s intervals for the abyss with the normal (red) and the elongated (blue) floor ($P = 0.0186$ for the normal and $P = 0.813$ for the elongated floor, Wilcoxon signed rank test). *d*, Same as panel *c*, for the mouse whose individual transients are presented in panel *b*. Running speeds before and after the abyss were significantly different with the normal but not with the elongated floor ($P = 0.002$ vs. $P = 0.232$, Wilcoxon signed rank test). (These figure panels are from **Extended Data Fig. 6** of the previous version of the manuscript).

In the Methods we now say:

“To support the need for stereovision, we conducted 3 different tests with an abyss, one with a flying bird, one with 3D oriented 2D patches, and three with 3D objects, where stereovision was tested for the following components of the 3D virtual corridor:

- ceiling in the flying bird test (**Extended Data Fig. 5**),
- ceiling in the discrimination test with floating 3D objects and non-grating patterns (**Ext. Data Fig. 8**),
- wall in the task-engagement test with 3D objects (**Extended Data Fig. 11, 12**),
- wall in the discrimination test with 3D objects (**Fig. 4d-g, Extended Data Fig. 13**)
- wall in the classical discrimination test with 3D oriented 2D frames with grating patterns (**Fig. 3**).
- walls versus floor in the elevated maze test (**Extended Data Fig. 9a-d**),
- floor and walls in the abyss test (**Fig. 1**)”

3.13.3. Mculus can provide faster learning than freely moving conditions (see also section 3.10)

It has been shown that at least 4 training sessions (4 days) are required for freely moving mice to learn to discriminate two different orientations (**Ref. Fig. 26**). However, head-restrained animals took longer to learn the task (9–25 sessions were required for head-fixed and 4-6 sessions for freely moving mice)⁴.

a

b

c

Redacted

We performed a similar visual discrimination experiment with water reward and found that discrimination between two different orientations with Mculus requires only 40 minutes, a single session (section 3.2.3, **Ref. Fig. 14, 17**), indicating that learning with Mculus was 144-fold faster than freely moving conditions (freely moving: 5,760 minutes, Mculus: 40 minutes). When reinforcement learning was based on punishment, as in the original visual discrimination experiment with Mculus, we observed a similar fast learning time (**Fig. 2** of the previous version of the manuscript). This faster learning speed with Mculus can be partially explained with the immersive nature of VR devices, which increase attention to the specific task. Supporting this, immersive 3D VR systems have been shown to accelerate learning even in humans^{53,54} (**Ref. Fig. 27**). They are widely used for educational purposes to enhance learning⁵⁵⁻⁵⁸, although some studies have shown only a near-significant trend in the correlation between immersion and learning⁵⁹.

Redacted

Faster learning of the contextual avoidance in the Mocus system compared to the conventional Monitor-based setup is a nice added benefit of the system. It would be nice to show how the behavioral performance grows for both mouse groups within a day across 40 min of training as well as across all days, not just day 1 and day 5, as current figure 2f shows and more than Ext Fig9g shows. It would be important to demonstrate that learning performance is staying high for the Mocus group and when it catches up in the Monitor group. Showing in which 10-minute training block each mouse is reaching the threshold of good performance would lead to more nuanced statistics than average over-time performance across all mice in a group. Ext Fig9g seems to show that the performance of the Mocus group mice is not very stable and they do not stay at a good performance on day 2 or even later. In fact, only 4 out of 6 mice reached high performance on day 5, and only 2 on day 2. This is in contrast to the shaded confidence plot that Fig2f is showing. At the same time, the Monitor group seems to be showing non-random below-chance performance. Both this unexplained “significant” bias in behavior (freezing in the aversive zone?) and very small group size (3 mice) prevent, in my view, any possibility for statistical comparison between the groups. Showing behavioral and neural responses for the Monitor mice (as Fig2e,g) would allow the reader to assess the type of responses that this group has and group statistics (as f,h) would be useful as well. It is not clear from the current exposition how and to what extent the neural engagement of V1 in the Monitor group is different even if they do not learn as fast. Finally, as Fig3 exposes, the behavior is highly variable across trials even during periods of apparently high performance on average (authors refer to stochasticity of visual perception, which I don’t find justified until other factors are ruled out). It would thus be necessary to expose if and why sensory experience in Mocus is different from conventional VR, what this variability is related to and if it contributes to the trial-average performance assessment.

3.14.1. Behavioral performance stability with negative reinforcers

We thank the reviewer for highlighting the importance of longer-term stability in addition to the speed of learning. We have investigated the temporal stability of the visual discrimination task, as detailed in the original version of manuscript, for both the Mocus and classical monitor systems (**Extended Data Fig. 9** of the previous version of the manuscript). Our most significant observation is that the fluctuation highlighted by the referee is related to our behavioral protocol rather than the device itself. We chose an aversive reinforcement protocol expecting faster visual learning. However, while in most cases the mice learned to escape from the aversive stimulus by increasing their running speed, in some cases they reacted with a contrary behavior form after a certain day: running speed dropped to zero in the aversive zone or decreased significantly compared to the control zone in a longer time period, and signs of freezing were detected by a camera. The switch between escape strategy and freezing can be explored by dividing the data in 10-minute training blocks following the referee’s question. **Ref. Fig. 28** show an exemplified mouse with stable behavioral responses in the 10-minute training blocks: when teleported to the aversive zone, mouse increased running speed to avoid airpuff stimuli, while a second exemplified mice (**Ref. Fig. 28**) showed an increasing freezing reaction in the last 10-minute block of measurements during the third day. This resulted in an opposite signal: a significant decrease in running speed compared to the control zone that persisted in the next experimental days. As the mice were observed with a camera, the freezing response was easily detected and the amplitude of the airpuff reduced to provide a softer aversive stimulus. Despite all these methodical efforts, fluctuation in behavior remained high when negative reinforcers were used both with Mocus and classical monitor systems (**Extended Data Fig. 9** of the previous version of the manuscript). To better represent the behavior change in **Ref. Fig. 29**, the distances between aversive and control responses (and shuffled curves) were calculated as differences between the average curves (in the interval -2 and -0.1 s relative

to the end of the zone). This preserved the direction of the difference, since the dynamic time wrapping (dtw) as originally used in the main manuscript is not sensitive to the direction of the difference. Despite the aforementioned issues with the behavioral protocol and negative reinforcement, training with Moccus still resulted in a faster learning speed training with a classical monitor system. However, we could not improve the stability of the measurements even with more precise airflow control. Therefore, we switched to positive reinforcement to increase stability of the measurements (see next section, and Ref. Fig. 17 and Ref. Fig. 18 for further details).

Ref. Fig. 28. Stability of behavioral responses during visual learning with a negative reinforcement (airpuff). An exemplified mouse with stable behavioral responses. The 1st, 2nd, and 3rd 10-minute periods of training during the first, third and fifth days. During the first training day, mouse learned to leave the aversive zone faster than the control zone. This behavioral performance remained stable during the remaining training days, of which only two are shown. The first column (i) shows average velocity (mean±SEM) in the control (blue) and aversive (red) zones. Transients were aligned to the time when the mouse left the given zone or received the first airpuff. The second column (ii) shows the average velocity calculated for the two zones between -1.5 and 0 s. The third column (iii) displays the results of bootstrapping analysis for the velocity difference. The histogram shows velocity difference calculated on the shuffled data, while the red column represents the difference measured between the aversive and the control zones. The P-values are calculated as the ratio of shuffled differences (black dashed lines). Insets (iv) present the (1-P) values from column iii for the three time periods.

Ref. Fig. 29. Stability of behavioral responses during visual learning with negative reinforcement (airpuff). Similar to Ref. Fig. 28 but with a mouse with unstable behavioral responses. Data representation is organized similarly to Ref. Fig. 28. By the third time period of the first training day, the mouse learned to discriminate between the aversive and control zones and increased its running (escape) speed in the aversive zone. However, during the third training day, the animal began to exhibit freezing behavior in the aversive zone, resulting in a decrease in running speed in that zone. This freezing behavior remained stable for the rest of the training.

3.14.1. Positive reinforcement increases response stability in a new set of measurements.

Although Moccus produced a faster learning speed compared to training with a classical monitor system (Fig. 3g, Extended Data Fig. 9 of the previous version of the manuscript), reinforcement learning with negative reinforcement showed high variability (see previous section, Extended Data Fig. 9 of the previous version of the manuscript). To show that the increased variability was not a consequence of using the new stereo device, we repeated the discrimination experiments with positive reinforcement. Task engagement, which can also be defined as a first phase of learning, was already faster and more reliable with positive reinforcement using 3D objects (section 3.2.2, Ref. Fig. 15, 16). We have developed two protocols for discriminative learning with positive reinforcement using 3D objects with grating (section 3.2.3, Ref. Fig. 14, 17) and non-grating patterns (section 3.2.4, Ref. Fig. 18). Learning was fast with both protocols (within 30 minutes) and data with positive reinforcement showed higher stability than with negative reinforcement (compare Ref. Fig. 14, 17, and 18 to Extended Data Fig. 9 of the previous version of the manuscript). In summary, the new set of measurements confirmed that, similarly to the negative reinforcers, positive reinforcement can generate rapid learning but with improved stability.

3.14.2. “authors refer to stochasticity of visual perception, which I don’t find justified until other factors are ruled out). It would thus be necessary to expose if and why sensory experience in Moccus is different from conventional VR, what this variability is related to and if it contributes to the trial-average performance assessment.”

Following the referee’s remark, we conducted new experiments to investigate whether stochastic visual perception is present in a classical VR system. We used a treadmill with a classical monitor

system to generate a linear VR maze with reward zones (**Ref. Fig. 30a**). We found that the licking rate increased before the reward zone after 10 days of training, showed a ramp-like amplitude increase, and fluctuated between two states at the level of individual trials (**Ref. Fig. 30b,c**). Trials with high and low licking rates before the reward were defined as high- and low-ramp trials, respectively, according to the amplitude histogram of the anticipatory licking rate. We found that the anticipatory licking rate had a larger ramp-like amplitude increase during high-ramp trials (**Ref. Fig. 30d**), and that velocity decreased more sharply before the reward zone and had a smaller amplitude at reward than during low-ramp trials (**Ref. Fig. 30e**). Moreover, anticipatory dendritic subthreshold voltage signals recorded using 3D AO microscopy with real-time motion correction (see below **Ref. Fig. 32**) had a larger amplitude during high-ramp trials than during low-ramp trials. We found similar data for the entire population of $n=7$ mice (**Ref. Fig. 30g-j**). In summary, these data indicate that visual perception can also be stochastic, not only in Mculus, but also in classical VR systems, and can fluctuate between two different states: where one of the states is associated with early perception and fast reaction time with large amplitude changes in the behavioral parameters, such as running and licking. On the other hand, in the second state, there is a delay in behavior and the amplitude of the responses is also smaller. These two perception states were also reflected in neuronal responses: anticipatory signals, which were subthreshold membrane potential increases, were characteristic only of the high perception state. However, further measurements are required (e.g., silencing neuronal activity in V1 with optogenetics) to understand the neuronal mechanisms underlying trial-to-trial fluctuation.

Following the Editor's advice, we will avoid extending the neurobiological part of the manuscript with this new set of measurements.

Ref. Fig. 30. Stochastic fluctuation in visual perception in a classical VR system. **a**, Schematic of the measurements from a classical linear maze with a water reward device. VR was realized with a classical monitor system. Training was carried out in 7-10 days. **b-f**, Exemplified measurement from a single cell.

b, Amplitude of the anticipatory licking signal measured as an integral before reward location as a function of time. *c*, Amplitude histogram of the anticipatory licking signals showed a bimodal distribution with the two peaks defined as low ramps (cyan) and high ramps (red). Curves indicate two Gaussian fits and the sum of the two Gaussian fits. *d*, Average anticipatory licking during high-ramp ($n=23$) and low-ramp ($n=18$) trials. The ramp-like anticipatory signal before the reward zone is indicated with a red arrow. *e*, Running speeds during high-ramp ($n=23$) and low-ramp ($n=18$) trials. *f*, 3D dendritic voltage signal recorded with the JEDI-2 sensor during high-ramp ($n=23$) and low-ramp ($n=18$) trials from HC pyramidal cells. *g-i*, The same as panel d-f but for $n=7$ mice. *j*, Average running speeds during high (red) and low (blue) ramps in $n=7$ mice.

On a more conceptual level, while this task is demonstrating rapid visio-spatial learning, the main claim is that without the advanced features of Moculus it would not be possible. However, for this particular task, a head-fixed setup is not required, as contextual avoidance paradigms are well established in freely-moving rodents and large scale electrophysiological (especially with the Neuropixel probes), 1PM (Insxopix, Miniscope) and even 2PM imaging are all possible. I could not discern from the paper why does one need Moculus for studying this specific learning paradigm.

At the request of referees, we want to highlight below in three sections (**3.15.1- 3.15.3**), the benefits of Moculus for studying this specific learning paradigm:

3.15.1. Miniaturization of Moculus for freely moving applications

We thank the referee for drawing our attention to this important conceptual question. We agree with the referee that rapid visio-spatial learning should take place in freely moving conditions as well (see sections **3.13.3**, **3.10**). Nevertheless, the advantage of Moculus is that it allows the use of large, heavy, complex optical and electrophysiological recording systems that are currently not available in miniaturized versions. We also agree with the referee that freely moving systems represent the future and the that the field is developing rapidly (see for example references:⁶⁰⁻⁶⁴). Fortunately, the Moculus concept can easily be adapted to freely moving systems in the near future, as the total weight of recently developed projectors could be less than 1 gram (see for example:

[REDACTED]). We are already working on the development of a freely moving Moculus concept that will be compatible with the currently existing freely moving microscopes⁶³. However, to achieve further miniaturization, an additional development time of more than 1.5 years is required. The advantage of virtual realities during freely moving conditions is discussed in the **3.13.3** and **3.11.2** sections, see also *Ref. Fig. 27*.

Therefore, in the manuscript we now say:

“Thanks to the low weight (< 1g) of recently developed microdisplays, Moculus-S can be further miniaturized and adapted to freely-moving experiments (see for example Weijian Zong et al.)⁶⁰⁻⁶⁴”

3.15.2. Combination of Moculus with large recording systems that are currently not available for freely moving recordings allows new learning paradigms and complex measurements in behaving mice.

Although freely-moving imaging is an outstanding concept⁶⁰⁻⁶³, there are many excellent imaging and electrophysiological technologies for different purposes (e.g., dendritic imaging or 3D photostimulation) that are too heavy and not yet miniaturized. For example, but not limited to, the

following technologies are overweight for freely moving but can easily be used in combination with Mocolus: Scanned Line Projection Microscopy (Kaspar Podgorski et al., SFN poster); Holographic Microscopy⁶⁵; Acousto-Optical microscopy^{16,46,66} fast, Multiplane Line-scan Confocal Microscopy using axially distributed slits⁶⁷; Swept, Confocally-aligned planar excitation⁶⁸); for a review see e.g.⁶⁹). There are also several advantages of combining Mocolus and a 3D acousto-optical microscope (as shown in our manuscript), which, to our knowledge, is not yet available in freely moving configuration, for example:

- **Random-access voltage imaging.** Although MEMS scanners are widely used in freely moving application because of their high resonant frequency (x: 2.6 kHz, y: 2.62 kHz, [REDACTED]), their x- and y-axis step responses are in the order of 1 ms, which corresponds to the 1-kHz random-access scanning speed in 2D (Ref. Fig. 31). 3D AO microscopes, on the other hand, have a 100 times higher random-access time (100 kHz), which also applies to 3D measurements. However, fast voltage imaging with the latest generation of voltage sensors (JEDI-2, ASAP), requires high (> 2 kHz/ROI) imaging speeds, which are not possible with currently available MEMS devices in multiple regions. AO technology can provide over 100 times higher random-access scanning speeds in 3D, which is just enough for fast voltage imaging with the new generation of genetically encoded voltage indicators (Ref. Fig. 32). Therefore, for fast voltage imaging, we need to combine 3D AO microscopy with Mocolus, as the 3D AO scanners are currently too heavy for freely moving recording conditions.
- **Fast voltage frame imaging.** Mini microscopes currently only have a frame rate of 15-40 Hz, which is about 50-133-fold lower than that needed for fast voltage imaging. In contrast, 3D AO imaging allows fast voltage imaging at over 2 kHz speed by using 50 lines, resulting, for example, in 5 frames located in any position in 3D. In addition, with ULOVE or 3D disc scanning, which allows imaging a small volume or a disc within a single AO cycle (in 10 μ s), simultaneous imaging of 50 frames (cells) at any position in 3D is possible at a speed of 2 kHz.
- **Fast 3D two-photon voltage imaging of dendrites** (related to the previous two points). The product of the signal collection efficiency and acquisition speed is improved by several orders of magnitude when using a 3D AO microscope compared to raster scanning¹⁵. This enables fast voltage imaging of small diameter dendrites by scanning along different 3D objects, such as 3D ribbons, 3D lines, Multiple frames^{15,45} using JEDI-2 or ASAP-5 sensors with an excellent signal-to-noise ratio in behaving mice (Ref. Fig. 32). Multiple dendritic segments, spines and corresponding somatic location, up to 100 regions, can be recorded simultaneously at high (up to more than 1 kHz) rates (see Katona et al. 2012, Szalay et al. 2016, Judák et. al. 2022)^{15,16,45} allowing fast 3D voltage imaging (Ref. Fig. 32).
- **Real-time motion correction** (related to the previous three points). The AO technology provides fast motion correction not only along the x,y-axis, but also along the z-axis at high speed (up to 3 kHz, but only 100 Hz is required in mice: Ref. Fig. 32). The fast closed-loop motion correction (Ref. Fig. 32b) effectively eliminates brain motion (Ref. Fig. 32b) caused by heartbeats, breathing or physical movement of the mice. The method is essential for fast voltage imaging in behaving mice, as it reduces the amplitude of noise caused by brain motion by two orders of magnitude, according to the Fourier spectra of the in vivo measurements (Ref. Fig. 32d). Motion correction allows imaging of small structures such as dendritic spines in behaving mice, see the previous point.
- **Simultaneous 3D imaging and 3D photostimulation of neuronal networks and processes.** Photoactivation is highly sensitive to brain movements, and even a sub-micrometer displacement can generate significant differences in response amplitudes. However, real-time motion correction can provide the required high, submicron spatial resolution for targeting somata, dendrites and spines in behaving mice (Ref. Fig. 32).
- **Simultaneous imaging of neuronal network and neuronal process (dendrites, spine and axons) in 3D in behaving mice.** With 3D AO scanning, dendrites can be recorded in an over 650 μ m z-scanning range^{15,45}. This allows simultaneous imaging of somata and dendrites of LII/LIII and LV neurons.

- **Fast z-focusing** (related to the previous points). Unlike liquid lenses and piezoelectric solutions used in today's advanced freely moving microscope systems, z-focusing is not limited by mechanical constraints of the device and can be done with orders of magnitude higher speeds (within 10 μ s) and in an extended fast z-scan range (650-1000 μ m, depending on labelling)^{15,45}.
- **Large field-of-view with high spatial resolution.** Unlike GRIN lenses and microlens solutions used in today's advanced freely-moving microscope systems, AO microscopy can use the advantage of the state-of-the-art high NA objectives, which are relatively large and heavy but offer a large field-of-view (over 700 μ m) at maximally preserved resolution. Even larger FOVs (up to 4000 μ m) are possible for the price of reduced resolution.
- **Temporal 'super-resolution' microscopy.** An AO microscope can scan with up to 100 kHz speed, which is a significant advantage during in vivo voltage imaging with currently available fast dyes (JEDI-2, ASAP-5). Using 3D AO scanning, we were already able to achieve about 50 μ s temporal resolution with Ca²⁺ sensors using two different methods [**Supplementary Figure 9 and 10** in¹⁶]. Thanks to real-time motion correction and to the improved AO scanning technology described in the current manuscript, we now have better than 50 μ s temporal resolution with voltage sensors in behaving mice (**Ref. Fig. 33**).
- **etc.**

In summary, we agree with the referee that freely moving microscopy is one of the most progressively developing fields and, therefore, we would also like to develop a 3D AO scanner compatible with the currently available advanced freely moving microscopes. Nevertheless, many different microscopy technologies that have been developed with unique properties for in vivo measurements in head-restrained animals but which are currently not available in miniaturized form for freely moving conditions. Therefore, Mocolus opens a new opportunity for these laboratories to use their complex technologies in new fast learning and behavior paradigms.

Redacted

Redacted

Redacted

3.15.3. Fast task engagement and learning with Mculus allows more sophisticated and longer learning paradigms

According to discussions with several[√] principal investigators, behavioral and learning protocols are increasingly challenging for young students and are therefore the main speed-limiting factors of publications for many laboratories. Even more challenging is the growing demand for more complex protocols lasting over 60 days (see for example **Fig 2B** in Xinyu Zhao et al.⁴⁷). One possible solution is to develop an automated platform for high-throughput behavior and physiology with voluntary head fixation^{70,71}. However, during these prolonged automatic protocols lasting for several days, several factors may hide and alter the network mechanisms associated with the particular behavior protocol. First, some conditions, such as the expression levels of genetically encoded sensors and tissue stability, can affect neuronal recordings. Second, neuronal networks exhibit prominent internal dynamics and, although cortical representations are relatively stable on shorter time-scales in the V1 region³⁶, cortical representations of sensory information drift and fluctuate over a timescale of days³⁷⁻⁴⁴. The high internal dynamics interferes with measurement of the direct impact of learning and any other long-term effect. As Mculus has made behavior experiments about 144-432-fold faster (see sections **3.2.2**, **3.2.3**, and **3.2.4** and **Ref. Fig. 14-18**), it allows elimination of these long-term effects to reveal the network mechanisms more directly.

In summary, Mculus serves as a cost-effective alternative and also as an efficiency-increasing supplementary method of automated platforms that can accelerate high-throughput behavior experiments for basic and pharmaceutical research. Moreover, the high speed of task engagement and learning allows more sophisticated experiments.

In the manuscript we now say (see also above):

“Moreover, these tests and experiments also validated that adding these more natural components of vision to the VR with Mculus (true 3D corridors, abysses, 3D objects in the middle part of the binocular zone), generated better visual immersion with a more than 200-fold faster task engagement than with classical monitor systems⁶. In addition, the use of true 3D corridors with 2D patterns oriented in 3D and perspective correction to match the geometry of the 3D virtual corridor has improved discriminative learning by a factor of about 144-432 compared to previous studies with 2D screens (Mculus: 30-40 minutes, 2D screens: 3-9 days¹⁻⁵. However, the best performance was achieved when multiple 3D features were simultaneously combined in a natural way: we used a true 3D virtual corridor as above, because mice in their natural habit live in tunnels, but also added 3D objects with more natural patterns than gratings placed to the upper middle part of the binocular zone, because

mice in their natural habitat approach objects from the front. The improved learning speed opens up the possibility of applying complex protocols that currently take more than 60 days⁴⁷.”

Moreover, there is no comparison of the discovered aspects of neural correlates of learning with the ground truth for the head-fixed VR - freely moving equivalent of the task. Is learning of the task faster than in the freely-moving analog? Are neural correlates matching those in the freely-moving analog?

3.16.1. *“Is learning of the task faster than in the freely moving analog?”* (related to sections **3.10**, **3.13.3**)

To respond to the referee’s question, we have reproduced the previously published visual discrimination learning protocol⁴, but instead of freely moving mice we used head-fixed animals with Mocus. Control and reward zones were cued with 3D objects decorated with grating patterns (sections **3.2.2**, **3.2.3** and **Ref. Fig. 14-17**) and non-grating patterns (sections **3.2.4**, **Ref. Fig. 18**). Task engagement was much faster with 3D objects than with the 2D frames with grating pattern on the wall of the 3D virtual corridor, and discriminative learning was significant within a single session and more stable than with the previous protocol. The fastest performance in visual discrimination learning was achieved with 3D objects floating close to the center of the binocular zone (sections **3.2.4**, **Ref. Fig. 18**).

It was shown previously that while at least 4 training sessions (4 days) were needed for freely moving mice to learn to discriminate two different orientations⁴ (see **Ref. Fig. 26**), head-fixed animals took longer to learn the same task (9–25 sessions were required for head-fixed and only 4–6 sessions for freely-moving mice)⁴. In contrast, we have shown that a single session is sufficient for discriminative learning with Mocus. This means a more than 144-fold faster learning speed than freely moving conditions (freely moving with gratings: 5,760 minutes, Mocus with gratings: 40 minutes) and a more than 144-432-fold faster speed than the previous head-fixed measurements (head-restrained: 12,960 minutes, Mocus with 3D objects: 30 minutes)⁴.

The benefits of virtual-reality systems to enhance learning have already been demonstrated in human applications⁵⁵⁻⁵⁸, which may explain why VR is also more effective in mice.

3.16.2. *“Are neural correlates matching those in the freely moving analog?”*.

The Z-scanning range limits measurement of neural correlates in freely moving conditions. Even the current cutting-edge head-mounted two-photon microscope has only about a 160 μm fast z-scanning range in vivo (see Fig. 3F, G in Weijian Zong et al.⁶³). In our study, however, a fast z scanning range of about 650 μm was shown and required to find the aversive and control zone coding clusters.

Current scanning speed is limited to measure neural correlates in freely moving conditions. To measure the spatiotemporal coding of the aversive and control zone-coding clusters, we had to include an average of 97 cells located in an $x = 600 \mu\text{m}$, $y = 600 \mu\text{m}$, $z = 600 \mu\text{m}$ volume (consisting of $x = 512$ pixels, $y = 512$ pixels, $z = 300$ pixels) using chessboard scanning. This solution provided an excellent signal-to-noise ratio and resulted in an acceptable volume rate of 17.14 Hz (see **Methods**) In contrast, scanning the same 3D volume with 97 cells using the most advanced MEMS scanner and z drive of a quartet μTlens of the head-mounted microscopes⁶³ is possible currently at 0.41 Hz (0.41 Hz = 40 Hz / 97 planes), which is a 41.8-times lower speed than with 3D AO scanning. To validate the need for speed in our study, we sub-sampled our measurements by a factor of 41.8 and repeated our analysis. However, we were unable to identify and measure the ramp-like components, the timing of the high-ramps and low-ramp transients, etc. at this rate. This is because the recording frequency of 0.41 Hz allows only 0.82 measurement points in a 2-second-long recording period, which was the characteristic length of the ramp-like amplitude components. Moreover, temporal differences identified in this study

were on the scale of hundreds of milliseconds (see, for example, **Ref. Fig. 40c,d**, and **Fig. 3c-g** and **Extended Data Fig. 13** of the previous version of the manuscript) and not at the temporal resolution of 2.43 seconds calculated from the 0.41 Hz measurement speed. As a solution, it would be theoretically possible to reduce the number of simultaneously recorded planes during freely moving conditions, but this would limit the simultaneous measurements of the members of the 3D clusters coding the zones.

Signal-to-noise ratio is partially limiting. To correctly answer this part of the referee's question, measurements with a similar high signal-to-noise ratio are needed with the head-restrained and freely moving conditions. The high signal-to-noise ratio of our measurements is also appreciated by the referee in section 3.6 (*"it is certainly providing high-quality population data"*). To achieve a comparable high signal-to-noise ratio during freely moving conditions we need 2P excitation, since 1P excitation produces a significantly lower signal [the peak $\Delta F/F$ was more than 100-fold smaller with 1P excitation than with 2P excitation, see **Fig. S6F, S6G** in Weijian Zong et al.⁶³]. Although we are developing a fast, two-photon, 3D, acousto-optical scanner that will be compatible with freely moving microscopes [e.g., with MINI2P⁶³], this development may take years. With the currently available cutting-edge technologies, it is not possible to reproduce in freely moving conditions the high signal-to-noise ratio of the AO measurements, which was just at the limit of detection of ramp-like activity even in neurons with small response amplitudes (see section 1.3 and **Ref. Figs. 2 and 3**).

The third limiting factor is the spatial resolution. The state-of-the-art head-mounted two-photon microscopes, with their large field of view, have an objective lens of about NA=0.5, yielding at least a twofold lower lateral and a fourfold lower axial resolution (approximately 17.8-12.8 μm). This would decrease the efficiency of detecting and subtracting neuropil contamination. In contrast, the 3D acousto-optical microscope focuses each chessboard on the center of each soma, minimizing neuropil contamination (see **Ref. Fig. 6**).

Is ramp-like firing rate increase associated with running velocity increase in the aversive zone compared to control? If well-established effect of velocity cannot be regressed out, I don't see how the term "encoding" for the aversive/control zones can be applied. Similarly, quantification of the time-dependent effect given the increase in velocity is not justified.

3.17. To respond to this complex question, we performed a new analysis (**3.17.1**), explained clearer how the well-established effect of velocity can be regressed out (**3.17.2.1**), performed new measurements and analysis to compute the time-dependent effect (**3.17.2.2**), and demonstrated that the decoder can encode aversive and control zones independent of motion (**3.17.2.3**).

3.17.1 *"Is ramp-like firing rate increase associated with running velocity increase in the aversive zone compared to control."*

Yes. We demonstrated in **Fig. 2h** of the previous manuscript version that there was a significant increase in the ramp-like firing rate within the aversive zone compared to the control zone after 30-40 minutes learning (**Ref. Fig. 34**). Following the referee's suggestion, we now also show the statistical comparison with the pre-learning data. This increase was only a trend during the first 10-minute period (aversive zone: paired t-test, $P=0.177$, Mann-Whitney test $U=0.095$; control zone: paired t-test, $P=0.18$, Mann-Whitney test $U=0.40$) that increased significantly compared to the pre-learning data in the aversive zone after 10-20 minutes training (paired t-test, $P=0.026$, Mann-Whitney test $U=0.021$), but remained non-significant in the control zone (paired t-test, $P=0.10$, Mann-Whitney test $U=0.21$, one tailed paired t-test, $P=0.051$, one tailed Mann-Whitney test $U=0.105$). However, after 20-30 minutes learning, both zones showed a significant increase compared to pre-learning data (aversive zone: one

tailed paired t-test, $P=0.02693$, one tailed Mann-Whitney test $U=0.03$; control zone: one tailed paired t-test, $P=0.0184$, Mann-Whitney test $U=0.00609$).

Ref. Fig. 25. This panel is based on Fig. 2h of the previous version of the manuscript. We now show the statistical comparison also with the pre-training data in both zones.

We added **Ref. Fig. 34** and the new statistical calculations to the manuscript as **Fig. 3h**.

3.17.2 “If well-established effect of velocity cannot be regressed out, I don’t see how the term “encoding” for the aversive/control zones can be applied. Similarly, quantification of the time-dependent effect given the increase in velocity is not justified”

3.17.2.1 Justification of the time-dependent effect

We thank the referee for the helpful comment. Increasing running speed affects V1 responses, which is why we treated time intervals before and after running separately during different quantification methods. Velocity can be ‘regressed out’ at the level of mean responses: Ca²⁺ responses can be separated from velocity. The average Ca²⁺ responses start more than 2,000 ms earlier than the average velocity increase (**Ref. Fig. 35a**).

Ref. Fig. 35. This is from Fig. 3b and 3c from the previous version of the manuscript. **a**, Average velocity and population responses after learning (30-40 minutes) are overlaid. Note that the neuronal activity increase in the aversive zone started several seconds earlier (red arrow) than the running speed elevation. **b**, Average (mean \pm SEM) Ca^{2+} responses of four example neurons at the beginning (0-10 minutes, gray) and end (30-40 minutes, red) of fast visual learning in the aversive zone. Transients were aligned to the end of the zones (0 s) before averaging. The corresponding average running speed is indicated in blue. The ramp-like component of neuronal activity in each cell started earlier, before the mouse began running.

The earlier onset of average Ca^{2+} responses versus average velocity is also evident at the level of individual neurons (**Ref. Fig. 35b**): in each of the four exemplified neurons, mean Ca^{2+} responses started before mean velocity increase with a variable time delay. These results were calculated for the average of trials, which is the gold standard method. However, the high signal-to-noise ratio of 3D AO imaging allowed a more accurate analysis at the level of individual trials. We were able to identify two groups of responses: high-ramp trials and low-ramp trials. In the case of high-ramp trials, average neuronal responses (**Ref. Fig. 36a₁**) and responses of individual neurons (**Ref. Fig. 36a₂**) started earlier than the corresponding velocity increases. Notably, there was a clear time-gap between velocity and individual and average neuronal Ca^{2+} responses in the rise-time phase at the level of individual trials (**Ref. Fig. 36**).

Ref. Fig. 36. a_1 , Average neuronal activity (red, $n=85$ cells) and running speed (blue) are shown during three representative high-ramp trials. **a_2** , Corresponding activity of a single cell with the overlaid running curve. Only one example cell is shown of the 85 neurons. Red arrows indicate that neuronal activity precedes running at the population average, single-trial, and single-cell levels during high-ramp trials. (These panels are from **Extended Data Fig. 13a** of the previous version of the manuscript).

The initiation of the Ca^{2+} responses, the onset of velocity increase, and the time-delay between Ca^{2+} responses and velocity fluctuated at the level of individual trials (note the variable length of the red arrows **Ref. Fig. 36**, see also **Extended Data Fig. 20c,f**, **Fig. 6f-j**). The trial-to-trial fluctuation logically blurred the rise-time period of both Ca^{2+} responses and velocity, when high-ramp trials were averaged but the temporal delay remained visible (**Ref. Fig. 37**). Note, for the majority of the high-ramp trials, that the increase in velocity induced a second increase in neuronal responses (first and last row in **Ref. Fig. 36**), which is also seen in averages as an inflexion point in the later phase of the increasing ramp-like neuronal activity (at the first dashed line **Ref. Fig. 37**), further confirming that neuronal ramp activity is also modulated with running.

Similarly to high-ramp trials, in case of low-ramp trials, the average neuronal responses (**Ref. Fig. 38a₁**) and the responses of individual neurons (**Ref. Fig. 38a₂**) were well separated from velocity increase at the level of the individual trials. However, the temporal delay had an opposite direction: motor activity preceded neuronal responses with a relatively fixed latency. The latency was not only opposite in sign but also showed less variance than high-ramp trials. Therefore, when the average of all low-ramp trials was calculated, the effect of blurring in the raising phase period was also smaller (**Ref. Fig. 39**) and thus the separation of neuronal responses and velocity remained better preserved than in case of high ramps.

Ref. Fig. 37. Average neuronal activity (red) and running speed (blue) during high ramps. (This panel is from **Extended Data Fig. 13a** of the previous version of the manuscript).

Ref. Fig. 38. The same as **Ref. Fig. 36** but for low-ramp trials: left, population mean responses during three exemplified low-ramp trials; right, activity of exemplified individual neurons during the same three trials. The blue arrows indicate that running precedes neuronal activity during low-ramp trials at the level of both single cells and the entire population (These panels are from **Extended Data Fig. 13b** of the previous version of the manuscript).

Ref. Fig. 39. Same as Ref. Fig. 37 but for low-ramp trials. (This panel is from the **Extended Data Fig. 13b** of the previous version of the manuscript).

High-ramp trials and low-ramp trials appeared as a trial-to-trial fluctuation and formed two distinct clusters (see Fig. 3d-g of the previous version of the manuscript) that remained separated when we plotted velocity as a function of ramps (i); delay as a function of timing of ramp (ii); timing of running as a function of timing of the ramp component (iii), etc. (see Fig. 3 and Extended Data Fig 13 of the previous manuscript).

3.17.2.2. New measurements and analysis for justifying time-dependent effect

To show more clearly that “velocity can be regressed out”, or in other words, to demonstrate more clearly the temporal separation between neuronal responses and velocity, we shifted each trial in time to velocity onset and averaged neuronal responses and velocity. For high-ramps trials, neuronal activity started more than 2,000 ms before the speed increase (Ref. Fig. 40a, $n=7$ mice), which is similar to the previous calculation (Ref. Fig. 37), but the temporal separation between neuronal responses and velocity is more pronounced. For low-ramps trials, we obtained results (Ref. Fig. 40b) that are more similar to the previous analysis (Ref. Fig. 39) than for high-ramp trials, as the jitter in the delay between speed and neuronal activity was lower than in the case of high ramps, decreasing the effect of the time shift before averaging. High ramps and low ramps were detected in each mouse ($n=7$ mice, Ref. Fig. 40c), and the time lag of high ramps had not only an opposite sign but also a larger amplitude and higher variance, reflecting that visual computation before triggering running requires more time.

Ref. Fig. 40. New data and analysis. Difference between high-ramp and low-ramp trials fluctuating trial-to-trial. **a**, High-ramp trials were shifted to velocity onset time before averaging neuronal responses (red) and running speed (blue, mean±SEM, n=9 session, n=7 mice). **b**, The same as panel **a** but for low-ramp trials. Note, the enlarged temporal scale. **c**, Average delay between neuronal response and velocity onset for high -amp and low-ramp trials after the time shift to velocity onset time (n=9 session, n=7 mice). **d**, Averages calculated from panel **c**. low ramps median: -0.43 s, high ramps median: 1.4 s, Wilcoxon signed-rank test: $w=0.0091$.

3.17.2.3 Defining a decoder independent of motion

Response to the second part of the comment: “how the term “encoding” for the aversive/control zones can be applied” and also related to referee comment: “If well-established effect of velocity cannot be regressed out, I don’t see how the term “encoding” for the aversive/control zones can be applied”. It is shown in sections 3.17.2.1 and 3.17.2.2 that velocity and neuronal responses can be well separated at the level of individual trials and averages. We have also demonstrated previously that encoding of the aversive and control zones appears before running (Ref. Fig. 41): the activity of a subpopulation of neurons was able to separate the aversive zone from the control zone before running was initiated (Ref. Fig. 41a). Moreover, there was a clear gap between the neuronal responses in the aversive and control zones of around minus 2,500 ms (see the right panel in Ref. Fig. 41a). The average response of the entire subpopulation was also significantly different in the aversive and control zones before the increase in running speed (Ref. Fig. 41b). These findings suggested that the subpopulation of cells likely separates the aversive and control zones at the level of individual trials. To validate this, we tried to define a decoder based on the activity of some neurons from this subpopulation. We found that the sum of the activity of only 3 neurons was sufficient to define a reliable decoder (Ref. Fig. 41c,d).

Ref. Fig. 41. Subpopulation of neurons encode control and aversive zone reliably. **a**, Left, mean activity of 18 individual neurons from the recorded 85 cells shown in Fig. 2b,d of the previous version of the manuscript in the aversive (red) and control zones (gray). Right, magnified view showing a clear gap in neuronal responses between the two zones at around -2.5 s, which precedes the increase in running (blue). **b**, Corresponding mean neuronal responses with clear separation before running. (These figure panels are from Extended Data Fig. 12a, b of the previous version of the manuscript.) **c**, Left, sum of the activity of three neurons (cells #25, #28, #59) from the 18 cells shown in panel a in the aversive (red) and control zones (green dashed curves). Right, the average of the summed activity of the three cells calculated from -4 s to -2.5 s was defined as a decoder that provided orthogonal output for the neuronal activity in the control and aversive zones. Note that the decoder was defined for the time interval before the running curve started to increase. Dark red and dark gray colors indicate trials where running was initiated at high speed after the decoding period. Light colors are trials where running was started at low speed after the decoding period. Note the improved performance of the decoder when running was initiated at high speed after the decoding period. **d**, Similar to c but only for trials with high running speed after the decoding period. Note the improved performance of the decoder. **e**, Correlation between running and decoder performance for all trials (Pearson's $r=0.44$) and for the high-ramp trials (Pearson's $r=0.70$). Running speed and decoder amplitude were calculated from the averages in the temporal interval from -2.5 s to -1 s and from -4 s to -2.5 s, respectively. (These figure panels are from Extended Data Fig. 12c-e of the previous version of the manuscript.)

Note that the decoder had reliably separated aversive and control zones at the level of individual trials in the temporal interval from -4 s to -2.5 s independently of running speed increase (Ref. Fig. 41c,d,e). This can be partially explained by the fact that the decoder was defined in an interval before the increase in the running speed (Ref. Fig. 41c,d,e).

Decoder functions were able to separate aversive and control-zone coding assemblies at the level of individual trials with high reliability based on the activity of only seven cells ($97.5 \pm 2.5\%$, $n=5$ mice). Although both control and aversive-zone codings recruited most of the neurons, these data suggest that the two codings have both orthogonal and overlapping components.

In summary, we have quantified and demonstrated at the level of

- whole population,
- individual low-ramp trials,
- average low-ramp trials,
- individual high-ramp trials,
- average high-ramp trials,
- individual cells (**Extended Data Fig. 20 d-f**),
- and using running speed triggered averages,
- and a zone-specific decoder, defined in a temporal interval before running speed increase,

that velocity can be regressed out and the term “encoding” for the aversive/control zones can be applied. To answer this part of the referee’s question: *“similarly, quantification of the time-dependent effect given the increase in velocity is not justified”*, we demonstrated quantification of the time-dependent effect in **13 figure panels** by showing the relationships between parameters of “delay”, “time of ramp”, “speed”, “ramp component”, “timing of running”, etc. (see **Fig. 3d,e,f,g, Extended Data Fig 20c,e,f**). Additionally, we have included a further figure with four additional panels (**Ref. Fig. 40**) as **Extended Data Fig. 24** to further clarify these relationships.

We have added **Ref. Fig. 40** as **Extended Data Fig. 24**.

In the manuscript we now say:

*“The trial-to-trial fluctuation of the causal relationship between V1 computation and motor activity was preserved when trials were time-shifted to velocity onset time before averaging velocity and neuronal responses (delay for high ramps: 1.38 ± 0.28 s and for low ramps: -0.42 ± 0.014 s; $\text{mean} \pm \text{SEM}$, t -test with Welch correction $P=2.38 \cdot 10^{-4}$, **Extended Data Fig. 24**). The ca. 20-fold lower variability during low-ramp trials (0.014 s vs. 0.28 s) also supported that V1 activity during low ramps is an indirect signal that is triggered by running with a relatively fixed delay.”*

All arguments for differential coding in Fig4 are also confounded majorly by the velocity modulation, which is not identical for different neurons, hence there is a slight difference btw aversive and control zones in activity rate (e.g. Fig4c). According to Fig2e till about 1 sec before the end zone the velocity is not different between aversive and control zones on average, but the gain in firing rate is maximal in exactly after this point (Fig4a,b), when velocity in the aversive zone starts increasing compared to control.

3.18.1 We fully agree with this referee's observation: after velocity onset, the firing rate increases further and, indeed, about 1 second before the end zone, the velocity was not different between aversive and control zones on average in Fig. 2e (of the previous version of the manuscript), but the gain in firing rate was maximal exactly after this point, when velocity in the aversive zone starts

increasing compared to control. These data suggest that after the onset of running, indirect activation of the V1 region by motor activity (which is described also during low ramps, e.g. **Ref. Fig. 38-40**) is mixed together with the local computation in V1, which is visible at the level of averages and also at the level of individual trials as a sudden increase in neuronal activity at the onset of running, as shown in **Ref. Fig. 42**. Therefore, we focused our analysis on the period prior to running and showed that encoding for the aversive/control zones appears before running (**Ref. Fig. 41**, section **3.17.2.3**): the average activity of a subpopulation of neurons was able to separate the aversive zone from the control zone before running was initiated (**Ref. Fig. 41a**). Moreover, there was a clear gap between the average responses of the neuronal subpopulation in the aversive and control zones around minus 2,500 ms (see right panel in **Ref. Fig. 41a**). Accordingly, the mean response of the whole subpopulation was also significantly different in the aversive and control zones before the increase in running speed (**Ref. Fig. 41b**). These data indicated that a subpopulation of cells is likely to be able to separate the aversive and control zones at the level of individual trials. We therefore wanted to determine a decoder based on the activity of a few neurons from the subpopulation. We found that the sum of the activity of only 3 neurons qualified to define a reliable decoder (**Ref. Fig. 41c,d**). Importantly, the decoder provided robust performance at the level of individual trials in the time interval from -4 s to -2.5 before running onset (note that running, blue trace, increased only after the decoding period in **Ref. Fig. 41c**): it was able to reliably separate each trial associated to aversive and control zones (**Ref. Fig. 41c,d**). This can be partly explained by the fact that the decoder was defined as an integral in the interval before the increase in running speed (**Ref. Fig. 41c**). These data indicate that the coding in V1 is already robust even before running is initiated.

Ref. Fig. 42. a1, Average neuronal activity (red) and running speed (blue) during high ramps. **Right**, average population responses during two exemplified low-ramp trials. **a1-a2**, The three yellow arrows indicate a sudden increase in neuronal activity, which occurred when the running speed showed a progressive increase immediately or after a short fluctuation. (These figure panels are from **Extended Data Fig. 13** of the previous version of the manuscript).

3.18.2 “hence there is a slight difference btw aversive and control zones in activity rate (e.g. Fig4c).”

Fig. 4c of the previous version of the manuscript shows two effects simultaneously:

- i) First, The average ramp-like component of individual neurons correlate between the aversive and control zones (Pearson’s $r=0.64$, $P=3 \cdot 10^{-11}$), indicating a high overlap between aversive and control zone coding.

- ii) Second, moving along the x-axis, the amplitude of the aversive responses grows monotonically in small increments, while responses from the same cells fluctuate in the high amplitude range in the control zone between zero and a maximum value, which can even be larger than the average response amplitude in the aversive zone of the neighboring cells of the histogram (see also **Ref. Fig 35f**).

This means that if we examine only a short subinterval of the entire histogram, we can find neurons that “try” to maximize their amplitude difference in the control zone relative to the aversive zone, indicating that coding has also an orthogonal component.

In summary, this means that Fig. 4c of the previous version of the manuscript shows simultaneously the overlapping (i) and the orthogonal component (ii) of the coding. To quantify the overlapping component, we calculated the linear correlation (see above), while the orthogonal component of coding was quantified by defining a decoder (see **Ref. Fig. 41** and section **3.17.2.3**).

We describe these details in the manuscript in section: “Decoder revealed orthogonality of the overlapping coding assemblies”

The logic of the exposition of the results is not very easy to follow. The authors start by presenting the average behavior (velocity) and neural activity (population Ca imaging) comparing aversive and control zones (Fig2); then they show that both velocity and neural activity fluctuate from trial to trial even at the end of 1st day learning and are associated with neural activity being pre- and postdictive to motor activity depending on the trial (Fig3); then they turn to the analysis of the neural variability based on the trial-average responses in aversive and control zones splitting cells into hub, good and poor learners (Fig4); then they link early trials neural responses to emergence of the ramp and, finally, attempt to capture neuron/trial/time variability using TCA. I did not get a sense of coherent and focused approach to dissect the neural phenomenon, but rather combination of many interesting and diverse results which do not compile into a coherent picture.

3.19 We agree with the referee. Our original aim was to show a combination of many interesting and diverse phenomena to validate that the new VR glass can be a tool to reveal multiple new effects. However, detailed neurobiological analysis and validation of the underlying mechanisms was not our original aim (see our response at **section 3.3**). Nevertheless, in response to the referee’s comment, we have increased the coherence:

The previous version of the manuscript contained two weakly connected main neuroscience topics: i) rapid formation of orthogonal neuronal assemblies during discriminative learning and ii) stochastic vision. We started with the first topic to align our study with previously published works, then moved to the second story to explain the stochastic nature of the first topic and then switched back to the first topic. Following the referee’s advice, for clarity we now keep the first topic in one block and have shifted the second topic with the stochastic nature of vision to the second place. Alternatively, the second topic with the stochastic nature of vision could be our starting point. Accordingly, we have tried this sequence, but without explaining the basic terms on average responses, such as the definition of the ramp-like component associated with learning, fluctuation could not be explained.

Following the referee’s advice, we have rewritten the last paragraph of the introduction, which serves now as a “table of content” for the manuscript as it lists and explains the logical order of the topics in the manuscript.

At the end of the introduction, we now say (see also above):

“To validate that Moccus and the new behavioral protocols can reveal novel circuit mechanisms, we focused on two main neurobiological topics: i) rapid and parallel formation of reinforcement-associated and control-zone coding neuronal assemblies during visual discrimination, and ii) the stochastic nature of vision. In the first topic, we have demonstrated, during the first phase of learning, that reinforcement and control cues that are simultaneously present in the VR and need to be separated by the V1 region of the cortex during a discrimination task, generate anticipatory signals with ramp-like increase in activity. These anticipatory signals are generated “de novo” independently of orientation tuning responses and, in contrast to previous studies, orchestrate the majority of V1 neurons (70-80%) into growing and partially overlapping spatiotemporal clusters centered around hub cells. These have earlier response onset times, higher amplitudes, and increased functional connectivity, and compete in representing behavior relevant information. Over time, the reinforcement cue-associated cluster dominates neuronal activity exactly when behavior also becomes significant and, in parallel, representations of control cues shrink to the level of sparse activity (low activation ratio). The rapid formation of the clusters is mediated by reinforcement: neurons that have elongated signals at the beginning of learning with a large amplitude reinforcement-associated second component will develop increased ramp-like anticipatory signals at the end of learning. This supports the idea that reinforcement in the early phase of learning correlates with the amplitude of learning at the level of individual neurons in the late phase of learning.

In the second topic, we have demonstrated that vision is a stochastic process: successes and failures alternate from trial-to-trial. This means that the ramp-like anticipatory signals in V1 are generated stochastically from trial-to-trial with a decreasing failure rate. In the case of failures, when anticipatory responses are not generated in the V1 region of the cortex, the direction of communication is reversed, and motion will trigger an indirect signal in the V1 region with short delay and low jitter. Vice versa, during successes, ramp-like neuronal activity increases with high amplitude are followed by running with a significant delay, thus indicating that anticipatory ramp-like signals represent local computation in the V1 that contributes to the generation of motion.”

While authors use many terms “saturation”, “sparse”, “orthogonal” etc , they often do not clearly link to the proper statistics or theory and rather distract from the understanding of results.

3.20 We have added the missing definitions.

The manuscript now says:

“we show that most or even all of the neurons can be activated to near-saturation (close to 100% activation ratio) with different response amplitudes in a short time period during visual discrimination”

*“Over time, the reinforcement cue-associated cluster dominates neuronal activity exactly when behavior also becomes significant, and in parallel, representations of control cues shrink to the level of **sparse activity** (low activation ratio).”*

*“Thus, the question arises whether population coding is sufficiently **orthogonal** (i.e., whether V1 activity can discriminate between the two visual patterns associated with the aversive and control zones).”*

Control and proper regression of velocity are coming much after the terms “coding” get applied to changes of the firing rate that well parallel changes in velocity.

3.21.1 We agree with the referee: *“proper regression of velocity are coming much after the terms “coding” get applied”*. First, we defined coding during learning with averages, using mean neuronal

activity and mean running speed during ramps. This “all-in-one” approach is generally used in the literature, but contains many “artifacts”, such as the trial-to-trial fluctuation of vision through which opposite effects were superimposed. After revealing these, we were able to redefine coding in a more precise form. For clarity, we now clearly separate the initial and late definition of coding in the manuscript as both definitions are required.

3.21.2 *“coding get applied to changes of the firing rate that well parallel changes in velocity.”*

We have quantified and demonstrated in section 3.17 at the level of the whole population (i), individual cells (**Extended Data Fig. 20 d-f**) (ii), individual low-ramp trials (iii), average low-ramp trials (iv), individual high-ramp trials (v), average high-ramp trials (vi), using running speed triggered averages (vii), and using a zone-specific decoder, defined in a temporal interval before running speed increase (viii), that velocity can be regressed out and the term “encoding” for the aversive/control zones can be applied.

Another example of rushed conclusions is that TCA analysis, that shows very few cells having non-zero loading on components with diverse temporal and equally sparse trial loadings. The meaning of these factors/components is very different from the interpretation authors make. It is not appearing as a robust low-dimensional and comprehensive summary of the large variability in the neural data/behavior, but an overfitting result. In general, the notion of coding and other theoretical concepts appears to be too strongly pushed throughout the paper with no validation or reference to theoretical work. It is only at Fig4 that contribution from locomotion becomes clear and is acknowledged and anticipation of reward or motor confounds can become separated.

3.22.1. *“very few cells having non-zero loading on components...”*. We thank the referee for the helpful comment. Indeed, data interpretation in **Fig. 5f** of the previous version of the manuscript was misleading: the y axis of the first TCA component was set to include the outlier cell that compressed weight parameters suggesting that neuronal component are close to zero (**Ref. Fig. 43a**). However, if we rescale the y-axis, a rather homogeneous distribution of the neuronal weight factors of the first TCA component becomes evident (**Ref. Fig. 43a**). Since the first TCA component resembles the average ramp-like neuronal responses in the aversive zone (it can therefore also be defined as the TCA component associated with learning), its homogeneous distribution is in line with the data presented in **Fig. 2**, **Fig. 4** and **Fig. 5** of the previous version of the manuscript, which also suggested that several cells simultaneously encoded the ramp-like component of learning.

Ref. Fig. 43. Weight factors analysis of the first TCA component. **a**, Left, weight factors of the first TCA component shown in Fig. 5f of the previous version of the manuscript. The red and black dots indicate weight factors with high and low amplitudes, respectively. Right, the same weight components but after rescaling the y axis to reveal the homogeneous distribution and components with small amplitude. **b**, Average responses of neurons with high and low weight factors shown in panel a. Responses were aligned to the end of the aversive zone (0 s). **c**, The same neuronal responses as in b, but a new baseline was calculated. The large peak at around 2 s was characteristic only of neurons with high weighting factors. In contrast, neurons with low weight factors showed a monotone amplitude decrease after airpuff. **d**, The same as b but the x axis was rescaled to show the ramp-like period. **e**, The same neuronal responses as in b, but responses were aligned with the time of entry to the aversive zone (0 s). Note the overlapping visual on responses at the beginning of the aversive zone for cells with high and low weight factors. **f**, The distribution of the relative increase of the learning-associated ramp-like responses of individual neurons in the last 20-minute period of learning in the control (grey) and aversive (red, and orange) zones. Neurons with high and low weight neuronal factors for TCA component #1 are shown in red and orange, respectively. The majority of the cells with high weight factors were located at the top of the amplitude distribution. **g**, Similar to panel f but for absolute amplitudes: average 3D Ca^{2+} responses in the aversive zone after (30-40 minutes) learning. (These panels are the modified version of Fig. 2d and 4c. of the previous version of the manuscript).

3.22.2. Neurons with high weight factors are neither the result of overfitting nor can they be considered outliers; they belong to a functionally distinct subpopulation.

Next, we investigated whether outliers (neurons with high weight factors in Ref. Fig. 43a) are the result of over-fitting or whether they can be eliminated as an artefact. We found that the mean response of the five neurons with the highest weight factors for the first TCA component produced a 2.34-fold higher average ramp-like component than the rest of the population (cells with high weight factors: 0.18 ± 0.04 , with low weight factors: 0.07 ± 0.12 , $\Delta F/F$, mean \pm SEM, Mann-Whitney test, $U=0.0057$, integral from -1.4 to -0.4 s, Ref. Fig. 43b,d). Moreover, the visual ON responses at the time of entering the aversive zone did not differ between cells with high and low weight factors (Ref. Fig. 43e), indicating that the higher ramp-like component of the cells with high weight factors does not reflect an overall response increase in this neuronal subgroup. Moreover, only cells with a high weight

factor for the first TCA component had a second component after reinforcement at around 2-3 s (**Ref. Fig. 43b,c**), previously identified as a 'second component' of the reinforcement signal (**Fig. 5** of the previous version of the manuscript). Since the second component of the reinforcement signal recorded at the beginning of learning (0-10 minutes) has been shown to correlate with learning induced 30 minutes later (30-40 minutes, **Fig. 6a-c**), these data indicate that the cells with high weight factors may still be learning. However, further measurements are needed to establish the clear functional role of this second component and the neurons with high weight factors. In summary, we conclude that cells with a high weight factor have kinetics different to the rest of the population: they have a similar visual ON component but a 2.34-fold higher ramp-like component than the rest of the population with a specific second component after reinforcement; therefore, they may represent a functionally distinct subpopulation instead of being considered as a result of overfitting.

3.22.3. The majority of cells with high weight factors belong to the hub cells

The distribution of the relative increase in the learning-associated ramp-like responses of individual neurons in the last 20-minute period of learning showed a jump at an amplitude threshold (triangle in **Fig. 4c** of the previous version of the manuscript) at the upper end of the amplitude range, and amplitude-sorted ramp-like components decayed rapidly below the amplitude threshold (**Ref. Fig. 43f**). These neurons above the amplitude threshold were defined in the previous version of the manuscript as hub cells and were shown to be spatially (i) and temporally clustered (ii), to have earlier (iii) and larger (iv) response amplitudes, and more functional connections (v) than the rest of the population. The increased mean and ramp-like response of the subpopulation of neurons with high weight factors (**Ref. Fig. 43b,d**) suggested that these cells could overlap with hub cells that were previously defined solely on the basis of temporal averages. Indeed, we found that the subpopulation of neurons with high weight factors was located mainly in the high-amplitude range of the amplitude histogram of the relative increases of the learning-associated ramp-like responses (**Ref. Fig. 43f**) where hub cells were located. Similar to the relative increases of the learning-associated ramp-like responses, the top of the amplitude range of the amplitude-sorted absolute responses overlapped with the subpopulation of neurons with high weight factors (**Ref. Fig. 43g**). These data confirm that neurons with high weight factors are neither the result of overfitting, nor can they be considered outliers, but represent a functional subgroup belonging mainly to the subpopulation of hub cells.

3.22.4. *"It is only at Fig4 that contribution from locomotion becomes clear and is acknowledged and anticipation of reward or motor confounds can become separated."*

We now first show locomotion at the beginning of the neurobiological text block, at the first and second main figures of the neurobiological block (**Fig. 3e,g, Fig. 4b,c**), and also at the end of the manuscript in **Fig. 6**, because following the referee's comment **3.3** we moved the block with the stochastic nature of vision to the end of the manuscript, to keep the first neurobiological topic in one block.

We added **Ref. Fig. 43** as **Extended Data Fig. 23**.

In the manuscript we say:

*"To rule out the possibility of overfitting, we examined outliers (neurons with high weight factors, **Extended Data Fig. 23a**). We found that the mean response of the five neurons with the highest weight factors for the first TCA component produced a 2.34-fold higher average ramp-like component than the rest of the population (cells with high weight factors: 0.18 ± 0.04 , with low weight factors: 0.07 ± 0.12 , $\Delta F/F$, mean \pm SEM, Mann-Whitney test, $U=0.0057$, integral from -1.4 to -0.4, **Extended Data Fig. 23d**). Moreover, while the visual ON response at the time of entering the aversive zone did not differ between cells with high and low weight factors (**Extended Data Fig. 23e**), only cells with a high*

weight factor for the first TCA component had a second component after reinforcement at around 2-3 s (Extended Data Fig. 23b,c, this 'second component' will be identified in the next section as a reinforcement signal). These different kinetics suggested that these neurons belong to a functional subgroup. Indeed, neurons with high weight factors were mainly located in the high amplitude range of the amplitude-sorted absolute responses (Extended Data Fig. 23g) and at the top end of the amplitude histogram of the relative increases of the learning-associated ramp-like responses (Extended Data Fig. 23f). In summary, these data indicate that neurons with high weight factors are neither the result of overfitting, nor can they be considered outliers, but represent a functional subgroup belonging mainly to the subpopulation of hub cells."

Authors observe in aversive and control stimulus areas significant trial-to-trial variability of the activation of V1 neurons, normally having orientation tuning in the retinal space. Could this be related to the trail-to-trial variability of the pupil orientation of the left and right eyes while mouse is running through the grating pattern which results in the differential orientation-tuning-based drive by distinct grating patterns? Some authors (e.g. A. Meyer, C. Neil) have demonstrated independent movement of the eyes by mice, which in the current paradigm would likely be comparable. It would be important to control whether and to what extent the degree and consistency of the eye movement by mice in the 2 monitor (or conventional dome VR setup) differs from that in the Mculus setup.

3.23.1 Trail-to-trial variability of high and low ramps is not the result of changes in pupil diameter or position (new measurements and analysis)

We thank the referee for the helpful comment. The time delay between the average neuronal responses at high- and low-ramp trials was 2,696 ms ($dt_{2-1}=2,696$ ms, $t_1=-2,405 \pm 35$ ms, $t_2=-291 \pm 51$ ms, **Ref. Fig. 44a**). When trials were shifted to velocity onset time before averaging, the relative delay between high ramps and low ramps was reduced (**Ref. Fig. 44a**). However, the visual ON responses at the beginning of the aversive zone during high- and low-ramp trials showed no significant differences in the raising phase ($dt_{2-1}=148.4$ ms, $t_1=468 \pm 23$ ms, $t_2=617 \pm 39$ ms, Wilcoxon signed-rank test: $w=0.38$, calculated from 400 to 800 ms, single cell, **Ref. Fig. 44c**). Similarly, the mean time difference of visual ON responses at the beginning of the aversive zone (126.63 ms), which was 21.3-fold lower than the delay between high-ramp and low-ramp trials ($21.3=2,696$ ms / 126.6 ms, **Ref. Fig. 44a**), was not significant at the population level ($dt_{2-1}=126.63$, $t_1=298 \pm 60$ ms, $t_2=423 \pm 22$ ms, Wilcoxon signed-rank test: $w=0.67$, paired t test, $P=0.58$, $n=9$ sessions and $n=7$ mice, **Ref. Fig. 44d**). In summary, the small non-significant time difference in visual ON responses during high and low ramps suggest that the 21.3-fold higher time delay (2,696 ms) between the average neuronal responses during high and low ramps is not mediated by a different eye position of the mice entering the zone.

To validate that the visual ON responses can properly report eye position in our measurements, we performed a new set of experiments. Two zones were defined with the same moving grating pattern with opposite phases. The two zones were displayed alternately, and responses were averaged (**Ref. Fig. 46**). We also added a water reward at 4,300 ms to keep the attention of the mice high during the task (**Ref. Fig. 46**). The phase difference between the two visual stimulation patterns resulted in a 20-degree view angle difference that was reflected in the corresponding population response as a significant delay of 309 ms (**Ref. Fig. 46**). This means that if we want to theoretically explain the non-significant difference between the high- and low-ramp trials in the onset time of the visual ON responses at the beginning of the aversive zone (126.63 ms, **Ref. Fig. 44c,d**) with eye movements, then the 126.63 ms delay would correspond to an angle difference of 8.17 degrees ($8.17^\circ=20^\circ \times 126.6$ ms / 309 ms). However, it is not even possible to theoretically explain the 2,696 ms time delay between the average neuronal responses during high and low ramps by eye movement as this time delay would correspond to 175.5 degree ($175.5^\circ=20^\circ \times 126.6$ ms / 309 ms), which is a higher value than reported previously for eye movements of rodents see e.g., ⁷². The average amplitude of

saccades reported for mice was $19.0^\circ \pm 1.6^\circ$ ⁷³, which is 9.23-fold lower than the 175.5 degree calculated above. In summary, these data suggest that the trial-to-trial variability with the high time delay between high- and low ramp-associated average neuronal responses ($dt_{2-1}=2,696$ ms) cannot be explained by eye movements alone.

A direct reading of the pupil position with Moccus during behavior experiments would provide more direct evidence for the role of eye position and pupil diameter, but simultaneous use of the display and monitoring of the eye position during behavioral experiments is not yet possible. We have successfully demonstrated that mouse retina can be imaged with the Fraunhofer display (**Extended Data Fig. 4h,i** of the previous version of the manuscript), by which eye position can be calculated by mirroring the surface of the retina on the main axis of the eye. However, new optomechanics, electronics, and several software components need to be developed to integrate the retinal imaging method into behavioral experiments with Moccus. Alternatively, it may be possible in the future that a separate but integrated optical pathway with built-in IR illumination and a camera system could illuminate and image the position and diameter of the pupil. However, the creation of a proper beam-splitting method is challenging due to the small size of Moccus and the relatively high divergence of the beams (**Fig. 1c**, **Extended Data Fig. 1g** of the previous version of the manuscript).

Ref. Fig. 44. The trial-to-trial fluctuation of the causal relationship between V1 computation and motor activity cannot be explained by different eye position (related to **Fig. 3** of the previous version of the manuscript). **a**, Mean visual ON responses when entering the aversive zone during high- and low-ramp trials ($n=340$ and $n=595$, trials \times cells, respectively) in a single mouse. The delay between the visual onset times associated with high and low ramps was 148.4 ms ($t_1=468 \pm 23$ ms, $t_2=617 \pm 39$ ms), but while high- and low-ramps trials were significantly different according to Wilcoxon signed rank test ($w=1.3 \times 10^{-6}$) throughout the interval from 0 to 1,300 ms, they were not in the rise time period (from 400 to 800 ms, $w=0.38$). **b**, Similar to **a** but for $n=9$ sessions in $n=7$ mice. The mean time delay of 126 ms was not significantly different according to Wilcoxon signed-rank ($w=0.67$) and paired t -tests ($P=0.58$).

Ref. Fig. 45. The effect of pupil position and diameter on visual stimulation-related neuronal responses. **a**, Experimental arrangement and representative visual stimulus-induced responses of three cells shown as an example. **b**, Pupil position and diameter varied during the experiment. **c**, Left, the traces represent fluorescence, pupil diameter, and x- and y-positions during visual stimulation with a given orientation. Right, mean amplitude, diameter and position during visual stimulation. **d**, Top,

fluorescence, pupil diameter, and x- and y-positions during the whole imaging session. Bottom, explained variance of the fluorescence traces in case of position and diameter explanatory variables.

3.23.2 Impact of pupil positioning and diameter on trial-to-trial variability in population activity

In a new series of measurements, we investigated the impact of pupil positioning and diameter on the trial-to-trial variability of population activity during visual stimulation using pupil tracking. This study was carried out by recording the activity of VIP interneurons in the V1 region of the cortex. Pupil movements were recorded with a Basler CMOS camera (Basler puA 1,600–60 μm) through a 4x Olympus objective lens (**Ref. Fig. 45b**). At the population level, significant trial-to-trial variability was observed even when considering responses to a specific direction (**Ref. Fig. 45c**). This variability correlated most strongly with pupil diameter (**Ref. Fig. 45d**, multidirectional stimuli, diameter and position traces were Gauss-filtered). Using lasso regression (that penalizes redundancy of the explanatory variables) to examine the extent to which pupil position and diameter account for the variance, we found that they explained on average $33 \pm 13\%$ of the variability. It is important to note that position and diameter also correlate with each other ($R^2=35 \pm 10\%$), making it difficult to estimate their individual effects separately. However, it can be stated that diameter possesses greater explanatory power (diameter: $R^2=25 \pm 13\%$ vs x-y position: $R^2=19 \pm 8\%$, $n=255/4/3$ neurons/sessions/mice, color codes from D upper panel).

Redacted

In summary, although we found, similarly to previous studies, that pupil diameter and position can affect neuronal responses (**Ref. Fig. 45**), the long delay between high- and low-ramp-associated average neuronal responses ($dt_{2-1}=2,696$ ms) cannot be explained by pupil diameter or orientation because: i) pupil position changes induced much smaller temporal differences (**Ref. Fig. 46**), and ii) visual onset responses showed a 21.3-fold shorter and non-significant temporal delay during high- and low-ramp trials (**Ref. Fig. 44**).

We have added **Ref. Fig. 44c,d** with the new data as **Extended Data Fig. 22a,b**.

In the manuscript we now say:

*“The trial-to-trial fluctuation of the causal relationship between V1 computation and motor activity was preserved when trials were time-shifted to velocity onset time before averaging velocity and neuronal responses (delay for high ramps: 1.38 ± 0.28 s and for low ramps: -0.42 ± 0.014 s; mean \pm SEM, t-test with Welch correction $P = 2.38 \cdot 10^{-4}$, **Extended Data Fig. 24**). The ca. 20-fold lower variability during low-ramp trials (0.014 s vs. 0.28 s) also verified that V1 activity during low ramps is an indirect signal that is triggered by running with a relatively fixed delay. The overlap between the mean visual ON responses when entering the aversive and control zones suggested that the average time difference between high- and low-ramp trials (2,896 ms) cannot be explained by different eye positions (**Extended Data Fig. 22**).”*

Another aspect of the specific implementation of the task that is not discussed, but could play a role, is that control zones are composed of ± 45 degree stripes orientation, while the aversive zone is composed of 0 degree orientation. This would result in a differential optic flow effect and, potentially, differential eye movement induced. Did the authors test whether a different, counterbalanced, design of the control/aversive orientations of the grating give rise to a comparable behavioral and neural effects?

3.24. We agree with the referee that zones composed of ± 45 -degree and 0-degree stripes may result in a differential optic flow effect, because these zones were shown on 2D surfaces on the wall of the 3D virtual corridor with high contrast, and the pattern on the floor with lower contrast may be not strong enough to compensate for the effect of the different relative distance between stripes with 45-degree and 0-degree orientations. However, according to the referee’s question # 3.2, we successfully repeated the visual discriminative learning protocol with 3D objects (section 3.2.2 and 3.2.3, **Ref. Fig. 14**) and achieved fast learning similar to that with the previous learning protocol. The use of real 3D objects has reduced the possibility of different optical flow affecting our measurements, as 3D objects have a more accurate location. Supporting this, mice in Mculus were able to accurately estimate the distance to various 3D obstacles, for example when they stopped exactly at the edge of the abyss. See **Extended Data Fig. 6b** of the previous version of the manuscript: mice stopped quickly and precisely at the edge of the abyss.

As a further validation, we successfully repeated the visual discrimination task with 3D objects with non-grating patterns and obtained similar results with fast visual learning (section **3.2.4**, and **Ref. Fig. 18**).

Does a non-grating pattern work equally well? It would be important to test the generality of the visual context when arguing for the principles of the visual learning neural correlates.

3.25. Yes. According to the referee’s question # 3.2, we successfully repeated the visual discriminative learning protocol with 3D objects with (section 3.2.2 and 3.2.3, **Ref. Fig. 14**) and without grating patterns (section **3.2.4**, and **Ref. Fig. 18**).

Frame and refresh rate, and cohesion with other senses are important factors that contribute to immersion into the VR. It would be worth discussing how miniaturization of the VR setup led to compromised (frame rate or contrast of the micro screen) performance of the VR in other

dimensions compared to state-of-the-art possible with larger performant projectors and screens.

The maximum frame rate, achieved with optimizing the scenes of the VR and using a strong video card (NVIDIA GeForce GTX 1660) was 364 Hz. The delay of the virtual reality pipeline is measured as photon-to-photon latency based on the method of Matthew Warburton²². A high frame-rate camera (Teladyne FLIR Integrated Imaging Solutions, BFS-U3-17S7M-C) captured the initial signal of the entire pipeline that occurs when the mouse moves on the treadmill and recorded the subsequent change in the virtual environment displayed on the screens. The differences in elapsed time between animal movement on the treadmill and the corresponding virtual image response was measured in frames and the measurement time of one frame was then calculated by considering the refresh rate (1 s/framerate). The photon-to-photon latency of the entire system was 22 ms. In the field of neuroscience, most published works do not address the latency of VR systems. The few exceptions we found are:

1. FreeMoVR, a widespread standard VR solution for freely moving animals developed by John R Stowers et. al.²³ with 60-75 ms of latency,
2. 'The Dome' for freely moving locomotion for rodents²⁴ produced by Manu S. Madhav et. al., where the latency was calculated to be 97 ms.

In addition, as stated by Thurley et al., the biological effect of latency in rodent effects is not clear:

"Also the time lag between the actions of the animal and the system's response is crucial to realistic virtual stimulation. This issue has been investigated in humans (see, e.g., Friston and Steed 2014 for an overview) but to our best knowledge there is no such study examining effects of VR lag on rodent perception and performance. This may be due to the difficulty of fully determining what appears realistic to animals, specifically to rodents²⁵."

The current value (22 ms) is compliant with the industry standards of state-of-the-art human head-mounted-display virtual reality solutions and the competing solutions in behavior neuroscience. For instance, the Oculus Rift is a reference in human virtual reality research with 20.8ms photon-to-photon latency, as is Vive with 30.8 ms or Valve Index with 38.5 ms²².

The rendering of the Moculus virtual environment for the latency measurement was performed on a binocular 90 Hz OLED displays (2160x1200) by an NVIDIA GeForce GTX1660 GPU with 13th Gen Intel(R) Core(TM) i7 13700K and 16GB DDR4 RAM. Reducing the response time of the visualization will be a part of future work by optimizing rendering with graphical acceleration and parallelization, as well as terminating the bottleneck of temporal data sampling of movement input to improve the immersive experience.

In the manuscript we say:

"The delay of the virtual reality pipeline is measured as photon-to-photon latency based on the method of Matthew Warburton²². A high frame-rate camera (BFS-U3-17S7M-C, Teladyne FLIR Integrated Imaging Solutions,) captured the initial signal of the entire pipeline that occurs when the mouse moves on the treadmill and recorded the subsequent change in the virtual environment displayed on the screens. The differences in elapsed time between animal movement on the treadmill and the corresponding virtual image response was measured in frames and then the measurement time of one frame was calculated with refresh rate as 1/framerate. The photon-to-photon latency of the entire Moculus system was 22 ms, which is similar to previously published works: FreeMoVR with 60-75 ms latency²³ and dome for freely moving rodents with 97 ms latency²⁴. The current value (22 ms) is also compliant with the industry standards of state-of-the-art human head-mounted-display virtual reality solutions and the competing solutions in behavior neuroscience (for example, the Oculus Rift is a reference in human virtual reality research with 20.8 ms photon-to-photon latency, like Vive with 30.8 ms or Valve Index with 38.5 ms)²²."

The rendering of the Moculus virtual environment for the latency measurement was performed on a binocular 90 Hz OLED displays (2160x1200) by an NVIDIA GeForce GTX1660 GPU with 13th Gen Intel(R) Core(TM) i7 13700K and 16GB DDR4 RAM. Reducing the response time of the visualization will be a part of future work by

optimizing rendering with graphical acceleration and parallelization, as well as terminating the bottleneck of temporal data sampling of movement input to improve the immersive experience.”

Minor comments:

Authors emphasize that whiskers are not affected by their optic contraption. But this is also the case for the standard head-fixed VR. Moreover, false tactile feedback from parts of the setup is no less false than a lack of feedback when the mouse approaches virtual boundaries. Thus optimized mechanical design is a nice feature, but its advantage needs to be demonstrated. If authors want to emphasize this feature there needs to be a demonstration of the behavioral paradigm that makes use of the whiskers in combination with visual VR.

Thank you for your feedback. We acknowledge your points regarding the importance of demonstrating the advantage of our optimized mechanical design, particularly in relation to whisker function in combination with visual VR. We will consider incorporating a demonstration of this feature in the future to better highlight its significance.

Ref 62 is incomplete.

We have corrected the incorrectly listed reference.

While the improved 2PM imaging technique is impressive, it doesn't appear to be a critical and integral component of the focus of the paper - Mculus. I would therefore move some of the details of the methods in this section to the results, unless they are important to understand the validation of the Mculus setup.

Thank you for your feedback. We appreciate your suggestion to streamline the presentation of the methods regarding the 2PM imaging technique. We will consider moving some of the details from this section to the results, focusing on those aspects that are crucial for understanding the validation of the Mculus setup.

Fig2d - please label x-axis. It appears to be a number of neurons, based on legends, but more explicit labeling would help.

Certainly, we have added labeling to the x-axis of Figure 2d to make it more explicit. Thank you for bringing this to our attention.

Results exposition jumps from Fig2 to Fig4 and then back to Fig3, consider swapping figures.

*Thank you for the suggestion. The previous version of the manuscript contained two, weakly connected main neuroscience topics: i) rapid formation of orthogonal neuronal assemblies during discriminative learning, and ii) stochastic vision. We began with the first topic in order to align our study with previously published works, then moved to the second story to explain the stochastic nature of the first topic and then switched back to the first topic. Following the referee's need for clarity, we now keep the first topic in one block and have shifted the second topic with the stochastic nature of vision to the end of the manuscript. See section **3.3**.*

This email has been sent through the Springer Nature Tracking System NY-610A-NPG&MTS

Confidentiality Statement:

This e-mail is confidential and subject to copyright. Any unauthorised use or disclosure of its contents is prohibited. If you have received this email in error please notify our Manuscript Tracking System Helpdesk team at <http://platformsupport.nature.com>.

Details of the confidentiality and pre-publicity policy may be found here <http://www.nature.com/authors/policies/confidentiality.html>

Privacy Policy | Update Profile

DISCLAIMER: This e-mail is confidential and should not be used by anyone who is not the original intended recipient. If you have received this e-mail in error please inform the sender and delete it from your mailbox or any other storage mechanism. Springer Nature America, Inc. does not accept liability for any statements made which are clearly the sender's own and not expressly made on behalf of Springer Nature America, Inc. or one of their agents.

Please note that neither Springer Nature America, Inc. or any of its agents accept any responsibility for viruses that may be contained in this e-mail or its attachments and it is your responsibility to scan the e-mail and attachments (if any).

References

- 1 Poort, J. *et al.* Learning Enhances Sensory and Multiple Non-sensory Representations in Primary Visual Cortex. *Neuron* **86**, 1478-1490, doi:10.1016/j.neuron.2015.05.037 (2015).
- 2 Khan, A. G. *et al.* Distinct learning-induced changes in stimulus selectivity and interactions of GABAergic interneuron classes in visual cortex. *Nat Neurosci* **21**, 851-859, doi:10.1038/s41593-018-0143-z (2018).
- 3 Goltstein, P. M., Reinert, S., Glas, A., Bonhoeffer, T. & Hubener, M. Food and water restriction lead to differential learning behaviors in a head-fixed two-choice visual discrimination task for mice. *PLoS One* **13**, e0204066, doi:10.1371/journal.pone.0204066 (2018).
- 4 Goldstein, P. M., Reinert, S., Bonhoeffer, T. & Hübener, M. Mouse visual cortex areas represent perceptual and semantic features of learned visual categories. *Nature Neuroscience* **24(10)**, 1441–1451., doi:10.1038/s41593-021-00914-5 (2021).
- 5 Havenith, M. N. *et al.* The Virtual-Environment-Foraging Task enables rapid training and single-trial metrics of attention in head-fixed mice. *Sci Rep* **8**, 17371, doi:10.1038/s41598-018-34966-8 (2018).
- 6 Pakan, J. M., Francioni, V. & Rochefort, N. L. Action and learning shape the activity of neuronal circuits in the visual cortex. *Curr Opin Neurobiol* **52**, 88-97, doi:10.1016/j.conb.2018.04.020 (2018).
- 7 Szadai, Z. *et al.* Cortex-wide response mode of VIP-expressing inhibitory neurons by reward and punishment. *Elife* **11**, doi:10.7554/eLife.78815 (2022).
- 8 Arieli, E., Younis, N. & Moran, A. Distinct Progressions of Neuronal Activity Changes Underlie the Formation and Consolidation of a Gustatory Associative Memory. *J Neurosci* **42**, 909-921, doi:10.1523/JNEUROSCI.1599-21.2021 (2022).
- 9 Xiang, J. Z. & Brown, M. W. Neuronal responses related to long-term recognition memory processes in prefrontal cortex. *Neuron* **42**, 817-829, doi:10.1016/j.neuron.2004.05.013 (2004).
- 10 Wang, D. V. *et al.* Neurons in the amygdala with response-selectivity for anxiety in two ethologically based tests. *PLoS One* **6**, e18739, doi:10.1371/journal.pone.0018739 (2011).

- 11 King, C. W., Ledochowitsch, P., Buice, M. A. & de Vries, S. E. J. Saccade-Responsive Visual
Cortical Neurons Do Not Exhibit Distinct Visual Response Properties. *eNeuro* **10**,
doi:10.1523/ENEURO.0051-23.2023 (2023).
- 12 Henschke, J. U. *et al.* Reward Association Enhances Stimulus-Specific Representations in
Primary Visual Cortex. *Curr Biol* **30**, 1866-1880 e1865, doi:10.1016/j.cub.2020.03.018 (2020).
- 13 Jurjut, O., Georgieva, P., Busse, L. & Katzner, S. Learning Enhances Sensory Processing in
Mouse V1 before Improving Behavior. *Journal of Neuroscience* **37**, 6460-6474,
doi:10.1523/JNEUROSCI.3485-16.2017 (2017).
- 14 Corbo, J., McClure, J. P., Jr., Erkat, O. B. & Polack, P. O. Dynamic Distortion of Orientation
Representation after Learning in the Mouse Primary Visual Cortex. *J Neurosci* **42**, 4311-4325,
doi:10.1523/JNEUROSCI.2272-21.2022 (2022).
- 15 Szalay, G. *et al.* Fast 3D Imaging of Spine, Dendritic, and Neuronal Assemblies in Behaving
Animals. *Neuron* **92**, 723-738, doi:10.1016/j.neuron.2016.10.002 (2016).
- 16 Katona, G. *et al.* Fast two-photon in vivo imaging with three-dimensional random-access
scanning in large tissue volumes. *Nature Methods*, 201–208, doi:10.1038/nmeth.1851
(2012).
- 17 Oldenburg, I. A. *et al.* The logic of recurrent circuits in the primary visual cortex. *Nat Neurosci*
27, 137-147, doi:10.1038/s41593-023-01510-5 (2024).
- 18 Belli, H. M., Bresee, C. S., Graff, M. M. & Hartmann, M. J. Z. Quantifying the three-
dimensional facial morphology of the laboratory rat with a focus on the vibrissae. *PLoS One*
13, e0194981, doi:10.1371/journal.pone.0194981 (2018).
- 19 Petersen, C. C. The functional organization of the barrel cortex. *Neuron* **56**, 339-355,
doi:10.1016/j.neuron.2007.09.017 (2007).
- 20 Petersen, C. C. H. Sensorimotor processing in the rodent barrel cortex. *Nat Rev Neurosci* **20**,
533-546, doi:10.1038/s41583-019-0200-y (2019).
- 21 Bonifazi, P. *et al.* GABAergic hub neurons orchestrate synchrony in developing hippocampal
networks. *Science* **326**, 1419-1424, doi:10.1126/science.1175509 (2009).
- 22 Warburton, M., Mon-Williams, M., Mushtaq, F. & Morehead, J. R. Measuring motion-to-
photon latency for sensorimotor experiments with virtual reality systems. *Behav Res*
Methods **55**, 3658-3678, doi:10.3758/s13428-022-01983-5 (2023).
- 23 Stowers, J. R. *et al.* Virtual reality for freely moving animals. *Nat Methods* **14**, 995-1002,
doi:10.1038/nmeth.4399 (2017).
- 24 Madhav, M. S. *et al.* The Dome: A virtual reality apparatus for freely locomoting rodents. *J*
Neurosci Methods **368**, 109336, doi:10.1016/j.jneumeth.2021.109336 (2022).
- 25 Thurley, K. & Ayaz, A. Virtual reality systems for rodents. *Curr Zool* **63**, 109-119,
doi:10.1093/cz/zow070 (2017).
- 26 Friston, S. & Steed, A. Measuring latency in virtual environments. *IEEE Trans Vis Comput*
Graph **20**, 616-625, doi:10.1109/TVCG.2014.30 (2014).
- 27 Poort, J. *et al.* Learning and attention increase visual response selectivity through distinct
mechanisms. *Neuron* **110**, 686-697 e686, doi:10.1016/j.neuron.2021.11.016 (2022).
- 28 Si, Y. G. *et al.* Emerging V1 neuronal ensembles with enhanced connectivity after associative
learning. *Front Neurosci* **17**, 1176253, doi:10.3389/fnins.2023.1176253 (2023).
- 29 Garner, A. R. & Keller, G. B. A cortical circuit for audio-visual predictions. *Nat Neurosci* **25**, 98-
105, doi:10.1038/s41593-021-00974-7 (2022).
- 30 Marks, T. D. & Goard, M. J. Author Correction: Stimulus-dependent representational drift in
primary visual cortex. *Nat Commun* **12**, 5486, doi:10.1038/s41467-021-25825-8 (2021).
- 31 S.W.Failor, M. C., K. D.Harris. Visuomotor association orthogonalizes visual cortical
population codes. *bioRxiv*, doi:10.1101/2021.05.23.445338 (2022).
- 32 Geiller, T. *et al.* Local circuit amplification of spatial selectivity in the hippocampus. *Nature*
601, pages105–109 (2022), doi:10.1038/s41586-021-04169-9 (2021).
- 33 Grienberger, C., Giovannucci, A., Zeiger, W.,Portera-Cailliau, C. Two-photon calcium imaging
of neuronal activity. *Nature*, doi:10.1038/s43586-022-00147-1 (2022).

- 34 Bittner, K. C., Milstein, A. D., Grienberger, C., Romani, S. & Magee, J. C. Behavioral time scale synaptic plasticity underlies CA1 place fields. *Science* **357**, 1033-1036, doi:10.1126/science.aan3846 (2017).
- 35 Choi, Y. *et al.* NGL-1/LRRC4C-Mutant Mice Display Hyperactivity and Anxiolytic-Like Behavior Associated With Widespread Suppression of Neuronal Activity. *Front Mol Neurosci* **12**, 250, doi:10.3389/fnmol.2019.00250 (2019).
- 36 Jeon, B. B., Swain, A. D., Good, J. T., Chase, S. M. & Kuhlman, S. J. Feature selectivity is stable in primary visual cortex across a range of spatial frequencies. *Sci Rep* **8**, 15288, doi:10.1038/s41598-018-33633-2 (2018).
- 37 Lutcke, H., Margolis, D. J. & Helmchen, F. Steady or changing? Long-term monitoring of neuronal population activity. *Trends Neurosci* **36**, 375-384, doi:10.1016/j.tins.2013.03.008 (2013).
- 38 Ziv, Y. *et al.* Long-term dynamics of CA1 hippocampal place codes. *Nat Neurosci* **16**, 264-266, doi:10.1038/nn.3329 (2013).
- 39 Chen, J. L. *et al.* Pathway-specific reorganization of projection neurons in somatosensory cortex during learning. *Nat Neurosci* **18**, 1101-1108, doi:10.1038/nn.4046 (2015).
- 40 Peron, S. P., Freeman, J., Iyer, V., Guo, C. & Svoboda, K. A Cellular Resolution Map of Barrel Cortex Activity during Tactile Behavior. *Neuron* **86**, 783-799, doi:10.1016/j.neuron.2015.03.027 (2015).
- 41 Rose, T., Jaepel, J., Hubener, M. & Bonhoeffer, T. Cell-specific restoration of stimulus preference after monocular deprivation in the visual cortex. *Science* **352**, 1319-1322, doi:10.1126/science.aad3358 (2016).
- 42 Berry, K. P. & Nedivi, E. Spine Dynamics: Are They All the Same? *Neuron* **96**, 43-55, doi:10.1016/j.neuron.2017.08.008 (2017).
- 43 LeMessurier, A. M. & Feldman, D. E. Plasticity of population coding in primary sensory cortex. *Curr Opin Neurobiol* **53**, 50-56, doi:10.1016/j.conb.2018.04.029 (2018).
- 44 Schoonover, C. E., Ohashi, S. N., Axel, R. & Fink, A. J. P. Representational drift in primary olfactory cortex. *Nature* **594**, 541-546, doi:10.1038/s41586-021-03628-7 (2021).
- 45 Judak, L. *et al.* Sharp-wave ripple doublets induce complex dendritic spikes in parvalbumin interneurons in vivo. *Nat Commun* **13**, 6715, doi:10.1038/s41467-022-34520-1 (2022).
- 46 Griffiths, V. A. *et al.* Real-time 3D movement correction for two-photon imaging in behaving animals. *Nat Methods* **17**, 741-748, doi:10.1038/s41592-020-0851-7 (2020).
- 47 Xinyu Zhao, R. G., Andrea Kozlosky, Angela Jacobs, Colin Morrow, Sarah Lindo, Nelson Spruston. Meta-learning in head fixed mice navigating in virtual reality: A Behavioral Analysis *bioRxiv*, doi:10.1101/2023.05.01.538936 (2023).
- 48 Mathematics, U. o. U. D. o. *The Geometry of Perspective Drawing on the Computer*, [REDACTED] (2024).
- 49 Bankova, A. I. Application of a method for calculating the sizes of perspective objects. *IOP Conference Series: Materials Science and Engineering*, doi:10.1088/1757-899X/1031/1/012128 (2021).
- 50 Reinert, S., Hubener, M., Bonhoeffer, T. & Goltstein, P. M. Mouse prefrontal cortex represents learned rules for categorization. *Nature* **593**, 411-417, doi:10.1038/s41586-021-03452-z (2021).
- 51 Jacob, G. *et al.* A naturalistic environment to study visual cognition in unrestrained monkeys. *Elife* **10**, doi:10.7554/eLife.63816 (2021).
- 52 I. Hirskyj-Douglas, J. C. R., B. Cassidy. A dog centred approach to the analysis of dogs' interactions with media on TV screens. *International Journal of Human-Computer Studies* **Volume 98**, Pages 208-220, doi:10.1016/j.ijhcs.2016.05.007 (2017).
- 53 Ulbrich, M. *et al.* Advantages of a Training Course for Surgical Planning in Virtual Reality for Oral and Maxillofacial Surgery: Crossover Study. *JMIR Serious Games* **11**, e40541, doi:10.2196/40541 (2023).

- 54 Emanuele Argento, G. P., Eva Baka, Michail Maniadakis, Panos Trahanias, Michael Sfakianakis & Ioannis Nestoros Augmented Cognition via Brainwave Entrainment in Virtual Reality: An Open, Integrated Brain Augmentation in a Neuroscience System Approach. *Augment Hum Res*, doi:10.1007/s41133-017-0005-3 (2017).
- 55 Millunchick, V. C. B. C. S. D. M. S. J. in *Frontiers in Education (FIE) Conference* (2018).
- 56 Murat Coban, Y. I. B., Idris Goksu. The potential of immersive virtual reality to enhance learning: A meta-analysis. *Educational Research Review*, doi:10.1016/j.edurev.2022.100452 (2022).
- 57 Bustillo, D. C. A. A review of immersive virtual reality serious games to enhance learning and training. *Multimedia Tools and Applications*, doi:10.1007/s11042-019-08348-9 (2019).
- 58 Sheng, H. H. M. a. L. S. Applying Situated Learning in a Virtual Reality System to Enhance Learning Motivation. *International Journal of Information and Education Technology* (2011).
- 59 Holgersson, E. P. M. *Do higher levels of immersion in driving simulators lead to faster learning?*, Linköping University (2023).
- 60 Klioutchnikov, A. *et al.* A three-photon head-mounted microscope for imaging all layers of visual cortex in freely moving mice. *Nat Methods* **20**, 610-616, doi:10.1038/s41592-022-01688-9 (2023).
- 61 Ghosh, K. K. *et al.* Miniaturized integration of a fluorescence microscope. *Nat Methods* **8**, 871-878, doi:10.1038/nmeth.1694 (2011).
- 62 Kim, T. H. & Schnitzer, M. J. Fluorescence imaging of large-scale neural ensemble dynamics. *Cell* **185**, 9-41, doi:10.1016/j.cell.2021.12.007 (2022).
- 63 Zong, W. *et al.* Large-scale two-photon calcium imaging in freely moving mice. *Cell* **185**, 1240-1256 e1230, doi:10.1016/j.cell.2022.02.017 (2022).
- 64 Zong, W. *et al.* Fast high-resolution miniature two-photon microscopy for brain imaging in freely behaving mice. *Nat Methods* **14**, 713-719, doi:10.1038/nmeth.4305 (2017).
- 65 Papagiakoumou, E., Ronzitti, E. & Emiliani, V. Scanless two-photon excitation with temporal focusing. *Nat Methods* **17**, 571-581, doi:10.1038/s41592-020-0795-y (2020).
- 66 Benjamin Mathieu, V. V., Walther Akemann, Stephan W. Evans, Mariya Chavarha, Jonathan Bradley, Dongqing Shi, Laurent Bourdieu, Michael Lin, and Stéphane Dieudonné. in *Biophotonics Congress: Biomedical Optics Congress 2018* (2018).
- 67 Tsang, J. M. *et al.* Fast, multiplane line-scan confocal microscopy using axially distributed slits. *Biomed Opt Express* **12**, 1339-1350, doi:10.1364/BOE.417286 (2021).
- 68 Schaffer, E. S. *et al.* The spatial and temporal structure of neural activity across the fly brain. *Nat Commun* **14**, 5572, doi:10.1038/s41467-023-41261-2 (2023).
- 69 Yang, W. & Yuste, R. In vivo imaging of neural activity. *Nat Methods* **14**, 349-359, doi:10.1038/nmeth.4230 (2017).
- 70 Scott, B. B., Brody, C. D. & Tank, D. W. Cellular resolution functional imaging in behaving rats using voluntary head restraint. *Neuron* **80**, 371-384, doi:10.1016/j.neuron.2013.08.002 (2013).
- 71 Aoki, R., Tsubota, T., Goya, Y. & Benucci, A. An automated platform for high-throughput mouse behavior and physiology with voluntary head-fixation. *Nat Commun* **8**, 1196, doi:10.1038/s41467-017-01371-0 (2017).
- 72 Wallace, D. J. *et al.* Rats maintain an overhead binocular field at the expense of constant fusion. *Nature* **498**, 65-69, doi:10.1038/nature12153 (2013).
- 73 Miura, S. K. & Scanziani, M. Distinguishing externally from saccade-induced motion in visual cortex. *Nature* **610**, 135-142, doi:10.1038/s41586-022-05196-w (2022).

Reviewer #1:

Remarks to the Author:

The revised manuscript “Competition of cortical clusters during on-demand visual learning in immersive virtual reality” by Judák, Dobos, Ócsai et al. addresses a significant amount of the comments from reviewers. The custom implementation of an alternative cell extraction method and detailed characterization of whisker contacts with the VR system are impressive. Two main issues remain, detailed here:

We thank the referee for her/his very positive comment on our manuscript.

1) The custom implementation of Suite2p on a “non-traditional” dataset was clearly not a trivial process, and the authors are applauded for their efforts. They report the result of using this cell extraction method: “In summary, our original method, the Suite2p algorithm without (87%) and with subtraction of neuropil contamination (84%), gave a neuronal activation ratio very similar to our original pipeline (97.6%, Ref. Fig. 5f).” However, I would argue that 84% and 97.6% are not very similar numbers, and that the results with Suite2p analysis appear closer to the trends apparent in the raw data from Figure 3D (the ramping effect is not visible in 100% of neurons). While the individual effect of neuropil subtraction was relatively small (~3%) on their measurements of activation ratio, a perceived lack of change in results by the authors may not be the best argument for use of custom code over a tool accepted by the field that promotes consistency across studies. Given the ability to apply this tool to their custom dataset, the use of this standard technique seems more appropriate than custom analyses.

We agree with the reviewer on the importance of using more standardized methods. Please also note that the Suite2p method is more recent than the original software (MES). Implementing all of these new methods into the core software requires considerable development time that is beyond the scope of the current manuscript. However, we already use Suite2p in our lab in combination with various 3D acousto-optical scanning methods (see **section 1.1** below). In addition, we have located all of the errors that made chessboard scanning incompatible with Suite2p and we now demonstrate the successful integration of Suite2p and chessboard scanning (see **section 1.2**). The relevance of this SW development is that fast 3D line and point scanning may be the fastest mode of 3D acousto-optical microscopy, but most laboratories use the significantly slower chessboard scanning because of its preserved visualization and the possibility of fast motion compensation. Therefore, combining Suite2p and chessboard scanning provides valuable new possibilities for these laboratories.

1.1. Suite2p is already used in combination with multiple 3D acousto-optical scanning methods

Although 3D acousto-optical microscopy acquires data along unique, precisely controlled 3D patterns, the data are stored in the widely used and well-documented HDF5 format. This allows Suite2p to run on the raw data without any special preprocessing. Accordingly, multiple scanning methods of 3D acousto-optical technology, such as multiple-frame scanning, 3D ribbon scanning, and high-speed arbitrary frame scanning, have been made compatible with Suite2p and are used in multiple laboratories worldwide. In addition, Suite2p seems to be an indispensable step in the analysis pipeline of 3D recorded dendritic voltage and calcium responses in in vivo conditions. For example, most of the ascending dendritic segments of LII/LIII and LV pyramidal cells in the V1 region of the cortex are organized in dendritic bundles: dendrites are intertwined and run together for relatively long distances. Therefore, classical analysis solutions can generate artifacts on data recorded with 3D ribbon scans along dendritic segments because different compartments of the dendritic bundle may be close to each other. However, Suite2p successfully separated these neighboring overlapping ROIs of dendritic bundles (**Ref. Fig. 1**, manuscript in prep.). Note that different segments of the dendritic bundle can generate very different ON and OFF signals and also respond differently to the stopping of moving grating stimulation.

Redacted

1.2. Making Suite2p compatible with chessboard scanning

Referee Figure 2. Manual and automatic cell detection using the original MES analysis pipeline and Suite2p. **a**, ROIs automatically detected with the Suite2p algorithm. Dendritic elements (also picked up by the algorithm) are omitted. **b**, The same chessboard scan from the MES analysis program with the contour of the manually selected neurons. Note that MES automatically omits pixels affected by motion correction, and thus they appear in black. These pixels are also omitted from further analysis. **c**, Enlarged view of the region marked with a dashed red line in panel **a**. Pixels considered by Suite2p as part of a given neuron and the surrounding neuropil region are labeled with red and green, respectively. Blue pixels were omitted because they moved out of the frame of the chessboard during imaging in the behaving mouse. **d**, Same as panel **c**, but for the region marked with a dashed red line in panel **b**.

Although manuscript data recorded with chessboard scanning using 3D acousto-optical microscopy are stored in the widely used and well-documented HDF5 format, which allows Suite2p to run on the raw data without any special preprocessing, we encountered some serious artifacts (e.g., the boundaries of the neuropil extended to the neighboring frames) that had to be corrected. **Ref. Fig. 2a** shows the result of the ROI detection performed by Suite2p on the measurement shown in **Fig. 3** in the current version of the manuscript. For comparison, **Ref. Fig. 2b** shows the manual cell selection used in the previous manuscript version. Although running Suite2p automatically was seamless with our data format, interpreting the results is a more complex issue. During the review process, we realized that there was something wrong with the algorithm as the signal-to-noise ratio of neuronal responses decreased: whereas 3D chessboard scanning showed a mean standard deviation of 0.023 ± 0.011 (mean \pm SD, $\Delta F/F$) with our analysis pipeline in the baseline period, Suite2p produced a larger mean standard deviation of 0.21 ± 0.13 (mean \pm SD, $\Delta F/F$) in the same baseline period. However, we could

not pinpoint the error because both software codes are very robust as they are the result of many years of development. Therefore, to compensate for the decrease in the signal-to-noise ratio, neurons with a standard deviation higher than $0.3 \Delta F/F$ were removed from the analysis. This approach preserved only 80.4% of the active neurons detected with the Suite2p algorithm in our previous analysis but still resulted in a much lower activation ratio (without neuropil subtraction: 87%, with subtraction of neuropil contamination: 84%) than with our standard MES protocol (97.6%). We agree with the referee that “84% and 97.6% are not very similar numbers”. Therefore, we conducted a comprehensive analysis to further discern the differences between results generated by Suite2p and MES. To identify and handle all potential issues, we repeated the entire pipeline from data export to the $\Delta F/F$ calculation step-by-step, systematically eliminating all possible sources of artifacts as detailed below:

i) Remove transients calculated from individual pixels that had zero value following motion correction

If ROIs move out of the imaging frame during scanning, due to brain movement, the ‘missing’ pixels are replaced with 0 values in the MES analysis program during motion correction. This motion correction is integrated into the MES and based on the ‘dftregistration.m’ function from Manuel Guizar-Sicairos et al. (*“Efficient subpixel image registration algorithms,”* 2008, Opt. Lett. 33, 156-158), which is a widely used algorithm in the field of functional imaging. However, in the case of full-field imaging, 0-value pixels appear at the edges of the images, representing only a small fraction of the total image and can be easily handled. In the case of chessboard scanning, pixels with zero values also appear in more central regions of the image close to the ROIs identified by Suite2p (see the vertical and horizontal black lines in **Ref. Fig. 2a** and **b**) and can be picked up, for example, as part of the neuropil or the ROI signal.

To handle this issue, we separately collected all the pixel coordinates determined to be part of the ROI by Suite2p (‘stat’ variable) and separately extracted the fluorescent transients corresponding to all individual pixels, instead of using the field averages of the ROIs in the ‘F.mat’ matrix from the outputs of Suite2p (**Ref. Fig. 3**). Some of these calcium transients reached zero value at certain points. To handle this artifact, transients that contained zero value were excluded from further analysis. Next, the mean raw Ca^{2+} response in a given ROI was calculated as the weighted average of the remaining pixels using the weight factor of ‘lem’ determined by the Suite2p algorithm. The excluded pixels are marked in blue in **Ref. Fig. 2**, while those that were part of the mean data are shown in red.

Referee Figure 3. Exemplified raw fluorescent transients from individual pixels. Note that some of the curves reached zero value after motion correction.

ii) Improved method for selecting neurons and associated neuropil regions during chessboard scanning

We realized that the majority of the differences between analysis with Suite2p and the original MES analysis pipeline originated from the boundaries of the neuropil of the detected ROIs determined by Suite2p: ROIs extended beyond the edges of the individual frames of the 3D chessboard scanning and, entered neighboring chessboard frames situated in different locations in 3D (**Ref. Fig. 4**). In more detail: in the first step, we searched in the code of Suite2p for the parameters determining the regions in which neuropil subtraction was performed (neuropil-ROI). In the next step, we plotted neuropil-ROIs on chessboard images and realized that i) they sometimes extended not only into one but into multiple neighboring frames, and ii) they could cover not only neuropil structures belonging to the center cell of the given chessboard frame but also a large part of the center somata of the adjacent chessboard frame situated in a different 3D position (**Ref Fig. 4**). To address this issue, we revisited the single pixel data from the raw measurements. Pixels for neuropil calculation were determined by the output of Suite2p. Pixels that were affected by motion correction or were outside of the chessboard frame of the given neuron were omitted from the analysis. The neuropil transient was then calculated as the average of the remaining pixels.

There is also a difference in the selection criteria between Suite2p and MES. Suite2p did not detect the neurons in 7 out of the 85 measured chessboard fields, although the cells were visible. This could be explained by the low activity and the relatively low responses of the neurons, which did not reach the detection threshold of Suite2p. These cells were omitted from further analysis.

Referee Figure 4. The output of the original Suite2p algorithm is visualized and overlaid on a chessboard scan. Pixels with detected neuronal activity are shown in red, while the borders of the regions with neuropil activity detected by Suite2p are shown in green. Note that most of the green circles cross the border of the frames (marked with dashed lines) and enter one or multiple adjacent chessboard

frames situated in different x, y, and z positions. Additionally, some of the green neuropil-ROIs partially cover neighboring somata, which could affect both raw neuronal and neuropil activity calculations.

iii) $\Delta F/F$ calculation

To compare different analysis approaches, we tested the following three methods on the same set of raw data:

- **The original analysis pipeline in MES (MES)**. This method is based on the manual selection of the cells, followed by neuropil subtraction using pixels outside of the selected ROI but within the frame of the chessboard of the ROI (**Ref. Fig. 2**).
- **Native output of Suite2p (original Suite2p)**. This method uses raw Ca^{2+} and neuropil data that Suite2p estimated from chessboard scans, and it was used in the previous version of the manuscript.
- **Suite2P customized for chessboard scanning (Suite2p for chessboard)**. This method averages the corrected pixels from the output of Suite2p to calculate corrected neuronal and neuropil components (see the details above).

The neuropil signal is accounted for by subtracting 0.7 times the raw neuropil curves from the raw Ca^{2+} transients before $\Delta F/F$ calculation (Chen TW et al., Nature, 2013, doi: 10.1038/nature12354). **Ref. Fig. 5** compares the different neuropil calculation methods. It is important to note that the native output of Suite2p can generate both negative (**Ref. Fig 5a**, blue curve) and positive errors (**Ref. Fig 5b**, blue curves) on chessboard scans. However, after eliminating pixels containing zero values after motion correction or because they extend into adjacent chessboards, the output (green trace, corrected Suite2p) was similar to the curve calculated by the original analysis pipeline in MES (red curves). The difference was also evident in average population responses (**Ref. Fig 5c**), although with a smaller amplitude, as negative and positive errors compensated each other.

Referee Figure 5. Comparison of the three different analysis methods: original analysis pipeline in MES, native output of the original Suite2p, and the version of Suite2p made compatible with chessboard scanning. a, Comparison of the corrected curves after neuropil subtraction for an example cell calculated with manual selection in MES (green), with automatic neuropil subtraction using the original version of Suite2p (blue), and after correction of the individual pixels for motion and

spatial overhang into an adjacent frame (Suite2p modified, red). Shaded regions represent mean \pm standard deviation. **b**, Same as panel **a**, but for a different cell. **c**, Same as **a**, but for the average of all neurons. Note the increased noise in the rising phase period for the original version of Suite2p. In contrast, responses calculated with the original analysis pipeline in MES overlap those calculated with the modified Suite2p.

To quantify the difference between the three different methods, cosine similarity was calculated between the transients calculated with the original MES analysis program (used throughout the manuscript) and both the original and the modified version of the Suite2p method (**Ref. Fig. 6a**). The cosine similarity to transients derived with the original MES analysis program increased (the ratio of cells with over 0.9 similarity increased from 68.6% to 74.3% after pixel correction). The increase was not significant at the level of the entire population ($p = 0.84$, Wilcoxon signed-rank test), but considering only the ‘problematic’ curves, below 0.95 in the cumulative distribution, the increase in the cosine similarity was significant ($p = 6.35 \times 10^{-7}$, Wilcoxon signed-rank test, **Ref. Fig. 6b,c**).

Referee Figure 6. Comparison of different neuropil calculation methods. **a**, Cosine similarity between the manual method compared to either the original Suite2p output or to the pixel-corrected calculation. Blue dots indicate individual cells, dashed red line represents the equality line. Points above the red line indicate cells with increased similarity. (X and Y axes are limited to the 0.75 to 1 range for better visibility, as neither distribution has values below 0.75.) **b**, Cumulative probability of the different methods. For large similarity, the change in the neuropil calculation method has a non-significant effect, as these curves are determined by the large neuronal responses and the neuropil has a relatively smaller effect on them. However, on curves with lower amplitudes, the effect of the neuropil is greater. **c**, If we consider only the ‘problematic’ curves, with cosine similarity, the increase in the cosine similarity is significant ($p = 6.35 \times 10^{-7}$, Wilcoxon signed-rank test) between the native Suite2p and the original curves based on manual selection, lower than 0.95.

iv) Calculation of the activation ratios.

Finally, we recalculated activation ratios. The neuronal activation ratio with the Suite2p method customized for chessboard scanning was 94.11% after learning (one-tailed t-tests, after 20-40 minutes training). When neuropil contamination was not subtracted, the activation ratio increased by 3% to 97.14% (one-tailed t-tests). According to the second referee’s question (see **section 2.2.2**), we recalculated activation ratios with the Wilcoxon signed-rank test, which resulted in 97.1% and 91.2% activation ratios without and with subtracting neuropil contamination ratios, respectively; these values are similar to our original pipeline (97.6%). This means that the increased cosine similarity between the chessboard scanning-customized Suite2P and our original MES analysis pipeline is also reflected in more similar activation ratios.

The small relative differences between the activation ratios calculated with and without subtracting the neuropil contamination (relative difference for t-test: 97.14% / 94.11% = 1.034 → **3.4%**, for Wilcoxon signed-rank test: 97.1% / 91.2% = 1.064 → **6.4%**) can be explained by the fact that AO microscopy allows precise adjustment of the z coordinates, which can result in lower neuropil contamination in the case of somatic measurements (**Ref. Fig. 7**).

Ref. Fig. 7. Illustration of the contribution of neuropil signals during laser scanning (This is Ref. Fig. 6 from the referee's response). **a**, Schematic of the recording of a model cell with a diameter of $10 \mu\text{m}$ using the PSF of a 3D AO microscope. The boundary between the model cell and the neuropil is indicated with a dashed red line. During laser scanning, the PSF is convolved with the model cell, which is illustrated in the x-z plane. As the PSF is asymmetric ($x=0.45 \mu\text{m} \times z=2.6 \mu\text{m}$), which is a characteristic of two-photon microscopy, this will result in an asymmetric distribution for regions with and without neuropil contamination. The pink color indicates the spatial distribution of the center of PSFs that partially excite the neuropil, while the spatial location of the centers of PSFs that do not reach the neuropil region is shown in green. **b**, The same spatial distribution of PSF centers with and without neuropil contamination as in panel a, but two imaging planes are also indicated. If the model cell is imaged in plane B, which is the center plane of the model cell, the neuropil contamination, which is the ratio of pink and green areas, is low. However, using a more distant plane (plane A) results in a high contamination ratio. **c**, Redacted

(See the detailed explanation in section 1.4.2 of the referee response)

In summary, we made 3D acousto-optical microscopy compatible with Suite2p by integrating the HDF5 file format. Suite2p is already widely used in various projects in 3D acousto-optical laboratories. Additionally, Suite2p seems to be an essential tool for analyzing dendritic signals recorded in behaving mice (**Ref. Fig. 1**). However, the original version of Suite2p was not compatible with chessboard scanning. With the modification we present here, it is now possible to analyze chessboard scanning data with Suite2p. Since chessboard scanning (Szalay et al. 2016) is a widely used and favored method in several laboratories, combining it with Suite2p can significantly advance research in this field.

2) The in-depth discussion of the King et al. paper is genuinely appreciated. In the course of this discussion, however, two issues arise. First, data are apparently being removed from the baseline period in the calculation of activation ratio ("intervals with spontaneous neuronal events were neglected" and "periods with spontaneous events (black stars in Ref. Fig. 2a) were removed before the baseline was calculated."). It is not clear that this is a scientifically justifiable practice and will certainly bias results toward "activation" of neurons since the recalculated baselines will inherently be lower.

2.1 We are grateful to the reviewer for highlighting this important issue. To respond to this question,

- i) we first explain the origin of the low neuronal activity in the baseline period, and then
- ii) demonstrate how spontaneous events were removed before the baseline was calculated.

i) Neuronal activity is low in the baseline period before ramp activity increases

In the previous referee's response, we calculated activation ratios as a function of time (**Extended Data Fig. 18a,c**, see also below). The average neuronal activity in the interval from -1.6 s to -0.6 s before the end of the aversive zone (0 s) was calculated and compared with the average response in a baseline period before ramp-like activity increases using a paired one-tail t-test. We then shifted the interval [-1.6 s to -0.6 s] in steps of one second along the x-axis in both directions and repeated the same paired t-test to calculate the activation ratio (**Extended Data Fig. 18c**). After a gradual increase in the ramping period, which corresponds to the average increase in neuronal activity shown in **Fig. 3f,h** of the current version of the manuscript, the activation ratio peaked at the time of the airpuff and then decreased to the baseline. However, the activation ratio also increased before the baseline interval and peaked at around 80-90% at about -30 s, before returning to the baseline value at about -45 s. **Fig. 4a** of the new version of the manuscript explains this phenomenon: motion in VR systems generally shows a periodic pattern, which means that mice repetitively return to the same zones where neuronal activation ratio is increasing, so that the peak at -30 s can be explained by an increase in activity when mice went to the previous zones. Consistent with this, the average spatial distribution of mice in the previous control and aversive zones peaked from -50 s to -28 s, and the end of this interval, it coincided with the peak in activation ratio at around -30 s. In summary, the presence in the previous zones explains the increase in activation ratio before the baseline period at about -30 s.

Behavior can be considered as a periodic pattern in VR systems as described above. Consequently, the activation ratio, which had a mean value of $38.3 \pm 4.0\%$, began oscillating between two states: it became high at the end of the zones to meet increasing computing demands during visual discrimination and after reinforcement, and became low between these periods. In the temporal interval shown in **Extended Data Fig. 18c**, the activation ratio reached three times the baseline value (indicated with red arrows) separating two intervals of the motion (defined as period #1 and #2 in **Extended Data Fig. 18c**). The average activation ratio calculated for the three baseline periods (red arrows in **Extended Data Fig. 18c**) was only $2.67 \pm 0.76\%$. This low activation ratio is also visible at the level of average neuronal responses at the end of the aversive zone after learning (30-40 minutes): the number of spontaneous events was low in the baseline period before minus 5 s (see the "*" marks in **Extended Data Fig. 18a**). The low activation ratio in the baseline periods and the consequently low number of events allowed a more precise definition of the baseline (see next section).

Extended Data Fig. 18a,c. Average neuronal responses and activation ratios at the end of the aversive zone after learning (30-40 minutes). *a*, Average neuronal responses at the end of the aversive zone from Fig. 3d. Responses were sorted according to their ramp-like component and are shown with a LUT set for small responses. Top, intervals with spontaneous activities (*) were excluded before baseline calculation. Bottom, average neuronal responses from the top (10 cells) and bottom (10 cells) end of the distribution shown in the top panel. *c*, The ratio of significantly active cells after 30-40 minutes of training calculated at 1-second intervals. The red arrow in the middle indicates the interval defined as the baseline. The two other red arrows indicate the intervals when activity had fallen to the baseline level. Zero time indicates the time when mice left the aversive zone. The activation ratio increased rapidly during the ramp period from a baseline activation ratio of 4.7% to a saturating activation ratio of 100%, peaked at the airpuff, and then returned to the baseline value. The periodic nature of the motion present in the previous zones explains the increase in activation ratio before the baseline period at about minus 30 s. The mean activation ratio in periods #1 and #2 was $38.3 \pm 4.0\%$ (red dotted line). The average activation ratio calculated for the three baseline periods (red arrows) was $2.67 \pm 0.76\%$. Therefore, to calculate a baseline activity corrected activation ratio, we can subtract 2.67% from the 97.6% and 100% activation ratios calculated above, giving baseline-corrected activation ratios of 94.93% and 97.33%, respectively.

The emergence of high and low activity states has also been described in previous studies. It has been shown that when a subset of neurons became responsive in the V1 region during visual learning, neuronal activity not only increases in the rewarded location but also decreases in the preceding baseline period (Referee Fig. 8. A,B, Pakan et al., 2018, <https://doi.org/10.1016/j.celrep.2018.08.010>). This rearrangement of network activity into high and low states is also visible at the level of individual cells: note the decreased spontaneous activity in the baseline period after learning (Referee Fig. 8B).

Redacted

Chase W. King et al. defined the activation ratio similarly to our method: they selected a short interval of 333 ms before and after the saccades, averaged the data, and compared the means using a pairwise statistical test (Wilcoxon signed-rank test). Similar to our study, there were conditions at the level of individual saccades when neuronal activity was high. In these cases, the neuronal activity in the preceding baseline period was low with a small number of individual events (indicated by red asterisks in **Ref. Fig. 9**). In contrast, when saccade-associated responses were low, the number of spontaneous events in the preceding baseline period was high (indicated by red asterisks in **Ref. Fig. 9**). This inverse relationship between the neuronal responses associated with saccades and the preceding baseline activity can also be observed at the level of average response groups: if, before time zero, the mean response of a group of saccades was below the baseline calculated by the bootstrapping method, the saccade-associated responses after 0 s were higher and vice versa (see the red and blue arrows in **Ref. Fig. 9**). In summary, similar to our study, previous works with periodic behavior protocols also validate the emergence of baseline periods with very low activation ratios. During these periods, the sparse activity of the neuronal network, characterized by a low number of spontaneous events, allows for a more precise definition of the baseline compared to the current methods based on averaging (see next section).

Redacted

ii) Method for more precisely calculating the baseline during low network activity states

Motivation: to more precisely calculate near-saturating activation ratios at the end of the aversive zone, we need a more accurate method for baseline calculation. As described in the previous section, the activation ratio was very low in the baseline period before ramp-like activity increases. Only a few trials of a few neurons showed spontaneous activity increases in the baseline period before -5 s (**Ref. Fig. 10**).

AVERAGE AVERSIVE RESPONSES

Referee Fig. 10. Top, Average neuronal responses at the end of the aversive zone from Fig. 3d (The same as Ref. Fig. 2 of the previous response). Spontaneous activity before the ramp-like activity increases in a baseline period indicated with black asterisks. Note the sparse distribution of the spontaneous activity. Bottom, individual transients from exemplified neurons that were marked with asterisks in the top panel. Red indicates transients with spontaneous activity. Magenta indicates transients that show activity that increases at the end of the baseline period from around minus 5 seconds. Note that before and after spontaneous activity increases, Ca^{2+} responses return to the baseline.

This means that individual spontaneous events were well separated from the baseline, which allowed the detection and elimination of these events from baseline calculations using different mathematical approaches. Here we demonstrate only one method: the Peak Analyzer (Origin, OriginLab). In the first step, we used the Asymmetric Least Squares Smoothing Method of the Peak Analyzer to fit a baseline curve (**Ref. Fig. 11b**). After subtracting the baseline curve, peaks were detected by the Local Maximum Method of the Peak Analyzer (**Ref. Fig. 11c**) and fitted with the Fit Peaks Method (**Ref. Fig. 11c**) by using a fixed $y_0=0$ value. Then, a cumulative fit curve was calculated in the Peak Analyzer as the sum of all individual fits. Intervals where the cumulative fit curve was larger than zero were excluded from the baseline recalculation (**Ref. Fig. 11d**). In the remaining intervals, in which the cumulative fit curve was zero (defined as BL period and indicated as “BL₁₋₃” in **Ref. Fig. 11**), the original trials – without Asymmetric Least Squares Smoothing and subtraction – were averaged and the baseline was reset to 0 during $\Delta F/F$ calculation. Intervals for baseline calculation with zero cumulative fit value were accepted only from -10 s to -5 s. If the cumulative fit curve was non-zero in the entire range, the baseline calculation period was extended from -15 s to -5 s. The entire process

was repeated on each trial of each individual cell. Data and activation ratios did not change when we kept only a 3-second BL period, the closest interval to the minus 5 s time.

Referee Fig. 11. Elimination of spontaneous activity from baseline calculation. **a**, An example cell from Ref. Fig. 10 with seven example trials before the end of the aversive zone. Only one of the trials (red) shows spontaneous activity in a baseline period before -5 s. **b**, The red transients as in panel **a** but on a wider temporal scale. Peak Analyzer (Origin, OriginLab) was used to detect spontaneous activity. The baseline was fitted with the Asymmetric Least Squares Smoothing Baseline method in Peak Analyzer. **c**, The same as panel **b** but with the baseline subtracted. Three peaks were detected at 15.58 s, 10.89 s and 0.66 s using the Local Maximum method with “both directions” settings in Peak Analyzer. Peaks were fitted using the Fit Peak method in Peak Analyzer ($\chi^2=1.04\times 10^{-3}$, Adj. R-Square= 9.4×10^{-1} , SS= 6.3×10^{-1} , $y_0=0$, OriginPro, OriginLab). The cumulative fit curve is shown in black. **d**, Intervals with non-zero cumulative fit curve (black trace) value (>0) were excluded from the baseline calculation (gray bars). This method identified 3 baseline intervals (BL₁, BL₂ and BL₃) from which the last (BL₃) was accepted as the closest interval to minus 5 s. Data for baseline subtraction during $\Delta F/F$ calculation were derived from the original raw data (before Asymmetric Least Squares Smoothing Baseline subtraction). This process was used for each trial. BL_i intervals were accepted from -15 s to -5 s to calculate the baseline for a given trial. The efficiency of subtraction of spontaneous events is also shown in Referee Fig. 12. **e**, Further validation of the peaks detected with Peak Analyzer in the baseline period (P₁, P₂). SD values were calculated in the corresponding BL_i interval before the peak. Peak amplitudes were above the $2\times SD$ threshold, supporting the need for the elimination of spontaneous activity related to P₁, P₂ peaks.

Thanks to the low activation ratios in the baseline periods, there are several alternative approaches to eliminate spontaneous activity and calculate a more precise baseline. For example, using hidden Markov chains, we developed MLSpikes (Thomas Deneux et al. 2016, Nature Comm., DOI: 10.1038/ncomms12190), a method that provides the unique spike train that maximizes the likelihood of obtaining the recorded fluorescence time series. To do so, we used a version of the Viterbi algorithm to estimate the optimal input (the spike train) by maximizing the posterior distribution probability. One of the main advantages of this method is that it can precisely calculate the baseline even in noisy conditions and during high-frequency

trains. However, it requires complex calibration. Fortunately, the low activation ratios in our measurements allowed for the use of simpler approaches.

Second, one-sample one-tailed t-tests are being used to calculate whether neurons show significant activation. The use of a one-tailed t-test inherently assumes that neural activity will go in one direction (in this case, up). Neural activity could go up, down, or show no change, and it is the authors' assumption that the response should go up, so the choice in statistical test could bias the result, and a two-tailed test would be more appropriate.

2.2. We are grateful to the reviewer for highlighting this important issue. To address this question, we first explain the use of a one-tailed t-test (**section 2.2.1**), then we will demonstrate a more stringent non-parametric statistical test (Wilcoxon signed-rank test) to consider both directions of activity changes, and also to minimize assumptions about the underlying distribution of the data (**section 2.2.2**).

2.2.1 The low number of spontaneous events in the baseline period allowed a more precise definition of the baseline and is associated with unidirectional neuronal activity increases that support the use of the one-tailed t-test.

We agree with the referee that in most studies both directions of neuronal activity changes need to be considered, because: i) some neurons can have high baseline activity at time zero when a given effect is measured, and ii) these cells can also receive inhibitory inputs at the same time. Therefore, these cells can decrease their activities: neuronal activity can also go down and this can be detected with GCaMP sensors. Indeed, this is also the case in the two exemplified studies: in **Ref. Fig. 8B** neuronal activity is high during the baseline period and drops at the time of the reward in the novice mouse. Similarly, in the second example: in case of some trial types (marked with a blue asterisk in **Ref. Fig. 9** bottom) neuronal activity is high in the baseline period and drops at time zero; this decrease is also visible in population averages (blue arrow in **Ref. Fig. 9**). However, in both cases, when neuronal network activity switched to a high activation state (Chase W. King et al.: marked with red asterisks in **Ref. Fig. 9**; Pakan et al. 2018: the bottom panel in **Ref. Fig. 8B** with the expert mouse; see also the previous point), the preceding baseline network activity was low: high frequency bursts were missing or did not fill the entire baseline period (marked with red asterisks in **Ref. Fig. 9**); therefore periods with zero – or sub-detection threshold – activity emerged clearly. These periods can then be used to re-define the baseline from which neuronal activity can deviate only into one direction: only upward.

Referee Fig. 12. a, Average neuronal responses at the end of the aversive zone from **Fig. 3d** (the same as **Ref. Fig. 2** of the previous response) with spontaneous activities (*) in the baseline period before ramp-like activity increases. **b**, The same as panel **a**, but spontaneous responses were subtracted in the baseline period before averaging using the method shown in **Referee Fig. 11**. Note that the remaining activity resulted in a smooth baseline.

Similarly to the two exemplified studies, we also recorded very low neuronal activity in the baseline period after learning with an activation ratio of $2.67 \pm 0.76\%$ and with a low number of spontaneous events in a few cells (**Ref. Fig. 10**). This allowed for the subtraction of spontaneous events (**Ref. Fig. 11**) and the definition of a new, spontaneous event-free baseline (**Ref. Fig. 12**). From this redefined baseline, activity could deviate only in one direction: upward. To further validate that the lack of continuous high-frequency firing in the baseline period would not interfere with our method for baseline recalculation, we used Suite2p to detect neuronal responses after 20–40 minutes of learning (**Ref. Fig. 13a**). Only one trial in one neuron (cell #24) showed burst-like elongated neuronal activity (marked with a black asterisk in **Ref. Fig. 13a**) that could potentially generate a negative transient at the time of ramp-like activity. Plotting other trials of cell #24 suggested that there is a spontaneous event-free baseline period before minus 11 s that can be used as a new baseline of the trial ($\approx BL_1$ in **Ref. Fig. 13b**). To validate this baseline period, we used Peak Analyzer (Origin, OriginLab) as above (see **section 2.2**). After Asymmetric Least Squares Smoothing (**Ref. Fig. 13c**) and baseline subtraction, peaks were detected and fitted with Peak Analyzer (**Ref. Fig. 13c**). We used the $y_0=0$ criterion for the cumulative sum to determine a baseline period (BL_1). Using this baseline period, the average of all trials yielded a ramp-like amplitude of 0.36 ± 0.05 (mean \pm SEM) which was significant in a one-tailed t-test ($p=0.00011$) and also in the Wilcoxon signed-rank test ($p=0.00915$). The ramp-like activity increase of cell #24 remained significant when the trial with

burst-like activity was removed (one-tailed t-test: $p=0.00043$; Wilcoxon signed-rank test: $p=0.014$).

Referee Fig. 13. **a**, Individual neuronal responses from nine trials at the end of the aversive zone after learning. Neuronal responses were detected with Suite2p and sorted as a function of cell number. Only one trial of one cell (cell #24), showed an elongated burst-like activity pattern (*). **b**, Individual trials of cell #24. The red trace shows the trial with a burst-like activity pattern, indicated with a black asterisk in panel **a**. **c**, The same trial with a burst-like activity pattern but on an expanded scale. The baseline was fitted with the Asymmetric Least Squares Smoothing Baseline method in Peak Analyzer (dashed blue line). **d**, Peaks (P_i) were detected using the Local Maximum method with “both directions” settings and were fitted using the Fit Peak method in Peak Analyzer (green traces). The sum of fits for all peaks is shown in black (cumulative fit peak). The zero value of the cumulative fit peak curve defined a baseline interval before -5 s, indicated with blue brackets (BL_1). The baseline of this trial was set to zero in the BL_1 interval during the $\Delta F/F$ calculation, using the original raw data before implementation of the Asymmetric Least Squares Smoothing Baseline method.

2.2.2 Using Suite2p and Wilcoxon signed-rank test in combination to calculate activation ratios

To respond to the referee’s request to consider both directions of activity changes and also to minimize assumptions about the underlying distribution of the data, we used the Wilcoxon signed-rank test to validate significant activity increases in individual neurons. We used the customized Suite2p method (described in **section 1.2**), which was corrected at the level of individual pixels for motion and for the over-exceeding neuropil-ROIs that spread to the neighboring chessboard frames situated in a different 3D position. The neuronal activation ratio (with Wilcoxon signed-rank test) was calculated in a broader temporal interval (20-40 minutes of training), giving 97.142% and 91.17% without and with subtracting neuropil contamination, respectively.

Furthermore, if one-sample t-tests will be used on every neuron, multiple comparisons will need to be accounted for to prevent false positives. In this case, with approximately 85 neurons, the significance level should be on the order of $p < 0.0006$ using Bonferroni correction.

3.1. Short answer

We thank the referee for this valuable comment on the level of significance of the statistical test. We are of the opinion that the Bonferroni correction can indeed be very useful in cases where additive type I errors affect the efficiency of the statistical test (What's wrong with Bonferroni adjustments. *BMJ*. 1998 Apr 18;316(7139):1236-8. doi: 10.1136/bmj.316.7139.1236. PMID: 9553006; PMCID: PMC1112991). Indeed, it is a useful adjustment applied to p-values when two or more statistical analyses have been performed on the same sample of data in order to avoid problems arising from the fact that the familywise type I error rate can be larger than the per-analysis error rate. However, it is very important to note that when calculating activation ratios, we did not compare the expected values of different groups (i.e., different cells) and did not perform the tests on the same sample of data. Our null hypothesis was not to have equal expected values of the increased activity between different cells. When calculating activation ratios, we did not make a claim by comparing the increase of different individual neurons, but we looked at the increased activities of each cell separately, relative to their own baseline (for $n=7$ trials per one cell), and thus obtained significant increases for 83 out of 85 cases (corresponding to a 97.64% activation ratio).

3.2. A more detailed explanation with examples

Following your example, we would like to demonstrate the significance of the Bonferroni correction with an example involving the statistical probability of events that can be defined as a product of multiple comparisons to have a number-independent statistical value. (Of course, the activities of neurons in close proximity to each other are not statistically independent events in reality, but for simplicity we will consider them as such and approximate their occurrence by the product of their occurrences.) Let us calculate whether there is any neuron that is activated during a baseline period. If a neuron is not active under certain conditions in a given statistical test with a given p value, it means that the probability of being activated is p and the probability of not being activated is $1-p$. Let us define $p = 0.0001$ as an example. The probability to have at least one active cell (defined as P) in the population is:

$$P = 1 - (\text{none of the cell is active}) = 1 - (\text{a cell is not active})^N = 1 - (1 - p)^N = 1 - 0.9999^N$$

The term:

$$P = 1 - (1 - p)^N$$

can be expressed in Taylor series as:

$$\begin{aligned} P = 1 - (1 - p)^N &= 1 - \sum_{k=0}^N \binom{N}{k} (-1)^k p^k = 1 - \left(1 - Np + \frac{N(N-1)}{2} p^2 - \frac{N(N-1)(N-2)}{6} p^3 + \dots p^4 + \dots \right) = \\ &= Np + \frac{N(N-1)}{2} p^2 - \frac{N(N-1)(N-2)}{6} p^3 + \dots p^4 + \dots \end{aligned}$$

As p is a small number, the quadratic ($p^2 = 0.0001^2 = 0.0000001$) and higher -order components ($p^3 = 0.0001^3 = 0.000000001\dots$) are typically neglected, which results in the following simple equation:

$$P = Np$$

This means that the probability P increases proportionally with the N value: with the number of cases. Therefore, the output of the statistical test can be easily 'manipulated' with the number of cases. For example, using the previously defined $p = 0.0001$ with 10 cells, the probability of having an active cell is $P = p \times 10 = 0.001$, which is 0.1%. However, with 100 cells, the probability is already 1%, and for 1000 cells, we have a 10% probability. The advantage of the Bonferroni correction is that it removes N dependency:

$$P_{Bonferroni} = \frac{Np}{N} = p$$

In this simplified example, the original hypothesis can be described as a product of individual hypotheses:

$$P = 1 - \prod_{i=1}^N \tilde{P}(p_i \leq \alpha)$$

which, following the Bonferroni correction, results in an N -independent value as it was revealed by the Taylor series. There are many similar statistical questions where the final probability function can be described similarly as a product of individual hypotheses and, therefore, would create the need for using the Bonferroni correction. For example, we could ask whether all neurons were activated during the ramp period in the aversive or control zones. However, we did not address this type of statistical question, but rather wanted to determine the average ratio of active cells, which can be expressed not as a product but as a sum:

$$P' = \frac{\sum_{i=1}^N \tilde{P}(p_i \leq \alpha)}{N}$$

Here normalization with N is 'inherently' included, and therefore the activation ratio is independent of N .

To put this oversimplified mathematical explanation into concrete numbers, we would like to highlight the mathematical problem with the use of the Bonferroni correction in our dataset:

- i) The activation ratio using our original analysis pipeline in MES was: 97.64% ($n=85$). If we use the Bonferroni correction, then $p' = 0.05/85 = 0.00058$, but the smallest p value of the t-tests for the 85 cells was higher: $p=0.00183$. Therefore, the activation ratio with the Bonferroni correction would result in zero activated cells. If we return to the logic described in the previous paragraph, this statistic with the Bonferroni correction would better characterize the statistical question that is related to the product of the probabilities and this question would sound like: "*Were all neurons activated?*" For this question the statistic with the Bonferroni correction responded with no.
- ii) Next, we simulated the recording of a smaller number of cells, by subsampling the number of recorded cells. If we recorded only 5 cells, then the activation ratio calculated from the data in Fig. 3 would be 14.4%, with 4 cells: 24.7%, with 3 cells: 38.8%, and with 2 cells: 64.7%. This means that in contrast to the original statistical analysis, the Bonferroni correction adds a strong N dependency to the statistical test, decreasing the activation ratio rapidly to zero (**Ref. Fig. 14**). In contrast, without the Bonferroni correction the average activation ratio is independent of the total number of recorded cells (**Ref. Fig. 14**). (This is a simplified explanation that does not include many effects, such as the change in the SNR as a function of the number of cells recorded).

Ref. Fig. 14. Neuronal activation ratio as a function of the number of recorded cells with and without Bonferroni correction.

In summary, as the neuronal activation ratio can be described as an average and not as a product of probability functions of multiple hypotheses tested, it inherently includes independence from the total number of recorded cells and the Bonferroni correction is not required.

In sum, there appears to be a very interesting and real result about changes in neural activity with very rapid learning here, however it is suggested that the authors use caution not to overinflate the incidence of this effect across the neuronal population and instead rely on standard statistical techniques to determine the prevalence of this signal in the population.

We thank the referee for her/his very positive comment on our manuscript. The statistical errors and interpretations have been corrected according to the referee's request.

Reviewer #2 (Remarks to the Author):

The authors have addressed the reviewers concerns comprehensively. I have no further suggestions for improvement

Thank you for your thorough review and for acknowledging our efforts in addressing the concerns. We appreciate your time and support.

Reviewer #3 (Remarks to the Author):

The authors made a significant improvement to the manuscript and addressed most of my concerns.

Thank you for her/his positive feedback and for supporting the publication of our manuscript. We greatly appreciate your comments and are glad that the revisions meet your expectations.

References

Barson, D. *et al.* Simultaneous mesoscopic and two-photon imaging of neuronal activity in cortical circuits. *Nat Methods* **17**, 107-113, doi:10.1038/s41592-019-0625-2 (2020).

Judak, L. *et al.* Sharp-wave ripple doublets induce complex dendritic spikes in parvalbumin interneurons in vivo. *Nat Commun* **13**, 6715, doi:10.1038/s41467-022-34520-1 (2022).

Guizar-Sicairos, M., Thurman, S. T. & Fienup, J. R. Efficient subpixel image registration algorithms. *Opt Lett* **33**, 156-158, doi:10.1364/ol.33.000156 (2008).

Pakan, J. M. P., Currie, S. P., Fischer, L. & Rochefort, N. L. The Impact of Visual Cues, Reward, and Motor Feedback on the Representation of Behaviorally Relevant Spatial Locations in Primary Visual Cortex. *Cell Rep* **24**, 2521-2528, doi:10.1016/j.celrep.2018.08.010 (2018).

King, C. W., Ledochowitsch, P., Buice, M. A. & de Vries, S. E. J. Saccade-Responsive Visual Cortical Neurons Do Not Exhibit Distinct Visual Response Properties. *eNeuro* **10**, doi:10.1523/ENEURO.0051-23.2023 (2023).

Deneux, T. *et al.* Accurate spike estimation from noisy calcium signals for ultrafast three-dimensional imaging of large neuronal populations in vivo. *Nat Commun* **7**, 12190, doi:10.1038/ncomms12190 (2016).

Perneger, T. V. What's wrong with Bonferroni adjustments. *BMJ* **316**, 1236-1238, doi:10.1136/bmj.316.7139.1236 (1998).